# Forced changes in the Pacific Walker circulation over the past millennium

Georgina Falster[1,2,5 ✉], Bronwen Konecky[2], Sloan Coats[3] & Samantha Stevenson[4]

The Pacific Walker circulation (PWC) has an outsized influence on weather and climate worldwide. Yet the PWC response to external forcings is unclear[1,2], with empirical data and model simulations often disagreeing on the magnitude and sign of these responses[3]. Most climate models predict that the PWC will ultimately weaken in response to global warming[4]. However, the PWC strengthened from 1992 to 2011, suggesting a significant role for anthropogenic and/or volcanic aerosol forcing[5], or internal variability. Here we use a new annually resolved, multi-method, palaeoproxy-derived PWC reconstruction ensemble (1200–2000) to show that the 1992–2011 PWC strengthening is anomalous but not unprecedented in the context of the past 800 years. The 1992–2011 PWC strengthening was unlikely to have been a consequence of volcanic forcing and may therefore have resulted from anthropogenic aerosol forcing or natural variability. We find no significant industrial-era (1850–2000) PWC trend, contrasting the PWC weakening simulated by most climate models[3]. However, an industrial-era shift to lower-frequency variability suggests a subtle anthropogenic influence. The reconstruction also suggests that volcanic eruptions trigger El Niño-like PWC weakening, similar to the response simulated by climate models.

The PWC is the zonal component of atmospheric circulation over the tropical Pacific. The PWC may be characterized by a sea-level pressure (SLP) gradient ($\Delta$SLP) across the equatorial Pacific, with deep convection over the Indo-Pacific warm pool, subsidence over the equatorial eastern Pacific, upper-tropospheric westerlies and surface easterlies (the Pacific trade winds). Tightly coupled to tropical Pacific sea-surface temperature (SST), the PWC forms the atmospheric component of the El Niño–Southern Oscillation (ENSO), the dominant mode of global interannual climate variability. Despite its importance to global climate, both the PWC's response to external radiative forcings and its intrinsic variability are poorly understood[2,6]. For example, no consensus has emerged as to whether anthropogenic forcing has strengthened the PWC[7,8], weakened it[9–11] or had no detectable influence[12]. Most observational datasets indicate that the PWC strengthened considerably between around 1992 to 2011, in a trend to more 'La Niña-like' conditions[5,13]. However, it is unknown if this strengthening was externally forced or the result of intrinsic variability[2,8], in part because the strengthening is consistently absent from climate model simulations[3,14].

The high intrinsic variability of the PWC is a substantial obstacle to detecting forced changes[6], as observational records are too short to robustly characterize the two[9]. Annually resolved ENSO reconstructions have allowed assessment of the response of ENSO to volcanic eruptions, that is, the largest preindustrial forcing of the past millennium[15]. However, the tropical Pacific SST response to volcanic forcing remains contentious[16], and similar assessments have not been possible for the PWC, as atmospheric variability is notoriously difficult

to reconstruct without complex proxy-system transformations[17,18]. Existing inferences of preindustrial PWC variability[19–21] are derived from approximately decadally resolved records that rely on a mix of proxy sensors sensitive to different aspects of hydroclimate (rather than atmospheric circulation directly) and are of too low resolution to assess interannual variability.

## PWC reconstruction approach

Here we contextualize observational-era PWC variability with a new annually resolved reconstruction of the PWC from 1200 to 2000, derived from 59 palaeoclimate proxy records and including 4,800 ensemble members that sample uncertainty from observational data, reconstruction method and record chronologies. Our target variable was anomalies in the trans-Pacific $\Delta$SLP (ref. 11), which has been used in many studies to quantify the PWC (Fig. 1b; Methods). $\Delta$SLP anomalies were calculated relative to 1960–1990. Higher $\Delta$SLP values represent a stronger PWC, which broadly corresponds to more 'La Niña-like' atmospheric conditions; lower $\Delta$SLP values represent a weaker PWC, or more 'El Niño-like' conditions.

The first mode of observed global interannual precipitation $\delta^{18}O$ over 1982–2015 is significantly ($P < 0.05$) correlated with and explains 55% of the $\Delta$SLP variance[22]. This is the case even though many individual precipitation $\delta^{18}O$ records are not highly or significantly correlated with $\Delta$SLP (ref. 22) and supports the use of a non-local reconstruction approach. The $\Delta$SLP imprint in global precipitation $\delta^{18}O$ arises from

[1]Australian Research Council Centre of Excellence for Climate Extremes, Canberra, Australian Capital Territory, Australia. [2]Department of Earth and Planetary Sciences, Washington University in St. Louis, St. Louis, MO, USA. [3]Department of Earth Sciences, University of Hawai'i at Mānoa, Honolulu, HI, USA. [4]Bren School of Environmental Science and Management, University of California, Santa Barbara, Santa Barbara, CA, USA. [5]Present address: Research School of Earth Sciences, Australian National University, Canberra, Australia. ✉e-mail: georgina.falster@anu.edu.au

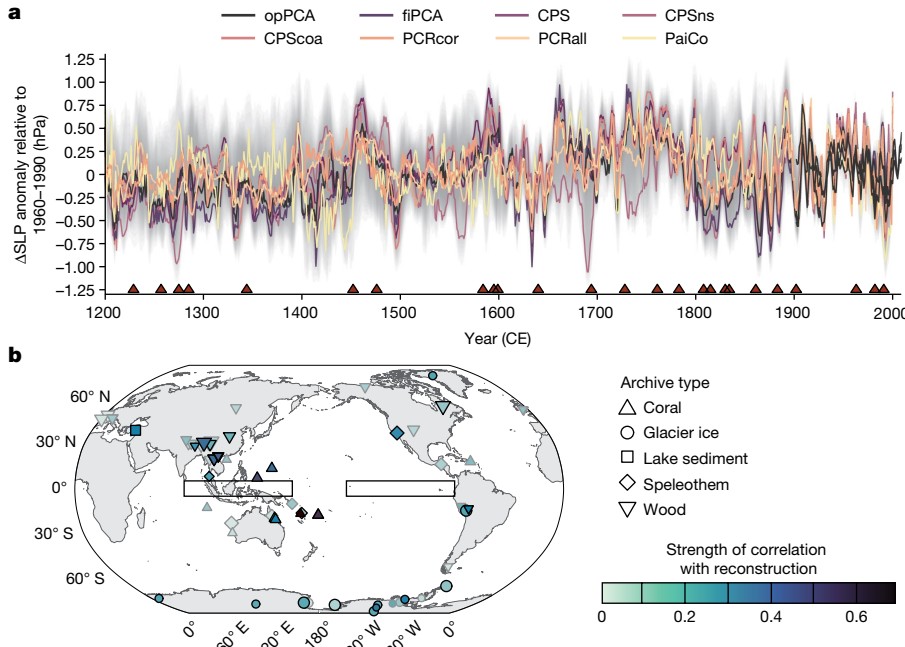

**Fig. 1 | Ensemble reconstruction of the PWC (in terms of the trans-Pacific ΔSLP) from 1200 to 2000 CE. a**, ΔSLP anomalies relative to 1960–1990, with a 5-year running mean applied. Grey shading represents the 2.5th/97.5th quantiles for the full ensemble ($n = 4,800$). Coloured lines show the ensemble median for each reconstruction method. Black lines show instrumental data for 1900–2010, from HadSLP[25], ICOADS[26] and ERA-20C (ref. 27). Triangles denote volcanic eruptions with reconstructed SAOD $\geq 0.05$ (ref. 35). CPS, composite plus scale; CPScoa, CPS using only records in a tropical Pacific 'centre of action'; CPSns, CPS using only records without a known seasonal bias; fiPCA, 'full-interval' principal component analysis; opPCA, 'overlap-period' principal component analysis; PaiCo, pairwise comparison; PCRall, principal component regression using all proxy records; PCRcor, principal component regression using only records significantly ($P < 0.1$) correlated with the training ΔSLP index in the calibration window. **b**, Locations of proxy records used in the ΔSLP reconstruction. Shapes correspond to archive type; fill shows the absolute correlation of that record with the ΔSLP reconstruction ensemble median across the interval in which that record contributed to the reconstruction (that is, the temporal segments; see Methods). Point size scales with record length. Black outline denotes that the proxy record is significantly ($P < 0.05$) correlated with the ΔSLP reconstruction ensemble. Black rectangles show regions used to calculate ΔSLP. Map created in R, using coastlines from Natural Earth.

several well-documented processes, including PWC-related changes in moisture source and transport length, and a PWC-driven or ENSO-driven 'amount effect' in tropical regions. Global precipitation δ¹⁸O variability is more strongly correlated with the PWC than with ENSO. This is probably because PWC-related changes in atmospheric circulation directly affect precipitation δ¹⁸O, whereas SST changes must be transmitted to precipitation δ¹⁸O by means of atmospheric processes[22].

We therefore reconstructed ΔSLP from 54 globally distributed annually or sub-annually resolved proxy records for the stable isotopic composition of precipitation and other meteoric waters ('water isotopes') and five annually resolved non-isotope-based palaeoclimate records that have a strong mechanistic relationship with the PWC or ENSO (Supplementary Table 1 and Extended Data Figs. 1 and 2; Methods). The reconstruction uses the Iso2k database[23], an innovative global synthesis of water-isotope proxy records. Iso2k includes data from diverse archive types and allows ready integration of water-isotopic signals into palaeoclimate reconstructions. Although not all water-isotope proxy records directly reflect precipitation δ¹⁸O variability, it is the primary driver of variability for most records used in this reconstruction[23]. The availability of continuous annually resolved records decreases rapidly back through time; to maximize information incorporated into our reconstruction, we performed the reconstruction in five temporal subsets (1200–2000, 1400–2000, 1600–2000, 1800–2000 and 1860–2000), in each case using proxy records with >66% coverage over that interval (Extended Data Figs. 1 and 2).

Palaeoclimate reconstructions are sensitive to both reconstruction method and the observational data used for training the reconstruction[24]. We therefore took a comprehensive, ensemble-based approach to reconstructing ΔSLP that accounts for these and other uncertainties.

We used five statistical methods: composite plus scale (CPS), principal component regression (PCR), pairwise comparison (PaiCo) and two variants of principal component analysis (PCA): (1) an 'overlap-period' PCA (opPCA), in which the first principal component of the proxy data is calculated over the calibration interval, then the loadings are projected over the full length of the time series, and (2) a 'full-interval' PCA (fiPCA), in which the first principal component of the proxy data is calculated over the full reconstruction interval. We performed the PCR reconstructions using (1) all proxy records and (2) the subset of proxy records correlated significantly ($P < 0.1$) with ΔSLP in the calibration window. We performed CPS reconstructions using the entire proxy dataset, as well as two subsets: (1) only proxy records in a broad tropical Pacific 'centre of action' (CPScoa) and (2) only proxy records that do not have a known bias to a particular season (CPSns). For each statistical method, we trained the reconstructions on ΔSLP calculated from three gridded SLP products: the Hadley Centre SLP dataset (HadSLP[25]), the International Comprehensive Ocean-Atmosphere Data Set SLP dataset (ICOADS[26]) and SLP from the ERA twentieth-century reanalysis (ERA-20C (ref. 27)). In all cases, we used a 1900–2000 calibration interval. We explicitly incorporated chronological uncertainty by sampling many realizations from a banded age–depth model ensemble for each record[28], thus propagating chronological uncertainty through subsequent analyses. For each iteration of the reconstruction ensemble, we randomly removed up to 15% of the available records to account for possible dependence of results on a particular subset of proxy records. Finally, we assessed reconstruction skill by creating a second set of reconstructions with a 1951–2000 calibration interval, then quantifying performance in an independent 1900–1950 interval (see Methods for a full description of the reconstruction methodology).

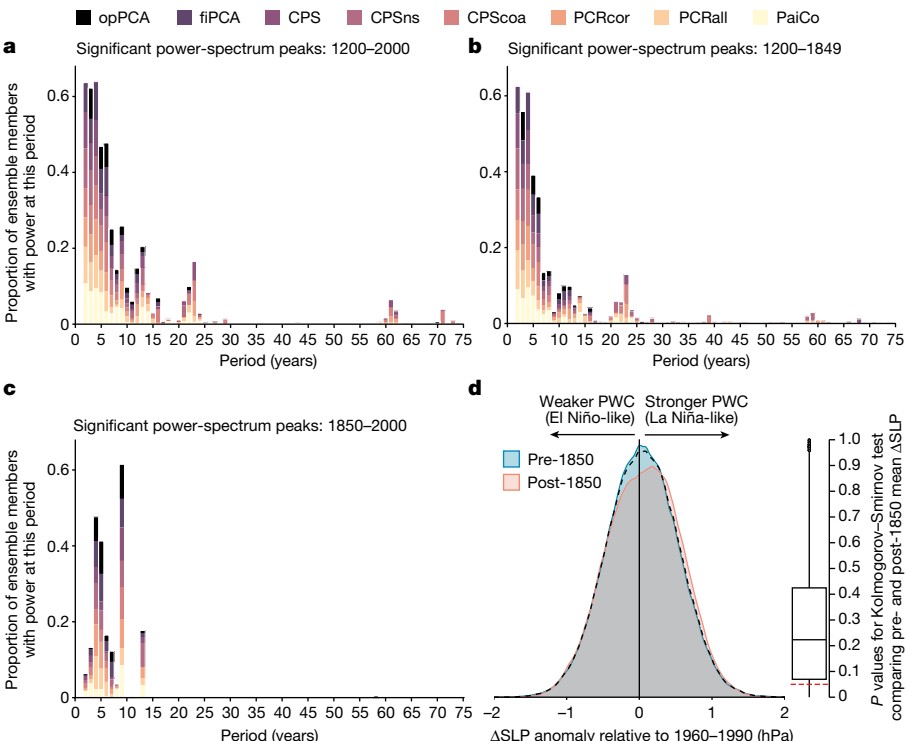

**Fig. 2 | Temporal characteristics of ΔSLP throughout 1200–2000 CE.**
**a**, Proportion of the 4,800 ΔSLP reconstruction ensemble members with significant (*P* < 0.05) power in periods from 1 to 75 years. Significance is evaluated against a power-law null (Methods). Colours denote reconstruction method. **b**, As per **a** but for 1200–1849, in all possible 150-year segments. The division into 150-year segments was to enable direct comparison with the power spectrum in the industrial era (Methods). **c**, As per **a** but for 1850–2000. **d**, Left: distribution of ΔSLP anomalies for 1200–2000 (summarizing all values from all individual reconstruction ensemble members). Cyan distribution shows preindustrial values (1200–1849). Salmon distribution shows industrial-era values (1850–2000). Dashed black line shows the distribution for the full reconstruction interval (1200–2000). Right: box plot summarizing *P* values from two-sample Kolmogorov–Smirnov tests of whether the post-1850 mean is different from the pre-1850 mean, performed on all 4,800 ΔSLP reconstruction ensemble members. Box shows median and interquartile range (IQR), whiskers show IQR × 1.5, and points show outliers. Dashed red line denotes *P* = 0.05.

The reconstruction closely tracks ΔSLP in the observational era (Extended Data Fig. 3); reconstruction ensemble median ΔSLP is highly correlated with mean ΔSLP from the three gridded SLP products (*r* = 0.81; Extended Data Table 1). The correlation remains high when assessing reconstructed ΔSLP against observed ΔSLP in an independent interval (*r* = 0.77; Extended Data Table 1). Uncertainty in the ensemble arises from uncertainties in the gridded SLP products (Extended Data Figs. 4a and 5a), as well as the statistical method used to calculate the reconstructions (Extended Data Figs. 4b and 5b). Skill decreases prior to around 1600 (Extended Data Figs. 4c and 5c); this decrease in skill back through time is because of decreased data coverage and increased chronological uncertainty (see Methods for a full accounting of reconstruction skill). We restrict our main findings to those robust relative to the reconstruction uncertainty.

## Preindustrial and industrial-era PWC variability

Our ΔSLP reconstruction demonstrates that large interannual to decadal variability has been a feature of the PWC throughout the past millennium (Fig. 1a). A weak positive ΔSLP trend from around 1200–1750 is followed by a slight decrease to around 1800, then a period of low inter-method agreement. Low inter-method agreement is also found in ENSO reconstructions over the same period[29]. In both cases, this disagreement may result from non-stationary climate covariation due to the presence of several volcanic eruptions over this period[29]. This in turn may drive inter-method differences owing to the different ways the reconstruction methods treat bias. The twentieth century is characterized by fluctuations around a stable mean, ending in a positive

trend over the past two decades (Fig. 1a). ΔSLP is weakly to moderately anticorrelated with reconstructions of ENSO over the past millennium (Extended Data Fig. 6c,d). When considering only significant peaks in the power spectrum, the PWC reconstruction has highest spectral power in the interannual (2–9-year) band (Fig. 2a), as expected from ENSO. Approximately 10% of ensemble members also have significant power in decadal (10–12-year) and multidecadal (21–24-year) bands, possibly indicating influence of the 11-year solar cycle[30]. The low spectral power at decadal to multidecadal timescales is reflected in a weak correlation with an ice-core-based reconstruction of the Inter-decadal Pacific Oscillation (IPO)[31] (Extended Data Fig. 6b). Notably, there is a shift to higher power at lower frequencies in the industrial era (1850–2000) relative to the preindustrial past millennium (1200–1849; 4–9-year rather than 2–9-year periods, with particularly high power in the 9-year band) (Fig. 2b,c; Methods). Both this shift and the low proportion of ensemble members with significant low-frequency variability are robust to our temporally nested reconstruction approach, although the proportion of ensemble members with power at each period is slightly different in a non-nested version of the reconstruction (Extended Data Fig. 7; Methods). The distribution of ΔSLP values in the industrial era is slightly skewed towards higher (more La Niña-like) values than in the preindustrial past millennium (Fig. 2d). However, the difference between preindustrial and industrial-era mean ΔSLP is not significant (*P* ≥ 0.05) in 81% of the 4,800 reconstruction ensemble members (Fig. 2d; Methods).

The lack of a significant PWC mean state change in response to anthropogenic forcing is an important result. Climate models suggest that the thermodynamic effect of greenhouse-gas-driven rising

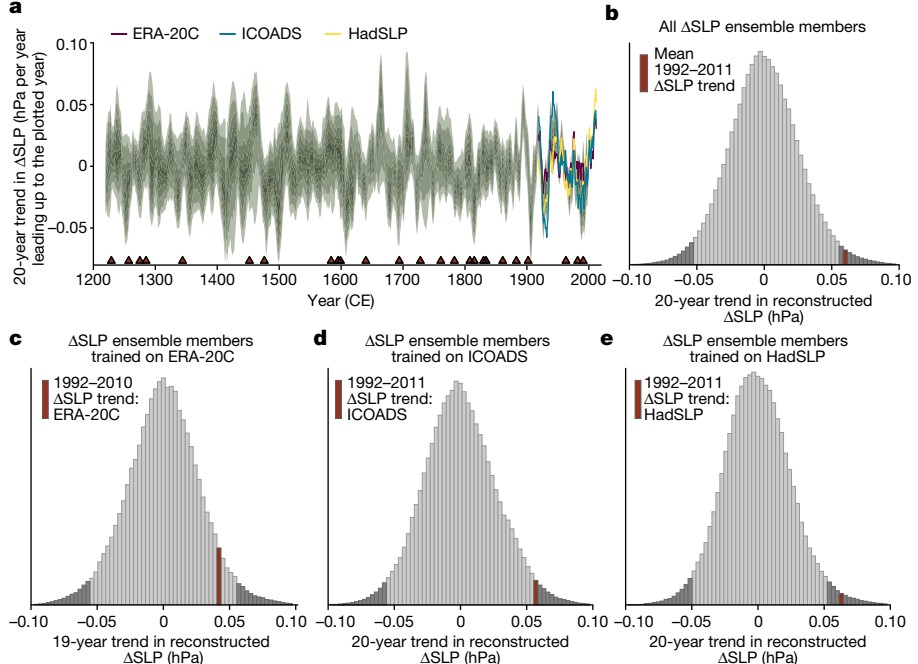

**Fig. 3 | Twenty-year trends in observed and reconstructed ΔSLP. a**, Green shading represents the 2.5th/97.5th quantiles of running 20-year trends throughout the 1200–2000 reconstruction interval, from the 4,800-member reconstruction ensemble (with each point showing the 20-year trend ending in that year). Coloured lines show running 20-year trends in ΔSLP for 1900–2011 for HadSLP[25] and ICOADS[26] and 1900–2010 for ERA-20C (ref. 27). **b**, Full distribution of the magnitude of 20-year trends in ΔSLP over 1900–2000 (from all individual reconstruction ensemble members). Dark grey tails show the 2.5th and 97.5th percentiles. Red bar shows the mean magnitude of the 1992–2011 ΔSLP trend for HadSLP and ICOADS. **c**, As per **b** but showing 19-year trends in reconstruction ensemble members trained on ΔSLP calculated from ERA-20C. Red bar shows the magnitude of the 1992–2010 ΔSLP trend in ERA-20C. **d,e**, As per **b** but only showing reconstruction ensemble members trained on ΔSLP calculated from ICOADS and HadSLP, respectively. Red bars show the magnitude of the 1992–2011 ΔSLP trend in ICOADS and HadSLP, respectively.

global mean surface temperature (GMST) should weaken the PWC by the end of the twenty-first century[11,32], and a negative ΔSLP trend is also present in historical simulations from most Coupled Model Intercomparison Project (CMIP5/6) models[3]. However, recent work suggests that global-warming-driven ocean–atmosphere dynamical changes accelerate the Pacific trade winds, resulting in a stronger PWC[14,33]. Our findings demonstrate that, during the industrial era, neither greenhouse-gas-driven effect is emergent from the large intrinsic variability of the PWC. Nevertheless, the industrial-era shift in PWC variability towards lower frequencies is intriguing and possibly a response to anthropogenic forcing that has not previously been identified.

## Recent strengthening not unprecedented

To determine whether the most recent PWC strengthening is anomalous relative to intrinsic variability, and hence potentially anthropogenically forced, we examined the 1992–2011 ΔSLP trend[13] from the gridded SLP products in the context of all possible 20-year trends throughout the 1200–2000 reconstruction period (Fig. 3a). Because ERA-20C data only extend to 2010, we compared the most recent 19-year trend in ERA-20C (1992–2010) to all 19-year trends in reconstruction ensemble members trained on ERA-20C data. Using trends calculated from ensemble members trained on HadSLP or ICOADS (with the 1992–2011 ΔSLP trend calculated using the same products), the 1992–2011 trend is unusually large (99th and 98th percentiles, respectively), although not unprecedented, in the context of the past millennium (Fig. 3d,e). Using ERA-20C, the 1992–2010 trend is less anomalous but still on the high end of the distribution (94th percentile; Fig. 3c). Comparing the 1992–2011 ΔSLP trend with the full reconstruction ensemble, the recent trend is again unusually large but not unprecedented (98th percentile; Fig. 3b).

Previous work using observational data and model simulations suggested that the recent multidecadal PWC strengthening may be attributable to either anthropogenic aerosol forcing or a slow recovery from a negative ΔSLP anomaly following the 1991 Mount Pinatubo eruption[5]. To resolve these possible drivers, we compared the 1992–2011 trend with the full distribution of 20-year trends following eruptions of Mount Pinatubo magnitude or greater (Methods). The 1992–2011 trend remains unusually large even in this context (Extended Data Fig. 8). Hence the 1992–2011 strengthening is probably not the result of volcanic forcing, making anthropogenic aerosols a more likely candidate if the trend is indeed a forced response.

## PWC response to volcanic forcing

Although the eruption of Mount Pinatubo did not likely force the 1992–2011 PWC strengthening, volcanic eruptions are the largest pre-industrial forcing of the past millennium and their impact on tropical Pacific climate is contentious[34]. We performed superposed epoch analysis (SEA) to test whether volcanic eruptions trigger a transient response in the PWC. SEA determines the median response to all volcanic eruptions over a defined interval (Methods). We identified volcanic eruption years using global mean stratospheric aerosol optical depth (SAOD), a dimensionless metric for the stratospheric scattering of solar radiation by volcanic aerosols, calculated in ref. 16 from the 'eVolv2k' reconstruction of Common Era volcanic sulfate aerosol loading[35]. Following recent work[36], we reassigned the major Kuwae eruption from 1458 to 1452. Proxy-based and model-based studies suggest that a tropical Pacific response to explosive volcanism only occurs when the eruption is of sufficient magnitude, so we restricted eruptions to those with SAOD ≥ 0.05 (that of the 1982 El Chichón eruption;

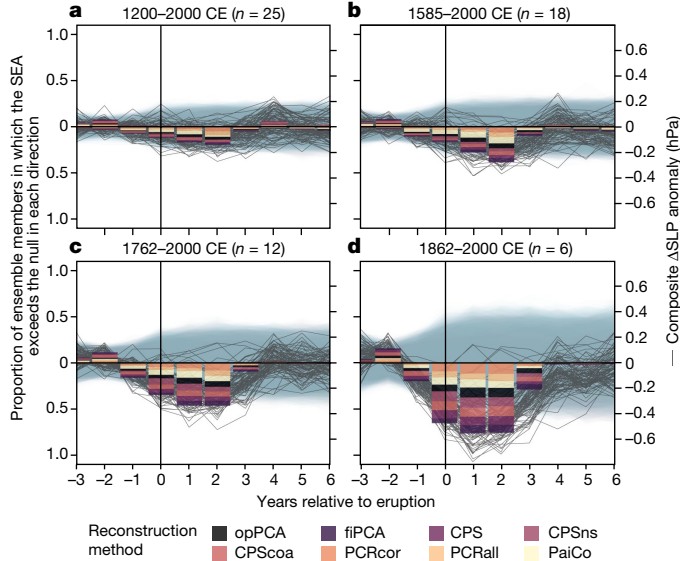

**Fig. 4 | SEA of ΔSLP reconstruction, with volcanic eruption years as defined in ref.** 35 **(with one exception; see text).** SEA averages the *n* volcanic eruptions events to provide a composite ΔSLP response to explosive volcanism. Bars show the proportion of the 4,800 ΔSLP reconstruction ensemble members that have a significant[37] positive (La Niña-like) or negative (El Niño-like) ΔSLP anomaly in the −3 to +6 years relative to each eruption composite (see Methods). Thin grey lines show composite ΔSLP anomalies for 100 randomly chosen ensemble members. Grey–blue envelopes show associated confidence intervals, calculated using random bootstrapping[37]. Vertical black line highlights the eruption year (year 0). **a**, All eruptions with SAOD ≥ 0.05 that intersect the 1200–2000 ΔSLP reconstruction (*n* = 25). **b**, As per **a** but only the 18 most recent volcanic eruptions with SAOD ≥ 0.05 (1585–2000). **c**, As per **a** but only the 12 most recent volcanic eruptions with SAOD ≥ 0.05 (1762–2000). **d**, As per **a** but only the six most recent volcanic eruptions with SAOD ≥ 0.05 (1862–2000). Significant (*P* < 0.05) responses are determined using a double-bootstrap approach[37]. Colour blocks on each bar show the proportion of significant responses from ensemble members calculated using each reconstruction method.

*n* = 25). We performed SEA on all 4,800 ΔSLP reconstruction ensemble members and report the proportion of ensemble members that have a significant[37] positive or negative ΔSLP response in the years following volcanic eruptions.

Figure 4 reveals a significant El Niño-like PWC weakening in the 0–2 years following large volcanic eruptions, with a rapid recovery to the pre-eruption state. This result is insensitive to the reconstruction method (colours in Fig. 4) and the observational product used to calculate the ΔSLP target index (colours in Extended Data Fig. 9). However, the PWC weakening in response to large eruptions is progressively obscured by including older eruptions—particularly those before the mid-nineteenth century (Fig. 4 and Extended Data Fig. 9). Chronological uncertainty is the probable source of this obfuscation, as it increases back through time, smoothing the ensemble-mean response to older eruptions (Extended Data Fig. 10). Further time-dependent uncertainty may arise from temporal non-stationarities between the PWC and some proxy records[38]. As also found in previous studies assessing SST in the Niño 3.4 region[16,34], the magnitude of the post-eruption ΔSLP response does not scale with eruption magnitude (Extended Data Fig. 11). Negative ΔSLP anomalies 1 year before and 3 years after eruptions (Fig. 4d) are probably due to the chronological uncertainty incorporated into the reconstruction ensemble. Positive ΔSLP anomalies 2 years before eruptions (Fig. 4c,d) are probably due to the narrower confidence intervals at this point (a feature of how these confidence intervals are calculated, with all composites centred on the pre-eruption mean[37]; Methods).

Importantly, El Niño events had initiated shortly before three of the twentieth-century eruptions (Mount Agung, 1963; El Chichón, 1982; Mount Pinatubo, 1991)[34,39]. This probably influences the results in Fig. 4, given that volcanic forcing causes an atmospheric response on a similar timescale as ENSO[39]. Nevertheless, in a SEA with these three eruptions excluded, the response is similar albeit muted (not shown). Therefore, the significant post-eruption PWC weakening seen in Fig. 4 is not driven entirely by the twentieth-century eruptions, for which the tropical Pacific may have already been in an El Niño state.

In climate model simulations, volcanic eruptions generally trigger an El Niño-like tropical Pacific SST response (see summary in ref. 15). To assess our findings in the context of this previous work, we used a suite of climate models to test whether an El Niño-like SST response to volcanic eruptions is associated with a significant negative ΔSLP anomaly, as observed in our reconstruction. We performed SEA on (1) ΔSLP and (2) SST anomalies in the Niño 3.4 region, using the most comprehensive single-model ensemble of simulations covering the reconstruction period: the Community Earth System Model Last Millennium Ensemble (CESM1 LME)[40], which produces an El Niño-like SST response to volcanic forcing in the ensemble mean[41]. We also analysed data from eight Paleoclimate Modelling Intercomparison Project (PMIP3/4) models with a past1000 experiment, including an extra single-model ensemble of simulations from GISS-E2-R (refs. 42,43). When applying the above SEA approach to the CESM1 LME (using the 25 strongest eruptions; Methods), nine of the 13 CESM1 LME members produce a significant[37] negative ΔSLP anomaly the year following a volcanic eruption (Fig. 5a), with ΔSLP anomaly magnitudes similar to those occurring during an average El Niño event. As previously identified for SST in the Niño 3.4 region[16], the number of CESM1 LME members producing a significant response increases as the eruption size threshold increases (Fig. 5b,c). Notably, the ΔSLP response in the CESM1 LME is more consistent than the SST response (Fig. 5d–f). Fewer CESM1 LME ensemble members have a significant Niño 3.4 SST response in the year following eruptions than have a ΔSLP response and there is greater spread in the SST response across ensemble members.

Among the PMIP3/4 models, high inter-model variability is evident in both the ΔSLP and SST responses to volcanism. Nevertheless, the ΔSLP response is again more consistent than the SST response, with seven PMIP models (including three GISS-E2-R ensemble members) having a significant ΔSLP response in the year following an eruption, versus five models (including one GISS-E2-R ensemble member) with a significant Niño 3.4 SST response (Fig. 5a,d). Recent palaeoclimate reconstructions covering this time period have not found a significant SST response to large volcanic eruptions[16,34] and this analysis suggests that ΔSLP may be more sensitive to volcanic aerosol forcing.

## Discussion

Our results demonstrate that the PWC has large intrinsic variability across timescales, highlighting the importance of a longer-term context when discussing trends in atmospheric circulation. Nevertheless, the two largest external forcings of the past millennium produce detectable PWC changes. Analysis of the ΔSLP reconstruction ensemble reveals a significant El Niño-like PWC weakening after volcanic eruptions. This response is reproduced in the CESM1 LME and PMIP3/4 models and is more consistent than the associated Niño 3.4 SST response. Although there is no significant PWC trend since the onset of anthropogenic forcing (around 1850), an anomalous PWC strengthening trend over the past couple of decades, as well as an industrial-era shift towards lower-frequency variability, suggests that the PWC may be responding to anthropogenic forcing, albeit in ways that are not consistently reproduced by climate model simulations.

Previous studies using observations and climate models identified a greenhouse-gas-driven PWC weakening through the twentieth and

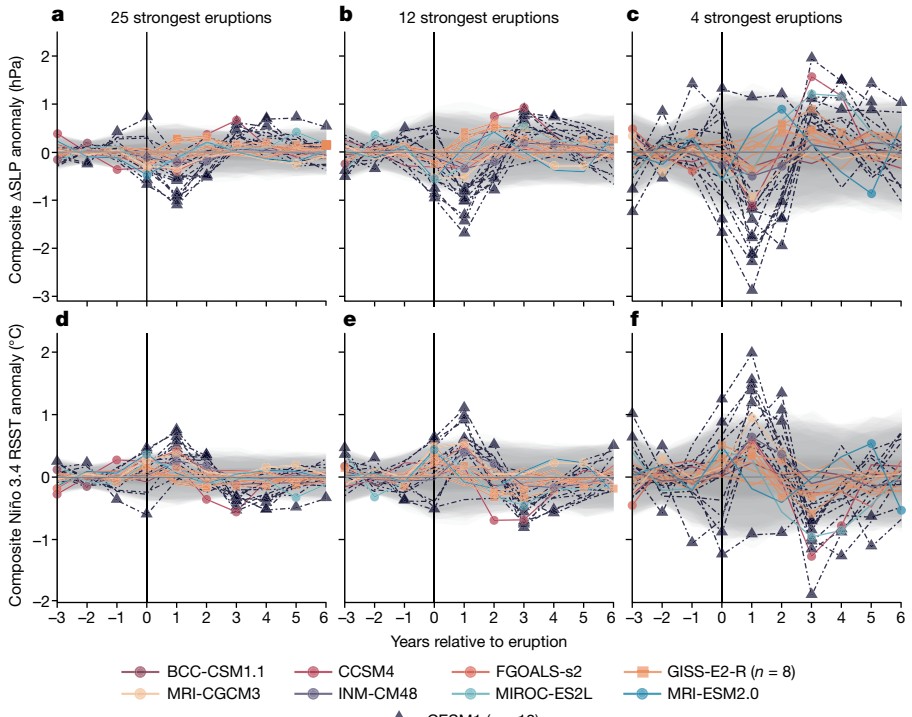

**Fig. 5 | SEA performed on PMIP3/4 simulations and all fully forced ensemble members from the CESM1 LME.** Volcanic eruption strength as per ref. 53 for CESM1 LME (ref. 40) and PMIP3 models[42], and as per ref. 35 for PMIP4 models[43]. Each line shows the composite response for one simulation in the −3 to +6 years relative to the included eruptions. Each line is associated with a grey band showing the threshold required for epochal anomalies to be deemed statistically significant ($P < 0.05$)[37]. A significant response (that is, when a line exceeds its confidence intervals) is highlighted by a point on the relevant line. **a**, SEA performed on ΔSLP calculated from the PMIP3/4 and CESM1 LME 'PSL' fields including the 25 strongest eruptions over the 1200–2000 interval. **b**, As per **a** but including only the 12 strongest eruptions. **c**, As per **a** but including only the four strongest eruptions. **d–f**, As per **a–c** but with SEA performed on relative SST (RSST) anomalies in the Niño 3.4 region (Methods).

twenty-first centuries[11,32], following a thermodynamically driven decline in vertical mass flux over the tropical Pacific. If this effect is emergent relative to internal variability, then we might expect GMST and ΔSLP to be anticorrelated in the industrial era, that is, the interval with the largest increase in GMST. However, our ΔSLP reconstruction reveals no industrial-era PWC weakening relative to the preceding 650 years (Fig. 2d and Extended Data Fig. 12). In fact, comparison with reconstructed GMST[44] reveals that PWC strength is not reliably anticorrelated with GMST across timescales, including correlation tests restricted to

the industrial era (Fig. 6a). A distribution of correlation coefficients between the two ensemble reconstructions over the full 1200–2000 interval shows only a weak anticorrelation (Fig. 6b). Our results therefore imply that, if there is a thermodynamic influence of GMST on the strength of the PWC: (1) it is obscured by competing forcings (for example, anthropogenic aerosol emissions[9,45]); (2) other thermodynamic and/or dynamic responses of the PWC to warming are operative as well[33,46]; or (3) the changes are too small to have emerged from intrinsic variability with the anthropogenic $CO_2$ increase experienced so far.

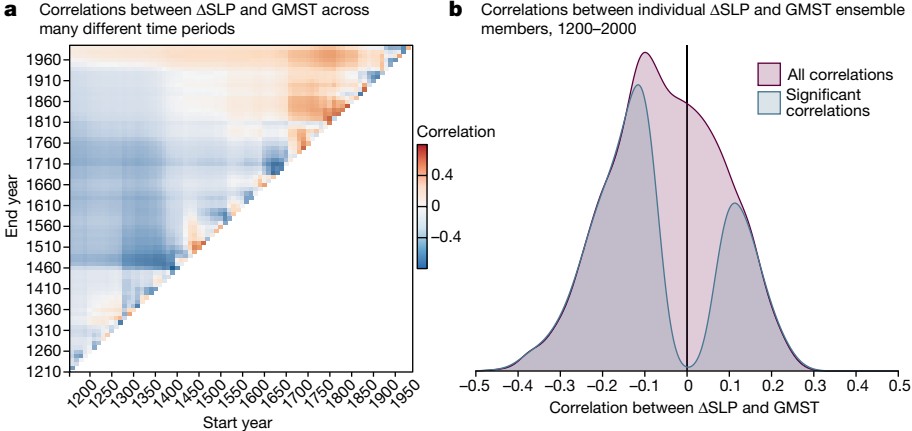

**Fig. 6 | Correlations between the ΔSLP reconstruction presented in this study and GMST[44].** Both are ensemble reconstructions ($n = 4,800$ for ΔSLP, $n = 7,000$ for GMST). **a**, Correlation of ΔSLP and GMST reconstruction medians across different time periods. The $x$ axis shows the start year of the interval across which the correlation was computed and the $y$ axis shows the end year.

Colours correspond to the strength and sign of the correlation. **b**, Distribution of correlation coefficients between 4,000 unique combinations of individual ΔSLP and GMST ensemble members, across the full ΔSLP reconstruction interval (1200–2000). Purple distribution shows all correlation coefficients and blue distribution shows only significant ($P < 0.05$) correlations.

Our findings do not discount the possibility that, with future changes in the relative magnitude of anthropogenic forcings (for example, a larger increase in atmospheric $CO_2$), a thermodynamically driven PWC weakening may yet emerge.

Of particular relevance to point (1) above is the unusually large 1992–2011 PWC strengthening, which is unlikely to solely represent a slow recovery from El Niño-like conditions following the Mount Pinatubo eruption. Model simulations suggest that anthropogenic aerosol emissions concentrated in the Northern Hemisphere drive a La Niña-like SST response[45,47,48]. Given that the anthropogenic aerosol forcing over the past few decades has been concentrated in the Northern Hemisphere, this could be expected to drive a multidecadal trend towards a stronger PWC. However, although the 1992–2011 PWC strengthening is unusual, it is not unprecedented in the past 800 years, so it may also be due to unforced decadal variability.

Although evidence for a PWC response to anthropogenic forcing is subtle, the response to volcanic forcing is comparatively clear. A significant El Niño-like ΔSLP anomaly occurs in the year of volcanic eruptions, probably associated with El Niño-like easterly surface wind anomalies over the equatorial Pacific[15]. The significant anomaly lasts until 2–3 years after the eruption (Fig. 4d). An El Niño-like ΔSLP response is also evident in climate model simulations, although the significant anomaly is strongest in the 1 and 2 years following eruption years, with large inter-model variation. Similar analyses performed on palaeo-ENSO records mostly suggest either no significant SST response[16,49] or a weak El Niño-like SST response[34], with some exceptions showing a strong El Niño-like response—generally from tree-ring-based ENSO reconstructions[50,51]. We offer three possible explanations. First, the tropical Pacific response to explosive volcanism seems to be stronger in the atmosphere than in SST (Fig. 5a–c versus Fig. 5d–f) and hence that intrinsic variability may mask the forced SST signal in some cases. Most studies investigating the tropical Pacific response to explosive volcanism use Pacific SST proxy records, whereas our reconstruction is based on globally distributed records that are directly affected by changes in atmospheric circulation[22]. Second, post-eruption El Niño-like temperature responses at individual proxy record locations may be masked by global cooling associated with volcanic eruptions[52]. Third, the signal is sensitive to loss of high-resolution signal back through time from the combined influences of increased reconstruction uncertainty and chronological uncertainty (Extended Data Fig. 10), which are not always accounted for.

Finally, our use of water-isotope proxy data to reconstruct atmospheric variability, including explicit incorporation of uncertainty from the training dataset, reconstruction method, age–depth models, and change in availability of proxy records back through time, allowed us to quantify the PWC response to the two largest external forcings of the past millennium—anthropogenic forcing and volcanic eruptions—and the magnitude and sources of uncertainty in these responses. To our knowledge, this is the first climate mode reconstruction that directly addresses each of these uncertainty sources, providing a robust tool for further analyses. Although diagnosis of the dynamics underlying forced responses and intrinsic variability in the PWC was beyond the scope of this paper, the ΔSLP reconstructions provide the necessary empirical foundation for such future investigations. Detailed data-model comparisons may also lead to increased understanding of model biases in forced and intrinsic tropical Pacific variability on interannual to multidecadal scales.

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

# Methods

To reconstruct PWC variability through 1200–2000, we took a multi-method, multi-proxy approach. Modern global precipitation $\delta^{18}O$ is highly correlated with the PWC, a result of various well-established mechanisms and teleconnections[22]. We leveraged that relationship using a globally distributed network of water-isotope proxy records from five different proxy archive types: glacier ice, wood, lake sediment, coral and speleothem. Using different archive types reduces the risk of archive-specific biases, for example, bias to 'warm' or 'wet' season values, while also allowing inclusion of the highest possible number of records. This is important, as networks of sites are more robust to non-stationary teleconnections than single sites[38,54]. We used eight statistical methods ('Reconstruction methods' section) to isolate the PWC signal, thereby accounting for method-specific biases. We also used several target datasets to account for the impact of observational uncertainties ('Observational data sources' section) and include a robust treatment of chronological uncertainty ('Incorporation of chronological uncertainty' section).

## Data

**Reconstruction target.** The reconstruction target was the trans-Pacific equatorial ΔSLP, defined as anomalies in the difference between the area-mean SLP over the central-eastern Pacific Ocean (160°–80° W, 5° S–5° N) and the western Pacific/eastern Indian oceans (80°–160° E, 5° S–5° N), relative to 1960–1990. ΔSLP is closely related to the strength of the PWC[8,9,11,22,47,55] and is highly correlated with more sophisticated circulation-based indices for the strength of the PWC, which are only available for 1950 to the present[56].

**Observational data sources.** We calculated ΔSLP using two gridded observational products (HadSLP and ICOADS) and one atmospheric reanalysis product (ERA-20C). HadSLP is available at 5° resolution spanning 1850 to the present and is derived from quality-controlled marine and terrestrial SLP observations[25]. For 1900–2004, we used the 'HadSLP2' product; from 2005 onwards, we used the 'HadSLP2r' product. SLP data from ICOADS are available at 2° resolution spanning 1800 to the present and is derived from surface marine observational data[26]. ERA-20C assimilates surface pressure and marine wind anomalies into an atmospheric general circulation model[27] and is available at approximately 1° resolution, spanning 1900 to 2010.

**Proxy data.** Most proxy data are from the Iso2k database ($n = 50$), a multi-archive compilation of proxy records for the stable isotopic composition of water[23]. Following a broad literature search, we sourced nine further records in which the authors describe a strong relationship between the proxy record and either the PWC or ENSO. Speleothem $\delta^{18}O$ data in this category were sourced from the SISAL database[57]. For Iso2k records, we only retained the designated 'primary' time series for each record[23] and then considered only annually or sub-annually resolved proxy records with data extending to at least 2000. Primary references for all datasets are described in Supplementary Table 1 (refs. 58–106).

**Model simulations.** We used data from the CESM1 LME (ref. 40) as well as five PMIP3 (ref. 42) and three PMIP4 (ref. 43) contributions, for comparison with our PWC reconstruction. We used past millennium simulations of the following CMIP5/PMIP3 models: BCC-CSM1.1 (ref. 107), CCSM4 (ref. 108), FGOALS-s2 (ref. 109), GISS-E2-R (ref. 110) and MRI-CGCM3 (ref. 111). We used past millennium simulations of the following CMIP6/PMIP4 models: INM-CM48 (ref. 112), MIROC-ES2L (ref. 113) and MRI-ESM2.0 (ref. 114).

We only used CESM1 LME members with all anthropogenic and natural external forcing factors applied, that is, fully forced ensemble members ($n = 13$). The PMIP3 data include an extra single-model ensemble (GISS-E2-R, $n = 8$).

## Reconstruction methods

We chose the 1200–2000 interval for reconstruction, as this struck the best balance between proxy data availability and sampling of long-term forced and internal variability.

**Calibration window.** We used two calibration windows. For the reconstructions presented in the main text, we used 1900–2000, to minimize the influence of non-stationary teleconnections[18]. 1900 is the earliest year covered by ERA-20C and an end year of 2000 provided the best balance of maximizing the calibration window length and the number of included proxy records.

We recalculated the full ΔSLP reconstruction ensemble using a shorter calibration window (1951–2000), providing a minimum estimate of reconstruction skill through independent validation tests performed over 1900–1950 ('Assessing reconstruction skill' section).

**Data preparation.** We reconstructed annual ΔSLP, allowing characterization of both long-term interannual PWC variability and lower-frequency variability. Most reconstruction methods require data on a common time step, so sub-annually resolved records were annually binned to calendar years (January to December). After binning, we retained records with data in two-thirds of the bins within the calibration window (1900–2000). We estimated any missing years in the calibration window using the Data Interpolating Empirical Orthogonal Functions (DINEOF) method, which interpolates missing values in such a way that underlying commonalities are maintained[115].

Three of our reconstruction methods require that contributing records are correlated with the target index. In this case, we retained only records significantly ($P < 0.1$) correlated with ΔSLP over the calibration window.

**Reconstructing ΔSLP from palaeoclimate proxy data. Reconstruction steps common to all methods.** Because the number of available records that extend to 2000 decreases with increasing record length (Extended Data Figs. 1 and 2), we performed all reconstructions over five temporal subsets: 1860–2000, 1800–2000, 1600–2000, 1400–2000, and 1200–2000, following the 'nested' approach of previous studies[24,116,117].

For each subset, we only included records with data in greater than or equal to two-thirds of the years spanning the entire interval. All methods except one require continuous data, so we interpolated missing data using the DINEOF method (Extended Data Fig. 1). To avoid spurious jumps when appending segments, we aligned each older segment (for example, 1400–2000) with the adjacent newer segment (for example, 1600–2000) by matching the mean of the first 20 years of the newer segment with the mean of the corresponding interval in the adjacent older segment (for example, 1600–1620). This nested approach allowed us to incorporate proxy records that do not span the full reconstruction interval.

All reconstructions except pairwise comparison (MATLAB) were performed in R (ref. 118).

**Incorporation of chronological uncertainty.** For each proxy record included in the reconstructions, we used the 'simulateBam' function from the geoChronR package[119] to calculate a 100-member banded age–depth model[28], assuming a 1% counting error. We explicitly incorporated this chronological uncertainty and its influence on the variance structure of the reconstruction by calculating the 800-year ΔSLP reconstruction 200 times, at each iteration randomly sampling one realization from the age–depth model ensemble for each record. This was done separately for each combination of reconstruction method and gridded product used to calculate the observational ΔSLP target index (hereafter 'target index'; eight reconstruction methods, three target indices, 200 age–depth model iterations = 4,800 ensemble members). This incorporates the probability distribution of the age–depth model ensembles, providing a robust treatment of age uncertainty.

**Uncertainty arising from outsized influence of particular records.** To incorporate uncertainty arising from the possibility that some records have an outsized influence on the reconstruction, before each iteration, we randomly removed up to 15% of all possible contributing records.

**Reconstruction methods.** To quantify uncertainty arising from the ΔSLP reconstruction method, we used eight different methods. These have various requirements for the input data—some require proxy records correlated with the target index ('Data preparation' section), whereas others use all available records. Records significantly correlated with the target indices in the calibration interval are denoted with a black outline in Extended Data Fig. 2.

*PCA*: Reconstructions based on PCA assume that the underlying gradient common to a group of time series significantly correlated with ΔSLP in the calibration window should be equivalent to ΔSLP (refs. 117,120,121). For PCA-based reconstructions, we therefore only used records that are correlated with ΔSLP in the calibration window.

For opPCA reconstructions, we performed PCA on the calibration window (that is, in which we know that the proxy records are correlated with the PWC) and then multiplied the loading of each proxy record on PC1 by the complete time series of the proxy records. The contribution of each record to the PCA was weighted according to the strength of its correlation with ΔSLP.

The direction of a PC axis is arbitrary. To align the temporal subsets, we flipped (if necessary) PC1 of the 1860–2000 subset to make it positively correlated with the target index and then aligned PC1 of subsequent temporal subsets to be positively correlated with their predecessors.

The fiPCA reconstructions were performed identically to the opPCA, except that PC1 was calculated over the full length of the proxy time series for each temporal subset.

*CPS*: The CPS method has been used in many multi-proxy palaeoclimate reconstructions[24,122–124]. In our implementation, all proxy records were scaled to unit variance and zero mean and then weighted according to their correlation with ΔSLP in the calibration window. The scaled and weighted records were composited and the composite was scaled to match the mean and variance of the ΔSLP in the calibration window. CPS reconstructions were performed using all available proxy records.

For CPSns, we repeated all steps for the CPS method but first filtering to only include records preserving an annually integrated signal (33 records; Supplementary Table 1). For Iso2k records, this determination was made on the basis of the 'isotopeInterpretation1_seasonality' metadata field[23]. For all other records, this was inferred from the primary publications. We only excluded records with a known (reported) seasonal bias.

For CPScoa, we repeated all steps for the CPS method but first filtering to only include records in a tropical Pacific 'centre of action', that is, only records between 40° S and 40° N, and 50° E and 50° W (42 records). This removes records with a higher potential for non-stationary teleconnections.

*PCR*: PCR is a multivariate regression method that has been used for palaeoclimate reconstructions of the past millennium[44,125,126]. PCR targets ΔSLP by performing PCA, calculating a linear regression of ΔSLP on the PCs and then retaining the minimum number of PCs required to maximize the correlation with ΔSLP. The number of retained PCs was determined using root mean squared error (RMSE) of prediction, estimated from cross-validation[127]. We chose the model with the fewest PCs that was still less than one sigma from the overall best model[128]. We performed PCR reconstructions using all proxy records (PCRall) and also on a subset that only included records significantly ($P < 0.1$) correlated with ΔSLP in the calibration window (PCRcor). We performed PCR reconstructions using the 'pls' R package[128]. Models were fitted to data in the calibration window and then values predicted for the full length of each temporal subset.

*PaiCo*: The non-linear PaiCo method was developed for use with multi-proxy palaeoclimate datasets[129]. The underpinning assumption of PaiCo is that an increase in a proxy record indicates an increase in the target index (ΔSLP) and the strength of agreement among proxy records on the change between two time points relates to the magnitude of reconstructed change in the target[129]. PaiCo reconstructions were performed in MATLAB, using records significantly ($P < 0.1$) correlated with ΔSLP in the calibration window.

**Post-reconstruction steps common to all methods.** The mean and variance of all reconstructed temporal subsets was adjusted so that:
- The mean variance across the reconstruction matches the variance of ΔSLP in the calibration window, and
- The mean ΔSLP of each reconstruction ensemble member during 1900–2000 matches the mean observational ΔSLP in the calibration window.

For ease of comparison, we adjusted all reconstruction time series to match the mean and variance of ΔSLP calculated from HadSLP, although the results are not sensitive to this choice. When adjusting the variance of each reconstruction time series, we applied a single variance-scaling factor to the entire time series. That is, temporal variability in variance was maintained, potentially allowing for similar changes as seen in reconstructions of tropical Pacific SST[130,131].

**Influence of trends in the calibration window.** We repeated all reconstruction steps but with all correlations calculated on detrended datasets. This did not make any meaningful difference to the ΔSLP reconstruction ensemble, reconstruction skill or post-reconstruction analyses.

**Assessing reconstruction skill. Reconstruction validation.** We calculated the following skill metrics for the reconstruction ensemble presented in the main text:
- Correlation coefficient ($r$),
- RMSE and
- Reduction of error (RE)[132].

We performed skill tests on all 4,800 ensemble members, which are reported by reconstruction method and ΔSLP index (Extended Data Fig. 4a,b).

For ease of comparison with existing reconstructions of tropical Pacific variability, we calculated all skill metrics for the reconstruction median (Extended Data Table 1), as well as $r$ for the median reconstructions for each reconstruction method and target index (Extended Data Fig. 3b). To estimate changes in reconstruction skill back through time, we calculated the same validation statistics for each temporal subset (Extended Data Fig. 5c).

To provide a minimum independent estimate of reconstruction skill, we calculated the same validation statistics across the 1900–1950 interval, using an otherwise exactly equivalent reconstruction ensemble calculated using a shorter calibration window (1951–2000) (Extended Data Fig. 5a,b and Extended Data Table 1). We also calculated the coefficient of efficiency for the reconstruction medians (Extended Data Table 1).

**Internal consistency.** To assess internal consistency among reconstruction ensemble members, we considered all possible combinations of reconstruction method and ΔSLP training data and calculated the 30-year running correlation among each pair of ΔSLP time series (Extended Data Fig. 4c). When agreement is high among reconstruction ensemble members, this probably reflects a strong ΔSLP signal in the proxy datasets regardless of reconstruction method and target index choice.

**Estimating contribution of each palaeoclimate record to the reconstruction.** To estimate the overall contribution of individual palaeoclimate records, we calculated the correlation of each component record (on its published chronology) with the ΔSLP reconstruction ensemble median across the interval to which that record contributed (Fig. 1b). Correlations were deemed significant if $P < 0.05$, and were calculated from the start of the earliest temporal segment to which each record contributed.

**Assessing temporal variability in the reconstructions.** We calculated the full distribution of values in the 4,800-member ΔSLP reconstruction ensemble as well as for the preindustrial (1200–1849) and industrial-era (1850–2000) sections of the reconstruction (Fig. 2d). We performed two-sample Kolmogorov–Smirnov tests on the preindustrial versus industrial-era segments of all 4,800 individual ensemble members. We adjusted the $P$ values to account for false discovery rate[133]. For 81% of ensemble members, the difference between the two time periods was not significant ($P \geq 0.05$; Fig. 2d).

**Spectral character.** We calculated the temporal power spectrum for each ensemble member and determined frequencies at which each ensemble member has significant ($P < 0.05$) power. Our spectral analysis was based on the geoChronR (ref. 119) implementation of multitaper spectral analysis, by means of the 'mtmPL' function from the R package 'astrochron' (ref. 134). Significance of spectral peaks was established through a power-law null[135]. We report the proportion of ensemble members with a significant peak at each period below 75 years. Beyond 75 years, a maximum of 3% of ensemble members have significant power at lower frequencies (maximum $n = 122$ ensemble members, at period length 148 years). For comparison, we performed the same analysis on instrumental ΔSLP (Extended Data Fig. 7b).

To determine whether the industrial-era power spectrum is different from that of the preindustrial, we assessed the distribution of spectral power in only the most recent 150 years of the reconstruction (1850–2000) (Fig. 2c). To ensure a fair comparison with spectral densities in the preindustrial, we compared this with the distribution of spectral power in all possible 150-year periods before 1850 (Fig. 2b), still showing the proportion of ensemble members with power in each period.

To assess whether the power spectra are influenced by the 'nesting' reconstruction approach, we repeated the above analysis across the 1600–2000 interval, using a reconstruction ensemble derived only from proxy records with full coverage across that interval (otherwise identically constructed). In this way, we test (1) whether our nesting approach dampens low-frequency (decadal to multidecadal) variability and (2) whether differences between the power spectra of the preindustrial and industrial era are because of changing contributions from different proxy records (Extended Data Fig. 7).

**Calculating distribution of 20-year trends.** To assess whether the 1992–2011 PWC strengthening[13] is anomalous, we calculated the distribution of 20-year trends in the 4,800-member ΔSLP reconstruction ensemble, for comparison with the observed trend from 1992–2011. We provide the full distribution, as well as individual distributions for reconstructions trained on each gridded SLP product. The observed 1992–2011 trend is shown as a red bar on each distribution in Fig. 3b–e. ERA-20C data only go to 2010, so for the ERA-20C-only distribution, we show the distribution of 19-year trends.

To isolate potential influence of the 1991 Mount Pinatubo eruption on the 1992–2011 strengthening, we also calculated the distribution of 20-year trends that start in the year following volcanic eruptions equal to or greater in magnitude than the Mount Pinatubo eruption. We similarly compared the recent observed trends with these post-eruption distributions (Extended Data Fig. 8). We identified volcanic eruption years using global mean SAOD, a dimensionless metric for the scattering of solar radiation by aerosol particles, calculated in ref. 16 from the 'eVolv2k' ice-core reconstruction of volcanic sulfate aerosol loading[35]. Eruption years are defined as the maximum of each SAOD peak. Following findings from several recent studies[36,136], we reassigned the year of the major Kuwae eruption to 1452 (as opposed to 1458 as per eVolv2k). The 1991 eruption of Mount Pinatubo had an estimated maximum SAOD of around 0.1.

**Preindustrial versus industrial-era trends.** For each ensemble member, we calculated the linear trend (regression coefficient) across two time intervals: 1200–1849 and 1850–2000. We show the distribution of trends in Extended Data Fig. 12; panel a shows the full distributions and panels b–d split the results according to the ΔSLP target index. We did not differentiate between significant and non-significant trends.

**Assessing the PWC response to volcanic eruptions.** To assess the PWC response to volcanic forcing, we composited the ΔSLP response to all large volcanic eruptions intersecting the reconstruction interval. This technique, known as SEA, treats volcanic eruptions as replicate cases of the same process. This allows assessment of whether the PWC responds in a consistent manner to volcanic forcing. SEA is commonly used to assess the ENSO response to volcanic eruptions[16,34].

For each time series (that is, each ΔSLP reconstruction ensemble member), we isolated 10-year segments spanning each eruption—3 years before and 6 years following each eruption. This resulted in $n$ ten-year segments, in which $n$ is the number of volcanic eruptions included in the SEA. We centred each 10-year segment according to its 3-year pre-eruption mean and then took the mean of all $n$ segments. This provided a single 10-year composite time series, in which any consistent response in a particular year relative to the eruptions is concentrated and intrinsic variability should cancel out to an anomaly around zero. This replicates the SEA parameters of ref. 16, although our results are insensitive to the addition of several years either side.

We identified eruption years using SAOD as described in the 'Calculating distribution of 20-year trends' section. We restricted eruptions to those with SAOD $\geq 0.05$. We performed SEA on all 4,800 ΔSLP reconstruction ensemble members and determined the significance of the results using the 'double-bootstrap' method of ref. 37. Specifically, we used the 'random-bootstrapping' approach, with confidence intervals generated from 1,000 pseudo-composite matrices. These pseudo-composites are also centred on the pre-eruption mean, resulting in relatively narrow confidence intervals before the eruption year. In Fig. 4 and Extended Data Figs. 9 and 11, we report the proportion of ensemble members with a significant ($P < 0.05$) positive or negative ΔSLP response to volcanic eruptions in each year of the analysis.

Twenty-five volcanic eruptions between 1203 and 1993 exceeded our 0.05 SAOD cutoff. We performed SEA using all 25 eruptions, as well as two sequences:

1. Sequentially removing the weakest eruption until only the six strongest eruptions remained (Extended Data Fig. 11).
2. Sequentially removing the oldest eruption until only the six most recent of the 25 eruptions remained (Fig. 4 and Extended Data Fig. 9).
   We also repeated sequence 2 but first removing the three most recent eruptions.

**Assessing the PWC response to volcanic eruptions in model simulations.** To directly compare the reconstructed and model-simulated tropical Pacific response to volcanic forcing, we replicated the analysis described in the previous section ('Assessing the PWC response to volcanic eruptions'), using data from all fully forced CESM1 LME members and eight PMIP3/4 models. ΔSLP calculated from climate models was scaled to match the variance of ΔSLP calculated from HadSLP. We performed SEA on three subsets of eruptions. In the first subset, we retained the same number of eruptions as input to the SEA performed on the ΔSLP reconstruction, that is, the 25 strongest eruptions during 1200–2000. We assessed two further subsets, the 12 strongest eruptions and the four strongest eruptions, allowing for comparison with the similar analysis performed in ref. 16, that is, Fig. 4B,C in that reference (although note that this reconstruction covered a different time interval). Eruption magnitudes were determined using the volcanic forcing reconstruction used to drive the model. For the CESM1 LME and PMIP3 models, this is ref. 53. For PMIP4 models, this is ref. 35.

We performed SEA on ΔSLP calculated from the PSL field of the atmospheric models, as well as relative SST (RSST) in the Niño 3.4 region (5° S–5° N, 170° W–120° W). RSST is the residual signal after removing mean tropical (20° N–20° S) SST anomalies from raw SST anomalies. We used RSST rather than raw SST anomalies because of the expectation that volcanic aerosols will cause cooling globally and

mask the tropical Pacific response[52,137]. This allowed us to compare our findings with previous work investigating the effect of explosive volcanism on ENSO (in terms of SST anomalies), as well as comparing the oceanic and atmospheric responses over the tropical Pacific. We acknowledge that SEA is a suboptimal method for assessing the climatic response to explosive volcanism in climate models, which have full spatial and temporal data coverage and hence allow more nuanced analyses. However, performing the same analysis on model-derived and proxy-derived ΔSLP allows us to directly compare results.

**Comparing palaeo-PWC with palaeo-GMST.** We evaluated the relationship of the PWC with GMST by comparing our ΔSLP reconstruction ensemble with the PAGES 2k multi-proxy, multi-method ensemble ($n = 7,000$) reconstruction of GMST throughout the Common Era[122]. To assess temporal variability in the relationship between ΔSLP and GMST, we calculated correlations between the ΔSLP and GMST ensemble medians in many different time periods, starting between 1200 and 1990, spanning 10 to 800 years in duration (Fig. 6a).

We assessed uncertainty in the long-term relationship between ΔSLP and GMST by computing correlations between 4,000 unique combinations of individual members from both ensembles, over the full 1200–2000 interval (Fig. 6b).

**Comparing palaeo-PWC with palaeo-ENSO and palaeo-IPO.** We compared our ΔSLP reconstruction with published annually resolved reconstructions of tropical Pacific variability extending back to at least 1600 (Extended Data Fig. 6). Reconstructed climate modes include ENSO[117,125,138–142] and the IPO[31]. ENSO reconstructions have different targets, for example, Niño 3, Niño 3.4 or ENSO indices incorporating several regions. If a study provided reconstructions of SST in several regions, we used the Niño 3.4 reconstruction. For the Last Millennium Reanalysis[139], we used the Niño 3.4 reconstruction median. We clipped reconstructions to their common time period 1600–1978. Note that reconstructions have different reconstruction target seasons. We calculated 30-year running correlations between each ENSO reconstruction and the ΔSLP reconstruction median (Extended Data Fig. 6c), as well as correlations between all reconstructions across the 1600–1978 interval (Extended Data Fig. 6d). To compare ΔSLP with the IPO, we applied a 13-year Gaussian kernel low-pass filter to all ΔSLP ensemble members (following ref. 31) and then calculated the correlation of each smoothed ensemble member with the IPO reconstruction (1) over 1200–2000 and (2) only 1900–2000. For comparison, we correlated mean smoothed observational ΔSLP (from ERA-20C, ICOADS and HadSLP) with observed IPO variability over the 1900–2000 period (Extended Data Fig. 6b). In Extended Data Fig. 6b, we only show significant ($P < 0.05$) correlations.

## Assessment of reconstruction skill
**Reconstruction skill scores.** The ensemble approach to this reconstruction allows estimation of reconstruction skill at several levels of detail. The simplest possible tests compare ensemble median reconstructed ΔSLP with mean ΔSLP from the three observational products (Extended Data Table 1). In this test, the reconstruction is highly correlated with observations. There is only a small difference in skill scores for tests on reconstructions using the entire calibration window ($r = 0.81$, $P < 0.05$) versus independent calibration-validation tests ($r = 0.77$, $P < 0.05$), whereby validation is performed on a 1900–1950 window, using reconstructions trained only on observational data from 1951–2000. The RMSE is low in both cases (0.27 for the full calibration window and 0.26 on the independent validation window). RE can range from negative infinity to one; reconstructions are generally considered skilful if RE > 0. RE is positive in all cases.

Comparing sub-ensemble medians for unique combinations of target index ($n = 3$) and reconstruction method ($n = 8$) reveals differences in correlations with the relevant target index and varying agreement between sub-ensemble medians (Extended Data Fig. 3b). For

all three target indices, the PCRall sub-ensemble median is the most highly correlated with observations. The fiPCA sub-ensemble median is consistently among the least correlated with observations. The PaiCo sub-ensemble median generally shows the lowest correlations with the other reconstruction medians.

There are minimal differences between skill-score distributions for ensemble members calculated using different training indices (Extended Data Fig. 4a) but larger differences among the reconstruction methods (Extended Data Fig. 4b). As seen in the sub-ensemble medians (Extended Data Fig. 3b), ensemble members calculated using PaiCo tend to perform worst, whereas ensemble members calculated using PCR-based methods tend to perform best. All other methods have similar medians and interquartile ranges, although PCA-based methods have the largest overall distributions (skewed to low scores).

When skill scores are calculated on an independent window (calibration 1951–2000, validation 1900–1950), the PCR-based methods still perform best (Extended Data Fig. 5b), but the two PCA-based methods perform worst, with particularly long low score tails (Extended Data Fig. 5b).

**Change in reconstruction skill through time.** By calculating skill scores for the individual temporal subsets contributing to the reconstruction ('Reconstruction steps common to all methods' section), we estimate the change in reconstruction skill through time (Extended Data Fig. 5c). Skill decreases with increasing age, which is not surprising given that proxy data availability drops off rapidly from around 1600 (Extended Data Figs. 1 and 2).

**Influence of proxy location and seasonality on skill.** By comparing skill scores for the CPS reconstruction method variants, we estimate the influence of (1) proxy archives that are located far from the tropical Pacific (hence relying heavily on teleconnections) and (2) proxies with known bias towards a particular season. When calculated across the full 1900–2000 interval, skill scores for sub-ensemble medians (that is, medians for reconstruction ensemble members calculated using CPS, CPScoa and CPSns) are very similar (Extended Data Table 1, first column). There are larger differences between skill scores when calculated on the independent 1900–1950 validation window (Extended Data Table 1, fourth column). Independent CPScoa reconstructions have higher $r$ and RE and lower RMSE than either of the other two variants. This suggests that incorporation of records far from the tropical Pacific may negatively influence reconstruction skill. However, exclusion of records that have a known seasonal bias does not improve reconstruction skill.

Notably, the CPSns reconstructions are typically the least similar to reconstructions from the other methods, often with a greater amplitude of variability, and sometimes showing change of opposite sign to reconstructions from other methods (Fig. 1a). This could be owing to: (1) substantial influence of record seasonality on the reconstructions; (2) loss of many records from a particular archive (tree cellulose); or (3) the reduced number of records contributing to the reconstruction.

**Estimating ΔSLP signal strength.** Extended Data Fig. 4c demonstrates the degree of agreement between ΔSLP reconstruction ensemble members changes through time, with a step change in intra-ensemble agreement at approximately 1600, coinciding with decreased proxy data availability (Extended Data Fig. 1). We can use this agreement to estimate the degree to which ΔSLP is recoverable from this combination of proxy data. Reasonably strong agreement between 1600 and 2000 suggests that the ΔSLP signal strongly underpins the proxy data during this interval. Before 1600, there is less agreement between ensemble members, possibly indicating the presence of temporal non-stationarities in the relationship between ΔSLP and some proxy records.

## Data availability
The ΔSLP reconstructions generated in this study are available at https://doi.org/10.5281/zenodo.7742760. All data used in this

manuscript are available from online repositories, with the exception of two palaeoclimate proxy datasets. Palaeoclimate proxy data (Supplementary Table 1) incorporated into the reconstruction are available from the following sources: Iso2k data from https://lipdverse.org/iso2k/current_version/; SISAL data from https://researchdata.reading.ac.uk/256/; Humanes-Fuente et al. (2020) from https://www.cr2.cl/datos-dendro-amazonas-peru/; Lough (2007) from https://www.ncei.noaa.gov/access/paleo-search/study/15188; Lough et al. (2015) from https://www.ncdc.noaa.gov/paleo/study/18917; Chen et al. (2016) and Pumijumnong et al. (2020): data available on request from the authors. Gridded observational and reanalysis datasets used in this study are available from the following sources: ERA-20C from https://www.ecmwf.int/en/forecasts/dataset/ecmwf-reanalysis-20th-century; ICOADS from https://icoads.noaa.gov/products.html; HadSLP from https://www.metoffice.gov.uk/hadobs/hadslp2/. Reconstructions of volcanic forcing are available from the following sources: Toohey and Sigl (2017) 'eVolv2k' from https://www.wdc-climate.de/ui/project?acronym=eVolv2k and Supplementary Material of Dee et al. (2020) https://www.science.org/doi/10.1126/science.aax2000; Gao et al. (2008) from http://climate.envsci.rutgers.edu/IVI2/. PAGES 2k reconstructions of GMST through the Common Era are available from https://www.ncei.noaa.gov/pub/data/paleo/pages2k/neukom2019temp/recons/. Reconstructions of tropical Pacific variability available from the following sources: Niño 3.4 from https://www.ncei.noaa.gov/access/paleo-search/study/8704, https://www.ncei.noaa.gov/access/paleo-search/study/11749, https://www.ncei.noaa.gov/access/paleo-search/study/29050 and https://atmos.washington.edu/~hakim/lmr/LMRv2/; Niño 3 from https://www.ncei.noaa.gov/access/paleo-search/study/6250; Niño 4 from https://www.ncei.noaa.gov/access/paleo-search/study/28417; 'Proxy ENSO' from https://www.ncei.noaa.gov/access/paleo-search/study/8409; IPO from https://data.aad.gov.au/metadata/AAS_4537_2000y-Interdecadal-Pacific-Oscillation-Reconstruction. PMIP3/CMIP5 and PMIP4/CMIP6 simulations are publicly available from Earth System Grid Federation nodes, https://esgf.llnl.gov/index.html. The CESM1 LME is available from the Earth System Grid, https://www.earthsystemgrid.org/. Processed time series are also provided in the repository associated with this submission, https://doi.org/10.5281/zenodo.7742760.

## Code availability

The code that supports the findings of this study is available at https://doi.org/10.5281/zenodo.7742760.

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

**Acknowledgements** This research was supported by US National Science Foundation (NSF) P2C2 grants AGS-1805141 to Washington University in St. Louis (B.K.), AGS-1805143 to the University of California, Santa Barbara (S.S.) and AGS-2041281 to the University of Hawai'i at Mānoa (S.C.). S.S. was also supported by NSF grant OCE-2202794. The CESM1 project is supported primarily by the NSF. G.F. was supported by the Australian Research Council Centre of Excellence for Climate Extremes. The Iso2k database is a contribution to Phases 3 and 4 of the PAGES 2k Network. PAGES receives support from the Swiss Academy of Sciences, the US NSF and the Chinese Academy of Sciences. This is School of Ocean and Earth Science and Technology (SOEST) publication no. 11701. We thank S. Eggins for feedback on the first draft of the manuscript, and J. Emile-Geay for guidance with the SEA.

**Author contributions** All authors contributed to conceptualization of the study. G.F. led the methodology design and development, with input from B.K., S.C. and S.S. G.F. and S.C. performed all formal analysis. G.F. performed all validation and data visualization, with input from B.K., S.C. and S.S. G.F. wrote the manuscript and created all figures, and all authors contributed to subsequent editing and review. B.K., S.C. and S.S. acquired funding to support this research.

**Competing interests** The authors declare no competing interests.

**Additional information**
**Correspondence and requests for materials** should be addressed to Georgina Falster.

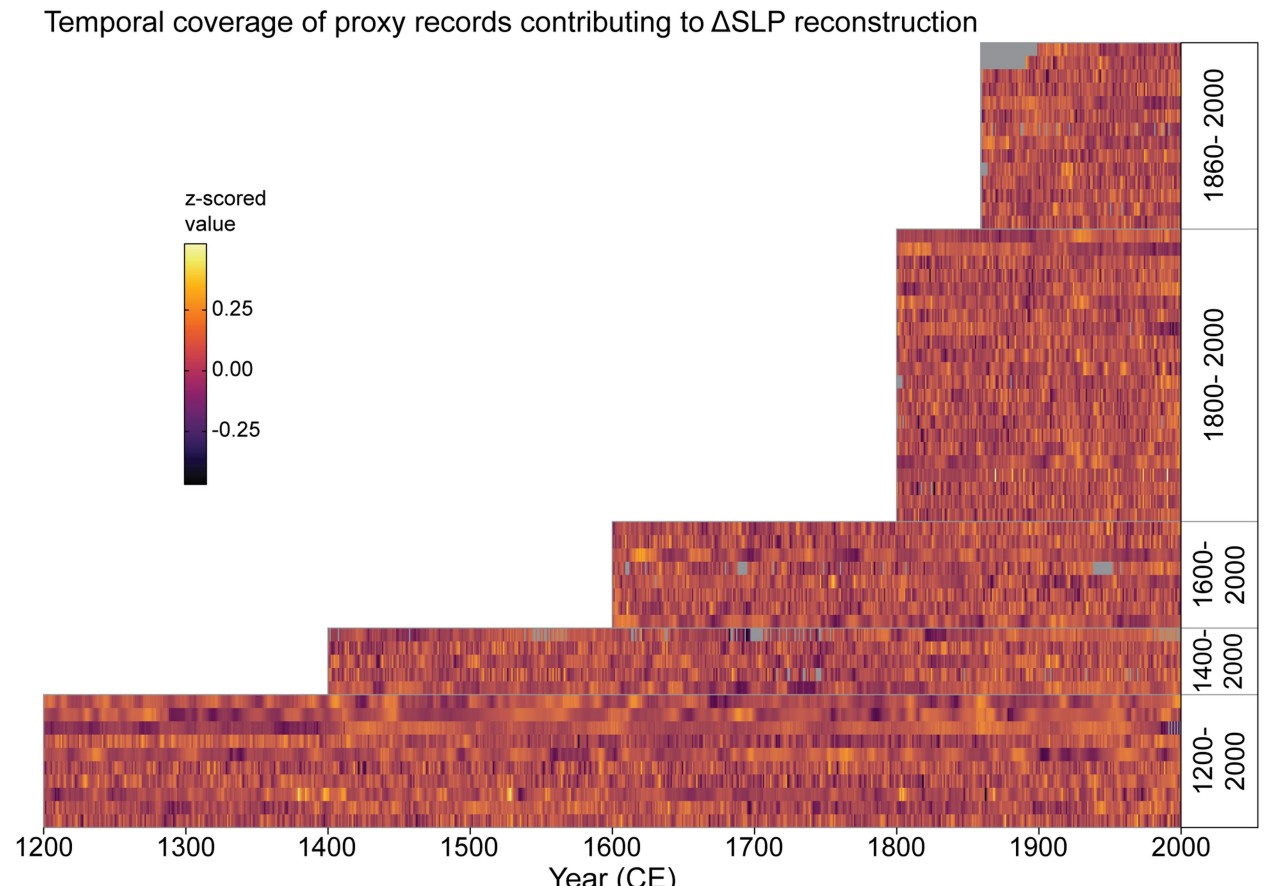

**Extended Data Fig. 1 | Temporal coverage of the 59 palaeoclimate proxy records contributing to the temporal segments of the ΔSLP reconstruction.** Data values shown for each time series have been scaled to zero mean and unit variance ('z-scores'). Grey space within time series denotes missing values; these gaps were filled using the DINEOF method before incorporation into the reconstruction (Methods). If a record extended past the start of a temporal segment, but not far enough to be included in the next-oldest segment, we truncated the values to match the oldest segment to which that record contributed.

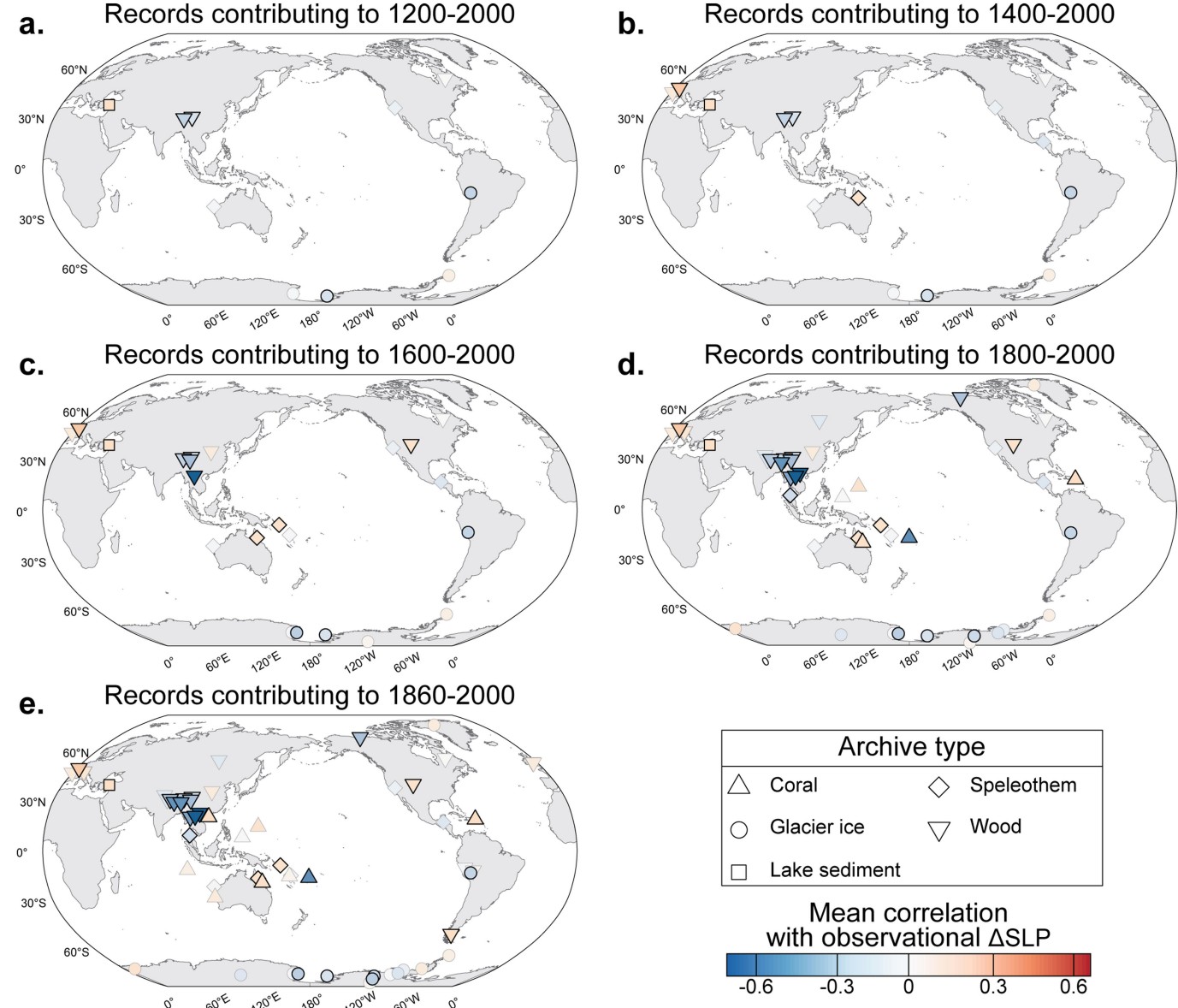

**Extended Data Fig. 2 | Maps showing the correlation of each component proxy record with instrumental ΔSLP in the calibration interval (1900–2000). a**, Location of proxy records contributing to the 1200–2000 section of the reconstruction. **b**, Location of proxy records contributing to the 1400–2000 section of the reconstruction. **c**, Location of proxy records contributing to the 1600–2000 section of the reconstruction. **d**, Location of proxy records contributing to the 1800–2000 section of the reconstruction.

**e**, Location of proxy records contributing to the 1860–2000 section of the reconstruction. Symbol colour corresponds to the mean correlation of the proxy record with ΔSLP calculated from HadSLP[25], ICOADS[26] and ERA-20C (ref. 27). Symbol shape denotes the proxy archive type. Black outline denotes that the proxy record is significantly (*P* < 0.1) correlated with instrumental ΔSLP. Maps created in R, using coastlines from Natural Earth.

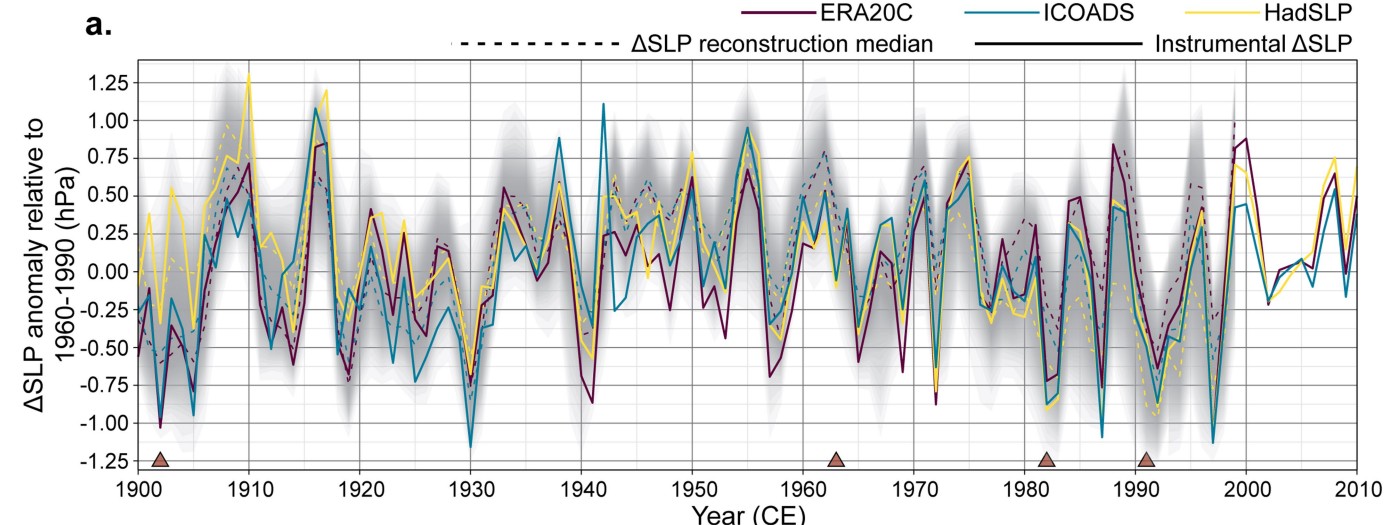

## b. Correlations in the calibration interval (1900-2000 CE)

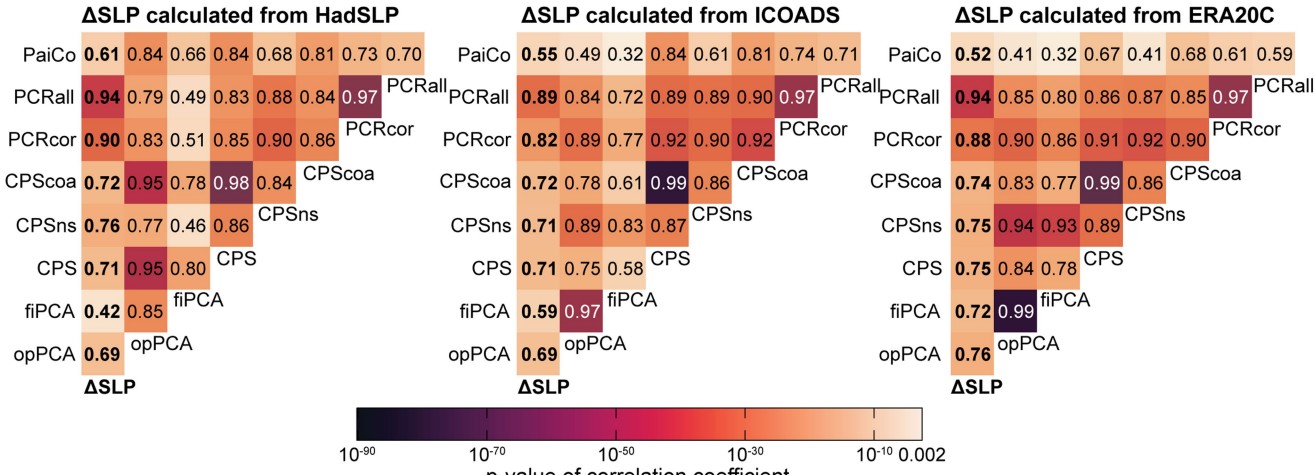

**Extended Data Fig. 3 | Correspondence between reconstructed ΔSLP and instrumental ΔSLP in the calibration interval (1900–2000). a**, Ensemble reconstruction of the PWC (in terms of the trans-Pacific SLP gradient; ΔSLP) from 1900 to 2000 (that is, Fig. 1a, zoomed in on the period of overlap with instrumental data). ΔSLP anomalies calculated with respect to 1960–1990. Grey shading represents the 2.5th/97.5th quantiles for the full ensemble (n = 4,800). Solid coloured lines show ΔSLP for 1900–2010, calculated from three gridded products ERA-20C (ref. 27), ICOADS[26] and HadSLP[25]. Dashed coloured lines show ΔSLP reconstruction sub-ensemble medians for ensemble members trained on each of those products, for example, dashed yellow line shows median ΔSLP from ensemble members trained on data from HadSLP (n = 1,600). Mean RMSE between observed ΔSLP and ensemble median ΔSLP is 0.27.

For comparison, mean RMSE between ΔSLP calculated from the three gridded products is 0.3. Triangles denote volcanic eruptions with reconstructed SAOD ≥ 0.05 (ref. 35). **b**, Correlation of the median of ΔSLP reconstruction ensemble members calculated with each reconstruction method with medians from each of the other reconstruction methods, as well as instrumental ΔSLP (first column, bold). Correlations are for the 1900–2000 interval and all are significant (P < 0.05). Correlations are shown for reconstruction method medians for ensemble members trained on ΔSLP calculated from HadSLP, ICOADS and ERA-20C (left to right). Colours scale with the significance of the correlation coefficients. Mean correlations of reconstructed versus instrumental ΔSLP for each reconstruction method are as follows: PaiCo, 0.56; PCRall, 0.92; PCRcor, 0.87; CPScoa, 0.73; CPSns, 0.74; CPS, 0.72; fiPCA, 0.58; opPCA, 0.71.

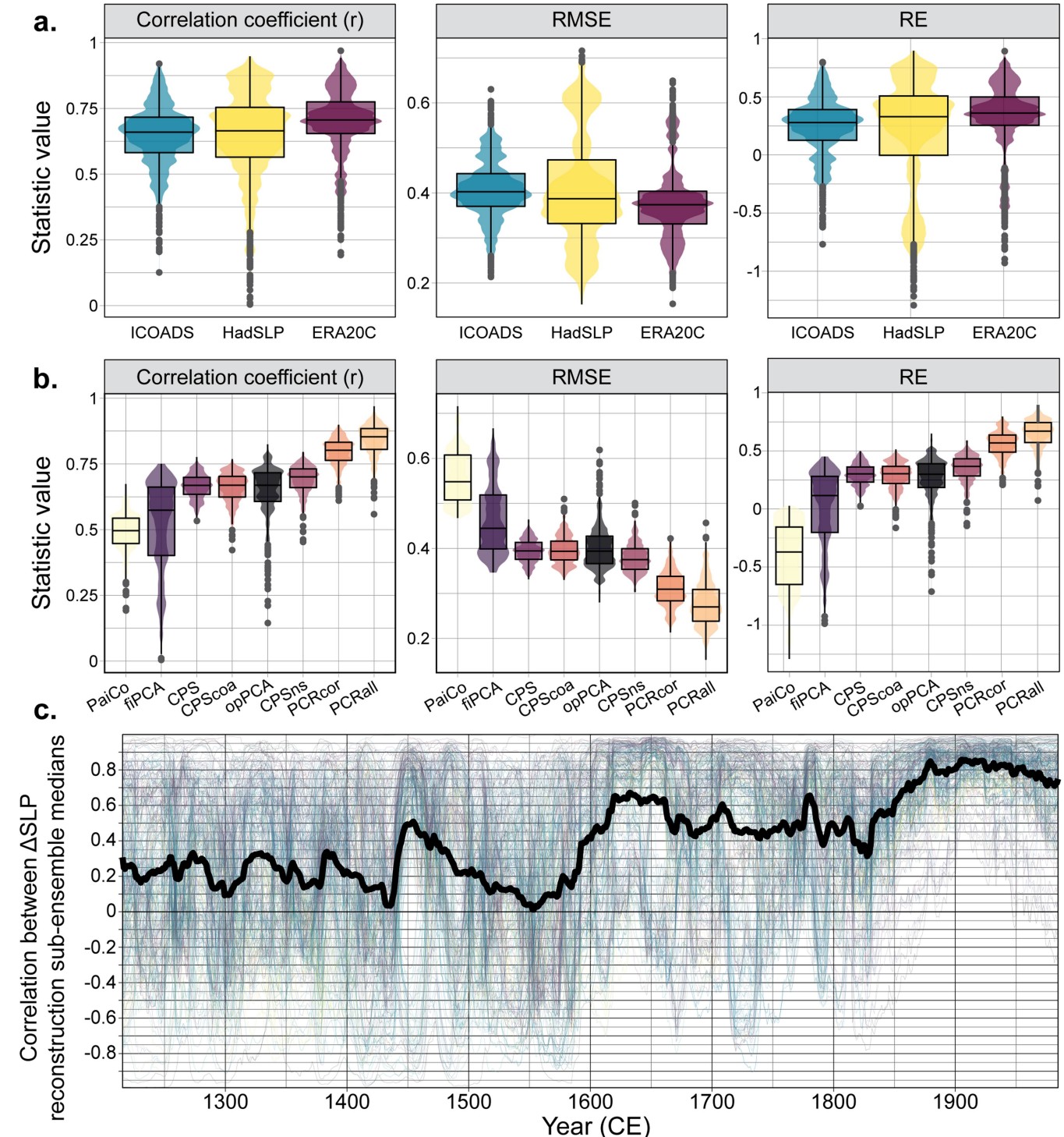

**Extended Data Fig. 4 | Comprehensive assessment of reconstruction skill.**
**a,b**, Violin-and-box plots ('voxplots') summarizing ΔSLP reconstruction skill in terms of correlation coefficient (*r*), RMSE and RE (ref. 132). These voxplots show the skill tests for the ΔSLP reconstruction ensemble shown in the main text. All tests were performed on all 4,800 ΔSLP reconstruction ensemble members. Voxplots show the distribution of scores; boxes shows median and interquartile range (IQR), whiskers show IQR × 1.5, points show outliers. Each individual ensemble member was assessed against ΔSLP used to train that particular ensemble member, that is, ΔSLP calculated from HadSLP, ICOADS or ERA-20C. **a**, Skill scores split according to the gridded SLP product used to calculate the ΔSLP training data. **b**, Skill scores split according to the reconstruction method. **c**, Running 30-year correlations between the ensemble medians for each possible combination of reconstruction method and gridded SLP product used to calculate the ΔSLP training data. Thick black line shows the median running correlation. Lines are coloured according to unique combinations of reconstruction method.

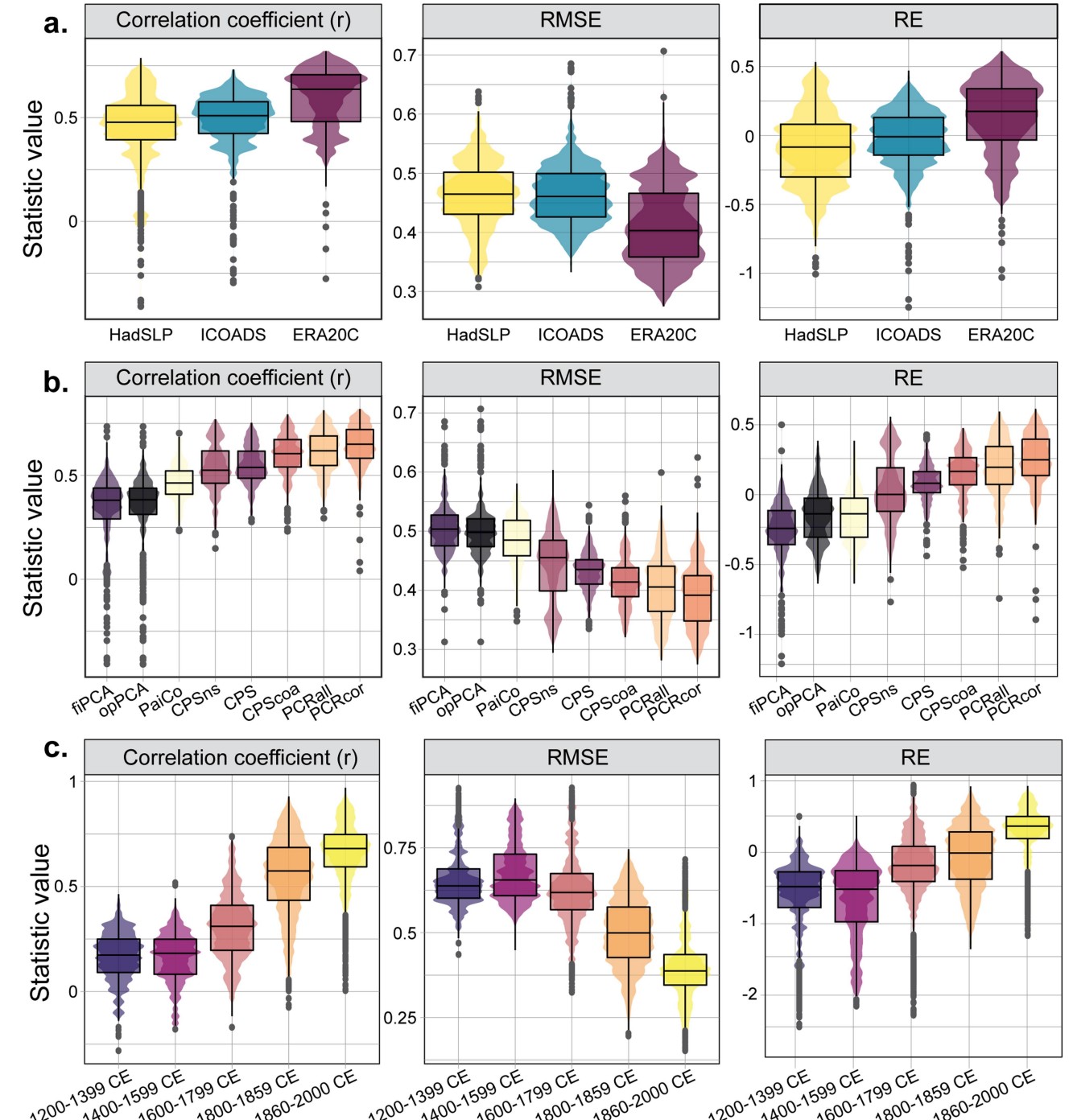

**Extended Data Fig. 5 | Comprehensive assessment of reconstruction skill, performed on separate calibration and validation intervals.** ΔSLP reconstruction ensemble members trained on a 1951–2000 calibration interval and assessed against instrumental ΔSLP in an independent (1900–1950) interval. These scores provide a minimum independent estimate of reconstruction skill; in reality, skill is probably higher, as we used a longer calibration interval. **a,b,** Violin-and-box plots ('voxplots') summarizing ΔSLP reconstruction skill in terms of correlation coefficient (*r*), RMSE and RE (ref. 132). All tests were performed on all 4,800 ΔSLP reconstruction ensemble members. Voxplots show the distribution of scores; boxes shows median and interquartile range

(IQR), whiskers show IQR × 1.5, points show outliers. Each individual ensemble member was assessed against ΔSLP used to train that particular ensemble member, that is, ΔSLP calculated from HadSLP, ICOADS or ERA-20C. **a,** Skill scores split according to the gridded SLP product used to calculate the ΔSLP training data. **b,** Skill scores split according to the reconstruction method. **c,** As per **a** and **b**, but with panels showing skill scores for each temporal subset that contributed to the full reconstruction interval. This provides an estimate of the decrease in reconstruction skill back through time (as the number of available proxy records decreases; see Extended Data Figs. 1 and 2).

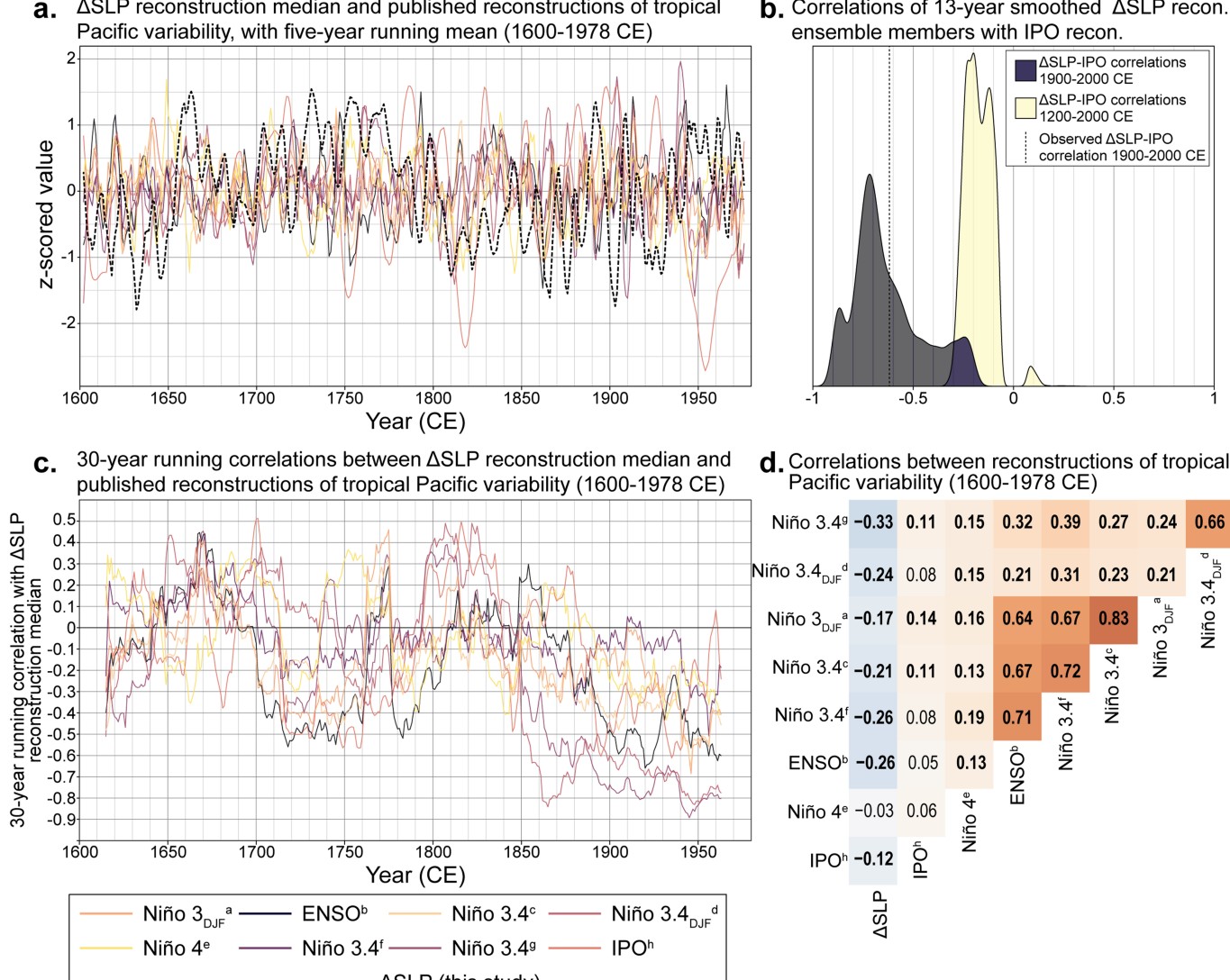

**a.** ΔSLP reconstruction median and published reconstructions of tropical Pacific variability, with five-year running mean (1600-1978 CE)

**b.** Correlations of 13-year smoothed ΔSLP recon. ensemble members with IPO recon.

- ΔSLP-IPO correlations 1900-2000 CE
- ΔSLP-IPO correlations 1200-2000 CE
- Observed ΔSLP-IPO correlation 1900-2000 CE

**c.** 30-year running correlations between ΔSLP reconstruction median and published reconstructions of tropical Pacific variability (1600-1978 CE)

**d.** Correlations between reconstructions of tropical Pacific variability (1600-1978 CE)

| | ΔSLP | IPOh | Niño 4e | ENSOb | Niño 3.4f | Niño 3.4c | Niño 3DJFa | Niño 3.4DJFd |
|---|---|---|---|---|---|---|---|---|
| Niño 3.4g | **−0.33** | 0.11 | 0.15 | **0.32** | **0.39** | **0.27** | **0.24** | **0.66** |
| Niño 3.4DJFd | **−0.24** | 0.08 | 0.15 | **0.21** | **0.31** | **0.23** | **0.21** | |
| Niño 3DJFa | **−0.17** | 0.14 | 0.16 | **0.64** | **0.67** | **0.83** | | |
| Niño 3.4c | **−0.21** | 0.11 | 0.13 | **0.67** | **0.72** | | | |
| Niño 3.4f | **−0.26** | 0.08 | **0.19** | **0.71** | | | | |
| ENSOb | **−0.26** | 0.05 | **0.13** | | | | | |
| Niño 4e | −0.03 | 0.06 | | | | | | |
| IPOh | **−0.12** | | | | | | | |

Legend:
- Niño 3DJFa
- ENSOb
- Niño 3.4c
- Niño 3.4DJFd
- Niño 4e
- Niño 3.4f
- Niño 3.4g
- IPOh
- ------ ΔSLP (this study)

**Extended Data Fig. 6 | Comparison of ΔSLP reconstruction with published reconstructions of SST-based tropical Pacific variability. a**, Coloured lines show published reconstructions of ENSO or the IPO across the common time period 1600–1978. Dashed black line shows our ΔSLP reconstruction median. For visualization purposes, all records have been scaled to zero mean and unit variance and had a 5-year running mean applied. **b**, Correlations of individual ΔSLP reconstruction ensemble members with a reconstruction of the IPO[31]. To match the IPO reconstruction, a 13-year Gaussian smoothing filter was applied to the ΔSLP reconstruction ensemble members. Yellow distribution shows significant ($P < 0.05$) correlations over the full 1200–2000 interval; purple–grey distribution shows significant correlations over the calibration interval (1900–2000). Dotted vertical line shows the correlation between instrumental IPO[143] and mean ΔSLP calculated from ERA-20C (ref. 27), ICOADS[26] and HadSLP[25], with a 13-year Gaussian smoothing filter applied, across 1900–2000 ($P < 0.05$). **c**, 30-year running correlations between the ΔSLP reconstruction ensemble median and published reconstructions of SST-based tropical Pacific variability. **d**, Correlations between reconstructions of tropical Pacific variability (ENSO, IPO, ΔSLP) across their common 1600–1978 interval. Correlations in bold are significant ($P < 0.05$). Reconstructions are as follows: [a]Niño 3 (DJF)[140]; [b]'proxy ENSO'[138]; [c]Niño 3.4 (ref. 142); [d]Niño 3.4 (ref. 125); [e]Niño 4 (ref. 141); [f]Niño 3.4 (DJF, running PC1)[117]; [g]Niño 3.4 (ensemble median)[139]; [h]IPO[31].

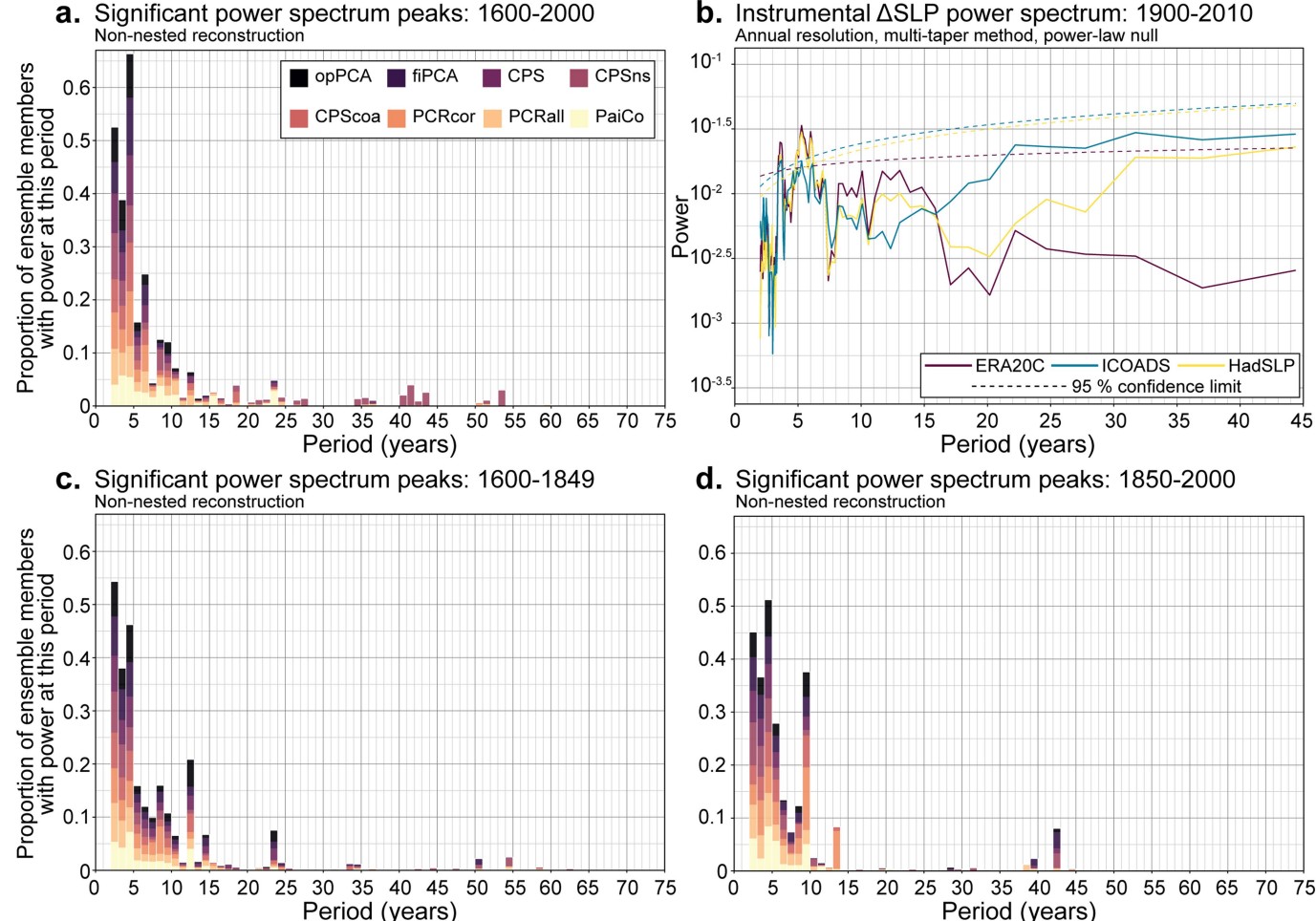

**Extended Data Fig. 7 | Power spectral densities of ΔSLP reconstruction (1600–2000, non-nested) and observations. a**, Proportion of the 4,800 ΔSLP reconstruction ensemble members with significant ($P < 0.05$) power in all periods from 1 to 75 years. Significance is evaluated against a power-law null[135]. Colours denote reconstruction method. ΔSLP reconstructed using only records with full coverage across 1600–2000, that is, without the temporal nesting approach (see Methods). **b**, Power spectra for ΔSLP calculated from ERA-20C (ref. 27), ICOADS[26] and HadSLP[25]. Solid lines show the power spectra calculated from annual ΔSLP (1900–2010); dashed lines show the 95% confidence limit. **c**, As per **a** but for 1600–1849, in all possible 150-year segments. The division into 150-year segments was to enable direct comparison with the power spectrum in the industrial era (Methods). **d**, As per **a** but for 1850–2000.

### All ΔSLP ensemble members

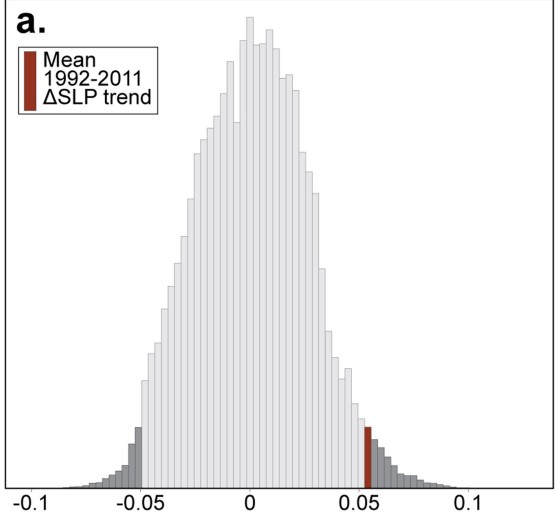

### ΔSLP ensemble members trained on ERA20C

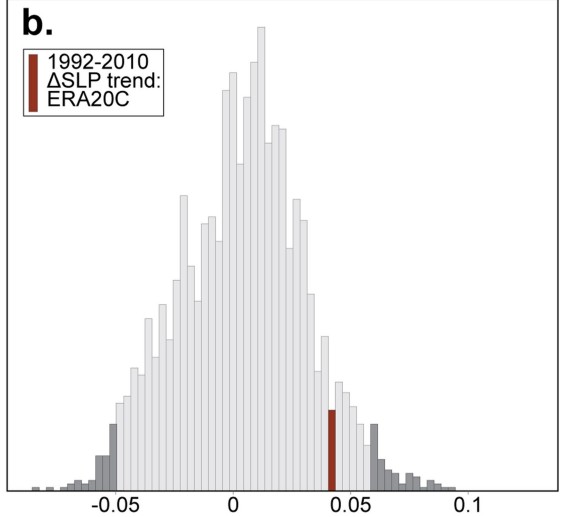

### ΔSLP ensemble members trained on ICOADS

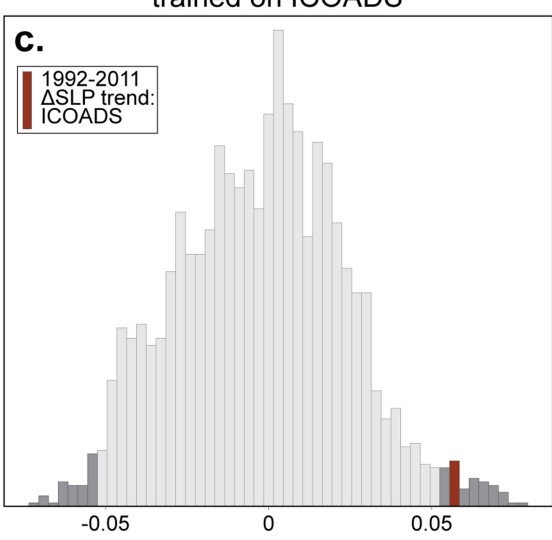

### ΔSLP ensemble members trained on HadSLP

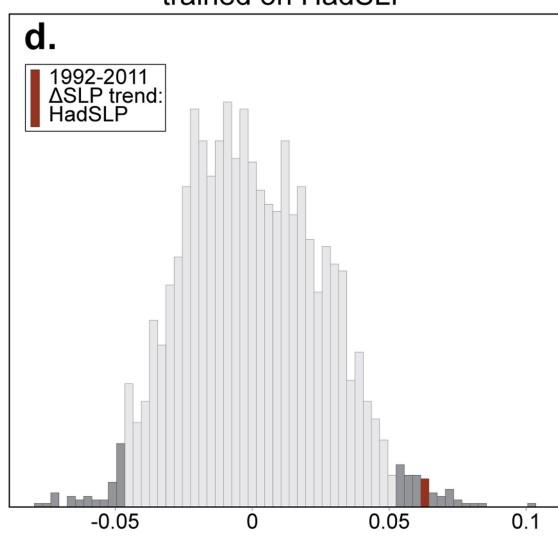

**Extended Data Fig. 8 | Distribution of 20-year trends in ΔSLP anomalies following all volcanic eruptions with SAOD equal to or greater than that of the 1991 Mount Pinatubo eruption (SAOD ≥ 0.1) during the 1200–2000 interval. a**, Full distribution of the magnitude of post-eruption 20-year trends in ΔSLP anomalies across 1900–2000 (from all individual reconstruction ensemble members, $n = 4,800$). Dark grey tails show the 2.5th and 97.5th percentiles. Red bar shows the mean magnitude of the 1992–2011 ΔSLP trend from instrumental data (following the 1991 eruption of Mount Pinatubo).

**b**–**d**, As per **a** but only showing trends in reconstruction ensemble members trained on ΔSLP calculated from ERA-20C (ref. 27), ICOADS[26] and HadSLP[25], respectively. Red bar shows the magnitude of the 1992–2011 ΔSLP trend (from HadSLP and ICOADS) or the 1992–2010 ΔSLP trend (from ERA-20C) calculated from instrumental data. Volcanic eruption years taken from the 'eVolv2k' ice-core sulfate-based reconstruction of Common Era volcanic sulfate aerosol loading[16,35]. Note that differences between the panels arise mainly from differences between the three observational products.

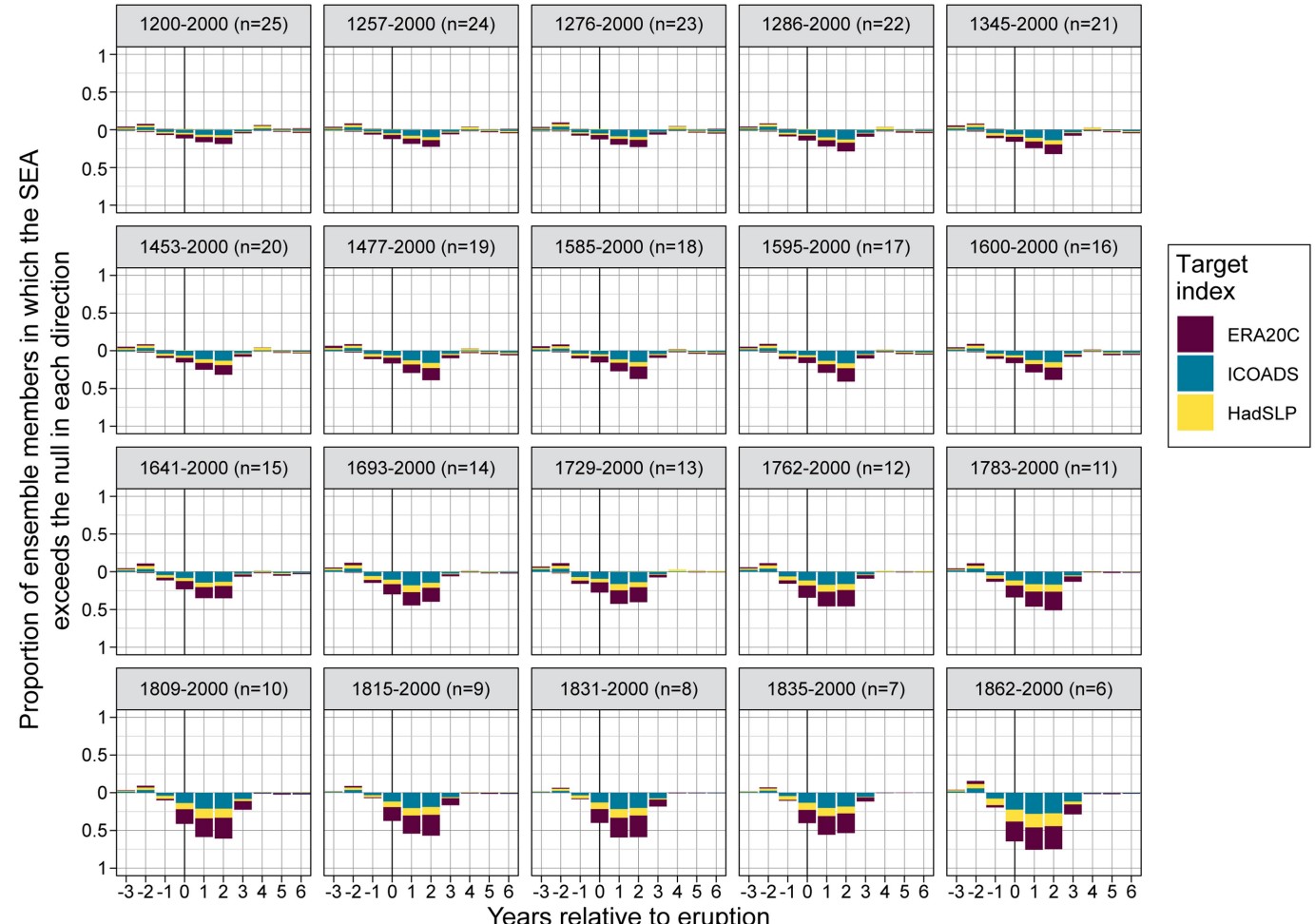

**Extended Data Fig. 9 | SEA for ΔSLP reconstruction, with volcanic eruption years as defined in the volcanic forcing reconstruction of ref.** 35. Bars show the proportion of the 4,800 ΔSLP reconstruction ensemble members that have a significant positive (La Niña-like) or negative (El Niño-like) ΔSLP anomaly in the −3 to +6 years relative to each eruption composite (see Methods). Starting with the 25 strongest eruptions of the 1200–2000 period, each panel shows results from the SEA calculated using a different number of these eruptions (showing the change in the result when removing progressively older eruptions). The top-left panel shows results from an SEA using all 25 eruptions. Going left to right row-wise and downward column-wise, the oldest eruption is sequentially removed, until the SEA is performed using only the six most recent of the 25 original eruptions (bottom right). Colour blocks on each bar show the proportion of responses from ensemble members calculated using each gridded SLP product used to calculate the ΔSLP training index.

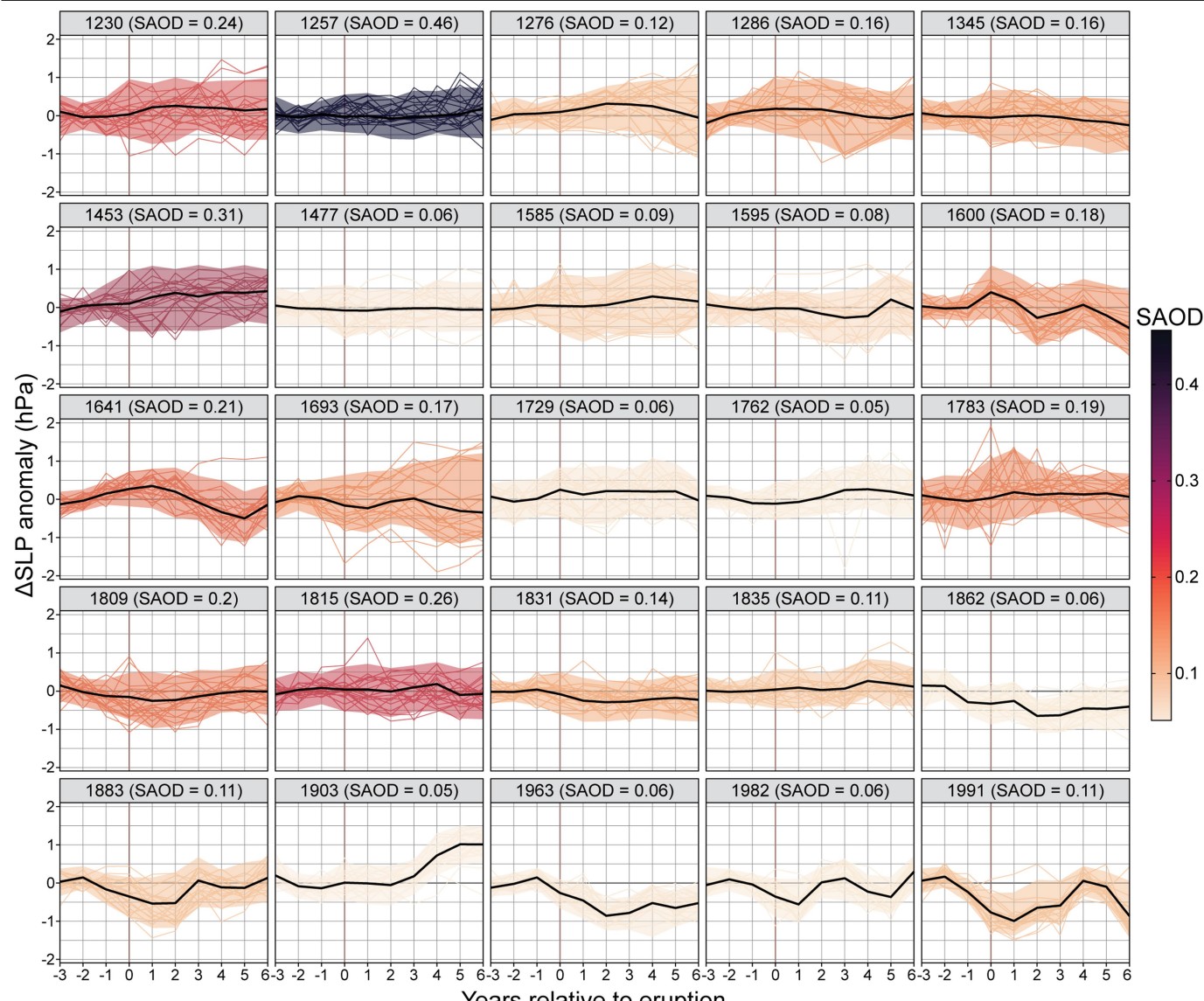

**Extended Data Fig. 10 | Ensemble ΔSLP response to the 25 strongest volcanic eruptions of the 1200–2000 period.** These are the 25 eruptions used in the SEA described in the main text. In each panel, the black line shows the ensemble median response to the eruption and the coloured windows show the upper and lower 5th percentiles of the ensemble response. We also show 20 randomly chosen individual ensemble members as thin coloured lines for comparison. Before calculating the summary statistics describing the ensemble response to each eruption, the responses of individual ensemble members were centred on the pre-eruption mean. The vertical red line on each panel shows the eruption year (listed in the title strip of each panel, along with the reconstructed SAOD[16,35]). Colours correspond to eruption magnitude.

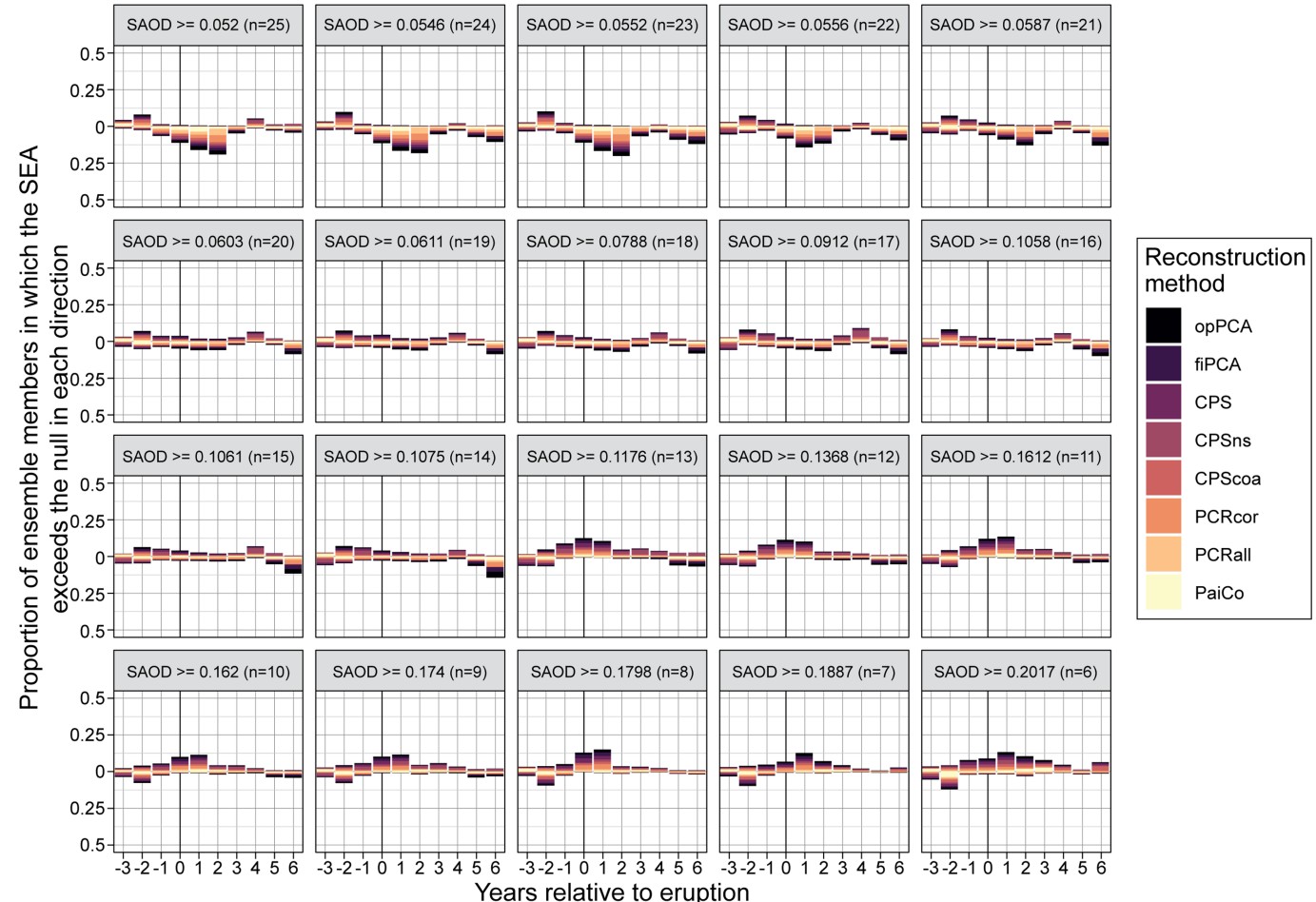

**Extended Data Fig. 11 | SEA for ΔSLP reconstruction, with volcanic eruption years as defined in the volcanic forcing reconstruction of ref.** 35. Bars show the proportion of the 4,800 ΔSLP reconstruction ensemble members that have a significant positive (La Niña-like) or negative (El Niño-like) ΔSLP anomaly in the −3 to +6 years relative to each eruption composite (see Methods). Starting with the 25 strongest eruptions of the 1200–2000 period (top left), each panel shows results from the SEA calculated using a different number of these eruptions (showing the change in the result with different SAOD thresholds).

Going left to right row-wise and downward column-wise, the weakest eruption is sequentially removed from the analysis (that is, the SAOD threshold for inclusion is raised) until the SEA is performed using only the six largest eruptions of the 1200–2000 interval (bottom right). Colour blocks on each bar show the proportion of responses from ensemble members calculated using each reconstruction method. Note that the *y*-axis scale is half that of Fig. 4, that is, showing a maximum proportion of 0.5.

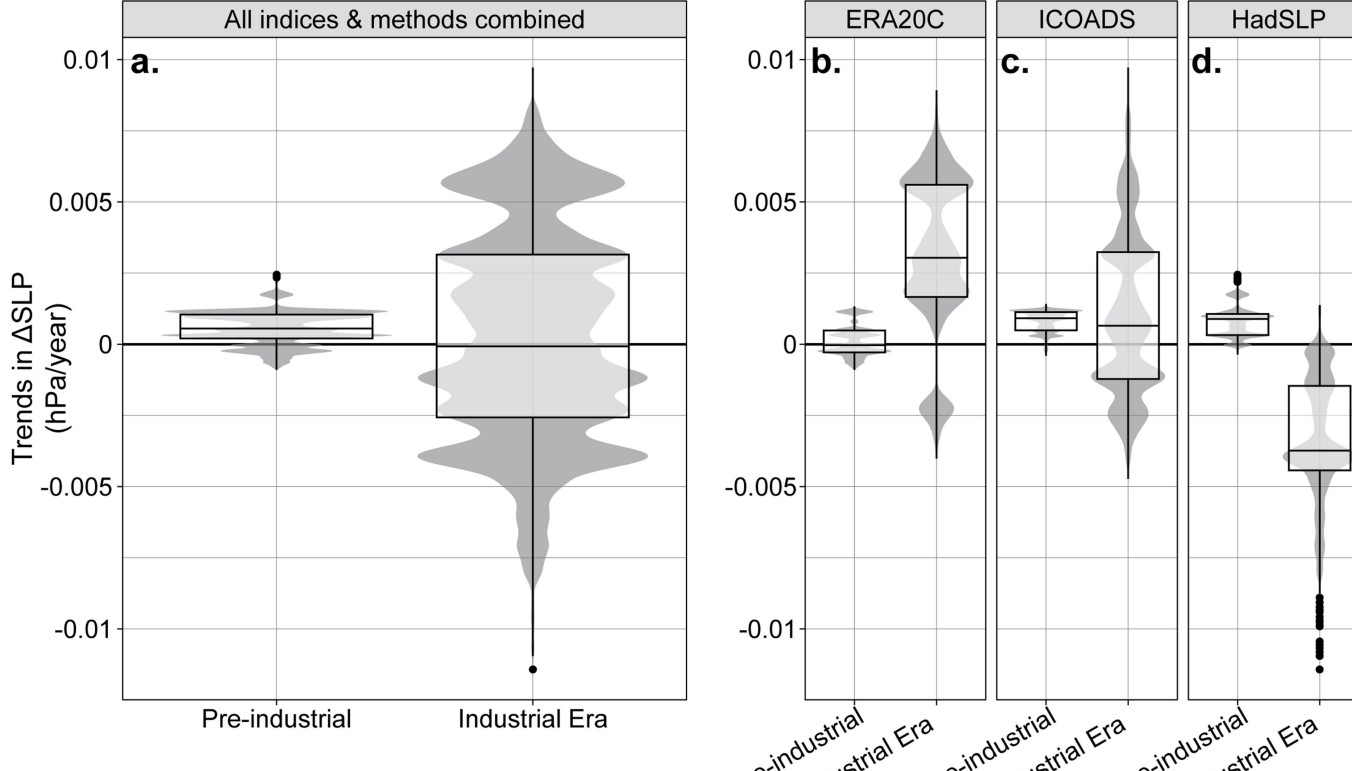

**Extended Data Fig. 12 | Violin-and-box plots ('voxplots') showing the distribution of linear trends (regression coefficients) in the 4,800-member ΔSLP reconstruction ensemble.** Voxplots show the distribution of trends across all ensemble members; boxes shows median and interquartile range (IQR), whiskers show IQR × 1.5, points show outliers. **a**, Linear trends for the preindustrial (1200–1849) and industrial-era (1850–2000) intervals of the full reconstruction. **b**, As per **a** but only showing trends from ensemble members trained on ΔSLP calculated from ERA-20C. **c**, as per **a** but only showing trends from ensemble members trained on ΔSLP calculated from ICOADS. **d**, As per **a** but only showing trends from ensemble members trained on ΔSLP calculated from HadSLP. These plots include all trends, regardless of significance.

**Extended Data Table 1 | Summary of ΔSLP reconstruction skill in terms of correlation coefficient (*r*), RMSE, RE (ref. 132) and coefficient of efficiency (CE)[144]**

## Summary of ΔSLP reconstruction skill

| Skill metric | Calibration 1900-2000 | Calibration 1951-2000 | Validation 1900-1950 |
|:---:|:---:|:---:|:---:|
| **Full reconstruction** | | | |
| r | 0.81 | 0.83 | 0.77 |
| RMSE | 0.27 | 0.27 | 0.26 |
| RE | 0.64 | 0.69 | 0.57 |
| CE | 0.64 | 0.69 | 0.57 |
| **CPS only** | | | |
| r | 0.72 | 0.67 | 0.72 |
| RMSE | 0.33 | 0.42 | 0.31 |
| RE | 0.46 | 0.22 | 0.45 |
| CE | 0.46 | 0.22 | 0.45 |
| **CPS 'centre of action' only** | | | |
| r | 0.73 | 0.71 | 0.77 |
| RMSE | 0.32 | 0.37 | 0.29 |
| RE | 0.48 | 0.39 | 0.51 |
| CE | 0.48 | 0.39 | 0.51 |
| **CPS 'no seasonal' only** | | | |
| r | 0.75 | 0.76 | 0.71 |
| RMSE | 0.31 | 0.41 | 0.31 |
| RE | 0.51 | 0.26 | 0.44 |
| CE | 0.51 | 0.26 | 0.44 |

Values in the second column are for the ΔSLP reconstruction ensemble shown in the main text, that is, calculated with the full 1900–2000 calibration interval. The third column shows values for the ΔSLP reconstruction ensemble calculated with the shorter (1951–2000) calibration interval. The fourth column shows validation values for the ΔSLP reconstruction ensemble calculated with the 1951–2000 calibration interval, assessed against instrumental ΔSLP in an independent interval (1900–1950). Rows 1 to 4 show skill scores calculated for the ΔSLP ensemble median, assessed against mean ΔSLP from HadSLP, ICOADS and ERA-20C. Rows 5 to 16 show scores for ensemble members calculated using variants on the CPS method (see Methods). Skill scores are calculated for ΔSLP ensemble medians, assessed against mean ΔSLP from HadSLP, ICOADS and ERA-20C.