## [Peer Review File · Nature]

Manuscript Title: Forced changes in the Pacific Walker Circulation over the past millennium

Reviewer Comments & Author Rebuttals

Reviewer Reports on the Initial Version:

Referees' comments:

Referee #1 (Remarks to the Author):

A. Summary of the key results

Falster et al.'s "Forced changes in the Pacific Walker Circulation over the past millennium" describes a highly original reconstruction and assessment of Walker Circulation variability using a network of long proxy records in the Iso2k network.

The chief conclusions of this analysis are: 1) the recent 1992-2011 trends in the Walker Circulation are not unprecedented over the last 800 years, 2) Volcanic eruptions prompt an observable weaker Walker response, 3) The relationship between global mean surface temperature and Walker Circulation is not as straightforward as many studies have suggested. The two important indices can vary considerably.

This manuscript will be of interest to readers of Nature who are interested in climate dynamics, paleoclimatology, and historical climate change. I recommend this manuscript for publication with minor revisions.

B. Originality and significance

This analysis is very novel and illustrates the utility of a new water isotope dataset (Konecky et al. 2020) to examine a long standing question in the scientific literature. This analysis does an admirable job of examining the uncertainty in their reconstructions. The authors examine the sensitivity of their method to dating errors, calibration dataset, temporal period of reconstruction and reconstruction methods—and clearly illustrate the sensitivity of their results to these choices. It is difficult to envision how the authors could have done more to test the robustness of their reconstruction to arbitrary methodological choices.

This work would be very relevant to several fields. This work also serves to highlight the utility of water isotope data.

One suggestion for the authors is to amplify the fact that this analysis uses such a novel water isotope dataset. This is *really* exciting (and novel!) and I am afraid that this important nuance might be lost on readers who have never heard of the Iso2k project.

C. Data and Methodology

The data used and methodology used in this manuscript are both very reasonable and very original. The new Iso2k dataset (Konecky et al. 2020) used heightens the originality and significance of this study.

D. Appropriate use of statistics

I did not attempt to repeat any statistical tests, but all tests and results used in the manuscript appear logical. All figures illustrate an admirable use of error bars and significance testing. Furthermore, the authors take supreme care to test the sensitivity of their results to reconstruction methods. As an example of this, a 1992-2011 was a key analysis period in their study (due to previously published work in the scientific literature), but this period begins just after Pinatubo. The authors examined the sensitivity of this trend to the volcanic eruption by examining the 20

year trends after all volcanic eruptions and concluding no strong signal.

My largest concern with this study is that the 1992-2011 period is of chief importance to the analysis, but is a period not covered by the authors reconstruction (800-200AD). The authors repeatedly compare the observed instrumental trend over the 1992-2011 period to the proxy reconstructed trends and draw major conclusions from this comparison. I understand that proxy records grow sparse after the year 2000, but it would be helpful to see just how well the reconstruction method compares with the instrumental observations over the 20th century. (Perhaps add a panel to Figure 3)? The authors state that there is high correlation between the two (Supplementary Table 2, $r \sim 0.8$), but it would be helpful to see the agreement between the reconstruction and all instrumental products over this important period. I think readers will question if the authors are indeed comparing apples to apples with the present discussion.

My second largest concern with the methodology used in the study regards the variance adjustment applied to the reconstructions to maintain the variance of the 20th century throughout the 800-2000 reconstruction interval. The motivations for this variance adjustment are highly logical—nearly all proxy reconstructions feature a loss in variance with time (as the available proxy fraction dwindles), but my concern lies in the documented change in ENSO variability over the last millennium and how that may influence the Walker Circulation reconstruction. Both Grothe et al. 2020 and Cobb et al. 2013 demonstrate an increase in Niño 3.4 variability over the last few decades. At the very least, the authors could comment on how the assumed 20th century variance may influence the results (and cite these studies).

Grothe, Pamela R., Kim M. Cobb, Giovanni Liguori, Emanuele Di Lorenzo, Antonietta Capotondi, Yanbin Lu, Hai Cheng et al. "Enhanced El Niño–Southern oscillation variability in recent decades." *Geophysical Research Letters* 47, no. 7 (2020): e2019GL083906.

Cobb, Kim M., Niko Westphal, Hussein R. Sayani, Jordan T. Watson, Emanuele Di Lorenzo, H. Cheng, R. L. Edwards, and Christopher D. Charles. "Highly variable El Niño–southern oscillation throughout the Holocene." *Science* 339, no. 6115 (2013): 67-70.

E. Conclusions

The conclusions of this manuscript are well grounded (aside from the concern about the reconstruction comparisons with the instrumental record over the 1992-2011 period). The authors describe their conclusions in the context of present literature and their results are easily interpretable.

I did not personally try to replicate any of the analysis, but the authors are very clear about the methodology and data used, thus their results seem reproducible. Furthermore, the authors state that they will provide access to their reconstruction upon publication—and offer to provide code upon request.

F. Suggested Improvements

I have major and minor suggestions for improvement and have listed them below.

Major Points

The first two points are discussed in the "Appropriate Use of Statistics" section, but I will briefly outline them again (apologies for repetition).

First: I worry about the comparison of the proxy reconstruction to the instrumental archives over the 1992-2011 period. I think that adding a figure to show the consistency of the reconstruction results to the HADSLP, ICOADS, and ERA20C reanalysis products over the 20th century would help abate the concern that it is not entirely appropriate to directly compare these two sources of data.

Secondly: Adjusting the variance of the reconstruction to that observed in the 20th century causes some concern. I suggest that the authors comment that they acknowledge that they may be

incorporating some bias with this approach.

Finally: The authors include some discussion about the utility of examining Walker Circulation Variability with a few different variables, but set up a contrast between examining the SST component of Walker Circulation change vs the atmospheric/ SLP component (eg. Line 195, but also in a few other places). I think a sentence or two to clarify that discussion would be helpful to readers. Few studies examining the climate response to volcanic eruptions use raw SST, instead they use a "relative SST" and examine the Nino3.4 SST after subtracting the tropical mean (due to the expectation that volcanic aerosols will cause cooling globally and mask the tropical Pacific response, eg. Khodri et al. 2017, Predybaylo et al. 2020, Zuo et al. 2018, many others). This is virtually impossible when using single paleoclimate proxy record that only monitors conditions at a single location.

Khodri, Myriam, Takeshi Izumo, Jérôme Vialard, Serge Janicot, Christophe Cassou, Matthieu Lengaigne, Juliette Mignot et al. "Tropical explosive volcanic eruptions can trigger El Niño by cooling tropical Africa." *Nature communications* 8, no. 1 (2017): 1-13.

Predybaylo, Evgeniya, Georgiy Stenchikov, Andrew T. Wittenberg, and Sergey Osipov. "El Niño/Southern Oscillation response to low-latitude volcanic eruptions depends on ocean pre-conditions and eruption timing." *Communications Earth & Environment* 1, no. 1 (2020): 1-13.

Zuo, M., Man, W., Zhou, T., and Guo, Z.: Different impacts of northern, tropical, and southern volcanic eruptions on the tropical Pacific SST in the last millennium, *J. Climate*, 31, 6729–6744, <https://doi.org/10.1175/JCLI-D-17-0571.1>, 2018.

Minor Points

Line 22—"the 4800 member ensemble" phrase is just confusing without introducing what the ensemble represents. How about something like, "We use a new paleoproxy-derived 4800 member ensemble"?

Line 40: A reference should consider adding to this discussion is the Seager et al. 2019 paper that illustrates some of this mismatch between observations and models may be entirely due to biases in climate models.

Seager, Richard, Mark Cane, Naomi Henderson, Dong-Eun Lee, Ryan Abernathey, and Honghai Zhang. "Strengthening tropical Pacific zonal sea surface temperature gradient consistent with rising greenhouse gases." *Nature Climate Change* 9, no. 7 (2019): 517-522.

Line 50: What is your specific definition for Walker Circulation? Suggest that the authors say "hydroclimate (rather than SLP directly)" or something like that for clarity.

Line 53: Again, be clear with what this 4800 member ensemble entails

Line 54: As the 1992-2011 period has been mentioned, it would help to clarify why the reconstruction does not extend to year 2100

Line 55: If specifically citing a study, its far less confusing for the reader if you write XX et al. 13 rather than just the superscripted number

Line 60: All readers would appreciate a little more information about "the dominant mode of global interannual precipitation d18O carries a strong imprint of delta SLP". Over what years? What was the proxy distribution? How do the authors define "strong imprint"? Perhaps a sentence or two more would help readers better appreciate the work you have put into this task!

Line 65: I would include a few more details about the proxy dataset in the main text. Are these

records at least annually resolved? What types of proxies compose these records?

Lines 70-90: Absolutely love that you did all of this sensitivity testing—and so clearly! This is one of my favorite parts of the paper. I was curious how you went from 59 proxies to a 4800 member ensemble. Could you provide the reader with this detail? We have 5 reconstruction methods, 3 calibration products, and some sort of age adjustment? Am I missing anything?

Line 95: How well do the instrumental products agree over the 1900-2022 interval?

Line 95: If you train the products on the 20th century instrumental record, aren't you forcing them to agree with the instrumental trends over the 20th century? Perhaps there is a way to check this influence

-- I see that you did this in the Supplemental Materials! Great! Clearly referencing that additional analysis here would be helpful!

Line 99: "this decrease in skill back through time is a common but rarely emphasized feature of paleoclimate reconstructions" – I'm not sure what you mean here. Could you clarify?

Line 105: Personally, I think some of this inter-method disagreement is really interesting. Why might these methods have strong disagreement over particular intervals? Changes in traditional covariance patterns between proxies and the Walker Circulation? Sanchez et al. 2021 documented similar disagreement in tropical Pacific reconstructions over the 1800-1850 periods when using coral-based paleo data assimilation and attributed the breakdown to different patterns of covariability during a period highly influenced by volcanic eruptions.

Sanchez, Sara C., Gregory J. Hakim, and Casey P. Saenger. "Climate model teleconnection patterns govern the Niño-3.4 response to early nineteenth-century volcanism in coral-based data assimilation reconstructions." *Journal of Climate* 34, no. 5 (2021): 1863-1880.

Line 107/Figure 2: How do some of these frequencies have zero power? Are the authors only plotting the periodicities with significant power? If so, be sure to say it! It is not immediately clear to the reader.

Line 114-115: Really happy the authors put in the sentence "the difference between the two distributions is within the range calculated between any other two intervals of equivalent length..."! I was curious if the PDFs had a more distinct difference if you compared slightly different intervals, eg. 1200-1900 vs 1900-2000?

Line 122: Could also cite the Seager et al. 2019 paper (mentioned earlier) here

Line 123: Could potentially discuss tropical Pacific Decadal Variability (eg. Power et al. 2021) here

Power, Scott, Matthieu Lengaigne, Antonietta Capotondi, Myriam Khodri, Jérôme Vialard, Beyrem Jebri, Eric Guilyardi et al. "Decadal climate variability in the tropical Pacific: Characteristics, causes, predictability, and prospects." *Science* 374, no. 6563 (2021): eaay9165.

Line 142-146: This is so cool! I had been worried about the potential influence of Pinatubo on the results given the importance of the 1992-2011 period, but I feel much more comfortable given this analysis. I am still a little worried about the potential apples to oranges comparison as detailed in the Major Points section.

Line 149: I again am wondering about the resolution of the proxies considered in this network. Are they all annual or sub annual?

Line 157: "reassigned major eruption"—this is great!

Line 165: how large is this trend?

Line 172: If specifically referencing a study, please use the name of the study and then superscript it instead of just using the superscripted numbers

Line 190-192: Consider re-writing this sentence for clarity "if all volcanic eruptions with..." I'm not totally sure I follow what you are saying.

Line 191: How is significance defined?

Line 195: Consider writing "Fewer LME ensemble members" to help the readers stay on point as you also call your proxy reconstruction an ensemble

Line 218: Again, this is so cool!

Line 225: Why pCO₂ instead of CO₂? I think CO₂ is more intuitive for readers

Line 248: Did the authors remove the tropical mean from this? See major point for more discussion

Line 259: This manuscript describes a very original and exciting analysis. However, I don't think that the authors have emphasized just how original this source of data is! Many scientists and working groups are interested in finding uses for water isotope data—and this is a great illustration! If there is space, I would definitely hammer on this point a little more.

Figure 2B. I would make two boxes to indicate just where your delta SLP index comes from

Figure 3. These distributions are very gaussian. Is that expected? How well do these indices agree with the reconstruction?

Figure 4: In many of these plots, it looks as though there is a consistent Stronger Walker state observed two years prior to the eruptions in all periods. This is wild! It is quite large in subsets B, C, and D. It might be worth commenting on this unexpected result.

Figure 5A/B/C: How large are these SLP anomalies relative to a traditional El Nino/ La Nina Event?

Figure 5D/E/F: Have you subtracted the tropical mean from these Nino 3.4 SST estimations? Most of the studies that look at the ENSO response do remove the global tropical mean from SSTs (something very difficult to do with a single proxy record)

Figure 5G/H/I: Love that you included this! So cool!

Line 498: It would be great if you included a table with each proxy record and cite the individual authors with the database. Also, what resolution are these proxies?

Line 509: "Model simulations" instead of "Model data"

Line 529—great! I don't think that I got this from the main text. Do you include a table of the skill?

Line 535: Do you use a January-December year or a tropical (May-April) year when converting records to annual resolution? It's very important to be clear on what was done.

Line 645: If referencing the study by name, please include the author's name, not just the superscript number.

Line 656: Matching the SLP variance to that observed over the twentieth century could be a big problem. I discuss this in the major points section.

Line 741: the years 1992-2011 are superscripted

Line 743: just mention the authors and then superscript if directly mentioning a study.

Line 826: Suggest adding the of the method here-- eg. PAGES2k GMST (#66)

G. References

The authors did a good job of including relevant, recent references. However, there were a few papers that merit mention in this discussion (mentioned previously in this review).

Seager, Richard, Mark Cane, Naomi Henderson, Dong-Eun Lee, Ryan Abernathy, and Honghai Zhang. "Strengthening tropical Pacific zonal sea surface temperature gradient consistent with rising greenhouse gases." *Nature Climate Change* 9, no. 7 (2019): 517-522.

Power, Scott, Matthieu Lengaigne, Antonietta Capotondi, Myriam Khodri, Jérôme Vialard, Beyrem Jebri, Eric Guilyardi et al. "Decadal climate variability in the tropical Pacific: Characteristics, causes, predictability, and prospects." *Science* 374, no. 6563 (2021): eaay9165.

Sanchez, Sara C., Gregory J. Hakim, and Casey P. Saenger. "Climate model teleconnection patterns govern the Niño-3.4 response to early nineteenth-century volcanism in coral-based data assimilation reconstructions." *Journal of Climate* 34, no. 5 (2021): 1863-1880.

Grothe, Pamela R., Kim M. Cobb, Giovanni Liguori, Emanuele Di Lorenzo, Antonietta Capotondi, Yanbin Lu, Hai Cheng et al. "Enhanced El Niño–Southern oscillation variability in recent decades." *Geophysical Research Letters* 47, no. 7 (2020): e2019GL083906.

Cobb, Kim M., Niko Westphal, Hussein R. Sayani, Jordan T. Watson, Emanuele Di Lorenzo, H. Cheng, R. L. Edwards, and Christopher D. Charles. "Highly variable El Niño–southern oscillation throughout the Holocene." *Science* 339, no. 6115 (2013): 67-70.

Khodri, Myriam, Takeshi Izumo, Jérôme Vialard, Serge Janicot, Christophe Cassou, Matthieu Lengaigne, Juliette Mignot et al. "Tropical explosive volcanic eruptions can trigger El Niño by cooling tropical Africa." *Nature communications* 8, no. 1 (2017): 1-13.

H. Clarity and Context

There are a few places where the authors could improve the clarity of their manuscript to help the readers understand the novelty of the work and significance of their results. I have generally commented on this in the Suggested Improvements Section.

However, there are a few points that would add a lot of clarity to the manuscript in general :

Firstly, A more in depth description of the proxies that make up the reconstruction. Table S1 cites the Iso2k database, but does not directly reference the individual authors who made many of these proxy datasets. This is a small point, but many paleo people that aren't familiar with the ISO2k database would get much more out of this table if you directly wrote "Bagnato et al. 2005 (14)" instead of #14 and the ISO2k ID number. I'm also a little confused about the "unknown" resolution or seasonality. Does this mean variable resolution?

Secondly, I think that it would help the readers if the authors included their specific definition of

Walker Circulation Variability in the main text (presently included in the methods). Suggestion that the authors add two boxes to Figure 1 to highlight the regions where SLP anomalies are considered. It'd really help the clarity if this was directly addressed this early on (I think early on in the manuscript the authors clarify that their index isn't related to hydroclimate or SST, but don't say exactly what it is). I also think that adding a sentence to describe just how these proxies mechanistically relate to Walker Circulation Variability would be extremely helpful for readers unfamiliar with the Falster et al. Journal of Climate paper.

Referee #2 (Remarks to the Author):

Based on the relationship of d18O in precipitation and Δ SLP, the authors used proxy records for d18O of precipitation and reconstructed the Δ SLP to represent the intensity of PWC. To reconstruct Δ SLP, they use five statistical methods to train the proxy records with three instrumental SLP datasets. The effort aims to minimise the uncertainties arising from the reconstruction method or dataset used for calibration. Furthermore, by comparing the pre-industrial (before 1850) and industrial era on the reconstructed Δ SLP, and examining the features of the composite of Δ SLP after the volcanic eruptions, the authors attempted to address the effects of anthropogenic and volcanic forcing on PWC.

The reconstructed PWC does not exhibit a significant trend during industrial-era as shown in GMAT, but shows a shift to lower frequency variability, which may be due to the anthropogenic forcing. The authors did not provide any concrete physical explanation for this shift. The composites from both reconstruction and model simulation show a weakening of PWC after the volcanic eruption, consistent with many previous studies.

The reconstruction method applied in this work is a helpful tool to minimise uncertainties. CESM last millennium simulations are used to support the effect of the volcanic eruption. There is no reference and more information about the model ensembles and the motivation for using the ensembles from one model instead of the multi-model ensembles from PMIP3/PMIP4 last millennium simulations. Some last millennium simulations have provided the continuous simulation for the industrial-era, which is helpful to investigate the anthropogenic forcing on PWC using the model simulation, for example, MPI-ESM last millennium simulations (<https://mpimet.mpg.de/en/science/projects/archive/millennium>). The continuous simulation into 21st century would overcome the initial condition issue that might affect the historical simulations.

Scientifically I don't find the results from this work are breakthrough novel and add new insight into the changes and variability of PWC during the last millennium. I would find it difficult to recommend publication in Nature. On the other hand, the reconstructed time series of Δ SLP is useful and could publish in a paleoclimate journal.

Minor comments.

In Fig5, I don't understand why it is necessary to apply the chronological uncertainty to model data. For the model experiment, we know that once the external forcing is given, the model uncertainty usually arises from the internal variability.

The threshold for SAOD is 0.05 for reconstruction and 0.07 for model simulations. I understand that 0.07 for CESM-LME follows Dee et al. (2020) to be comparable. So why not use 0.07 for the reconstruction instead of 0.05? Would that yield different results?

Referee #3 (Remarks to the Author):

Review of Falster and co-authors "Forced changes in the Pacific Walker Circulation over the past millennium". In this study the authors attempt to reconstruct the Pacific Walker Circulation (PWC) over the past millennium relying on various statistical methods applied to annually-resolved proxy records. They then explore the potential influence of external forcings (natural and anthropogenic) on the PWC inter annual to decadal variability and trends over the last 800 years.

Distinguishing between forced and unforced variability is critical in the context of the impact of the ongoing anthropogenic forcing on climate. A large body of recent papers have explored such an issue for the Pacific sector including the period prior to 1850. So far, most studies have attempted to reconstruct sea surface temperature (SST) inter annual to decadal variability over the Common Era, to address the relative influence of radiative forcings respectively to internal variability processes in shaping the Pacific Ocean past dynamical changes. From this point of view, the present work is very novel, as the authors attempt to reconstruct the sea level pressure (SLP) zonal asymmetry driven by the Walker Circulation variability, allowing to capture the coupled dynamics (as opposed to SST which are often blurred by the direct radiative forcing).

Based on a large ensemble of PWC reconstructions, the authors explore the potential influence of the largest natural forcing of the last millennium, namely the volcanic forcing, and try to put in a longer term perspective the variability and trends observed over the historical period. They conclude that the recent PWC trend is not anomalous in the context of the last 800 years. They also conclude for a significant PWC weakening in the 3 years following the largest eruptions of the reconstructed period, suggesting an El-Niño like response which appears roughly consistent with previous studies based on SST reconstructions or climate models.

The paper writing needs to be significantly improved as it refers constantly to SI or extended material that are central to the study, so that the reader has to stop and go back and fourth through various documents at almost every paragraph to follow the results and reasoning presented in the main manuscript. The scientific question mentioned in the abstract and in the introduction is also not specific enough. The reader has often to guess when the authors discuss the "recent strengthening" or "weaken" of the PWC in response to anthropogenic forcing without mentioning clearly the related timescales and related specific scientific question. For example, at the decadal timescale it is likely that the variability is largely aliased by the Interdecadal Pacific Oscillation (internal to the climate system, an issue that is not discussed or quantified as such by the authors), while the secular trends are more likely to be dominated by external radiative forcing, at least over the industrial period. The confusion between the underlying processes of decadal variability and secular trends, is present in the abstract and persists throughout the manuscript so that the reader does not have a clear understanding about the scientific challenges that are addressed in the study. There are also many caveats that have to be clarified in the statistical approaches and experimental protocol. As it stands, these caveats leave questions and large uncertainties in the obtained reconstructions limiting any robust interpretation and conclusions.

I however acknowledge that a lot of work has been developed in an attempt to use appropriate various statistical methods to analyse a large set of proxy records. Yet the major issues listed bellow have to be addressed to assess what is robust and what is not. Provided that the authors can make significant improvements (in the writing presenting clearly the scientific question and hypothesis, the methodology to obtain robust reconstructions and analyses), such work could potentially allow for significant progresses in our understanding of tropical Pacific natural variability.

Major comments:

a. The authors selected a set of 54 globally distributed "water isotopes" proxy records and five non isotope records (listed in Table S1) based on their "high" correlation with the PWC or ENSO but no specific analyses are shown sustaining such statement. What is "high"? Is there a specific threshold or statistical analyses made to make such proxy records selection? The authors mention

in the SI that the “primary” age-model timeseries is retained during the selection process based on the record authors claims. It might be a proper way to go, but how consistent are the statistical analyses criterias among the various original publications? It is important to clarify this especially that the rest of the study explores the sensitivity to the age model, in an ensemble approach to reconstruct the PWC. I suggest that the selection criteria (level of significance based on various metrics) to be defined first by the authors and then used on the ensemble of possible age models to retain only the proxy timeseries (across the possible age models) that have the best scores. This will allow for a consist treatment of available records from various source and authors and a more objective way to perform the selection.

b. To test the robustness of the results depending on the records availability, the authors performed reconstructions on five temporal sections, i.e. 1200-2000, 1400-2000, 1600-2000 and 1800-2000, using in each case proxy records with more than 66% of temporal coverage. Records are then interpolated with the DINEOFs method to fill in the missing data. I understand that for each reconstruction 15% of the available proxy are removed to account for possible dependence of a particular subset of proxys. This approach raises 2 issues that might alter the derived reconstructions. Is the same set of records used for each period? If not, then changing the set of records from one temporal section to an other might still hide proxy-dependant sensitivity and introduce artificial discontinuities in the variances and variability content, raising the issue of the proxy records resampling. A way to test this, would be to perform reconstructions over the 5 periods (keeping the 15% random sampling) using the same set of proxy records available throughout the longest period, i.e. 1200-2000. I understand that the number of available records over the whole period will be lower but at least this will allow to have a reference with the same dataset to test the robustness in any shift of frequency and trends over the last 800 years. The second issue concerns the DINEOFs interpolation method. I suspect that it has a larger influence when we go further back in time since there are more and more gaps to fill. Can the authors provide a figure showing the number of records with gap filling and their length throughout the 800 years to assess the influence of such an approach in the uncertainties of the derived reconstructions?

c. As a follow up on the previous comment, Table S2 and Extended Figure 1 rely on correlation, RMSE and binomial sign test computed over the 1900-2000 period, but no scores are shown regarding the score related to the amplitude of the variance. The scores are computed comparing the 4800-member ensemble reconstructions median (n=4800) against the mean of 3 observational datasets (n=3). It would be good when assessing the variance amplitude to randomly select 3-member ensemble reconstructions to allow for a proper comparison to observations. The question of the variance has been overlooked in the assessment of the reconstructions while they are used to discuss frequency content and trends compared to the recent historical period.

d. The authors state on lines 104 to 111 « A weak positive Δ SLP trend from \sim 1200-1750 CE is followed by a slight decrease to \sim 1800 CE, then a period of low inter-method agreement. ... Approximately 10% of ensemble members also have significant power in \sim decadal (10-12 year) and multidecadal (21-24 year) bands. Interestingly, there is a shift to higher power at lower frequencies in the industrial period (since 1850 CE) relative to the pre-industrial last millennium (4-9 rather than 2-9 year periods, with particularly high power in the 9-year band)»

Why only 10% of ensemble member have a significant power in the decadal and multidecadal band ? This number is low, how robust is it related to the other 90% of the members without such variability. Is it linked to the method, the age models errors, the proxies used,... ? Since 15% of the proxies are ignored it is perhaps related to the types of proxies considered? I feel like most of the discussion on the frequency changes is not robust or depends strongly on the method/proxy dataset. These questions prevent drawing any firm conclusions based on only 10% of the reconstructions. In addition, an apparent discontinuity also exists in the timeseries at year 1800 respectively to the previous period and at 1900 respectively to the following period. This is even

more important as the 1800-1900 period bears the largest inter-method disagreement, which is counterintuitive as there are many available proxy records over this time window so the method should not have such a strong weight on the uncertainties. The resulting large disagreements prevent putting the recent instrumental period into perspective regarding the secular trends or decadal variability. Addressing the comments listed previously (records selections, changing records sets from one temporal window to another, the gap filling, scores on variance) might help reduce these uncertainties and pin down what is robust and what is not.

e. Figure 2A and extended Figure 3: In line with my previous comment, cumulating individual reconstructions for which a very small proportion of members (0.1 each?) with a significant power at a given period is misleading. It gives an artificial perception of robustness. Are the reconstructions with significant multi-decadal variability the same as those with significant decadal variability? Providing the form of the frequency content for at least reconstructions with a significant signal or by type of method would be helpful. Is it the same set of proxies for the industrial period (since 1850 CE) relative to the pre-industrial periods? Could it be due to the influence of proxies resampling, the more important role of DINEOFs interpolation method when going back in time? Would it be possible to add instrumental observations on these figures to see how the well reconstructions performs over the observational period?

f. Lines 112-115: « The distribution of Δ SLP values in the industrial period is skewed toward higher (more La Niña-like) values than in the pre-industrial last millennium (1200-1849 CE; Fig. 2b). However, the difference between the two distributions is within the range of that calculated between any other two intervals of equivalent length across the reconstruction (Methods). »

This means that the differences before and after 1850 is not significant but the writing is not clear enough and suggests a tendency towards more La Niña-like over the historical period. Please rephrase to avoid any misleading messages.

g. Lines 117-124: « The lack of a significant PWC mean state shift in response to anthropogenic forcing is an important result. Climate models suggest the thermodynamic effect of GHG-driven rising global mean surface temperature (GMST) should ultimately weaken the PWC 15,34 and a negative Δ SLP trend is present in historical simulations from most CMIP5/6 models 3. However, recent work suggests that the global warming-driven slowdown of the Atlantic Meridional Overturning Circulation accelerates the Pacific trade winds, resulting in a stronger PWC 35. Our findings demonstrate that neither GHG-driven effect is emergent. Nevertheless, the post-industrial shift in PWC variability toward lower frequencies is intriguing, and possibly a response to anthropogenic forcing that has not previously been identified. ».

Considering that the uncertainties in the reconstructed decadal variability and secular trends are much larger than the signal the authors are looking for, I don't think such a statement can be made. The results shown so far simply illustrate that the errors and uncertainties in the reconstructions do not allow putting the recent changes in perspective.

h. Figure 3 (B,C,D) : The trend in the observation can potentially be more marked than in the reconstructions only because of the bias in the method which underestimates the real variability. It would be nice to indicate the median and maybe max, min (or 97.5 and 2.5 percentile) trend of the reconstructions over the period 1992-2011. And given that the historical period is the period with largest available proxies, do we not have the strongest trends marked at 20-year resolution because we lack proxies in the past ? Are these results robust if we use the same set of proxy over the full period (see previous comments a and b above)?

i. Figure 4: Again cumulating individual very small number of reconstruction fraction per method is misleading. Would it be possible to add a y-axis to the right of panels A, B, C and D with perhaps the median anomaly and the 97.5 and 2.5 percentile for each method?

j. Figure 1A and line 564-566: Visually the inter-method uncertainty over the period 1800-1900 seems especially strong and marked in 1800-1810. And then line 564-566: If I understand correctly the oldest period is drawn towards the most recent segment to make a continuous timeseries from sliced reconstructions? Isn't this an important source of error, especially if the connection is made in 1800 in view of the important inter-method uncertainties at this period in figure 1A?

k. Line 562-563: It would be useful to have a curve showing the number of proxies available over time with the proportion of real vs. interpolated proxies. The importance of the interpolation method in the reconstruction obtained certainly varies over time and it would be troublesome to have a proportion of interpolated proxies too high in some years (e.g. 8 out of 10 proxies used in the year 1350 are interpolated) and where the reconstruction would be largely related to the interpolation method. Making a figure showing that if at some point 90% of the available proxies are interpolated will allow when there might be a problem or a very large uncertainty?

l. Figure 5: There is an error in the caption g-i instead of g-h. I am not sure about the relevance of the 5g-i panels. I understand the idea of perturbing the age model by + or - 1% to compare with the reconstructions. But isn't this perturbation for model outputs more "violent" than what we have in the proxies? There is a notion of detecting the major signal from the reconstructions which use a set of proxies, whereas if I understood correctly the error is here directly applied on the model outputs? Is it appropriate ?

Referee #4 (Remarks to the Author):

Dear authors of the manuscript "Forced changes in the Pacific Walker Circulation over the past millennium", the ENSO response to volcanic eruption is a hot topic, but remains unresolved due to uncertainties in proxy data reconstructions and climate model simulations. This manuscript is novel and fruitful because the atmospheric Pacific Walker Circulation (PWC) is more sensitive to external forcing than the East Pacific Sea Surface Temperature (ENSO). Isotope records are more sensitive to climate than other records. Methods including CPS, PCR and Pairwise comparison are often used in the proxy data reconstruction, and these methods are mature and effective. Most of the analysis and results are reasonable and fruitful. However, data screening does not meet all methods. The comparison between the PWC reconstruction and the previous ENSO reconstructions is missing, and the physical process of the PWC to the external forcings is still unclear. Therefore, I recommend revising it before accepting it.

Main Comments:

1. Regarding the reconstruction method, there are four concerns. First, the criteria for screening proxy records should apply to all methods. e.g., there are four cases, a) all proxy records, b) the subsets of proxy records significantly correlated with the target index in the calibration window, c) proxy records in a tropical Pacific 'centre of action', and d) proxy records for specific seasons only, with no known biases. Second, the first principal component of the proxy data is only retained in two variants of PCA, leading to a system bias in partly missing the low-frequency signals (McShane and Wyner, 2011). I suggest that that some first PCs should be assessed, which is similar with the PCR method. Third, there is no age uncertainty in woody records. Iterations of age-depth models for wood records would introduce new systematic biases. Fourth, a reason for the sensitivity of these isotopic records to atmospheric circulation is suggested. e.g., how do the ice-core isotope records in the two poles respond to the PWC?
2. The Pacific Walker circulation constitutes the atmospheric component of the ENSO, so it is suggested that the PWC reconstruction be compared with previous ENSO reconstructions to emphasize the novelty of this work. e.g., the isotope records used in PWC reconstructions reduce inconsistencies in other types of proxy records.
3. The climate dynamics of the PWC to the external forcings are missing. It is recommended to analyze signal forcing experiments in CESM LME simulations to show physical processes. Moreover,

what kind of the internal variability is responsible to the PWC? e.g., the Interdecadal Pacific Oscillation? How to know it? Finally, how to use a new PWC reconstruction with pre-1600 CE too low skill and an old climate simulation to answer the question "Yet the PWC response to external forcings is unclear, with empirical data and model simulations often disagreeing on the magnitude and sign of these responses".

Specific Comments:

1. Pages 1-2, lines 25-26. What kind of the internal variability? Interdecadal Pacific Oscillation?
2. Page 2, lines 43-48. How to deal with the same difficulty of the ENSO's response to volcanic eruptions in this work?
3. Page 3, line 53. It is important to show the number of the proxy records and not the number of members. The reason is that the number of the members is arbitrary.
4. Page 3, line 55. The reference format does not meet the requirements of the journal. e.g., 15 reference.
5. Page 3, line 56. The anomalous period is usually to 1961-1990 CE.
6. Page 5, lines 123-124. The difference in low-frequency signals between the post-industrial and pre-industrial eras may be due to the different lengths of the analyzed datasets. Moreover, the current PCA methods partly lost low-frequency signals, and the proxy records only with annual resolution also impact the low-frequency signals. Therefore, it is recommended to use more records with low resolution and use a modified PCA method with more PCs to assess the frequency difference.
7. Page 6, lines 134-136. The three datasets show different results. A new reanalysis dataset could be included to confirm this result. e.g., ECMWF reanalysis v5.
8. Page 6, lines 144-145. The three datasets show different results in the Extended Data Figure 4. How to get the consistent result?
9. Page 8, lines 197-200. How to understand the strong SLP response but the SST response? The possible reason is suggested.
10. Page 8, lines 204-205. How to know the PWC reconstruction has large internal variability? Which internal variability? Pacific Decadal Oscillation?
11. Page 9, lines 218-222. The Extended Data Fig. 9 shows the significant relationship between the GMST and the Δ SLP reconstruction. A sensitive experiment using the climate model is suggested to test this thermodynamical process.
12. Page 10, line 245. "one year in the CESM LME".
13. Page 10, line 264. "the first climate mode" should be replaced with "the first atmospheric mode", because the previous ENSO reconstruction is also a climate mode.
14. Pages 11-14. The reference format should be improved according to requirements of the journal.
15. Page 15. It is interesting to show the results of the 2001-2021 CE period.
16. Page 16. Extended Data Figure 3 should be included in main Figure 2, because the frequency difference between pre-1850 and post-1850 is a novel result.
17. Page 18, line 438. "See Methods".
18. Page 19. The size of volcanic eruptions in the climate model simulation should be consistent with the proxy reconstruction.
19. Page 20, lines 464-466. It is suggested to briefly explain how to screen proxy records. How these records were selected from 759 isotope records (Konecky et al., 2020)? It is well known that there are many proxy records associated with ENSO variability. How to screen them?
20. Page 21, lines 495-497. The ERA20C could be extended with ERA5 during the 2011-2021 CE period.
21. Page 21, lines 506-508. This condition is too strict for the non-dendro records because the glacier ice, lake sediment, and speleothem cannot satisfy annual resolution.
22. Page 22, lines 515-516. The signal-forcing (volcanic and anthropogenic forcing) experiments are suggested to confirm the results.
23. Page 24, line 571. "MATLAB".
24. Page 24, lines 574-583. The wood records have no dating uncertainty.
25. Pages 24-25, lines 592-612. Keeping only the first PC leads to systematic bias because PC1

mainly captures high-frequency while losing low-frequency signals (McShane and Wyner, 2011).
26. Page 26, lines 631-633. This screening records are suggested to use in the other methods.
27. Page 26, line 650. "MATLAB".
28. Page 27, line 672. The reduction of error and the coefficient of efficiency (Cook et al., 2010) are suggested in the reconstruction validation.
29. Page 28, lines 701-704. The correlation coefficients before 1600 CE in the Extended Data Figure 1b are too small to show the robustness of the Δ SLP signal.
30. Page 32, line 801. The signal forcing (volcanic eruption) experiment in the CESM LME is suggested to assess the PWC response to volcanic eruptions.
31. Page 34, lines 839. All codes and datasets should be archived in the publish database to help the reader reproduce results.

References

Cook, E.R., Anchukaitis, K.J., Buckley, B.M., D'Arrigo, R., Jacoby, G.C., Wright, W.E., 2010. Asian monsoon failure and megadrought during the last millennium. *Science* 328, 486-489.
Konecky, B.L., McKay, N.P., Churakova, O.V., Comas-Bru, L., Dassié, E.P., DeLong, K.L., Falster, G.M., Fischer, M.J., Jones, M.D., Jonkers, L., Kaufman, D.S., Leduc, G., Managave, S.R., Martrat, B., Opel, T., Orsi, A.J., Partin, J.W., Sayani, H.R., Thomas, E.K., Thompson, D.M., Tyler, J.J., Abram, N.J., Atwood, A.R., Cartapanis, O., Conroy, J.L., Curran, M.A., Dee, S.G., Deininger, M., Divine, D.V., Kern, Z., Porter, T.J., Stevenson, S.L., von Gunten, L., Iso2k Project, M., 2020. The Iso2k database: a global compilation of paleo- $\delta^{18}\text{O}$ and $\delta^2\text{H}$ records to aid understanding of Common Era climate. *Earth Syst. Sci. Data* 12, 2261-2288.
McShane, B.B., Wyner, A.J., 2011. Rejoinder: A statistical analysis of multiple temperature proxies: Are reconstructions of surface temperatures over the last 1000 years reliable? *Ann. Appl. Stat.* 5, 99-123.

Referee #5 (Remarks to the Author):

Falster et al. use a synthesis of 54 globally distributed water isotope (e.g., $\delta^{18}\text{O}$) proxy records from corals, trees, speleothems, lake sediment, and glacial ice to reconstruct the strength of the Pacific Walker Circulation (PWC) from 1200-2000 CE. This study builds upon previous work in Falster et al. (2021) that found a strong imprint of the PWC in interannual precipitation $\delta^{18}\text{O}$. In this study, the target climate variable for the reconstruction is the trans-Pacific sea level pressure gradient (Δ SLP).

This study investigates how the PWC varied during the Common Era and its response to both volcanic and anthropogenic forcings (e.g., greenhouse gases and anthropogenic aerosols). The authors find that although the more recent PWC strengthening is anomalous (i.e., the recent trend falls within the positive tails of the full distributions), it is not outside the range of natural variability observed in the 800-year-long reconstruction. The recent trend could thus be attributed to internal variability. The high intrinsic variability in the PWC reconstruction is a notable result that highlights the importance of having a longer-term context when discussing recent trends in atmospheric circulation. Additionally, this work uses superposed epoch analysis, to identify a link between the strength of the PWC and volcanic forcing. This study finds that strong volcanic eruptions trigger an El Niño-like PWC weakening in the subsequent years, a relationship more clearly identified since the 1860s.

Overall, the ensemble-based approach to reconstructing Δ SLP is comprehensive and statistically sound. The authors' decisions to use 5 different statistical reconstruction techniques, 3 observational products as a training set, and a Monte Carlo-style approach for adding chronological uncertainty in the proxy records, addresses many of the most common uncertainties and challenges/issues in these types of paleoclimate studies. The study is well-written, the figures are well-presented, and I feel the results will be of broad interest to the climate community. Below, I

provide some recommendations to further clarify the results and strengthen the paper, as well as some questions/comments to address in a future revision. These comments largely focus on the influence of seasonality in the PWC reconstruction, the relationship between the PWC and volcanic forcing, and the role of internal PWC variability.

Main Comments and Questions:

1. In this study, superposed epoch analysis is used to identify the relationship between PWC variability and volcanic forcing. The authors find that strong volcanic eruptions trigger an El Niño-like PWC weakening in subsequent years. This is a surprising result given how contentious the response of Pacific SST variability to volcanic forcing is in the literature.

A similar PWC weakening after large volcanic eruptions is identified in the fully forced ensemble members from the CESM Last Millennium Ensemble. That said, the simulated relationship between the strength of the PWC and volcanic forcing does not hold when 1% 'chronological uncertainty' is added to the climate model simulations (Fig. 5 G, H, I). As a reviewer, I found this to be a very intriguing result that was not thoroughly discussed in the text. L193-194 states that 1% 'chronological uncertainty' obscures the signal, but additional details explaining the importance of this result are not provided. A more thorough discussion of this result will improve the manuscript especially since a significant PWC-volcanic forcing relationship is identified in the proxy-inferred reconstructions in Fig. 1 that have accounted for chronological uncertainty. I am curious to hear the authors' comments about why the PWC-volcanic forcing relationship may persist when adding chronological uncertainty in the proxy-inferred reconstructions, but not in the climate model simulations.

2. The colors in Fig. 1B shows the median correlation of each proxy records with the Δ SLP ensemble over the period in which that record contributed to the reconstruction. To clarify this calculation, was each individual proxy record correlated with the ensemble median for each reconstruction method (i.e., the 8 colored lines in Fig. 1A) over their common interval of overlap? Then the median of those 8 correlation coefficients was plotted in Fig. 1B? Some additional clarification in the main text, methods and/or the figure caption would be a helpful addition.

I am curious why the strength of correlation with the reconstruction is shown in Fig. 1B as opposed to the correlation with the observational Δ SLP training index as in Extended Data Figure 10. As an alternative Fig. 1B, I recommend showing a condensed version of Extended Data Figure 10 (e.g., a single map including all the proxy records) in the main text so the reader can clearly identify how well-correlated the individual records are with instrumental Δ SLP. Although many of the proxy records are significantly correlated with instrumental Δ SLP over the calibration interval, the absolute magnitude of some of the correlation coefficients are quite low and worth noting.

3. I recommend including an additional supplementary figure that shows the time series of the three observational SLP products (HadSLP, ICOADS, and ERA20C) over their common 1900-present interval of overlap. The manuscript discusses that differences between the observational products impact the reconstruction skill, as mentioned in L71 and L95 and shown in Extended Data Fig. 1Ai. The observed time series are shown by the three black lines in Fig. 1A, but it is difficult to discern any differences. I suggest including a zoomed in version of the observed Δ SLP time series that clearly differentiates the three products.

As a related point, gridded observational products typically become more uncertain further back in time due to a reduced number of observations. This is well documented for SST observations (e.g., Fig. 3 of Deser et al. 2010). Are there any measures of uncertainty provided in the HadSLP and ICOADS products? Do the different gridded products have better-agreement during the more recent decades, compared to the early 20th century due to a denser network of observations?

Reference: Deser, C., Alexander, M. A., Xie, S.-P. & Phillips, A. S. Sea surface temperature

variability: Patterns and mechanisms. *Annu. Rev. Marine. Sci.* 2, 115–143 (2010).

4. Why is the composite plus scale method using only records without a known seasonal bias (CPSns, pink curve in Fig. 1A) typically the most different compared to the other 7 reconstructions? It appears that the amplitude of the Δ SLP variations is often larger compared to the other methods. For example, there are notable differences during the mid 1500s, the late 1600s, and the 1700-1800s, in some cases the median CPSns reconstruction shows Δ SLP changes of opposite sign to the other median reconstructions.

Do these results suggest that seasonality is indeed an important factor to consider when reconstructing the PWC? Or are the differences in the CPSnc reconstruction mostly attributed to the reduced number of proxy records available, and is thus more uncertain? Additional discussion about the impact of seasonality would strengthen the manuscript.

5. This study uses the CESM Last Millennium Ensemble (CESM LME) full forcing simulations to investigate the simulated response of the PWC to volcanic forcing. I am curious how the CESM LME control simulations (e.g., the 1850 or 850 control) could be used to provide context about the range of internal variability in the PWC. This would likely strengthen the study's conclusions about internal PWC variability.

For example, how does the variance of Δ SLP in a control simulation compare to the variance of the fully forced simulations? The trend analysis for the PWC reconstruction and the PDFs in Figure 3 are interesting and could presumably be repeated for the CESM LME. In the simulations, are there any significant 20-year trends, and/or changes in decadal/multi-decadal variability in Δ SLP that can arise completely in the absence of external forcing?

6. L543-545 states that each reconstruction only retained proxy records significantly correlated with Δ SLP over the calibration window. There are then, three proxy data subsets, one for each observational product. Are there large differences between the three subsets of proxy records? For example, is one of the observational data products consistently better correlated with a larger number of records? Some additional justification for this approach would strengthen the manuscript.

Minor Comments:

- The figure caption of Extended Data Figure 10 states that the black outline denotes that the proxy record is significantly correlated with instrumental Δ SLP (Extended data, L110-111). In my version of the document, all the proxy records are outlined in black, but some have a darker black outline.

- Extended Data Figure 10 shows the spatial distribution of proxy records for different time intervals, but a time series would also provide helpful context. I imagine a similar figure to Figure 2b in the Konecky et al. (2020) Iso2k database paper, but only including the records selected for this study.

- L93: Does this refer to a specific supplementary table? What are the traditional (non-ensemble) reconstructions referred to here? I recommend providing the references.

- Fig 2b caption: Indicate that a KS test was used to demonstrate that the pre- and post- industrial Δ SLP distributions are statistically significantly different. From visual inspection alone, it doesn't look like the two PDFs would be statistically different.

- How do the 1200-2000 CE reconstructions for each reconstruction technique compare to the full 'nested' reconstructions stitched together with the different temporal segments?

Author Rebuttals to Initial Comments:

Response:

We have revised the manuscript according to suggestions from the five referees, with additional layout changes as suggested above. All the referees' suggestions—including additional tests of statistical robustness—strengthened our original findings.

To summarise the major changes that we have made to the manuscript, as suggested by the referees:

- Incorporation of the original Extended Data Fig. 3 into main text Fig. 2 (spectral analyses performed separate on the pre- and post-industrial)
- Addition of data from PMIP3/4 models to Fig. 5 (Superposed Epoch Analysis performed on model data; originally just CESM, new figure has 8 additional models)
- Promotion of the original Extended Data Fig. 9 to main text Fig. 6, given particular interest in those point from several referees (comparison of Δ SLP with GMST)
- Addition of new Extended Data Figures, showing:
 - spectral analysis performed on non-nested reconstructions
 - agreement between instrumental and reconstructed Δ SLP in the calibration period
 - proxy data availability through time (including intervals of missing data)
 - relationships with published ENSO reconstructions
- Addition of 'reduction of error' skill test
- Provision of code in a Zenodo repository, allowing readers to replicate findings (repository is set up, and has a DOI, but is not yet public {REDACTED})
- Promotion of the original Supplementary Tables 2 & 3 into the Extended Data

These revisions, along with additional explanations and clarifications in the text, address referee concerns regarding methodological details, and statistical robustness. The main text is now ~3600 words, and the Methods are now ~4600 words.

We also made many minor changes, including the requested edits to the text for clarity (particularly in the Methods), additional explanations throughout, and minor figure edits - these address the concerns of Referee #3 about clarity of presentation.

Regarding use of climate model simulations to provide additional mechanistic insight into the phenomena identified here: this extends well beyond the scope of this empirically-focussed manuscript. Although there were several excellent suggestions from referees as to follow-up analyses (many of which were already being undertaken by the authorship team or their students), these would fit better in one or more model-focussed manuscripts. We have clarified the logic behind our use of model simulations in the

assessing the Δ SLP response to volcanic eruptions, and added the following statement in the concluding paragraph of the manuscript: “*Although diagnosis of the dynamics underlying forced responses and intrinsic variability in the PWC was beyond the scope of this paper, the Δ SLP reconstructions provide the necessary observational foundation for such future investigations*”.

Referees' comments:

Referee #1 (Remarks to the Author):

A. Summary of the key results

Falster et al.'s "Forced changes in the Pacific Walker Circulation over the past millennium" describes a highly original reconstruction and assessment of Walker Circulation variability using a network of long proxy records in the Iso2k network.

The chief conclusions of this analysis are: 1) the recent 1992-2011 trends in the Walker Circulation are not unprecedented over the last 800 years, 2) Volcanic eruptions prompt an observable Weaker Walker response, 3). The relationship between global mean surface temperature and Walker Circulation is not as straightforward as many studies have suggested. The two important indices can vary considerably.

This manuscript will be of interest to readers of Nature who are interested in climate dynamics, paleoclimatology, and historical climate change. I recommend this manuscript for publication with minor revisions.

B. Originality and significance

This analysis is very novel and illustrates the utility of a new water isotope dataset (Konecky et al. 2020) to examine a long standing question in the scientific literature. This analysis does an admirable job of examining the uncertainty in their reconstructions. The authors examine the sensitivity of their method to dating errors, calibration dataset, temporal period of reconstruction and reconstruction methods—and clearly illustrate the sensitivity of their results to these choices. It is difficult to envision how the authors could have done more to test the robustness of their reconstruction to arbitrary methodological choices.

Response:

Thanks - we feel that this is one of the key strengths & novelties of the paper :).

This work would be very relevant to several fields. This work also serves to highlight the utility of water isotope data.

One suggestion for the authors is to amplify the fact that this analysis uses such a novel water isotope dataset. This is *really* exciting (and novel!) and I am afraid that this important nuance might be lost on readers who have never heard of the Iso2k project.

Response:

We have added text throughout the manuscript to emphasise the novelty of the water isotope dataset (and its use in paleoclimate reconstruction). In particular, at Line 71 we now state *“The reconstruction leverages the recently created Iso2k database, a highly innovative global synthesis of water isotope proxy records. Iso2k includes data from diverse archive types, and allows ready integration of water isotopic signals into palaeoclimate reconstructions.”*

C. Data and Methodology

The data used and methodology used in this manuscript are both very reasonable and very original. The new Iso2k dataset (Konecky et al. 2020) used heightens the originality and significance of this study.

D. Appropriate use of statistics

I did not attempt to repeat any statistical tests, but all tests and results used in the manuscript appear logical. All figures illustrate an admirable use of error bars and significance testing. Furthermore, the authors take supreme care to test the sensitivity of their results to reconstruction methods. As an example of this, a 1992-2011 was a key analysis period in their study (due to previously published work in the scientific literature), but this period begins just after Pinatubo. The authors examined the sensitivity of this trend to the volcanic eruption by examining the 20 year trends after all volcanic eruptions and concluding no strong signal.

My largest concern with this study is that the 1992-2011 period is of chief importance to the analysis, but is a period not covered by the authors reconstruction (800-2000AD). The authors repeatedly compare the observed instrumental trend over the 1992-2011 period to the proxy reconstructed trends and draw major conclusions from this comparison. I understand that proxy records grow sparse after the year 2000, but it would be helpful to see just how well the reconstruction method compares with the instrumental observations over the 20th century. (Perhaps add a panel to Figure 3)? The authors state that there is high correlation between the two (Supplementary Table 2, $r \sim 0.8$), but it would be helpful to see the agreement between the reconstruction and all instrumental products over this important period. I think readers will question if the authors are indeed comparing apples to apples with the present discussion.

Response:

We have added the suggested figure (Extended Data. Fig. 3), showing correspondence between the reconstruction and observed Δ SLP over the 20th century. This also shows how observed Δ SLP continues past the end of the reconstructions.

My second largest concern with the methodology used in the study regards the variance adjustment applied to the reconstructions to maintain the variance of the 20th century throughout the 800-2000 reconstruction interval. The motivations for this variance adjustment are highly logical—nearly all proxy reconstructions feature a loss in variance with time (as the available proxy fraction dwindles), but my

concern lies in the documented change in ENSO variability over the last millennium and how that may influence the Walker Circulation reconstruction. Both Grothe et al. 2020 and Cobb et al. 2013 demonstrate an increase in Niño 3.4 variability over the last few decades. At the very least, the authors could comment on how the assumed 20th century variance may influence the results (and cite these studies).

Response:

When constructing the reconstruction methodology, we sought to balance true variance changes through time with artificial variance loss back through time due to chronological uncertainty and proxy-specific effects. We performed extensive sensitivity testing on this point. However, changes in variance through time are not strongly affected by the variance correction, as a variance scaling is applied to the *entire* timeseries. That is, any variance changes through time are maintained. We have edited the Methods text to make this clearer (L620-624).

Additionally, following related concerns from several referees about the temporal nesting methodology and how that might affect variance, we repeated two key analyses, using a non-nested version of the reconstruction, i.e., a reconstruction using the same set of proxies throughout. All conclusions remain robust in this context (see below), and we have added one of these analyses as a new Extended Data Figure (ED Fig. 7, L673-693).

Grothe, Pamela R., Kim M. Cobb, Giovanni Liguori, Emanuele Di Lorenzo, Antonietta Capotondi, Yanbin Lu, Hai Cheng et al. "Enhanced El Niño–Southern oscillation variability in recent decades." *Geophysical Research Letters* 47, no. 7 (2020): e2019GL083906.

Cobb, Kim M., Niko Westphal, Hussein R. Sayani, Jordan T. Watson, Emanuele Di Lorenzo, H. Cheng, R. L. Edwards, and Christopher D. Charles. "Highly variable El Niño–southern oscillation throughout the Holocene." *Science* 339, no. 6115 (2013): 67-70.

E. Conclusions

The conclusions of this manuscript are well grounded (aside from the concern about the reconstruction comparisons with the instrumental record over the 1992-2011 period). The authors describe their conclusions in the context of present literature and their results are easily interpretable.

I did not personally try to replicate any of the analysis, but the authors are very clear about the methodology and data used, thus their results seem reproducible. Furthermore, the authors state that they will provide access to their reconstruction upon publication—and offer to provide code upon request.

Response:

We now provide all code necessary to replicate our findings. Code will be available from the following Zenodo repository: doi.org/10.5281/zenodo.7742761. {REDACTED}

F. Suggested Improvements

I have major and minor suggestions for improvement and have listed them below.

Major Points

The first two points are discussed in the “Appropriate Use of Statistics” section, but I will briefly outline them again (apologies for repetition).

First: I worry about the comparison of the proxy reconstruction to the instrumental archives over the 1992-2011 period. I think that adding a figure to show the consistency of the reconstruction results to the HADSLP, ICOADS, and ERA20C reanalysis products over the 20th century would help abate the concern that it is not entirely appropriate to directly compare these two sources of data.

Response:

We have added this figure as a new Extended Data item (ED Fig. 3).

Secondly: Adjusting the variance of the reconstruction to that observed in the 20th century causes some concern. I suggest that the authors comment that they acknowledge that they may be incorporating some bias with this approach.

Response: R40

We have added details and additional commentary on the variance adjustment to the Methods.

Finally: The authors include some discussion about the utility of examining Walker Circulation Variability with a few different variables, but set up a contrast between examining the SST component of Walker Circulation change vs the atmospheric/ SLP component (eg. Line 195, but also in a few other places). I think a sentence or two to clarify that discussion would be helpful to readers. Few studies examining the climate response to volcanic eruptions use raw SST, instead they use a “relative SST” and examine the Nino3.4 SST after subtracting the tropical mean (due to the expectation that volcanic aerosols will cause cooling globally and mask the tropical Pacific response, eg. Khodri et al. 2017, Predybaylo et al. 2020, Zuo et al. 2018, many others). This is virtually impossible when using single paleoclimate proxy record that only monitors conditions at a single location.

Response:

We have re-done the SEA on model SST in the Nino 3.4 region, but now using the relative SST as suggested (results shown in Fig. 5). We have added text describing this approach in the Methods (L764-768). We have also added additional discussion of this point to the main text (L284-285).

Khodri, Myriam, Takeshi Izumo, Jérôme Vialard, Serge Janicot, Christophe Cassou, Matthieu Lengaigne, Juliette Mignot et al. "Tropical explosive volcanic eruptions can trigger El Niño by cooling tropical Africa." *Nature communications* 8, no. 1 (2017): 1-13.

Predybaylo, Evgeniya, Georgiy Stenchikov, Andrew T. Wittenberg, and Sergey Osipov. "El Niño/Southern Oscillation response to low-latitude volcanic eruptions depends on ocean pre-conditions and eruption timing." *Communications Earth & Environment* 1, no. 1 (2020): 1-13.

Zuo, M., Man, W., Zhou, T., and Guo, Z.: Different impacts of northern, tropical, and southern volcanic eruptions on the tropical Pacific SST in the last millennium, *J. Climate*, 31, 6729–6744, <https://doi.org/10.1175/JCLI-D-17-0571.1>, 2018.

Minor Points

Line 22—"the 4800 member ensemble" phrase is just confusing without introducing what the ensemble represents. How about something like, "We use a new paleoproxy-derived 4800 member ensemble"?

Response:

We added 'paleoproxy-derived' to the introductory sentence (L22).

Line 40: A reference should consider adding to this discussion is the Seager et al. 2019 paper that illustrates some of this mismatch between observations and models may be entirely due to biases in climate models.

Response:

We have added this reference (L42).

Seager, Richard, Mark Cane, Naomi Henderson, Dong-Eun Lee, Ryan Abernathy, and Honghai Zhang. "Strengthening tropical Pacific zonal sea surface temperature gradient consistent with rising greenhouse gases." *Nature Climate Change* 9, no. 7 (2019): 517-522.

Line 50: What is your specific definition for Walker Circulation? Suggest that the authors say "hydroclimate (rather than SLP directly)" or something like that for clarity.

Response:

We have edited the text for clarity (L51-52).

Line 53: Again, be clear with what this 4800 member ensemble entails

Response:

We added a brief statement introducing the sources of uncertainty that make up the ensemble (L56-58).

Line 54: As the 1992-2011 period has been mentioned, it would help to clarify why the reconstruction does not extend to year 2100

Response:

Assuming that the referee means 'extend to the year 2011', we added some clarifying text as to this choice in the Methods (L492-493). In short, it is because many of the proxy records stop around 2000, such that the proxy data availability would have been greatly decreased.

Line 55: If specifically citing a study, its far less confusing for the reader if you write XX et al. 13 rather than just the superscripted number

Response:

We have changed all instances of this in the text.

Line 60: All readers would appreciate a little more information about "the dominant mode of global interannual precipitation $\delta 18O$ carries a strong imprint of ΔSLP ". Over what years? What was the proxy distribution? How do the authors define "strong imprint"? Perhaps a sentence or two more would help readers better appreciate the work you have put into this task!

Response:

We have added extra text to provide more information. Specifically, from L64: "*The dominant mode of observed global interannual precipitation $\delta 18O$ over 1982–2015 carries a strong imprint of ΔSLP , even though many individual records are not strongly correlated with ΔSLP . This imprint arises from multiple*

well-documented processes, including PWC-related changes in moisture source and transport length, and a PWC- or ENSO-driven ‘amount effect’ in tropical regions.”

Line 65: I would include a few more details about the proxy dataset in the main text. Are these records at least annually resolved? What types of proxies compose these records?

Response:

We have added more details about the proxy dataset to the main text, including more information about the Iso2k database (L67-74).

Lines 70-90: Absolutely love that you did all of this sensitivity testing—and so clearly! This is one of my favorite parts of the paper. I was curious how you went from 59 proxies to a 4800 member ensemble. Could you provide the reader with this detail? We have 5 reconstruction methods, 3 calibration products, and some sort of age adjustment? Am I missing anything?

Response:

It’s one of our favorite parts too :) . That information is provided in the Methods (Section 2.3.1.1) - we have added a pointer at this location to refer the reader to the Methods (L99-100), and edited the text to make this more clear to the reader.

Line 95: How well do the instrumental products agree over the 1900-2022 interval?

Response:

This is an excellent question, and one that we spent a lot of time looking into at the start of this project. The short answer is ‘not as well as one might hope’, which is also the case for other gridded observational products (e.g., SST). A full discussion of this question is outside the scope of this paper, but to allow readers to assess this for themselves, we have added Extended Data Fig. 3 showing the 1900-2010 CE period (the limits of ERA20C), with the Δ SLP from a) the reconstruction ensemble, and b) the three instrumental products. We also state this in the text (L102).

Line 95: If you train the products on the 20th century instrumental record, aren’t you forcing them to agree with the instrumental trends over the 20th century? Perhaps there is a way to check this influence -- I see that you did this in the Supplemental Materials! Great! Clearly referencing that additional analysis here would be helpful!

Response:

We have added this information to the main text (L104-105).

Line 99: “this decrease in skill back through time is a common but rarely emphasized feature of paleoclimate reconstructions” – I’m not sure what you mean here. Could you clarify?

Response:

We have edited the wording for clarity (L108-110) - *“this decrease in skill back through time is a challenge for most palaeoclimate reconstructions, due to decreased data coverage and increased chronological uncertainty”*.

Line 105: Personally, I think some of this inter-method disagreement is really interesting. Why might these methods have strong disagreement over particular intervals? Changes in traditional covariance patterns between proxies and the Walker Circulation? Sanchez et al. 2021 documented similar

disagreement in tropical Pacific reconstructions over the 1800-1850 periods when using coral-based paleo data assimilation and attributed the breakdown to different patterns of covariability during a period highly influenced by volcanic eruptions.

Response:

Agreed! We assume that the periods of lower/higher agreement are due at least in part to changes in the covariances between the proxies and the Walker circulation - that is one of the reasons we used the different reconstruction methods. Many thanks for the reference - we have added this comparison with Sanchez et al's work to the main text (L116-118).

Sanchez, Sara C., Gregory J. Hakim, and Casey P. Saenger. "Climate model teleconnection patterns govern the Niño-3.4 response to early nineteenth-century volcanism in coral-based data assimilation reconstructions." *Journal of Climate* 34, no. 5 (2021): 1863-1880.

Line 107/Figure 2: How do some of these frequencies have zero power? Are the authors only plotting the periodicities with significant power? If so, be sure to say it! It is not immediately clear to the reader.

Response: R11

Yes, that is correct - we are only showing periodicities with significant power. This is stated in the figure caption, but we have clarified it in the text.

Line 114-115: Really happy the authors put in the sentence “ the difference between the two distributions is within the range calculated between any other two intervals of equivalent length...”! I was curious if the PDFs had a more distinct difference if you compared slightly different intervals, eg. 1200-1900 vs 1900-2000?

Response:

Prior to submitting the paper, we thoroughly investigated this exact point, including stepping that ‘split point’ progressively further toward the present to see the change in the differences between the two distributions (see below). We agree that this is a really interesting analysis, however we do not have space (in text or figures) to cover all the nuance of this particular point, and instead distilled this analysis into just the information relevant to our question of whether there is a detectable anthropogenic influence on the PWC.

Additionally, we re-did the statistical analysis addressing this point such that the results are more intuitive (Fig. 2d, L132-134), and added the option for readers to play around with similar tests using the provided code.

Line 122: Could also cite the Seager et al. 2019 paper (mentioned earlier) here

Response:

We have added this citation (L141).

Line 123: Could potentially discuss tropical Pacific Decadal Variability (eg. Power et al. 2021) here

Response:

We have added some discussion of the Pacific decadal variability in the paragraph above (L124-126), and also added Extended Data Fig. 6b comparing our reconstruction with Vance et al.'s (2022) ice core-based reconstruction of the Interdecadal Pacific Oscillation.

Power, Scott, Matthieu Lengaigne, Antonietta Capotondi, Myriam Khodri, Jérôme Vialard, Beyrem Jebri, Eric Guilyardi et al. "Decadal climate variability in the tropical Pacific: Characteristics, causes, predictability, and prospects." *Science* 374, no. 6563 (2021): eaay9165.

Line 142-146: This is so cool! I had been worried about the potential influence of Pinatubo on the results given the importance of the 1992-2011 period, but I feel much more comfortable given this analysis. I am still a little worried about the potential apples to oranges comparison as detailed in the Major Points section.

Response:

Comparison of observed or modelled modern/future changes to a paleoclimate reconstruction is reasonably common. In this particular case, we argue that the demonstrated strong agreement between the reconstruction and the observations in the calibration period means that this comparison is valid. To make this more clear, we have added Extended Data Fig. 3 zoomed in on this period of overlap, where the reader can more clearly see the close correspondence between the observations and the reconstruction.

Line 149: I again am wondering about the resolution of the proxies considered in this network. Are they all annual or sub annual?

Response:

Yes, they are all sub-annually or annually resolved. This is mentioned in the Methods, and we have now added it to the main text (L68).

Line 157: “reassigned major eruption”—this is great!

;))

Line 165: how large is this trend?

Response:

At the suggestion of Referee 3, we have added a second axis to the panels of Fig. 4, showing a random sample of the ensemble PWC response to volcanic eruptions. This shows the magnitude of the response.

Line 172: If specifically referencing a study, please use the name of the study and then superscript it instead of just using the superscripted numbers

Response:

We have changed all instances of this in the text.

Line 190-192: Consider re-writing this sentence for clarity “if all volcanic eruptions with...” I’m not totally sure I follow what you are saying.

Response:

We have re-written this paragraph for better clarity (and also added PMIP3/4 models to this analysis) - L216-220 “*When applying the above SEA approach to the CESM1 LME (using the 25 strongest eruptions; Methods), 9 of the 13 CESM1 LME ensemble members produce a significant negative Δ SLP anomaly the year following a volcanic eruption (Fig. 5a), with Δ SLP anomaly magnitudes similar to those occurring during an average El Niño event*”.

Line 191: How is significance defined?

Response:

Significance is defined the same way as for the reconstruction. We have added the citation of the method we used to determine significance to make this clear (L218).

Line 195: Consider writing “Fewer LME ensemble members” to help the readers stay on point as you also call your proxy reconstruction an ensemble

Response:

We have made the suggested edit (L223).

Line 218: Again, this is so cool!

Response:

We have promoted the associated Extended Data figure to the main text to emphasise this point (now Fig. 6).

Line 225: Why pCO₂ instead of CO₂? I think CO₂ is more intuitive for readers

Response:

We have removed the rho in both instances (L258, L260).

Line 248: Did the authors remove the tropical mean from this? See major point for more discussion

Response:

We have re-done this analysis, now assessing relative SSTs in all cases (L764-768, L284-285).

Line 259: This manuscript describes a very original and exciting analysis. However, I don't think that the authors have emphasized just how original this source of data is! Many scientists and working groups are interested in finding uses for water isotope data—and this is a great illustration! If there is space, I would definitely hammer on this point a little more.

Response: R18

We have added text to remind readers how useful water isotope proxy data is. For example, at line 290 *“Finally, our novel use of water isotope proxy data to reconstruct atmospheric variability, including explicit incorporation of uncertainty from the training dataset, reconstruction method, and age-depth models....”*.

Figure 2B. I would make two boxes to indicate just where your delta SLP index comes from

Response:

We have added the two boxes to Figure 1.

Figure 3. These distributions are very gaussian. Is that expected? How well do these indices agree with the reconstruction?

Response:

The distributions include data from all 4800 ensemble members. Given the large number of samples, we expect the distributions to tend to a normal distribution. When examining skill scores, both individual ensemble members (Extended Data Fig. 4a) and the reconstruction ensemble mean (Extended Data Fig. 3b) agree very well with the target indices. With the new Extended Data Fig. 3, readers can now see this for themselves.

Figure 4: In many of these plots, it looks as though there is a consistent Stronger Walker state observed two years prior to the eruptions in all periods. This is wild! It is quite large in subsets B, C, and D. It might be worth commenting on this unexpected result.

Response:

We have added text with a proposed explanation for this results (L195-197).

Figure 5A/B/C: How large are these SLP anomalies relative to a traditional El Nino/ La Nina Event?

Response:

They are of very similar magnitude: Δ SLP anomalies during El Niño events in the CESM-LME range from around -0.8 to -4 hPa. We have added that information to the main text (L219-220).

Figure 5D/E/F: Have you subtracted the tropical mean from these Niño 3.4 SST estimations? Most of the studies that look at the ENSO response do remove the global tropical mean from SSTs (something very difficult to do with a single proxy record)

Response:

We have re-done this analysis, first subtracting the tropical mean from the Niño 3.4 SST anomalies. We have also added text explaining and discussing this point (L764-768, L284-285). Many thanks for pointing out this error in our original calculations.

Figure 5G/H/I: Love that you included this! So cool!

Line 498: It would be great if you included a table with each proxy record and cite the individual authors with the database. Also, what resolution are these proxies?

Response:

This table is unfortunately too large to be included in the Extended Data, but is available (including authors of each individual dataset) as supplementary information. All records are annually or sub-annually resolved, and we have added this information to the main text (L68).

Line 509: “Model simulations” instead of “Model data”

Response: R19

We have made the suggested change (L479).

Line 529—great! I don’t think that I got this from the main text. Do you include a table of the skill?

Response:

That table was in the supplementary materials, but has now been promoted to Extended Data Table 1.

Line 535: Do you use a January-December year or a tropical (May-April) year when converting records to annual resolution? It’s very important to be clear on what was done.

Response:

We used a calendar year (Jan-Dec) to match the other proxy data. We have added this clarification to the text (L508).

Line 645: If referencing the study by name, please include the author’s name, not just the superscript number.

Response:

We have changed all instances of this in the text.

Line 656: Matching the SLP variance to that observed over the twentieth century could be a big problem. I discuss this in the major points section.

Response:

This is a valid concern, and one that we spent a lot of time investigating. The method we used is a commonly-used approach for variance scaling in palaeoclimate reconstructions. Additionally, variance

changes through time are preserved, and we have added text to the Methods expanding on this point (L623-624).

Line 741: the years 1992-2011 are superscripted

Response:

We have fixed this formatting error.

Line 743: just mention the authors and then superscript if directly mentioning a study.

Response:

We have changed all instances of this in the text.

Line 826: Suggest adding the of the method here-- eg. PAGES2k GMST (#66)

Response:

We added the in-text reference to PAGES 2k (L778).

G. References

The authors did a good job of including relevant, recent references. However, there were a few papers that merit mention in this discussion (mentioned previously in this review).

Response:

Many thanks for these excellent suggestions. We are limited by the 50-reference limit, but have added most of the below references, in place of some of the references originally included.

Seager, Richard, Mark Cane, Naomi Henderson, Dong-Eun Lee, Ryan Abernathy, and Honghai Zhang. "Strengthening tropical Pacific zonal sea surface temperature gradient consistent with rising greenhouse gases." *Nature Climate Change* 9, no. 7 (2019): 517-522.

Power, Scott, Matthieu Lengaigne, Antonietta Capotondi, Myriam Khodri, Jérôme Vialard, Beyrem Jebri, Eric Guilyardi et al. "Decadal climate variability in the tropical Pacific: Characteristics, causes, predictability, and prospects." *Science* 374, no. 6563 (2021): eaay9165.

Sanchez, Sara C., Gregory J. Hakim, and Casey P. Saenger. "Climate model teleconnection patterns govern the Niño-3.4 response to early nineteenth-century volcanism in coral-based data assimilation reconstructions." *Journal of Climate* 34, no. 5 (2021): 1863-1880.

Grothe, Pamela R., Kim M. Cobb, Giovanni Liguori, Emanuele Di Lorenzo, Antonietta Capotondi, Yanbin Lu, Hai Cheng et al. "Enhanced El Niño–Southern oscillation variability in recent decades." *Geophysical Research Letters* 47, no. 7 (2020): e2019GL083906.

Cobb, Kim M., Niko Westphal, Hussein R. Sayani, Jordan T. Watson, Emanuele Di Lorenzo, H. Cheng, R. L. Edwards, and Christopher D. Charles. "Highly variable El Niño–southern oscillation throughout the Holocene." *Science* 339, no. 6115 (2013): 67-70.

Khodri, Myriam, Takeshi Izumo, Jérôme Vialard, Serge Janicot, Christophe Cassou, Matthieu Lengaigne, Juliette Mignot et al. "Tropical explosive volcanic eruptions can trigger El Niño by cooling tropical Africa." *Nature communications* 8, no. 1 (2017): 1-13.

H. Clarity and Context

There are a few places where the authors could improve the clarity of their manuscript to help the readers understand the novelty of the work and significance of their results. I have generally commented on this in the Suggested Improvements Section.

However, there are a few points that would add a lot of clarity to the manuscript in general :

Firstly, A more in depth description of the proxies that make up the reconstruction. Table S1 cites the Iso2k database, but does not directly reference the individual authors who made many of these proxy datasets. This is a small point, but many paleo people that aren't familiar with the ISO2k database would get much more out of this table if you directly wrote "Bagnato et al. 2005 (14)" instead of #14 and the ISO2k ID number. I'm also a little confused about the "unknown" resolution or seasonality. Does this mean variable resolution?

Response:

We have added more informative citations to the table. For seasonality, 'unknown' refers to records where the authors of the primary dataset did not claim any particular seasonality for their data, but also did not explicitly state that the record represents an annually integrated signal. We have added clarifying text to the table caption.

Secondly, I think that it would help the readers if the authors included their specific definition of Walker Circulation Variability in the main text (presently included in the methods). Suggestion that the authors add two boxes to Figure 1 to highlight the regions where SLP anomalies are considered. It'd really help the clarity if this was directly addressed this early on (I think early on in the manuscript the authors clarify that their index isn't related to hydroclimate or SST, but don't say exactly what it is). I also think that adding a sentence to describe just how these proxies mechanistically relate to Walker Circulation Variability would be extremely helpful for readers unfamiliar with the Falster et al. *Journal of Climate* paper.

Response:

We have added the index reference boxes to the map in Figure 1. We also added text summarising the main findings of the Falster et al. *J Clim* paper, in terms of how water isotopes relate to the Walker circulation (L64-67).

Referee #2 (Remarks to the Author):

Based on the relationship of d18O in precipitation and Δ SLP, the authors used proxy records for d18O of precipitation and reconstructed the Δ SLP to represent the intensity of PWC. To reconstruct Δ SLP, they use five statistical methods to train the proxy records with three instrumental SLP datasets. The effort aims to minimise the uncertainties arising from the reconstruction method or dataset used for calibration. Furthermore, by comparing the pre-industrial (before 1850) and industrial era on the reconstructed Δ SLP,

and examining the features of the composite of Δ SLP after the volcanic eruptions, the authors attempted to address the effects of anthropogenic and volcanic forcing on PWC.

The reconstructed PWC does not exhibit a significant trend during industrial-era as shown in GMAT, but shows a shift to lower frequency variability, which may be due to the anthropogenic forcing. The authors did not provide any concrete physical explanation for this shift. The composites from both reconstruction and model simulation show a weakening of PWC after the volcanic eruption, consistent with many previous studies.

Response:

As far as we know, this is the first time this post-industrial shift to lower-frequency variability has been noted in the literature. A mechanistic explanation for this shift is outside the scope of this paper, which rather identifies the phenomenon first the first time, and reports the finding. We have stated this caveat in the concluding paragraph of the main text (L296-298) - *“Although diagnosis of the dynamics underlying forced responses and intrinsic variability in the PWC was beyond the scope of this paper, the Δ SLP reconstructions provide the necessary observational foundation for such future investigations”*.

Additionally, this is now the topic of ongoing research in our group (using state-of-the-art climate models).

The reconstruction method applied in this work is a helpful tool to minimise uncertainties. CESM last millennium simulations are used to support the effect of the volcanic eruption. There is no reference and more information about the model ensembles and the motivation for using the ensembles from one model instead of the multi-model ensembles from PMIP3/PMIP4 last millennium simulations. Some last millennium simulations have provided the continuous simulation for the industrial-era, which is helpful to investigate the anthropogenic forcing on PWC using the model simulation, for example, MPI-ESM last millennium simulations (<https://mpimet.mpg.de/en/science/projects/archive/millennium>). The continuous simulation into 21st century would overcome the initial condition issue that might affect the historical simulations.

Response:

We had included the reference for the CESM LME in the Methods, but have now added it to the main text (L213). The CESM LME simulations continue to 2005, and this data was used in these analyses. A complete characterisation of the PWC in PMIP3/4 models and the CESM LME is beyond the scope of this paper, which rather focuses on empirical evidence of PWC changes, with model data provided as a comparison with previous work. We have clarified this in the main text (L208-211) *“To assess our findings in the context of this previous work...”*. Detailed investigation of the anthropogenic and other external forcings’ influences on model PWC is currently underway in our group.

Nevertheless, to build on our original use of a single-model ensemble to assess our findings in the context of previous studies investigating the model ENSO response to volcanic eruptions, we have now added data from eight additional PMIP3/4 models to Figure 5 (including eight GISS-E2-R ensemble members) - we have added new text throughout to incorporate this

addition, but particularly L225-229. This allows assessment of inter-model variability in the Δ SLP response to volcanic eruptions, as well as further contextualisation in terms of the similar work that has been done previously, assessing the tropical Pacific SST response to eruptions in PMIP models.

Scientifically I don't find the results from this work are breakthrough novel and add new insight into the changes and variability of PWC during the last millennium. I would find it difficult to recommend publication in Nature. On the other hand, the reconstructed time series of Δ SLP is useful and could publish in a paleoclimate journal.

Minor comments.

In Fig5, I don't understand why it is necessary to apply the chronological uncertainty to model data. For the model experiment, we know that once the external forcing is given, the model uncertainty usually arises from the internal variability.

Response:

We have removed these panels from Figure 5.

The threshold for SAOD is 0.05 for reconstruction and 0.07 for model simulations. I understand that 0.07 for CESM-LME follows Dee et al. (2020) to be comparable. So why not use 0.07 for the reconstruction instead of 0.05? Would that yield different results?

Response:

The reason we used 0.05 as our threshold was to ensure inclusion of eruptions during the temporally relatively well-constrained 19th/20th centuries. Rather than change this, we have now updated the model analysis to match the parameters of the reconstruction analysis - using the 25 strongest eruptions of the 1200-2000 period (L216-217, L751-762).

Referee #3 (Remarks to the Author):

Review of Falster and co-authors "Forced changes in the Pacific Walker Circulation over the past millennium". In this study the authors attempt to reconstruct the Pacific Walker Circulation (PWC) over the past millennium relying on various statistical methods applied to annually-resolved proxy records. They then explore the potential influence of external forcings (natural and anthropogenic) on the PWC inter annual to decadal variability and trends over the last 800 years.

Distinguishing between forced and unforced variability is critical in the context of the impact of the ongoing anthropogenic forcing on climate. A large body of recent papers have explored such an issue for the Pacific sector including the period prior to 1850. So far, most studies have attempted to reconstruct sea surface temperature (SST) inter annual to decadal variability over the Common Era, to address the relative influence of radiative forcings respectively to internal variability processes in shaping the Pacific Ocean past dynamical changes. From this point of view, the present work is very novel, as the authors attempt to reconstruct the sea level pressure (SLP) zonal asymmetry driven by the Walker Circulation

variability, allowing to capture the coupled dynamics (as opposed to SST which are often blurred by the direct radiative forcing).

Based on a large ensemble of PWC reconstructions, the authors explore the potential influence of the largest natural forcing of the last millennium, namely the volcanic forcing, and try to put in a longer term perspective the variability and trends observed over the historical period. They conclude that the recent PWC trend is not anomalous in the context of the last 800 years. They also conclude for a significant PWC weakening in the 3 years following the largest eruptions of the reconstructed period, suggesting an El-Niño like response which appears roughly consistent with previous studies based on SST reconstructions or climate models.

The paper writing needs to be significantly improved as it refers constantly to SI or extended material that are central to the study, so that the reader has to stop and go back and fourth through various documents at almost every paragraph to follow the results and reasoning presented in the main manuscript.

Response:

We have now incorporated Supplementary Information into the Methods or Extended Data items, and have promoted two key Extended Data figures to the main text. We have also added extra methodological details into the main text. If published, the online paper will be formatted such that Extended Data items are easily viewed using the sidebar.

The scientific question mentioned in the abstract and in the introduction is also not specific enough. The reader has often to guess when the authors discuss the “recent strengthening” or “weaken” of the PWC in response to anthropogenic forcing without mentioning clearly the related timescales and related specific scientific question. For example, at the decadal timescale it is likely that the variability is largely aliased by the Interdecadal Pacific Oscillation (internal to the climate system, an issue that is not discussed or quantified as such by the authors), while the secular trends are more likely to be dominated by external radiative forcing, at least over the industrial period. The confusion between the underlying processes of decadal variability and secular trends, is present in the abstract and persists throughout the manuscript so that the reader does not have a clear understanding about the scientific challenges that are addressed in the study. There are also many caveats that have to be clarified in the statistical approaches and experimental protocol. As it stands, these caveats leave questions and large uncertainties in the obtained reconstructions limiting any robust interpretation and conclusions.

Response:

This paper specifically targets the PWC, rather than the IPO or the PDO (both of which are low-frequency). Of course it is possible that the IPO/PDO signals alias into the internal variability of the PWC, which is one of the reasons for doing the reconstruction as it is, at annual resolution. The reconstruction provides a longer record of the PWC so that we can test if the post-industrial variability and trends are distinct from intrinsic and naturally externally forced variability related to things like the IPO/PDO (as during the pre-industrial last millennium there is no anthropogenic forcing). Additionally, although interannual- to decadal-scale variability is likely mostly internal variability, as stated in the text a) anthropogenic aerosol emissions potentially affect the global climate system on decadal scales, and b) volcanic eruptions affect the global climate system on interannual timescales and interact with internal modes of variability. We have edited the abstract (L23-24) and main text to make that clearer, and also

have edited the text throughout to be more specific in our language regarding ‘recent strengthening’ et.c. (e.g. L20, 25, 138, 170, 262).

We have addressed the referee’s statistical concerns below.

I however acknowledge that a lot of work has been developed in an attempt to use appropriate various statistical methods to analyse a large set of proxy records. Yet the major issues listed below have to be addressed to assess what is robust and what is not. Provided that the authors can make significant improvements (in the writing presenting clearly the scientific question and hypothesis, the methodology to obtain robust reconstructions and analyses), such work could potentially allow for significant progresses in our understanding of tropical Pacific natural variability.

Major comments:

a. The authors selected a set of 54 globally distributed “water isotopes” proxy records and five non isotope records (listed in Table S1) based on their “high” correlation with the PWC or ENSO but no specific analyses are shown sustaining such statement. What is “high”? Is there a specific threshold or statistical analyses made to make such proxy records selection?

Response:

As described in the Methods, the specific set of records included for each reconstruction method differs - this part of testing the sensitivity of the reconstruction to the reconstruction methodology (the method includes not only the statistical method used to extract the PWC signal, but also the data input to the statistical method). In line with the findings of Falster et al (2021), which forms the theoretical foundation for this reconstruction, for the relevant methods (CPS, PCA, PCR) we used **all** available water isotope proxy records, regardless of their correlation with the target index. This is the common approach for using each of those methods in paleoclimatic reconstruction (e.g., similar methods used in the PAGES 2k GMST reconstruction). Nevertheless, we performed a test as to the sensitivity of the results to including a) all records, and b) only records significantly ($p < 0.1$, as stated in the Methods) correlated with the target index - you can see those results in the ‘PCR-all’ versus ‘PCR-cor’ sub-ensembles. We have now added that threshold to the main text (L89).

For the non-Iso2k records, we used records where the authors claimed a mechanistic relationship with tropical Pacific variability. We have clarified the wording in the Methods (L472) “*Following a broad literature search, we sourced 9 additional records from within the tropical Pacific, or where authors describe a strong relationship between the proxy record and either the PWC or the El Niño-Southern Oscillation (ENSO)*”.

For the reconstruction methods that require a correlation with the target index, we still filtered the full 59-record dataset to only include records significantly ($p < 0.1$) correlated with the target index. Records that are/are not significantly correlated with the target index are shown in Extended Data Figure 2 (denoted by a black outline). We recognise that this outline was difficult to distinguish from the grey outline of the non-correlated records, so we have edited this figure accordingly. We have also added a reference to the Methods section at Line 71, and clarified our wording at L70 “*five annually resolved non isotope-*

based palaeoclimate records that have a strong mechanistic relationship with the PWC or ENSO”.

To assuage potential similar concerns from readers, we have taken some methodological information from the Methods and added it to the main text (e.g. L56-58, 66-67, 70-71).

The authors mention in the SI that the “primary” age-model timeseries is retained during the selection process based on the record authors claims. It might be a proper way to go, but how consistent are the statistical analyses crieterias among the various original publications? It is important to clarify this especially that the rest of the study explores the sensitivity to the age model, in an ensemble approach to reconstruct the PWC. I suggest that the selection criteria (level of significance based on various metrics) to be defined first by the authors and then used on the ensemble of possible age models to retain only the proxy timeseries (across the possible age models) that have the best scores. This will allow for a consist treatment of available records from various source and authors and a more objective way to perform the selection.

Response:

This is a good suggestion, and it is something that we considered when designing our reconstruction methodology. Essentially, it is an additional layer of sensitivity testing - to what degree are the correlations of the proxy records with the target index in the 1900-2000 CE interval dependent on chronological uncertainty? The reason that we did not include this layer is that it makes very minimal difference. For each of these records, chronological uncertainty increases back through time - for most archive types, the assumed chronological error was 1 year per 100 years. Therefore, uncertainty in the reconstruction ensemble attributable to chronological uncertainty increases back through time, but in the calibration interval it is negligible.

Aside from this, the record selection is objective: the full suite of records (both correlated and uncorrelated with the target index) is simply all Iso2k records that meet the temporal resolution and coverage requirements outlined in the Methods, as well as non-isotope records (from a broad literature search) that are in or immediately adjacent to the Pacific Ocean, or where the authors specifically claim a strong correlation with the PWC or ENSO. When performing reconstructions using only records that are significantly correlated with the target index, this filtering is applied to all records. We have edited the main text and methods to emphasise these points (e.g. L89, 472-474, 553-556).

Regarding “*how consistent are the statistical analyses crieterias among the various original publications?*” - this relates only to the non-Iso2k records that we included (n=9), that are (according to our criteria) “*from within the tropical Pacific, or where authors describe a strong relationship between the proxy record and either the PWC or the El Niño-Southern Oscillation (ENSO)*”. We did not perform this assessment, but rather trusted the conclusions of the authors. In all cases, they provided mechanistic links between ENSO/PWC and variability in their proxy record.

b. To test the robustness of the results depending on the records availability, the authors performed reconstructions on five temporal sections, i.e. 1200-2000, 1400-2000, 16000-2000 and 1800-2000, using in each case proxy records with more than 66% of temporal coverage. Records are then interpolated with the DINEOFs method to fill in the missing data. I understand that for each reconstruction 15% of the

available proxy are removed to account for possible dependence of a particular subset of proxies. This approach raises 2 issues that might alter the derived reconstructions. Is the same set of records used for each period? If not, then changing the set of records from one temporal section to another might still hide proxy-dependant sensitivity and introduce artificial discontinuities in the variances and variability content, raising the issue of the proxy records resampling.

Response:

To answer the referee's question: no, the same set of records is not used for each period. This is described in the Methods (L549, L521-524), and can be seen in Extended Data Figures 1 and 2.

Regarding “*changing the set of records from one temporal section to another might still hide proxy-dependant sensitivity and introduce artificial discontinuities in the variances and variability content, raising the issue of the proxy records resampling*” - artificial (non-climatic) changes in variance are an inherent feature of palaeoclimate proxy records - this is a well-known phenomenon. In general, variance decreases back through time, due to chronological and other proxy/archive-dependent uncertainties. A common approach to dealing with this is to scale the variance of the entire reconstruction to match that of the target index, which still allows for changes in variance through time. Going a step further, to relatively state-of-the-art reconstructions using the temporal nesting approach, the variance of subsections is similarly adjusted (e.g., Freund et al. (2019) *Nature Geoscience*, Dätwyler et al. (2019) *International Journal of Climatology*). Acknowledging that this is a methodological choice that must be made, with no ‘perfect’ option, we performed extensive sensitivity testing to determine the optimal method for adjusting the variance of adjacent sections - the main parameters we tested were the degree to which the variance of adjacent sections was matched to the variance of the antecedent section, versus the variance of the calibration interval.

In the process of addressing several of the referee's specific concerns below, we repeated two key analyses, using a non-nested version of the reconstruction, i.e., a reconstruction using the same set of proxies throughout. All conclusions remain robust in this context (see below), and we have added one of these analyses as the new Extended Data Figure 7 (L128-130, L688-693).

A way to test this, would be to perform reconstructions over the 5 periods (keeping the 15% random sampling) using the same set of proxy records available throughout the longest period, i.e. 1200-2000. I understand that the number of available records over the whole period will be lower but at least this will allow to have a reference with the same dataset to test the robustness in any shift of frequency and trends over the last 800 years. The second issue concerns the DINEOFs interpolation method. I suspect that it has a larger influence when we go further back in time since there are more and more gaps to fill. Can the authors provide a figure showing the number of records with gap filling and their length throughout the 800 years to assess the influence of such an approach in the uncertainties of the derived reconstructions?

Response:

Regarding “*A way to test this, would be to perform reconstructions over the 5 periods (keeping the 15% random sampling) using the same set of proxy records available throughout the longest period, i.e. 1200-2000*” – this is an excellent suggestion. Please see below for the use cases where we have now incorporated this test (e.g. the new Extended Data Fig. 7).

Regarding “*Can the authors provide a figure showing the number of records with gap filling and their length throughout the 800 years...*” - We have added Extended Data Fig. 1, showing this information. This clearly demonstrates that the referee’s concern - “*I suspect that it has a larger influence when we go further back in time since there are more and more gaps to fill*” - is not the case. If palaeoclimate proxy records have large gaps, these tend to be concentrated at the ends, although even this is not common for the sub-annually to annually resolved records used in this reconstruction. In those few cases, a record may have been filtered out from one temporal subset, but made it through the filter for the next-youngest temporal subset. What remains after this temporal-coverage filtering are records with the more common issue of occasional missing years - this can be clearly seen on the new figure. It is highly unlikely that this makes a meaningful difference to the final reconstructions. Hopefully this new figure assuages doubts that the gap-filling biases the results.

c. As a follow up on the previous comment, Table S2 and Extended Figure 1 rely on correlation, RMSE and binomial sign test computed over the 1900-2000 period, but no scores are shown regarding the score related to the amplitude of the variance. The scores are computed comparing the 4800-member ensemble reconstructions median (n=4800) against the mean of 3 observational datasets (n=3). It would be good when assessing the variance amplitude to randomly select 3-member ensemble reconstructions to allow for a proper comparison to observations. The question of the variance has been overlooked in the assessment of the reconstructions while they are used to discuss frequency content and trends compared to the recent historical period.

Response:

In Extended Data Figs. 4 & 5, we show the skill scores for **all individual ensemble members** relative to the instrumental data used to train that particular ensemble member. In Panel B of Extended Data Fig. 3, we also show the correlation of the sub-ensembles trained on each instrumental Δ SLP with that index. As the referee correctly implies, it would be incorrect to only perform comparisons between the reconstruction and instrumental Δ SLP medians. The only case where we compare ensemble median Δ SLP with mean instrumental Δ SLP is in Extended Data Table 1, and this is mostly to provide a means of comparison with the skill for previous reconstructions of tropical Pacific variability (few of which provide such exhaustive ensemble-based skill testing). We argue that by showing the full distribution of skill scores for all individual ensemble members, we accurately report the range of reconstruction skill for the ensemble.

Regarding the variance: as part of the reconstruction methodology, all reconstructions were scaled such that the variance in the 20th century matches the variance of the observational data (L620-624). However, we now incorporate the reduction of error test into our skill assessment - this assesses both mean and variance, and is a preferred measure of skill in climate reconstructions (e.g., Mann et al. (2007) *JGR Atmospheres*) - this is shown in Extended Data Figs. 4-5, and Extended Data Table 1.

d. The authors state on lines 104 to 111 « A weak positive Δ SLP trend from ~1200-1750 CE is followed by a slight decrease to ~1800 CE, then a period of low inter-method agreement. ... Approximately 10% of ensemble members also have significant power in ~decadal (10-12 year) and multidecadal (21-24 year) bands. Interestingly, there is a shift to higher power at lower frequencies in the industrial period (since 1850 CE) relative to the pre-industrial last millennium (4-9 rather than 2-9 year periods, with particularly high power in the 9-year band)»

Why only 10% of ensemble member have a significant power in the decadal and multidecadal band ? This number is low, how robust is it related to the other 90% of the members without such variability. Is it linked to the method, the age models errors, the proxies used,... ? Since 15% of the proxies are ignored it is perhaps related to the types of proxies considered? I feel like most of the discussion on the frequency changes is not robust or depends strongly on the method/proxy dataset. These questions prevent drawing any firm conclusions based on only 10% of the reconstructions. In addition, an apparent discontinuity also exists in the timeseries at year 1800 respectively to the previous period and at 1900 respectively to the following period. This is even more important as the 1800-1900 period bears the largest inter-method disagreement, which is counterintuitive as there are many available proxy records over this time window so the method should not have such a strong weight on the uncertainties. The resulting large disagreements prevent putting the recent instrumental period into perspective regarding the secular trends or decadal variability. Addressing the comments listed previously (records selections, changing records sets from one temporal window to an other, the gap filling, scores on variance) might help reduce these uncertainties and pin down what is robust and what is not.

Response:

The referee's concerns about gap filling and variance have been addressed with additional analyses and explanations. We address other specific points below.

Following the logic of the referee's suggestion, we repeated the analysis leading to this conclusion of a shift to lower-frequency variability in the industrial era, but using *only* the records with continuous data from 1600-2000 CE (L688-693). That is, not using the nested approach. We judged this the best interval for this test, as the sacrifice of half of the overall reconstruction interval allows inclusion of many more records for this sensitivity test (compared with performing the test on the 1200-2000 CE interval). The main finding is not changed when using reconstructions without the nesting approach, i.e., there is still a distinct shift to significance at lower-frequencies in the post-industrial compared with the pre-industrial. We are therefore confident that the nested reconstruction approach has not biased this result. We have added Methods text describing this test (L688-693), incorporated this new figure as Extended Data Fig. 7, and added main text at L128: "*Both this shift, and the low proportion of ensemble members with significant low-frequency variability, are robust to our temporally-nested reconstruction approach (Extended Data Fig. 7)*". Critically, as stated in this sentence, this also provides confidence that we are not artificially dampening low-frequency (decadal to multi-decadal) variability with our temporally-nested reconstruction approach.

Regarding "*Why only 10% of ensemble member have a significant power in the decadal and multidecadal band ? This number is low, how robust is it related to the other 90% of the members without such variability. Is it linked to the method, the age models errors, the proxies used,... ?*". We chose quite a strict cutoff for definition of 'significant' power at a certain frequency - not just the threshold ($p < 0.05$), but the method of identifying peaks (the multi-taper method with a power-law null). This greatly reduces the likelihood of false positives, particularly at lower frequencies (e.g., Mann et al. (2020) *Nature Communications*). So we are confident that where a signal has been identified, it is robust. When using a less stringent method of identifying peaks (or indeed assigning significance at lower p-values), then the proportion of ensemble members with significant peaks at lower frequencies is higher, but we elected instead to use the most robust method.

Regarding “*I feel like most of the discussion on the frequency changes is not robust or depends strongly on the method/proxy dataset*” - readers can see from the colour bars of Fig. 2a and Extended Data Fig. 7 that the reconstruction method does not strongly influence the results. The new Extended Data Fig. 7 demonstrates that the result is also not affected by changing contributions from different proxies.

Regarding “*These questions prevent drawing any firm conclusions based on only 10% of the reconstructions*” - in fact, we argue that our stringent approach to assigning significance to spectral peaks makes the results more robust. We could have used a less conservative method, thereby identifying ‘significant’ frequencies in a higher proportion of ensemble members (or just using the ensemble median), however what would make our conclusions less robust. We have added a statement to make it clear that we are only considering significant peaks in the power spectrum (L121).

Regarding “*In addition, an apparent discontinuity also exists in the timeseries at year 1800 respectively to the previous period and at 1900 respectively to the following period. This is even more important as the 1800-1900 period bears the largest inter-method disagreement, which is counterintuitive as there are many available proxy records over this time window so the method should not have such a strong weight on the uncertainties...*” - we have added extra discussion of the 1800-1900 interval of lower inter-method agreement, following a very useful literature reference from Referee 1, observing that the same period of low inter-method agreement is present in tropical Pacific SST reconstructions (L116-118). This provides additional confidence that this stems from a climatic feature (possibly the result of several small volcanic eruptions over this time), rather than a purely methodological bias. Regarding the ‘apparent discontinuities’ at 1800 and 1900 CE, after the results of many sensitivity tests (including the new analysis shown here), we are confident that our temporal nesting approach does not bias our main findings. Additionally, 1900 is not a temporal nest boundary.

e. Figure 2A and extended Figure 3: In line with my previous comment, cumulating individual reconstructions for which a very small proportion of members (0.1 each?) with a significant power at a given period is misleading. It gives an artificial perception of robustness. Are the reconstructions with significant multi-decadal variability the same as those with significant decadal variability? Providing the form of the frequency content for at least reconstructions with a significant signals or by type of method would be helpful. Is it the same set of proxies for the industrial period (since 1850 CE) relative to the pre-industrial periods? Could it be due to the influence of proxies resampling, the more important role of DINEOFs interpolation method when going back in time? Would it be possible to add instrumental observations on these figures to see how the well reconstructions performs over the observational period?

Response:

We have edited the text for clarity along these lines.

Regarding the referee’s concerns “*Is it the same set of proxies for the industrial period (since 1850 CE) relative to the pre-industrial periods? Could it be due to the influence of proxies resampling, the more important role of DINEOFs interpolation method when going back in time?*” - the first concern is addressed by the non-nested spectral analysis figure (Extended Data Fig. 7), which demonstrates that this phenomenon is present even when the same set of proxy records is used for the full interval. That is, the nesting approach does not affect low-frequency variability, nor does having more proxy data available for

the post-1850 interval. The second concern is addressed by the new Extended Data Figure 1, demonstrating that very little interpolation is required, and only at random intervals.

Regarding “*Providing the form of the frequency content for at least reconstructions with a significant signals or by type of method would be helpful*” - we are not entirely certain what the referee is requesting, but interpret this as a request for power spectrum plots for individual ensemble members.

We have endeavoured throughout this paper to avoid reducing our results to single timeseries (e.g. a median), as that fails to show the uncertainties associated with each analysis - indeed we would argue that this is one of the particular strengths and novelties of this paper. However, we now provide the power spectrum of instrumental Δ SLP in the new Extended Data Figure 7, showing the power-law null used to determine significance.

f. Lines 112-115: « The distribution of Δ SLP values in the industrial period is skewed toward higher (more La Niña-like) values than in the pre-industrial last millennium (1200-1849 CE; Fig. 2b). However, the difference between the two distributions is within the range of that calculated between any other two intervals of equivalent length across the reconstruction (Methods). »

This means that the differences before and after 1850 is not significant but the writing is not clear enough and suggests a tendency towards more La Niña-like over the historical period. Please rephrase to avoid any misleading messages.

Response:

According to a Kolmogorov-Smirnov test, the two distributions *are* significantly different. However, we acknowledge that perhaps this was not the correct test for this situation i.e., with several thousand ensemble members, it is very easy to have a statistical difference between means. We have therefore redone this analysis, but instead performing K-S tests on all **individual** ensemble members, and reporting the distribution of p-values i.e., the proportion of ensemble members where the pre- and post-industrial are significantly different (L667-670). We report this proportion in the main text (L133, L669), and have added a boxplot of K-S test p-values to Fig. 2. We have also rephrased this sentence accordingly: “...*the difference between pre- and post-industrial mean Δ SLP is insignificant ($p >= 0.05$) in 81 % of the 4800 reconstruction ensemble members*” (L132-134).

g. Lines 117-124: « The lack of a significant PWC mean state shift in response to anthropogenic forcing is an important result. Climate models suggest the thermodynamic effect of GHG-driven rising global mean surface temperature (GMST) should ultimately weaken the PWC 15,34 and a negative Δ SLP trend is present in historical simulations from most CMIP5/6 models 3. However, recent work suggests that the global warming-driven slowdown of the Atlantic Meridional Overturning Circulation accelerates the Pacific trade winds, resulting in a stronger PWC 35. Our findings demonstrate that neither GHG-driven effect is emergent. Nevertheless, the post-industrial shift in PWC variability toward lower frequencies is intriguing, and possibly a response to anthropogenic forcing that has not previously been identified. ».

Considering that the uncertainties in the reconstructed decadal variability and secular trends are much larger than the signal the authors are looking for, I don't think such a statement can be made. The results shown so far simply illustrate that the errors and uncertainties in the reconstructions do not allow putting the recent changes in perspective.

Response:

The additional analyses suggested by the referee—particularly spectral analysis on non-nested reconstructions, now incorporated into Extended Data Figure 7—demonstrate that the lack of decadal-scale variability in the reconstruction is not an artefact of the temporal nesting methodology.

We challenge the referee’s suggestion that we cannot make the statement in this paragraph. Our analyses clearly show that if there is any anthropogenic influence on the PWC, then it is too small to be detected relative to intrinsic variability. This is equivalent to the same analyses performed using model simulations, where secular changes are only considered emergent (or significant) once a signal exceeds the range of the model (or model ensemble)’s intrinsic variability (e.g., from a pre-industrial control simulation or initial condition large ensemble). This also supports recent high-impact publications based only on observational data, suggesting there is no significant PWC trend during the industrial era. We have confirmed this finding by placing it into a longer-term, and thus we argue more robust, context.

h. Figure 3 (B,C,D) : The trend in the observation can potentially be more marked than in the reconstructions only because of the bias in the method which underestimates the real variability. It would be nice to indicate the median and maybe max, min (or 97.5 and 2.5 percentile) trend of the reconstructions over the period 1992-2011. And given that the historical period is the period with largest available proxies, do we not have the strongest trends marked at 20-year resolution because we lack proxies in the past ? Are these results robust if we use the same set of proxy over the full period (see previous comments a and b above)?

Response:

The logic underpinning Figure 3 (and indeed all analyses presented in this study) is that with an ensemble reconstruction, the ‘real’ answer will be contained somewhere in the ensemble. Hence we have avoided, where possible, reducing the ensemble to measures of central tendency.

To answer the referee’s question about bias arising from the temporally-nested reconstruction approach, we repeated the analyses presented in Figure 3 with the same 1600-2000 CE non-nested reconstruction ensemble as used to address the referee’s concerns about bias in the power spectrum. The results of this test are shown below, clearly demonstrating that the key result presented in the main text (that the 1992-2011 PWC strengthening is unusual but not anomalous in a multi-centennial context) is not influenced by the temporal nesting approach.

Regarding “*bias in the method which underestimates the real variability*” - we have added a skill test to Extended Data Figs. 4 & 5, describing the ensemble skill in reconstructing variance (‘reduction of error; L636).

Regarding “*median and maybe max, min (or 97.5 and 2.5 percentile) trend of the reconstructions over the period 1992-2011*” - unfortunately we are unable to do this as the reconstructions go to 2000 CE. As stated throughout the text, the 1992-2011 trend is taken from observations. The new Extended Data Figure 3 clearly demonstrates good agreement between the reconstruction ensemble and the observations in the interval of overlap, providing confidence that this is a reasonable comparison.

i. Figure 4: Again cumulating individual very small number of reconstruction fraction per method is misleading. Would it be possible to add a y-axis to the right of panels A, B, C and D with perhaps the median anomaly and the 97.5 and 2.5 percentile for each method?

Response:

We have added a second axis to the figure, showing the composite Δ SLP anomaly for 100 randomly-chosen ensemble members.

j. Figure 1A and line 564-566: Visually the inter-method uncertainty over the period 1800-1900 seems especially strong and marked in 1800-1810. And then line 564-566: If I understand correctly the oldest period is drawn towards the most recent segment to make a continuous timeseries from sliced reconstructions? Isn't this an important source of error, especially if the connection is made in 1800 in view of the important inter-method uncertainties at this period in figure 1A?

Response:

This is a correct summary of how the temporal subsets were joined, and this is an approach that has been used previously (e.g., Dätwyler et al. 2019, Emile-Geay et al. 2013, Freund et al. 2019 as cited in the Methods). Although inter-method disagreement is indeed high for the first half of the 19th century, this is a phenomenon that has been observed in other reconstructions of tropical Pacific variability, as pointed out to us by Referee 1. We have added text expanding on this point (L116-118). We also note that this period of higher inter-method disagreement resolves around 1900, which is not at the boundary of temporal subsets, suggesting it is not a purely methodological bias.

k. Line 562-563: It would be useful to have a curve showing the number of proxies available over time with the proportion of real vs. interpolated proxies. The importance of the interpolation method in the reconstruction obtained certainly varies over time and it would be troublesome to have a proportion of interpolated proxies too high in some years (e.g. 8 out of 10 proxies used in the year 1350 are interpolated) and where the reconstruction would be largely related to the interpolation method. Making a figure showing that if at some point 90% of the available proxies are interpolated will allow when there might be a problem or a very large uncertainty?

Response:

This is a valid concern; fortunately it is not the case with the set of proxy data that we used. As described above, annually-resolved records tend not to have large gaps. We have now added the new Extended Data Fig. 1 that clearly shows the temporal coverage of each proxy record contributing to the reconstruction. As can be seen from the new figure, there is particular interval where data from multiple proxies are interpolated rather than ‘real’. Additionally: given the decrease in reconstruction skill back through time,

we have been careful throughout the manuscript to avoid making large claims based on the earlier section of the reconstruction. Rather, we have restricted the main points in the text to ‘big-ticket’ takeaways that are robust to data availability issues. We have added that statement to the main text: “*We restrict our main findings to those robust relative to the reconstruction uncertainty*”, L107-111.

1. Figure 5: There is an error in the caption g-i instead of g-h. I am not sure about the relevance of the 5g-i panels. I understand the idea of perturbing the age model by + or - 1% to compare with the reconstructions. But isn't this perturbation for model outputs more “violent” than what we have in the proxies? There is a notion of detecting the major signal from the reconstructions which use a set of proxies, whereas if I understood correctly the error is here directly applied on the model outputs? Is it appropriate ?

Response:

We have removed these panels from Figure 5.

Referee #4 (Remarks to the Author):

Dear authors of the manuscript “Forced changes in the Pacific Walker Circulation over the past millennium”, the ENSO response to volcanic eruption is a hot topic, but remains unresolved due to uncertainties in proxy data reconstructions and climate model simulations. This manuscript is novel and fruitful because the atmospheric Pacific Walker Circulation (PWC) is more sensitive to external forcing than the East Pacific Sea Surface Temperature (ENSO). Isotope records are more sensitive to climate than other records. Methods including CPS, PCR and Pairwise comparison are often used in the proxy data reconstruction, and these methods are mature and effective. Most of the analysis and results are reasonable and fruitful. However, data screening does not meet all methods. The comparison between the PWC reconstruction and the previous ENSO reconstructions is missing, and the physical process of the PWC to the external forcings is still unclear. Therefore, I recommend revising it before accepting it.

Main Comments:

1. Regarding the reconstruction method, there are four concerns. First, the criteria for screening proxy records should apply to all methods. e.g., there are four cases, a) all proxy records, b) the subsets of proxy records significantly correlated with the target index in the calibration window, c) proxy records in a tropical Pacific ‘centre of action’, and d) proxy records for specific seasons only, with no known biases.

Response:

Our use of different subsets of the data for the different reconstructions is a central component of the uncertainty quantification—specifically, quantifying uncertainty in the reconstruction method, of which both the statistical method *and* the input dataset are components. This was previously described in the opening paragraph of the Methods, and again in Section 2.2, but we have added additional Methods text at the start of Section 2.3.2 to clarify this point (L553-556). Regarding a) versus b), whether or not proxy records are required to be correlated with the target index is an important component of the various methods - for example, CPS uses all available records, regardless of their correlation with the target index, whereas the PCA-based methods require a significant correlation. Regarding c) and d) - these are sensitivity tests, which we regard as central to the robustness of our reconstruction ensemble. That is, we wanted to assess the likelihood that 1) the degree to which proxy data are ‘remote’ from the tropical Pacific (i.e., reliance on teleconnections); and 2) individual records’ known biases to particular seasons,

affect the results. Hence the additional versions of the CPS reconstruction sub-ensemble, along with Extended Data Table 1 describing these results.

Second, the first principal component of the proxy data is only retained in two variants of PCA, leading to a system bias in partly missing the low-frequency signals (McShane and Wyner, 2011). I suggest that that some first PCs should be assessed, which is similar with the PCR method.

Response:

If this is a source of bias in the PC1-based reconstructions, then it is offset by the other six sets of reconstructions that are not based on a PC1 approach (e.g. the PCR method, which uses multiple PCs). We have added text clarifying this point in the Methods (L553-556).

Third, there is no age uncertainty in woody records. Iterations of age-depth models for wood records would introduce new systematic biases.

Response:

Uncertainty in the age of tree-ring records arises most commonly from double rings and missing rings, although there are other sources of uncertainty. Similarly to coral records, given the small degree of the age uncertainty (~1 %, or one year per century), this is often ignored in data analysis, but nevertheless is a true source of uncertainty. See, e.g., Ricker et al (2020) *Plos One* <https://doi.org/10.1371/journal.pone.0239052>

Fourth, a reason for the sensitivity of these isotopic records to atmospheric circulation is suggested. e.g., how do the ice-core isotope records in the two poles respond to the PWC?

Response: R6

We have added extra text to provide more information about how globally-distributed water isotope records provide information about the PWC - “*The dominant mode of observed global interannual precipitation $\delta^{18}O$ over 1982–2015 carries a strong imprint of ΔSLP , even though many individual records are not strongly correlated with ΔSLP . This imprint arises from multiple well-documented processes, including PWC-related changes in moisture source and transport length, and a PWC- or ENSO-driven ‘amount effect’ in tropical regions*” - L64-67.

Regarding the ice core records specifically: there are many examples in the literature of a tropical Pacific (ENSO or PWC) influence on Antarctic ice cores - for a few examples, Vance et al (2013) *Journal of Climate*, Patterson et al (2005) *Annals of Glaciology*, Tariq et al (2022) *Frontiers in Earth Science*, Naik et al (2010) *Journal of Earth System Science*, Meyerson et al (2002) *Annals of Glaciology*. A tropical Pacific influence on Greenland ice cores has also been identified, e.g., Li et al (2019) *Journal of Climate*, Werner et al (2002) *JGR-Atmospheres*. Although the signal of the tropical Pacific in ice core records may be less prominent than more proximal modes of variability, the methodology in this paper (and the paper that formed the theoretical basis for this paper, Falster et al 2021 *Journal of Climate*) is designed to capture the dominant signal shared by *all* component records, which in this case is the PWC. We have added summary text to this effect (L65).

2. The Pacific Walker circulation constitutes the atmospheric component of the ENSO, so it is suggested that the PWC reconstruction be compared with previous ENSO reconstructions to emphasize the novelty

of this work. e.g., the isotope records used in PWC reconstructions reduce inconsistencies in other types of proxy records.

Response:

We have performed extensive comparisons with existing reconstructions of tropical Pacific variability (in terms of temporal character and relationships, and spectral character), that were not originally included in this submission. Although we assessed all existing relevant reconstructions, we focussed on comparison of our Δ SLP reconstruction ensemble with two key SST-based index reconstructions: 1) ENSO (Nino 3.4) reconstructed from the Last Millennium Reanalysis v2 (LMR), and 2) the remote (ice core-based) IPO reconstruction from Vance et al (2022). We performed particularly detailed comparisons with the LMR, which is a data-assimilation based ensemble reconstruction, using CCSM4 for the model prior. These included analysis of the temporal relationships between the two reconstructions ensembles, and comparison of spectral character. This detailed investigation is beyond the scope of this particular paper, but will be the topic of an upcoming publication.

To address the referee's suggestion, we have added additional text (L119-120), and the new Extended Data Figure 6, comprising the following components:

1. Timeseries of existing reconstructions of tropical Pacific variability
2. Correlations of Δ SLP reconstruction ensemble with Vance et al (2022) IPO reconstruction - 13-year Gaussian smooth applied to Δ SLP to match IPO reconstruction
3. Running correlations between Δ SLP reconstruction median, and existing reconstructions of tropical Pacific variability
4. Static correlations between Δ SLP reconstruction median, and existing reconstructions of tropical Pacific variability

We only use reconstructions of tropical Pacific variability with data extending back to at least 1600 CE. We further clip the reconstructions to a common interval of 1600-1978 CE; 1977 being the youngest year of one of the reconstructions. Note that across the reconstructions we compare to our new Δ SLP ensemble, there are various target seasons. For the purposes of this simple comparison, we do not differentiate these. For this simple comparison, we also reduce the LMR Nino 3.4 reconstruction ensemble to a median.

A comparison of reconstruction skill with previous ENSO reconstructions was previously summarised in the Supplementary Text, however due to word count limits this new analysis has taken the place of that summary.

Additionally, we have added text stating the novelty of our water-isotope-based approach (e.g. L71-74, L281-284, L290).

3. The climate dynamics of the PWC to the external forcings are missing. It is recommended to analyze signal forcing experiments in CESM LME simulations to show physical processes. Moreover, what kind of the internal variability is responsible to the PWC? e.g., the Interdecadal Pacific Oscillation? How to know it? Finally, how to use a new PWC reconstruction with pre-1600 CE too low skill and an old climate simulation to answer the question "Yet the PWC response to external forcings is unclear, with empirical data and model simulations often disagreeing on the magnitude and sign of these responses".

Response:

Regarding “*It is recommended to analyze signal forcing experiments in CESM LME simulations to show physical processes*”: this work is currently underway in our research group, and will likely be published this year. However, it is outside the scope of this paper, which rather sought to 1) develop an empirical reconstruction of the PWC, 2) assess temporal variability, and 3) determine whether we can detect forced responses. The in-preparation publication will diagnose and explore these mechanisms with the requisite nuance that would not be possible if combined with the detailed empirical analyses presented in this paper. We have clarified in the text that the model analyses are largely for contextualisation (L208-209), and have added a statement in the concluding paragraph that such detailed diagnoses are beyond the scope of this paper “*Although diagnosis of the dynamics underlying forced responses and intrinsic variability in the PWC was beyond the scope of this paper, the Δ SLP reconstructions provide the necessary observational foundation for such future investigations*” - L296-298.

Regarding “*Moreover, what kind of the internal variability is responsible to the PWC? e.g., the Interdecadal Pacific Oscillation? How to know it?*”: in the figure provided to address the comments above, we have compared our Δ SLP reconstruction with an existing reconstruction of the IPO (Vance et al 2022) (methodology described in L798-801). The correlations are negative, as expected, although weak. However, if there was a strong IPO signal, we would have expected to see a stronger spectral signal at decadal to multidecadal periodicities. We have stated this in the main text (L124-126).

Regarding “*Finally, how to use a new PWC reconstruction with pre-1600 CE too low skill and an old climate simulation*”: we have now added data from PMIP3/4 simulations to Figure 5. This allows further contextualisation of our work in terms of previous studies investigating the modelled tropical Pacific SST response to volcanic eruptions. Given the distinct decrease in reconstruction skill back through time, we have been careful throughout the manuscript to avoid making large claims based on the earlier section of the reconstruction. Rather, we have restricted the main points in the text to ‘big-ticket’ takeaways that are robust to data availability issues, and clearly stated this pre-1600 skill decrease in several places throughout the text. We have added further clarifying statements (e.g. L110-111).

Specific Comments:

1. Pages 1-2, lines 25-26. What kind of the internal variability? Interdecadal Pacific Oscillation?

Response:

Internal variability does not necessarily need to be assigned to a particular decadal-mode - it may be inherent to the PWC. We have added text and analyses comparing this reconstruction with a remote (ice core-based) reconstruction of the IPO (Vance et al., 2022) (L124-126, Extended Data Fig. 6).

2. Page 2, lines 43-48. How to deal with the same difficulty of the ENSO's response to volcanic eruptions in this work?

Response:

As highlighted in the Discussion, we suggest that the relatively clear volcanic signal in the PWC compared with ENSO is primarily due to a) a stronger atmospheric than SST response; and b) our incorporation of chronological uncertainty in the reconstruction. The previous difficulty of reconstructing atmospheric modes is part of the novelty of this work - using water isotope proxy records to capture

atmospheric variability. We have edited the text to emphasise this point (L279-288, particularly L281-284).

3. Page 3, line 53. It is important to show the number of the proxy records and not the number of members. The reason is that the number of the members is arbitrary.

Response:

We have added the number of proxy records contributing to the reconstruction (L56).

4. Page 3, line 55. The reference format does not meet the requirements of the journal. e.g., 15 reference.

Response:

We have fixed this reference.

5. Page 3, line 56. The anomalous period is usually to 1961-1990 CE.

Response:

For palaeoclimate reconstructions, there is no single common interval used to calculate anomalies - this ranges from various ~30-year windows, to the entire instrumental period, or the reconstruction period. The exception is reconstructions of global temperature, which tend to use the same base period. In this case, adding one year to the interval anomalies are presented relative to would not affect our findings.

6. Page 5, lines 123-124. The difference in low-frequency signals between the post-industrial and pre-industrial eras may be due to the different lengths of the analyzed datasets. Moreover, the current PCA methods partly lost low-frequency signals, and the proxy records only with annual resolution also impact the low-frequency signals. Therefore, it is recommended to use more records with low resolution and use a modified PCA method with more PCs to assess the frequency difference.

Response: R31

As stated in the Methods (Section 2.6.1), we performed the pre- versus post-industrial comparison using datasets of the same length, to avoid exactly this bias (L682-686). We have added a pointer to the Methods in case of similar reader concerns. This explanation is also in the caption for Extended Data Figure 3 - promotion of this figure to the main text (as suggested below) makes this information more readily available to readers (L362-363).

The issue with reconstructing using only the first principal component is ameliorated by our use of methods that do not do this (composite plus scale, PCR, pairwise comparison). It is evident from the colour bars in Fig. 2 and Extended Data Fig. 7 that the reconstruction method does not bias the spectral signature. Therefore, the lack of power at low frequencies is unlikely to be due to the PCA-based loss of low frequencies.

Our use of only annually-resolved records is common with most annually-resolved climate mode reconstructions. Use of lower-resolution records may indeed introduce low-frequency variability into the reconstruction, but it is possible that this would be statistically artefactual.

7. Page 6, lines 134-136. The three datasets show different results. A new reanalysis dataset could be included to confirm this result. e.g., ECMWF reanalysis v5.

Response:

Unfortunately ERA5 only extends back to 1950, whereas our calibration interval was 1900-2000. Figure 3 compares all possible 20-year trends in the sub-ensembles calculated from each observational product with the observed 1992-2011 trend from that same observational product.

8. Page 6, lines 144-145. The three datasets show different results in the Extended Data Figure 4. How to get the consistent result?

Response: R32

The difference between the panels in Extended Data Fig. 4 (now Extended Data Fig. 8) is due to differences in the observational products, which are propagated to differences in the reconstruction ensemble members. We have added text explaining this to the figure caption (L1120-1121).

9. Page 8, lines 197-200. How to understand the strong SLP response but the SST response? The possible reason is suggested.

Response:

This is discussed in detail several paragraphs later, in the Discussion (L279-288).

10. Page 8, lines 204-205. How to know the PWC reconstruction has large internal variability? Which internal variability? Pacific Decadal Oscillation?

Response:

Internal variability does not necessarily need to be assigned to a particular decadal-mode - it may be inherent to the PWC. We have edited the wording for clarity (L234-235).

11. Page 9, lines 218-222. The Extended Data Fig. 9 shows the significant relationship between the GMST and the Δ SLP reconstruction. A sensitive experiment using the climate model is suggested to test this thermodynamical process.

Response:

As described in the response to Main Comment 3, this in-depth mechanistic analysis is beyond the scope of this paper (which has an empirical focus), but this work is well underway in our research group. Use of climate models to determine the mechanisms driving the changes we describe will take at least one additional paper, and likely more. As it is central to the Discussion, we have promoted Extended Data Fig. 9 to the main text (now Fig. 6).

12. Page 10, line 245. "one year in the CESM LME".

Response:

We have made the suggested change, accounting for the addition of PMIP3/4 models to this analysis (L276).

13. Page 10, line 264. "the first climate mode" should be replaced with "the first atmospheric mode", because the previous ENSO reconstruction is also a climate mode.

Response: R31

We are not sure which ENSO reconstruction the referee is referring to. The ENSO reconstruction from the Last Millennium Reanalysis derives uncertainty estimates using different model priors, but does not incorporate chronological uncertainty, or uncertainty arising from modern observational products. The Dätwyler et al. (2019) ENSO reconstruction also did not incorporate chronological uncertainty, or

uncertainty arising from modern observational products. We have edited the wording to make this clear (L295).

14. Pages 11-14. The reference format should be improved according to requirements of the journal.

Response:

We have checked through the re-generated reference list.

15. Page 15. It is interesting to show the results of the 2001-2021 CE period.

Response:

In the new Extended Data Figure 3, showing a close-up of the reconstruction calibration interval, we also now include data from the observational products to 2010 (the extent of the ERA20C data).

16. Page 16. Extended Data Figure 3 should be included in main Figure 2, because the frequency difference between pre-1850 and post-1850 is a novel result.

Response:

We have incorporated the original Extended Data Figure 3 into the main text Figure 2.

17. Page 18, line 438. "See Methods".

Response:

We have fixed this typographical error (L393).

18. Page 19. The size of volcanic eruptions in the climate model simulation should be consistent with the proxy reconstruction.

Response:

We have adjusted the parameters for the SEA performed on model simulations to instead match the SEA performed on the reconstruction (L752-762).

19. Page 20, lines 464-466. It is suggested to briefly explain how to screen proxy records. How these records were selected from 759 isotope records (Konecky et al., 2020)? It is well known that there are many proxy records associated with ENSO variability. How to screen them?

Response:

The initial screening was not for correlation with the target indices, but rather for resolution and temporal coverage. That is, records had to overlap with the calibration interval, and be at least annually resolved. This is described in Methods Section 1.2 (L475-478).

20. Page 21, lines 495-497. The ERA20C could be extended with ERA5 during the 2011-2021 CE period.

Response:

These two reanalysis products have different input data and underlying models. Unfortunately, it would therefore not be physically or statistically sound to append ERA5 data to ERA20C.

21. Page 21, lines 506-508. This condition is too strict for the non-dendro records because the glacier ice, lake sediment, and speleothem cannot satisfy annual resolution.

Response:

We do incorporate information from glacier ice, lake sediment, and speleothems (Table S1). The requirement for annually-resolved data is common amongst annually-resolved paleoclimate reconstructions.

22. Page 22, lines 515-516. The signal-forcing (volcanic and anthropogenic forcing) experiments are suggested to confirm the results.

Response:

We agree that this analysis is necessary and (very!) interesting, however it is beyond the scope of this paper. The work is underway in our research group, and we have added a statement to the concluding paragraph of the main text that such diagnoses are beyond the scope of this empirically-focussed paper (L296-298).

23. Page 24, line 571. "MATLAB".

Response: R36

We have made this correction (L534).

24. Page 24, lines 574-583. The wood records have no dating uncertainty.

Response:

Please see description of uncertainty in tree-ring chronologies in the response to Main Comment 1.

25. Pages 24-25, lines 592-612. Keeping only the first PC leads to systematic bias because PC1 mainly captures high-frequency while losing low-frequency signals (McShane and Wyner, 2011).

Response:

As stated in the response to Main Comment 1, any bias arising from the two PC1-based reconstruction methods is a component of methodological uncertainty. All six other methods will not suffer from this possible frequency bias.

26. Page 26, lines 631-633. This screening records are suggested to use in the other methods.

Response:

This greatly reduces the number of available records, and hence forms a sensitivity test - what is the trade-off between less reliance on teleconnections, and more records contributing information (hence buffering against potential non-stationary teleconnections)? The results are shown in Table S3 (now promoted to Extended Data Table 1).

27. Page 26, line 650. "MATLAB".

Response: R37

We have made this correction (L611).

28. Page 27, line 672. The reduction of error and the coefficient of efficiency (Cook et al., 2010) are suggested in the reconstruction validation.

Response:

Reduction of error and coefficient of efficiency tests are similar in calculation and interpretation. We have therefore replaced the binomial skill test with the reduction of error test (Extended Data Figs. 4-5, Extended Data Table 1, L636).

29. Page 28, lines 701-704. The correlation coefficients before 1600 CE in the Extended Data Figure 1b are too small to show the robustness of the Δ SLP signal.

Response:

We have clearly presented the decrease in reconstruction skill back through time. Accordingly, we have been careful throughout the manuscript to avoid making large claims based on the earlier section of the reconstruction. Rather, we have restricted the main points in the text to ‘big-ticket’ takeaways that are robust to data availability issues, and clearly stated this pre-1600 skill decrease in the text (L107-111).

30. Page 32, line 801. The signal forcing (volcanic eruption) experiment in the CESM LME is suggested to assess the PWC response to volcanic eruptions.

Response:

This assessment has been performed for Niño 3.4 by Dee et al (2020) <https://www.science.org/doi/10.1126/science.aax2000>. This paper is not focussed on volcanic eruptions in models, but rather on developing the reconstruction and assessing internal variability and external forcing. Nevertheless, in response to suggestions from other referees, we have added data from additional PMIP3/CMIP5 & PMIP4/CMIP6 models. We have also more clearly stated the scope of the paper, and our reasoning for including the model analyses (L208-211).

31. Page 34, lines 839. All codes and datasets should be archived in the publish database to help the reader reproduce results.

Response:

We now provide code allowing readers to reproduce the results. Code will be available from the following repository doi.org/10.5281/zenodo.7742761. {REDACTED}.

References

- Cook, E.R., Anchukaitis, K.J., Buckley, B.M., D'Arrigo, R., Jacoby, G.C., Wright, W.E., 2010. Asian monsoon failure and megadrought during the last millennium. *Science* 328, 486-489.
- Konecky, B.L., McKay, N.P., Churakova, O.V., Comas-Bru, L., Dassié, E.P., DeLong, K.L., Falster, G.M., Fischer, M.J., Jones, M.D., Jonkers, L., Kaufman, D.S., Leduc, G., Managave, S.R., Martrat, B., Opel, T., Orsi, A.J., Partin, J.W., Sayani, H.R., Thomas, E.K., Thompson, D.M., Tyler, J.J., Abram, N.J., Atwood, A.R., Cartapanis, O., Conroy, J.L., Curran, M.A., Dee, S.G., Deininger, M., Divine, D.V., Kern, Z., Porter, T.J., Stevenson, S.L., von Gunten, L., Iso2k Project, M., 2020. The Iso2k database: a global compilation of paleo- $\delta^{18}\text{O}$ and $\delta^2\text{H}$ records to aid understanding of Common Era climate. *Earth Syst. Sci. Data* 12, 2261-2288.
- McShane, B.B., Wyner, A.J., 2011. Rejoinder: A statistical analysis of multiple temperature proxies: Are reconstructions of surface temperatures over the last 1000 years reliable? *Ann. Appl. Stat.* 5, 99-123.

Referee #5 (Remarks to the Author):

Falster et al. use a synthesis of 54 globally distributed water isotope (e.g., $\delta^{18}\text{O}$) proxy records from corals, trees, speleothems, lake sediment, and glacial ice to reconstruct the strength of the Pacific Walker Circulation (PWC) from 1200-2000 CE. This study builds upon previous work in Falster et al. (2021) that found a strong imprint of the PWC in interannual precipitation $\delta^{18}\text{O}$. In this study, the target climate variable for the reconstruction is the trans-Pacific sea level pressure gradient (ΔSLP).

This study investigates how the PWC varied during the Common Era and its response to both volcanic and anthropogenic forcings (e.g., greenhouse gases and anthropogenic aerosols). The authors find that although the more recent PWC strengthening is anomalous (i.e., the recent trend falls within the positive tails of the full distributions), it is not outside the range of natural variability observed in the 800-year-long reconstruction. The recent trend could thus be attributed to internal variability. The high intrinsic variability in the PWC reconstruction is a notable result that highlights the importance of having a longer-term context when discussing recent trends in atmospheric circulation. Additionally, this work uses superposed epoch analysis, to identify a link between the strength of the PWC and volcanic forcing. This study finds that strong volcanic eruptions trigger an El Niño-like PWC weakening in the subsequent years, a relationship more clearly identified since the 1860s.

Overall, the ensemble-based approach to reconstructing ΔSLP is comprehensive and statistically sound. The authors' decisions to use 5 different statistical reconstruction techniques, 3 observational products as a training set, and a Monte Carlo-style approach for adding chronological uncertainty in the proxy records, addresses many of the most common uncertainties and challenges/issues in these types of paleoclimate studies. The study is well-written, the figures are well-presented, and I feel the results will be of broad interest to the climate community. Below, I provide some recommendations to further clarify the results and strengthen the paper, as well as some questions/comments to address in a future revision. These comments largely focus on the influence of seasonality in the PWC reconstruction, the relationship between the PWC and volcanic forcing, and the role of internal PWC variability.

Main Comments and Questions:

1. In this study, superposed epoch analysis is used to identify the relationship between PWC variability and volcanic forcing. The authors find that strong volcanic eruptions trigger an El Niño-like PWC weakening in subsequent years. This is a surprising result given how contentious the response of Pacific SST variability to volcanic forcing is in the literature.

A similar PWC weakening after large volcanic eruptions is identified in the fully forced ensemble members from the CESM Last Millennium Ensemble. That said, the simulated relationship between the strength of the PWC and volcanic forcing does not hold when 1% 'chronological uncertainty' is added to the climate model simulations (Fig. 5 G, H, I). As a reviewer, I found this to be a very intriguing result that was not thoroughly discussed in the text. L193-194 states that 1% 'chronological uncertainty' obscures the signal, but additional details explaining the importance of this result are not provided. A more thorough discussion of this result will improve the manuscript especially since a significant PWC-volcanic forcing relationship is identified in the proxy-inferred reconstructions in Fig. 1 that have accounted for chronological uncertainty. I am curious to hear the authors' comments about why the PWC-

volcanic forcing relationship may persist when adding chronological uncertainty in the proxy-inferred reconstructions, but not in the climate model simulations.

Response:

We suspect that this is because—unlike the proxy data, which have inherent chronological uncertainty—the model PWC response to explosive volcanism occurs in the ‘correct’ years. Hence the addition of chronological uncertainty only degrades the signal, whereas by incorporating chronological uncertainty into the reconstruction ensemble, we ensure at least some ensemble members have the response in the correct year. However, in response to comments from two other referees, we have removed these panels.

2. The colors in Fig. 1B shows the median correlation of each proxy records with the Δ SLP ensemble over the period in which that record contributed to the reconstruction. To clarify this calculation, was each individual proxy record correlated with the ensemble median for each reconstruction method (i.e., the 8 colored lines in Fig. 1A) over their common interval of overlap? Then the median of those 8 correlation coefficients was plotted in Fig. 1B? Some additional clarification in the main text, methods and/or the figure caption would be a helpful addition.

Response:

Each individual proxy record was correlated with the median of the **full** ensemble, over the period of the temporal sub-section that they contributed. We have clarified this in the figure caption (L328-330) and the relevant Methods section (L658-662).

I am curious why the strength of correlation with the reconstruction is shown in Fig. 1B as opposed to the correlation with the observational Δ SLP training index as in Extended Data Figure 10. As an alternative Fig. 1B, I recommend showing a condensed version of Extended Data Figure 10 (e.g., a single map including all the proxy records) in the main text so the reader can clearly identify how well-correlated the individual records are with instrumental Δ SLP. Although many of the proxy records are significantly correlated with instrumental Δ SLP over the calibration interval, the absolute magnitude of some of the correlation coefficients are quite low and worth noting.

Response:

We are showing two different things with these two figures - both of which we thought were relevant for readers. The map in Fig. 1B is more of an ‘overview’ figure, with an estimate of which records contributed ‘more’ versus ‘less’ to the reconstruction. In contrast, the maps in Extended Data Fig. 10 (now Extended Data Fig. 2) show, in more detail, the correlation of each record with the target index, as well as the combination of records contributing to each temporal subset. We could have reversed the metrics shown in these two figures, i.e., shown the target index correlations on a single map in Fig. 1, and then the record contributions split out by temporal subsection in Extended Data Fig. 2. However, we judged that showing the (arguably more important) index correlation information in a more detailed form would allow readers to better assess the strengths of the correlations, compared with condensing that information to a single map. Hence we feel that the figures are more appropriate in their current form.

3. I recommend including an additional supplementary figure that shows the time series of the three observational SLP products (HadSLP, ICOADS, and ERA20C) over their common 1900-present interval of overlap. The manuscript discusses that differences between the observational products impact the reconstruction skill, as mentioned in L71 and L95 and shown in Extended Data Fig. 1Ai. The observed time series are shown by the three black lines in Fig. 1A, but it is difficult to discern any differences. I

suggest including a zoomed in version of the observed Δ SLP time series that clearly differentiates the three products.

Response:

We have added this figure as a new Extended Data item (Extended Data Fig. 3).

As a related point, gridded observational products typically become more uncertain further back in time due to a reduced number of observations. This is well documented for SST observations (e.g., Fig. 3 of Deser et al. 2010). Are there any measures of uncertainty provided in the HadSLP and ICOADS products? Do the different gridded products have better-agreement during the more recent decades, compared to the early 20th century due to a denser network of observations?

Response:

That information is now available to readers from the new figure as suggested above. It quite clearly shows distinct periods of low inter-product agreement, and it was this that prompted us to include the observational data as a source of uncertainty in the reconstruction ensemble.

Reference: Deser, C., Alexander, M. A., Xie, S.-P. & Phillips, A. S. Sea surface temperature variability: Patterns and mechanisms. *Annu. Rev. Marine. Sci.* 2, 115–143 (2010).

4. Why is the composite plus scale method using only records without a known seasonal bias (CPSNs, pink curve in Fig. 1A) typically the most different compared to the other 7 reconstructions? It appears that the amplitude of the Δ SLP variations is often larger compared to the other methods. For example, there are notable differences during the mid 1500s, the late 1600s, and the 1700-1800s, in some cases the median CPSNs reconstruction shows Δ SLP changes of opposite sign to the other median reconstructions.

Do these results suggest that seasonality is indeed an important factor to consider when reconstructing the PWC? Or are the differences in the CPSnc reconstruction mostly attributed to the reduced number of proxy records available, and is thus more uncertain? Additional discussion about the impact of seasonality would strengthen the manuscript.

Response:

This is a very good point - as you say, there are multiple possible reasons for the differences between the CPSNs and other reconstructions. There are indeed fewer records, possibly resulting in a weaker signal, although the skill tests performed on an independent interval are not significantly different to the other CPS reconstructions (Extended Data Fig. 5). It is also possible that the seasonality of the records does indeed make a difference. This would be an interesting future avenue of research (although probably over a much shorter time period, or using climate model outputs, given the relative data sparsity). The discussion of reconstruction skill, incorporating a section discussing seasonality, has been promoted from the Supplementary Information to the Methods, and we have added additional discussion of this specific point (L853-857).

5. This study uses the CESM Last Millennium Ensemble (CESM LME) full forcing simulations to investigate the simulated response of the PWC to volcanic forcing. I am curious how the CESM LME control simulations (e.g., the 1850 or 850 control) could be used to provide context about the range of internal variability in the PWC. This would likely strengthen the study's conclusions about internal PWC variability.

For example, how does the variance of Δ SLP in a control simulation compare to the variance of the fully forced simulations? The trend analysis for the PWC reconstruction and the PDFs in Figure 3 are interesting and could presumably be repeated for the CESM LME. In the simulations, are there any significant 20-year trends, and/or changes in decadal/multi-decadal variability in Δ SLP that can arise completely in the absence of external forcing?

Response:

We agree that this would be an interesting analysis, but do not see that it would strengthen the conclusions of this paper—in part because there are existing papers analysing Δ SLP in climate models, but (until now) no empirically-based reconstruction to provide a ‘test’ of model accuracy. The model analysis was provided 1) as a ‘reader-accessible’ comparison with previous SST-based work (e.g. Dee et al, 2020), and 2) to show that potentially some of the previous model-proxy disagreement is because the volcanic response appears to be stronger in Δ SLP than in Niño 3.4. The model analyses are not a central or diagnostic component of the study, and nor do we feel that this paper is the right place for a detailed assessment of Δ SLP in the CESM LME. Similar experiments have already been performed in the CESM LME, assessing the probability of El Niño and La Niña events for post-eruption years, versus their probabilities in the pre-industrial control (Stevenson et al., 2016 *Journal of Climate* Table 2). Future work will focus on further assessing Δ SLP in climate models, in light of the features we have revealed with this reconstruction. We have added the following statement to the concluding paragraph of the main text, stating that such diagnoses are beyond the scope of this empirically-focussed paper: “*Although diagnosis of the dynamics underlying forced responses and intrinsic variability in the PWC was beyond the scope of this paper, the Δ SLP reconstructions provide the necessary observational foundation for such future investigations*” (L296-298).

6. L543-545 states that each reconstruction only retained proxy records significantly correlated with Δ SLP over the calibration window. There are then, three proxy data subsets, one for each observational product. Are there large differences between the three subsets of proxy records? For example, is one of the observational data products consistently better correlated with a larger number of records? Some additional justification for this approach would strengthen the manuscript.

Response:

There are not major differences in those three subsets - the number of records significantly correlated with each observational data product is the same. We have clarified our wording in the methods (L515-517).

Minor Comments:

- The figure caption of Extended Data Figure 10 states that the black outline denotes that the proxy record is significantly correlated with instrumental Δ SLP (Extended data, L110-111). In my version of the document, all the proxy records are outlined in black, but some have a darker black outline.

Response:

We have edited this figure to make the difference more obvious.

- Extended Data Figure 10 shows the spatial distribution of proxy records for different time intervals, but a time series would also provide helpful context. I imagine a similar figure to Figure 2b in the Konecky et al. (2020) Iso2k database paper, but only including the records selected for this study.

Response:

We have added a timeseries figure showing record temporal availability (Extended Data Fig. 1).

- L93: Does this refer to a specific supplementary table? What are the traditional (non-ensemble) reconstructions referred to here? I recommend providing the references.

Response:

This was a reference to a supplementary discussion of reconstruction skill, which has now been promoted to the Methods - the main text has been adjusted accordingly. The full references are provided in that discussion, but unfortunately we have hit the reference limit for the main text.

- Fig 2b caption: Indicate that a KS test was used to demonstrate that the pre- and post- industrial Δ SLP distributions are statistically significantly different. From visual inspection alone, it doesn't look like the two PDFs would be statistically different.

Response:

According to a K-S test, the two distributions *are* significantly different. However, we acknowledge that perhaps this was not the correct test for this situation i.e., with several thousand ensemble members, it is very easy to have a statistical difference between means. We have therefore re-done this analysis, but instead performing K-S tests on all **individual** ensemble members, and reporting the distribution of p-values i.e., the proportion of ensemble members where the pre- and post-industrial are significantly different (L667-670). We report this proportion in the main text (L132-133), and have added a boxplot of K-S test p-values to Fig. 2.

- How do the 1200-2000 CE reconstructions for each reconstruction technique compare to the full 'nested' reconstructions stitched together with the different temporal segments?

Response:

We have repeated several key analyses on a non-nested version of the reconstruction (1600-2000, using only records covering that full interval). One of these is now incorporated into the manuscript as the new Extended Data Figure 7 (L688-693), and the other is shown above in the response to Referee 3. We have also added text explaining that this does not appear to be a source of bias in the reconstructions (L128-130).

Reviewer Reports on the First Revision:

Referees' comments:

Referee #2 (Remarks to the Author):

Dear authors,

I would like to revisit my initial assessment of your manuscript's suitability for publication in Nature. After carefully reviewing your responses to my comments and those of other reviewers and the revisions you have made, I have reevaluated my initial judgment.

Your clarifications and the additional analyses you have incorporated, such as including data from eight additional PMIP3/4 models, strengthen the comparison and provide a better understanding of inter-model variability in the Δ SLP response to volcanic eruptions.

I appreciate your acknowledgement that the underlying physical explanation for the observed shift in PWC frequency is beyond the scope of this paper and that it is now a topic of ongoing research in your group. I understand that the main focus of your work is to provide empirical evidence of PWC changes and use model data to compare with previous work. Your effort to minimise uncertainties in the Δ SLP reconstruction using multiple statistical methods and datasets is commendable, as it strengthens the basis for studying the dynamics of PWC variability and its response to anthropogenic and volcanic forcings.

It is good to know that detailed investigations of the anthropogenic and other external forcings' influences on modelled PWC are currently underway in your group. I believe this will provide valuable insights into the complex interactions driving PWC variability.

Given these improvements and considering the potential impact of your reconstructed Δ SLP time series on future research, I am confident your paper is well-suited for publication in Nature. I hope you understand that my revised opinion is based on the strength of your responses and the revisions made to the manuscript.

I look forward to seeing your work published in Nature and its contribution to our understanding of PWC dynamics.

Referee #3 (Remarks to the Author):

The authors have addressed my comments, I am happy with the revised manuscript and therefore recommend its publication as it is.

Referee #4 (Remarks to the Author):

Dear authors of the manuscript "Forced changes in the Pacific Walker Circulation over the past millennium", I appreciate your diligent efforts in revising the manuscript and addressing my comments. The manuscript is now significantly clearer than before. I understand that the focus of this manuscript is to explore the Pacific Walker Circulation (PWC) variability using proxy records. The innovative aspect of this manuscript lies in utilizing the same indicator with different types of proxy records to reconstruct this circulation index. Analyzing the attribution of PWC variability trends during 1992-2011 CE from a paleoclimate perspective is an excellent example of "The past is the key to the present". Moreover, the significant application of this PWC reconstruction is to demonstrate the El Niño-like response to volcanic eruptions, which is also a hotspot issue. I recommend this manuscript is accepted for publication after a minor revision.

Minor comments:

1. How should one interpret the good negative correlation between PWC and ENSO during the instrumental period, but a more complicated relationship before the instrumental period, as shown in Figure 6c? Previous studies show the PWC is coupled with ENSO variability (e.g., (Rasmusson and Carpenter, 1982)).
2. Why do these records primarily reflect PWC variability but not other indicators? The authors cited previous studies to explain it, but it is still suggested to provide a more detailed explanation about "PWC-related changes in moisture source and transport length, and a PWC- or ENSO-driven 'amount effect' in tropical regions." In fact, some of these proxy records (e.g., the $\delta^{18}\text{O}$ records of coral and tree-ring) also show a strong correlation with the ENSO index during the instrumental period. Additionally, I agree that the dominant mode of observed global interannual precipitation may be strongly related to PWC, however, the $\delta^{18}\text{O}$ records from different proxies cannot be equated to precipitation $\delta^{18}\text{O}$. In other words, the climate significance of different oxygen isotope indicators is not uniform and clear (e.g., (Cheng et al., 2019; Shi et al., 2022; Xu et al., 2021)). It is suggested to clarify the climate significance of oxygen isotopes of different proxies.
3. Lines 116-118, how should this reason be understood? Shouldn't volcanic activity as a strong external forcings improve the consistency of results?
4. Lines 126-128, is there an issue when comparing two power spectra of sequences with different time periods? Different lengths of two sequences will result in different power spectra.
6. Since PWC may be related to Interdecadal Pacific Oscillation (IPO) variability, it is suggested to compare it with multiple Pacific Decadal Oscillation (PDO) reconstructions, as the IPO includes the PDO, and the PDO has a better expression from middle-low latitude proxies than just one IPO reconstruction from high-latitude proxies.
7. Solar activity, as a source of Earth's energy, should also be mentioned in the text.

Following the editor's requirement, I also assessed the response to referee #1.

Referee #1's primary concern is that the 1992-2011 CE period is not covered in the reconstruction (800-2000 CE). I believe this concern is insignificant, as there is a significant relationship between the reconstructed PWC and the instrumental PWC during the instrumental period. The authors have also included a new figure comparing them, which sufficiently addresses this concern. The second concern from Referee #1 is important. I am unable to understand the statement "changes in variance through time are not strongly affected by the variance correlation." I believe variance scaling should be applied to the instrumental period, not the entire time series, and then any variance changes before the instrumental period can be maintained relative to the variance of the instrumental data. Regardless, the non-nested version of the reconstruction does not affect the conclusion, which is sufficient to address this concern.

3. Except for the first two main comments, the following minor comments are revised according to the Referee #1's comments. All responses are acceptable to me.

References

- Cheng, H., Zhang, H., Zhao, J., Li, H., Ning, Y., Kathayat, G., 2019. Chinese stalagmite paleoclimate researches: A review and perspective. *Sci. China Earth Sci.* 62, 1489-1513.
- Rasmusson, E.M., Carpenter, T.H., 1982. Variations in tropical sea surface temperature and surface wind fields associated with the Southern Oscillation/El Niño. *Mon. Weather Rev.* 110, 354-384.
- Shi, F., Goosse, H., Li, J., Yin, Q., Ljungqvist, F.C., Lian, T., Sun, C., Wang, L., Wu, Z., Li, J., Zhao, S., Xu, C., Liu, W., Liu, T., Nakatsuka, T., Guo, Z., 2022. Interdecadal to multidecadal variability of East Asian Summer Monsoon over the past half millennium. *J. Geophys. Res. [Atmos.]* 127, e2022JD037260.
- Xu, C., Zhao, Q., An, W., Wang, S., Tan, N., Sano, M., Nakatsuka, T., Borhara, K., Guo, Z., 2021. Tree-ring oxygen isotope across monsoon Asia: Common signal and local influence. *Quat. Sci. Rev.*

Referee #5 (Remarks to the Author):

Summary:

Falster et al. use a network of water isotope records to reconstruct the strength of the Pacific Walker Circulation (PWC) from 1200-2000 CE. The main findings are 1) the recent (1992-2011 CE) PWC strengthening is anomalous but not outside the range of natural variability, 2) strong volcanic eruptions trigger an El Niño-like PWC weakening in subsequent years, and 3) there is not a clear relationship between PWC strength and global mean surface temperature (GMST).

A key strength of this study is how thoroughly the authors address many sources of uncertainty inherent in paleoclimate reconstructions. As mentioned in my first review, this study uses 5 statistical reconstruction techniques, 3 observational products as a training set, and a Monte Carlo-style approach for adding chronological uncertainty in the proxy records. This approach is comprehensive and addresses many of the most common uncertainties in paleoclimate reconstructions. The resulting Δ SLP reconstructions provides important context about the range of PWC variability and will likely support many follow-up studies that seek to better understand the mechanisms driving PWC variability and its response to external forcing. Overall, the analysis is novel and will of broad interest to readers interested in paleoclimate, tropical climate variability and change, and climate dynamics.

Review:

The authors did substantial work to address the comments from the 5 referees and the editor. This included restructuring the manuscript and extended data to improve readability, reducing the amount of material in the supporting information to only 1 table, clarifying the text (particularly in the methods), promoting the GMST analysis to the main text, and adding model data from PMIP3/PMIP4 Last Millennium simulations to Figure 5. The codes for the analyses were made privately available to the reviewers and a public DOI will be available upon publication.

In the revised manuscript and response to the referees, the authors clearly state that the goal of this study is to provide evidence of PWC changes through time as opposed to a holistic mechanistic understanding of why PWC responds to anthropogenic and volcanic forcings. Given the amount of effort that went into quantifying uncertainties, generating the Δ SLP reconstructions, and characterizing PWC changes through time (and its response to external forcing), I agree with the authors' assessment that the current scope of work is sufficiently comprehensive for publication.

The authors thoroughly addressed my 6 major comments as well as the minor comments/suggestions. They did this by clarifying text in the main text and the methods, adding Extended Data Fig. 3 (observed Δ SLP with the proxy-inferred reconstructions) and Extended Data Fig. 1 (proxy availability through time), and answering my questions about seasonality in the proxy records and uncertainty in the observed Δ SLP products. They also modified the statistical testing in Fig. 2D by performing K-S tests on all the individual ensemble members. They find that the pre- and post-1850 mean Δ SLP is not significantly different in 81% of the 4800 ensemble members. The boxplot of p-values in Fig. 2D is a welcome addition that nicely summarizes these findings. In response to my comments and those of Referee #3, the authors also repeated several analyses using a non-nested version of the Δ SLP reconstruction. I was also pleased to see the addition of the PMIP3/4 simulations in the superposed epoch analysis, and the general consistency of the results in Fig. 5.

In summary, the above additions and methodological changes strengthened the study and further supported the main conclusions. The layout changes, particularly the restructured methods, extended data, and supplementary information, also greatly improved the overall readability of the manuscript. The manuscript is well-written, and the analyses/figures remain high-quality.

Referee #6 (Remarks to the Author)

Original comments from the editor and referees are in black text; our responses are in **provided in red text**. **Reviewer notes on R1 comments and author responses are in blue text**, with those in my view requiring further rebuttal and revision in purple text.

Response:

We have revised the manuscript according to suggestions from the five referees, with additional layout changes as suggested above. All the referees' suggestions—including additional tests of statistical robustness—strengthened our original findings.

To summarise the major changes that we have made to the manuscript, as suggested by the referees:

- Incorporation of the original Extended Data Fig. 3 into main text Fig. 2 (spectral analyses performed separate on the pre- and post-industrial)
- Addition of data from PMIP3/4 models to Fig. 5 (Superposed Epoch Analysis performed on model data; originally just CESM, new figure has 8 additional models)
- Promotion of the original Extended Data Fig. 9 to main text Fig. 6, given particular interest in those point from several referees (comparison of Δ SLP with GMST)
- Addition of new Extended Data Figures, showing:
 - spectral analysis performed on non-nested reconstructions
 - agreement between instrumental and reconstructed Δ SLP in the calibration period
 - proxy data availability through time (including intervals of missing data)
 - relationships with published ENSO reconstructions
- Addition of 'reduction of error' skill test

- Provision of code in a Zenodo repository, allowing readers to replicate findings (repository is set up, and has a DOI, but is not yet public. {REDACTED})
- Promotion of the original Supplementary Tables 2 & 3 into the Extended Data

These revisions, along with additional explanations and clarifications in the text, address referee concerns regarding methodological details, and statistical robustness. The main text is now ~3600 words, and the Methods are now ~4600 words.

We also made many minor changes, including the requested edits to the text for clarity (particularly in the Methods), additional explanations throughout, and minor figure edits - these address the concerns of Referee #3 about clarity of presentation.

Regarding use of climate model simulations to provide additional mechanistic insight into the phenomena identified here: this extends well beyond the scope of this empirically-focussed manuscript. Although there were several excellent suggestions from referees as to follow-up analyses (many of which were already being undertaken by the authorship team or their students), these would fit better in one or more model-focussed manuscripts. We have clarified the logic behind our use of model simulations in the assessing the Δ SLP response to volcanic eruptions, and added the following statement in the concluding paragraph of the manuscript: *“Although diagnosis of the dynamics underlying forced responses and intrinsic variability in the PWC was beyond the scope of this paper, the Δ SLP reconstructions provide the necessary observational foundation for such future investigations”*.

Referees' comments:

Referee #1 (Remarks to the Author):

A. Summary of the key results

Falster et al.'s "Forced changes in the Pacific Walker Circulation over the past millennium" describes a highly original reconstruction and assessment of Walker Circulation variability using a network of long proxy records in the Iso2k network.

The chief conclusions of this analysis are: 1) the recent 1992-2011 trends in the Walker Circulation are not unprecedented over the last 800 years, 2) Volcanic eruptions prompt an observable Weaker Walker

response, 3). The relationship between global mean surface temperature and Walker Circulation is not as straightforward as many studies have suggested. The two important indices can vary considerably.

This manuscript will be of interest to readers of Nature who are interested in climate dynamics, paleoclimatology, and historical climate change. I recommend this manuscript for publication with minor revisions.

B. Originality and significance

This analysis is very novel and illustrates the utility of a new water isotope dataset (Konecky et al. 2020) to examine a long standing question in the scientific literature. This analysis does an admirable job of examining the uncertainty in their reconstructions. The authors examine the sensitivity of their method to dating errors, calibration dataset, temporal period of reconstruction and reconstruction methods—and clearly illustrate the sensitivity of their results to these choices. It is difficult to envision how the authors could have done more to test the robustness of their reconstruction to arbitrary methodological choices.

Response:

Thanks - we feel that this is one of the key strengths & novelties of the paper :).

As indicated specifically below, I think there are opportunities for independent validation exercise results to be further described and discussed.

This work would be very relevant to several fields. This work also serves to highlight the utility of water isotope data.

One suggestion for the authors is to amplify the fact that this analysis uses such a novel water isotope dataset. This is **really** exciting (and novel!) and I am afraid that this important nuance might be lost on readers who have never heard of the Iso2k project.

Response:

We have added text throughout the manuscript to emphasise the novelty of the water isotope dataset (and its use in paleoclimate reconstruction). In particular, at Line 71 we now state *“The reconstruction leverages the recently created Iso2k database, a highly innovative global synthesis of water isotope proxy records. Iso2k includes data from diverse archive types, and allows ready integration of water isotopic signals into palaeoclimate reconstructions.”*

OK.

C. Data and Methodology

The data used and methodology used in this manuscript are both very reasonable and very original. The new Iso2k dataset (Konecky et al. 2020) used heightens the originality and significance of this study.

D. Appropriate use of statistics

I did not attempt to repeat any statistical tests, but all tests and results used in the manuscript appear logical. All figures illustrate an admirable use of error bars and significance testing. Furthermore, the authors take supreme care to test the sensitivity of their results to reconstruction methods. As an example of this, a 1992-2011 was a key analysis period in their study (due to previously published work in the

scientific literature), but this period begins just after Pinatubo. The authors examined the sensitivity of this trend to the volcanic eruption by examining the 20 year trends after all volcanic eruptions and concluding no strong signal.

My largest concern with this study is that the 1992-2011 period is of chief importance to the analysis, but is a period not covered by the authors reconstruction (800-2000AD). The authors repeatedly compare the observed instrumental trend over the 1992-2011 period to the proxy reconstructed trends and draw major conclusions from this comparison. I understand that proxy records grow sparse after the year 2000, but it would be helpful to see just how well the reconstruction method compares with the instrumental observations over the 20th century. (Perhaps add a panel to Figure 3)? The authors state that there is high correlation between the two (Supplementary Table 2, $r \sim 0.8$), but it would be helpful to see the agreement between the reconstruction and all instrumental products over this important period. I think readers will question if the authors are indeed comparing apples to apples with the present discussion.

Response:

We have added the suggested figure (Extended Data. Fig. 3), showing correspondence between the reconstruction and observed Δ SLP over the 20th century. This also shows how observed Δ SLP continues past the end of the reconstructions.

OK. But also: define the colors in EDF3B in the caption (suggestion: use them to denote significance given effective degrees of freedom; also in other instances with similar presentation noted in specific comments below); put into labeling or caption the root mean square difference (RMSD) between ERA20C, ICOADS, HadSLP dSLP, and for the RMSD between reconstructed dSLP and 20th century observed dSLP. This will also meet the second largest concern by showing that the calibrated residual amplitude is no bigger than the mean amplitude difference between historical dSLP estimates. The authors could report the median across estimates for the different reconstruction methods in EDF3, and for the 20y trend in dSLP in Fig 3. See below for other instances in which significance tests are missing.

My second largest concern with the methodology used in the study regards the variance adjustment applied to the reconstructions to maintain the variance of the 20th century throughout the 800-2000 reconstruction interval. The motivations for this variance adjustment are highly logical—nearly all proxy reconstructions feature a loss in variance with time (as the available proxy fraction dwindles), but my concern lies in the documented change in ENSO variability over the last millennium and how that may influence the Walker Circulation reconstruction. Both Grothe et al. 2020 and Cobb et al. 2013 demonstrate an increase in Nino 3.4 variability over the last few decades. At the very least, the authors could comment on how the assumed 20th century variance may influence the results (and cite these studies).

Response:

When constructing the reconstruction methodology, we sought to balance true variance changes through time with artificial variance loss back through time due to chronological uncertainty and proxy-specific effects. We performed extensive sensitivity testing on this point. However, changes in variance through time are not strongly affected by the variance correction, as a variance scaling is applied to the *entire* timeseries. That is, any variance changes through time are maintained. We have edited the Methods text to make this clearer (L620-624).

Although I disagree with the reviewer about the legitimacy of variance adjustment, because it hides uncertainty arising from changes in data availability, I think the author's rebuttal is good. It would be better if it were performed in validation rather than calibration, in other words, comparison of reconstructed with observed dSLP for a period not used for calibrating the reconstruction. Are validation-period variance changes successfully reconstructed? This could be demonstrated by adding validation scores and coefficient of efficiency (CE) to Extended Data Table 1 for the 1951-2000 calibration/1900-1950 validation exercise. For CE, the difference is referencing to the validation rather than the calibration mean; this is relevant for assessing skill at decadal timescale mean changes, in this case.

Additionally, following related concerns from several referees about the temporal nesting methodology and how that might affect variance, we repeated two key analyses, using a non-nested version of the reconstruction, i.e., a reconstruction using the same set of proxies throughout. All conclusions remain robust in this context (see below), and we have added one of these analyses as a new Extended Data Figure (ED Fig. 7, L673-693).

Comparing Fig 2AB to EDF7A and considering EDF7B, and the interpretation that the significant reconstructed spectral power is at interannual frequencies, I agree.

But it seems that the Fig 2 results are sensitive to nesting. This is from comparing Fig 2C (1850-2000, nested) with EDF7D (1850-2000, non-nested). The proportions of ensemble members are roughly half as large as with the nested results, despite the fact that plotting proportions should adjust for different sample sizes, and despite the fact that the first nest is for 1860-2000, and despite the fact that the calibration interval is 1900-2000, or $\sim 2/3$ of the period of analysis. And the relative proportions within the interpreted 2-9yr periodicities are different. This needs some clarification and further analysis of validation results.

Grothe, Pamela R., Kim M. Cobb, Giovanni Liguori, Emanuele Di Lorenzo, Antonietta Capotondi, Yanbin Lu, Hai Cheng et al. "Enhanced El Niño–Southern oscillation variability in recent decades." *Geophysical Research Letters* 47, no. 7 (2020): e2019GL083906.

Cobb, Kim M., Niko Westphal, Hussein R. Sayani, Jordan T. Watson, Emanuele Di Lorenzo, H. Cheng, R. L. Edwards, and Christopher D. Charles. "Highly variable El Niño–southern oscillation throughout the Holocene." *Science* 339, no. 6115 (2013): 67-70.

E. Conclusions

The conclusions of this manuscript are well grounded (aside from the concern about the reconstruction comparisons with the instrumental record over the 1992-2011 period). The authors describe their conclusions in the context of present literature and their results are easily interpretable.

I did not personally try to replicate any of the analysis, but the authors are very clear about the methodology and data used, thus their results seem reproducible. Furthermore, the authors state that they will provide access to their reconstruction upon publication—and offer to provide code upon request.

Response:

We now provide all code necessary to replicate our findings. Code will be available from the following Zenodo repository: doi.org/10.5281/zenodo.7742761. {REDACTED}.

Good. The url exists and is a downloadable zipfile. I didn't want to compromise my anonymity by accessing it, and actually testing the code/data repository entry goes beyond the scope of the review Request remit I was given.

F. Suggested Improvements

I have major and minor suggestions for improvement and have listed them below.

Major Points

The first two points are discussed in the “Appropriate Use of Statistics” section, but I will briefly outline them again (apologies for repetition).

First: I worry about the comparison of the proxy reconstruction to the instrumental archives over the 1992-2011 period. I think that adding a figure to show the consistency of the reconstruction results to the HADSLP, ICOADS, and ERA20C reanalysis products over the 20th century would help abate the concern that it is not entirely appropriate to directly compare these two sources of data.

Response:

We have added this figure as a new Extended Data item (ED Fig. 3).

OK. See notes above, this could be improved, and validation results added and discussed. But changes in means across decades is not where the skill of the reconstruction lies (Fig 2).

Secondly: Adjusting the variance of the reconstruction to that observed in the 20th century causes some concern. I suggest that the authors comment that they acknowledge that they may be incorporating some bias with this approach.

Response: R40

We have added details and additional commentary on the variance adjustment to the Methods.

OK.

Finally: The authors include some discussion about the utility of examining Walker Circulation Variability with a few different variables, but set up a contrast between examining the SST component of Walker Circulation change vs the atmospheric/ SLP component (eg. Line 195, but also in a few other places). I think a sentence or two to clarify that discussion would be helpful to readers. Few studies examining the climate response to volcanic eruptions use raw SST, instead they use a “relative SST” and examine the Nino3.4 SST after subtracting the tropical mean (due to the expectation that volcanic aerosols will cause cooling globally and mask the tropical Pacific response, eg. Khodri et al. 2017, Predybaylo et al. 2020, Zuo et al. 2018, many others). This is virtually impossible when using single paleoclimate proxy record that only monitors conditions at a single location.

Response:

We have re-done the SEA on model SST in the Nino 3.4 region, but now using the relative SST as suggested (results shown in Fig. 5). We have added text describing this approach in the Methods (L764-768). We have also added additional discussion of this point to the main text (L284-285).

OK. But the description of the results at l. 274, "This is evident in both our reconstruction ensemble and climate model simulations, although the significant anomaly lasts ~three years in the reconstruction compared with one year in the model simulations." doesn't seem consistent with what's actually shown in those figures. The abstract doesn't reflect that very well either. Revise.

Khodri, Myriam, Takeshi Izumo, Jérôme Vialard, Serge Janicot, Christophe Cassou, Matthieu Lengaigne, Juliette Mignot et al. "Tropical explosive volcanic eruptions can trigger El Niño by cooling tropical Africa." *Nature communications* 8, no. 1 (2017): 1-13.

Predybaylo, Evgeniya, Georgiy Stenchikov, Andrew T. Wittenberg, and Sergey Osipov. "El Niño/Southern Oscillation response to low-latitude volcanic eruptions depends on ocean pre-conditions and eruption timing." *Communications Earth & Environment* 1, no. 1 (2020): 1-13.

Zuo, M., Man, W., Zhou, T., and Guo, Z.: Different impacts of northern, tropical, and southern volcanic eruptions on the tropical Pacific SST in the last millennium, *J. Climate*, 31, 6729–6744, <https://doi.org/10.1175/JCLI-D-17-0571.1>, 2018.

Minor Points

Line 22—"the 4800 member ensemble" phrase is just confusing without introducing what the ensemble represents. How about something like, "We use a new paleoproxy-derived 4800 member ensemble"?

Response:

We added 'paleoproxy-derived' to the introductory sentence (L22).

OK.

Line 40: A reference should consider adding to this discussion is the Seager et al. 2019 paper that illustrates some of this mismatch between observations and models may be entirely due to biases in climate models.

Response:

We have added this reference (L42).

OK.

Seager, Richard, Mark Cane, Naomi Henderson, Dong-Eun Lee, Ryan Abernathey, and Honghai Zhang. "Strengthening tropical Pacific zonal sea surface temperature gradient consistent with rising greenhouse gases." *Nature Climate Change* 9, no. 7 (2019): 517-522.

Line 50: What is your specific definition for Walker Circulation? Suggest that the authors say "hydroclimate (rather than SLP directly)" or something like that for clarity.

Response:

We have edited the text for clarity (L51-52).

Revise sentence at l 33 to clearly defined the PWC.

Line 53: Again, be clear with what this 4800 member ensemble entails

Response:

We added a brief statement introducing the sources of uncertainty that make up the ensemble (L56-58).

OK.

Line 54: As the 1992-2011 period has been mentioned, it would help to clarify why the reconstruction does not extend to year 2100

Response:

Assuming that the referee means 'extend to the year 2011', we added some clarifying text as to this choice in the Methods (L492-493). In short, it is because many of the proxy records stop around 2000, such that the proxy data availability would have been greatly decreased.

OK. But even better would have been to say how many or what fraction of the records extend to 2011. How many is it? If ~14, this is similar to what was available for 1860-2000, maybe worth the exercise.

Line 55: If specifically citing a study, its far less confusing for the reader if you write XX et al. 13 rather than just the superscripted number

Response:

We have changed all instances of this in the text.

OK.

Line 60: All readers would appreciate a little more information about "the dominant mode of global interannual precipitation $\delta 18O$ carries a strong imprint of ΔSLP ". Over what years? What was the proxy distribution? How do the authors define "strong imprint"? Perhaps a sentence or two more would help readers better appreciate the work you have put into this task!

Response:

We have added extra text to provide more information. Specifically, from L64: "*The dominant mode of observed global interannual precipitation $\delta 18O$ over 1982–2015 carries a strong imprint of ΔSLP , even though many individual records are not strongly correlated with ΔSLP . This imprint arises from multiple well-documented processes, including PWC-related changes in moisture source and transport length, and a PWC- or ENSO-driven 'amount effect' in tropical regions.*"

"strong imprint" remains undefined. Define it in the first revised sentence at l. 64.

Line 65: I would include a few more details about the proxy dataset in the main text. Are these records at least annually resolved? What types of proxies compose these records?

Response:

We have added more details about the proxy dataset to the main text, including more information about the Iso2k database (L67-74).

OK. the rebuttal is most reflected in EDFs 1,2; cite them here.

Lines 70-90: Absolutely love that you did all of this sensitivity testing—and so clearly! This is one of my favorite parts of the paper. I was curious how you went from 59 proxies to a 4800 member ensemble. Could you provide the reader with this detail? We have 5 reconstruction methods, 3 calibration products, and some sort of age adjustment? Am I missing anything?

Response:

It's one of our favorite parts too :). That information is provided in the Methods (Section 2.3.1.1) - we have added a pointer at this location to refer the reader to the Methods (L99-100), and edited the text to make this more clear to the reader.

At l. 96, if 'simulateBAM' is based on Comboul et al (2014), this reference should be cited and the specific "banded" added before "age-depth" on l. 96.

Comboul, M., Emile-Geay, J., Evans, M. N., Mirnateghi, N., Cobb, K. M., and Thompson, D. M.: A probabilistic model of chronological errors in layer-counted climate proxies: applications to annually banded coral archives, *Clim. Past*, 10, 825–841, <https://doi.org/10.5194/cp-10-825-2014>, 2014.

Line 95: How well do the instrumental products agree over the 1900-2022 interval?

Response:

This is an excellent question, and one that we spent a lot of time looking into at the start of this project. The short answer is ‘not as well as one might hope’, which is also the case for other gridded observational products (e.g., SST). A full discussion of this question is outside the scope of this paper, but to allow readers to assess this for themselves, we have added Extended Data Fig. 3 showing the 1900-2010 CE period (the limits of ERA20C), with the Δ SLP from a) the reconstruction ensemble, and b) the three instrumental products. We also state this in the text (L102).

OK, but see suggestions previously for adding information to this figure in support of uncertainty analysis.

Line 95: If you train the products on the 20th century instrumental record, aren't you forcing them to agree with the instrumental trends over the 20th century? Perhaps there is a way to check this influence -- I see that you did this in the Supplemental Materials! Great! Clearly referencing that additional analysis here would be helpful!

Response:

We have added this information to the main text (L104-105).

Incomplete; see notes prior and later on expanding discussion of validation results.

Line 99: “this decrease in skill back through time is a common but rarely emphasized feature of paleoclimate reconstructions” – I'm not sure what you mean here. Could you clarify?

Response:

We have edited the wording for clarity (L108-110) - *“this decrease in skill back through time is a challenge for most palaeoclimate reconstructions, due to decreased data coverage and increased chronological uncertainty”*.

I think it's better to base this on the results presented and simply say "this decrease in skill back through time is due to decreased data coverage and increased chronological uncertainty".

Line 105: Personally, I think some of this inter-method disagreement is really interesting. Why might these methods have strong disagreement over particular intervals? Changes in traditional covariance patterns between proxies and the Walker Circulation? Sanchez et al. 2021 documented similar disagreement in tropical Pacific reconstructions over the 1800-1850 periods when using coral-based paleo data assimilation and attributed the breakdown to different patterns of covariability during a period highly influenced by volcanic eruptions.

Response:

Agreed! We assume that the periods of lower/higher agreement are due at least in part to changes in the covariances between the proxies and the Walker circulation - that is one of the reasons we used the different reconstruction methods. Many thanks for the reference - we have added this comparison with Sanchez et al's work to the main text (L116-118).

Sanchez, Sara C., Gregory J. Hakim, and Casey P. Saenger. "Climate model teleconnection patterns govern the Niño-3.4 response to early nineteenth-century volcanism in coral-based data assimilation reconstructions." *Journal of Climate* 34, no. 5 (2021): 1863-1880.

I think the reviewer is confusing two aspects of covariance: covariance in the climate system, e.g. between a forcing and a response; and in the paleoclimatic reconstruction method, e.g. the use of covariance in a regression coefficient estimate. In the data assimilation of Sanchez et al (2021), these two things are represented independently by a climate model and a data model, respectively.

In the authors' work, nonlocal regression statistically conflates climate and data models. The reconstruction methods used in the manuscript all require covariance estimates and the stationarity assumption, so the author's response is insufficient here.

I would revise the rebuttal and text to say that covariation within the climate system may vary with forcing (Sanchez et al 2021), and reconstructed differences may arise from changes in covariance in the climate system or differences between reconstruction algorithms and how they treat bias (e.g. the distinction of opPCA vs FiPCA). A citation for this could be

J Gergis, R Neukom, AJE Gallant, DJ Karoly, Australasian temperature reconstructions spanning the last millennium, *Journal of Climate* 29 (15), 5365-5392. 2016, doi: 10.1175/JCLI-D-13-00781.1

or another reference of their choice that uses a variety of statistical methods to perform a nonlocal statistical paleoclimate reconstruction of a target time series.

It's also unclear from the Methods how the number of retained PCs for PCR was determined without a stopping criterion.

Minor, but helpful: at l. 83 it would be better to use the existing abbreviation for pairwise comparison based paleoclimate reconstruction, PaiCo, instead of PC, which is too similar to PC and PCR.

Line 107/Figure 2: How do some of these frequencies have zero power? Are the authors only plotting the periodicities with significant power? If so, be sure to say it! It is not immediately clear to the reader.

Response: R11

Yes, that is correct - we are only showing periodicities with significant power. This is stated in the figure caption, but we have clarified it in the text.

OK.

Line 114-115: Really happy the authors put in the sentence “ the difference between the two distributions is within the range calculated between any other two intervals of equivalent length...”! I was curious if the PDFs had a more distinct difference if you compared slightly different intervals, eg. 1200-1900 vs 1900-2000?

Response:

Prior to submitting the paper, we thoroughly investigated this exact point, including stepping that ‘split point’ progressively further toward the present to see the change in the differences between the two distributions (see below). We agree that this is a really interesting analysis, however we do not have space (in text or figures) to cover all the nuance of this particular point, and instead distilled this analysis into just the information relevant to our question of whether there is a detectable anthropogenic influence on the PWC.

Additionally, we re-did the statistical analysis addressing this point such that the results are more intuitive (Fig. 2d, L132-134), and added the option for readers to play around with similar tests using the provided code.

OK, but what could the 1900-2000 calibration period contribute to this problem?

Line 122: Could also cite the Seager et al. 2019 paper (mentioned earlier) here

Response:

We have added this citation (L141).

OK.

Line 123: Could potentially discuss tropical Pacific Decadal Variability (eg. Power et al. 2021) here

Response:

We have added some discussion of the Pacific decadal variability in the paragraph above (L124-126), and also added Extended Data Fig. 6b comparing our reconstruction with Vance et al.'s (2022) ice core-based reconstruction of the Interdecadal Pacific Oscillation.

Power, Scott, Matthieu Lengaigne, Antonietta Capotondi, Myriam Khodri, Jérôme Vialard, Beyrem Jebri, Eric Guilyardi et al. "Decadal climate variability in the tropical Pacific: Characteristics, causes, predictability, and prospects." *Science* 374, no. 6563 (2021): eaay9165.

OK. But in EF6b, the significance of the correlations should be indicated and the text revised accordingly; for example if a correlation is not significant it should not be described as a weak correlation; or else its p-value should be estimated.

Line 142-146: This is so cool! I had been worried about the potential influence of Pinatubo on the results given the importance of the 1992-2011 period, but I feel much more comfortable given this analysis. I am still a little worried about the potential apples to oranges comparison as detailed in the Major Points section.

Response:

Comparison of observed or modelled modern/future changes to a paleoclimate reconstruction is reasonably common. In this particular case, we argue that the demonstrated strong agreement between the reconstruction and the observations in the calibration period means that this comparison is valid. To make this more clear, we have added Extended Data Fig. 3 zoomed in on this period of overlap, where the reader can more clearly see the close correspondence between the observations and the reconstruction.

As for EF6B, EF3B ought to be controlled for statistical significance. Here I expect all but a few of these correlations over the calibration period are significant, and might indicate that the differences between the reconstruction techniques, which are all related to one another, are not.

Line 149: I again am wondering about the resolution of the proxies considered in this network. Are they all annual or sub annual?

Response:

Yes, they are all sub-annually or annually resolved. This is mentioned in the Methods, and we have now added it to the main text (L68).

OK.

Line 177: “reassigned major eruption”—this is great!

;))

Line 165: how large is this trend?

Response:

At the suggestion of Referee 3, we have added a second axis to the panels of Fig. 4, showing a random sample of the ensemble PWC response to volcanic eruptions. This shows the magnitude of the response.

OK.

Line 172: If specifically referencing a study, please use the name of the study and then superscript it instead of just using the superscripted numbers

Response:

We have changed all instances of this in the text.

OK.

Line 190-192: Consider re-writing this sentence for clarity “if all volcanic eruptions with...” I’m not totally sure I follow what you are saying.

Response:

We have re-written this paragraph for better clarity (and also added PMIP3/4 models to this analysis) - L216-220 “*When applying the above SEA approach to the CESMI LME (using the 25 strongest eruptions; Methods), 9 of the 13 CESMI LME ensemble members produce a significant negative Δ SLP anomaly the year following a volcanic eruption (Fig. 5a), with Δ SLP anomaly magnitudes similar to those occurring during an average El Niño event*”.

OK, but see next note.

Line 191: How is significance defined?

Response:

Significance is defined the same way as for the reconstruction. We have added the citation of the method we used to determine significance to make this clear (L218).

Too cryptic a response. Explain in rebuttal what the 'double-bootstrap' method of Rao et al (2019) works and how it's used, because you spend some time explaining later in rebuttal and revised text how it might identify significant warmings prior to volcanic forcing.

Line 195: Consider writing “Fewer LME ensemble members” to help the readers stay on point as you also call your proxy reconstruction an ensemble

Response:

We have made the suggested edit (L223).

OK.

Line 218: Again, this is so cool!

Response:

We have promoted the associated Extended Data figure to the main text to emphasise this point (now Fig. 6).

I think the authors mean Fig 5, as referenced in the text at and near l. 218? OK - except that the significance test is again important here, and needs description in rebuttal and I think in the text.

Line 225: Why pCO₂ instead of CO₂? I think CO₂ is more intuitive for readers

Response:

We have removed the rho in both instances (L258, L260).

OK. rho? p? I leave this up to the copy editors.

Line 248: Did the authors remove the tropical mean from this? See major point for more discussion

Response:

We have re-done this analysis, now assessing relative SSTs in all cases (L764-768, L284-285).

Hard to evaluate without seeing a side-by-side comparison of original and revised results. Did the interpretation change? Please explain in rebuttal if and how the interpretation changed as a result.

Line 259: This manuscript describes a very original and exciting analysis. However, I don't think that the authors have emphasized just how original this source of data is! Many scientists and working groups are interested in finding uses for water isotope data—and this is a great illustration! If there is space, I would definitely hammer on this point a little more.

Response: R18

We have added text to remind readers how useful water isotope proxy data is. For example, at line 290 *“Finally, our novel use of water isotope proxy data to reconstruct atmospheric variability, including explicit incorporation of uncertainty from the training dataset, reconstruction method, and age-depth models....”*.

OK but IMHO unnecessary.

Figure 2B. I would make two boxes to indicate just where your delta SLP index comes from

Response:

We have added the two boxes to Figure 1.

OK. This is valuable because it shows that not a single observation comes from the target areas. Worth noting that this suggests the use of a nonlocal reconstruction approach, and why the Iso2k network is nonetheless expected to produce skillful reconstructions for dSLP from these regions. See also note below.

Figure 3. These distributions are very gaussian. Is that expected? How well do these indices agree with the reconstruction?

Response:

The distributions include data from all 4800 ensemble members. Given the large number of samples, we expect the distributions to tend to a normal distribution. When examining skill scores, both individual ensemble members (Extended Data Fig. 4a) and the reconstruction ensemble mean (Extended Data Fig. 3b) agree very well with the target indices. With the new Extended Data Fig. 3, readers can now see this for themselves.

I suspect the reason the reconstructions are Gaussian is because (a) the observations are gaussian or have been standardized, and (b) the reconstruction methods, being covariance-based, impose this. For the skill scores (EF4a, EF3b) this suggests the differences between results using reconstruction methods and different target SLP fields are not very important. It also suggests more analysis and testing using independent validation experiments, e.g. 1951-2000 calibration, reporting the 1900-1950 validation scores in EDT1, and other tests, e.g. with independent subsets of proxies (I suspect independent reconstruction targets are not available.)

Figure 4: In many of these plots, it looks as though there is a consistent Stronger Walker state observed two years prior to the eruptions in all periods. This is wild! It is quite large in subsets B, C, and D. It might be worth commenting on this unexpected result.

Response:

We have added text with a proposed explanation for this results (L195-197).

This further illustrates the need for a detailed description of the significance testing approach, and for more reliance on independent validation.

Figure 5A/B/C: How large are these SLP anomalies relative to a traditional El Niño/ La Niña Event?

Response:

They are of very similar magnitude: Δ SLP anomalies during El Niño events in the CESM-LME range from around -0.8 to -4 hPa. We have added that information to the main text (L219-220).

OK, no way for me to evaluate.

Figure 5D/E/F: Have you subtracted the tropical mean from these Niño 3.4 SST estimations? Most of the studies that look at the ENSO response do remove the global tropical mean from SSTs (something very difficult to do with a single proxy record)

Response:

We have re-done this analysis, first subtracting the tropical mean from the Niño 3.4 SST anomalies. We have also added text explaining and discussing this point (L764-768, L284-285). Many thanks for pointing out this error in our original calculations.

OK. Please add the same level of explanation in the text to the Rao et al (2019) based significance testing.

Figure 5G/H/I: Love that you included this! So cool!

Line 498: It would be great if you included a table with each proxy record and cite the individual authors with the database. Also, what resolution are these proxies?

Response:

This table is unfortunately too large to be included in the Extended Data, but is available (including authors of each individual dataset) as supplementary information. All records are annually or sub-annually resolved, and we have added this information to the main text (L68).

OK; I don't have the SI.

Line 509: "Model simulations" instead of "Model data"

Response: R19

We have made the suggested change (L479).

OK.

Line 529—great! I don't think that I got this from the main text. Do you include a table of the skill?

Response:

That table was in the supplementary materials, but has now been promoted to Extended Data Table 1.

If I read the EDT1 caption correctly, this doesn't include 1900-1950 validation scores, but it should: please add, discuss if necessary the presence or absence of evidence for artificial skill.

Line 535: Do you use a January-December year or a tropical (May-April) year when converting records to annual resolution? It's very important to be clear on what was done.

Response:

We used a calendar year (Jan-Dec) to match the other proxy data. We have added this clarification to the text (L508).

OK: it would have been better to use a tropical year definition, but this may not make much difference. Put the resolution or averaging for the records in question into the appropriate location in the SI.

Line 645: If referencing the study by name, please include the author's name, not just the superscript number.

Response:

We have changed all instances of this in the text.

OK.

Line 656: Matching the SLP variance to that observed over the twentieth century could be a big problem. I discuss this in the major points section.

Response:

This is a valid concern, and one that we spent a lot of time investigating. The method we used is a commonly-used approach for variance scaling in palaeoclimate reconstructions. Additionally, variance changes through time are preserved, and we have added text to the Methods expanding on this point (L623-624).

OK; previously assessed.

Line 741: the years 1992-2011 are superscripted

Response:

We have fixed this formatting error.

OK.

Line 743: just mention the authors and then superscript if directly mentioning a study.

Response:

We have changed all instances of this in the text.

OK.

Line 826: Suggest adding the of the method here-- eg. PAGES2k GMST (#66)

Response:

We added the in-text reference to PAGES 2k (L778).

OK.

G. References

The authors did a good job of including relevant, recent references. However, there were a few papers that merit mention in this discussion (mentioned previously in this review).

Response:

Many thanks for these excellent suggestions. We are limited by the 50-reference limit, but have added most of the below references, in place of some of the references originally included.

OK.

Seager, Richard, Mark Cane, Naomi Henderson, Dong-Eun Lee, Ryan Abernathy, and Honghai Zhang. "Strengthening tropical Pacific zonal sea surface temperature gradient consistent with rising greenhouse gases." *Nature Climate Change* 9, no. 7 (2019): 517-522.

Power, Scott, Matthieu Lengaigne, Antonietta Capotondi, Myriam Khodri, Jérôme Vialard, Beyrem Jebri, Eric Guilyardi et al. "Decadal climate variability in the tropical Pacific: Characteristics, causes, predictability, and prospects." *Science* 374, no. 6563 (2021): eaay9165.

Sanchez, Sara C., Gregory J. Hakim, and Casey P. Saenger. "Climate model teleconnection patterns govern the Niño-3.4 response to early nineteenth-century volcanism in coral-based data assimilation reconstructions." *Journal of Climate* 34, no. 5 (2021): 1863-1880.

Grothe, Pamela R., Kim M. Cobb, Giovanni Liguori, Emanuele Di Lorenzo, Antonietta Capotondi, Yanbin Lu, Hai Cheng et al. "Enhanced El Niño–Southern oscillation variability in recent decades." *Geophysical Research Letters* 47, no. 7 (2020): e2019GL083906.

Cobb, Kim M., Niko Westphal, Hussein R. Sayani, Jordan T. Watson, Emanuele Di Lorenzo, H. Cheng, R. L. Edwards, and Christopher D. Charles. "Highly variable El Niño–southern oscillation throughout the Holocene." *Science* 339, no. 6115 (2013): 67-70.

Khodri, Myriam, Takeshi Izumo, Jérôme Vialard, Serge Janicot, Christophe Cassou, Matthieu Lengaigne, Juliette Mignot et al. "Tropical explosive volcanic eruptions can trigger El Niño by cooling tropical Africa." *Nature communications* 8, no. 1 (2017): 1-13.

H. Clarity and Context

There are a few places where the authors could improve the clarity of their manuscript to help the readers understand the novelty of the work and significance of their results. I have generally commented on this in the Suggested Improvements Section.

However, there are a few points that would add a lot of clarity to the manuscript in general :

Firstly, A more in depth description of the proxies that make up the reconstruction. Table S1 cites the Iso2k database, but does not directly reference the individual authors who made many of these proxy datasets. This is a small point, but many paleo people that aren't familiar with the ISO2k database would get much more out of this table if you directly wrote "Bagnato et al. 2005 (14)" instead of #14 and the

ISO2k ID number. I'm also a little confused about the "unknown" resolution or seasonality. Does this mean variable resolution?

Response:

We have added more informative citations to the table. For seasonality, 'unknown' refers to records where the authors of the primary dataset did not claim any particular seasonality for their data, but also did not explicitly state that the record represents an annually integrated signal. We have added clarifying text to the table caption.

OK. I don't have Table S1, cannot comment further.

Secondly, I think that it would help the readers if the authors included their specific definition of Walker Circulation Variability in the main text (presently included in the methods). Suggestion that the authors add two boxes to Figure 1 to highlight the regions where SLP anomalies are considered. It'd really help the clarity if this was directly addressed this early on (I think early on in the manuscript the authors clarify that their index isn't related to hydroclimate or SST, but don't say exactly what it is). I also think that adding a sentence to describe just how these proxies mechanistically relate to Walker Circulation Variability would be extremely helpful for readers unfamiliar with the Falster et al. Journal of Climate paper.

Response:

We have added the index reference boxes to the map in Figure 1. We also added text summarising the main findings of the Falster et al. JClim paper, in terms of how water isotopes relate to the Walker circulation (L64-67).

This needs revision and more explanation. The reviewer stated it well: " adding a sentence to describe just how these proxies mechanistically relate to Walker Circulation Variability would be extremely helpful for readers unfamiliar with the Falster et al. Journal of Climate paper." As noted previously, this is important, as all the proxies are nonlocal to the target dSLP regions indicated by the black boxes.

Author Rebuttals to First Revision:

Response:

We have addressed the residual concerns of Referee 4, as well as additional comments from Referee 6, and the Editor's requirements.

In summary, we have:

- Made all formatting changes requested by the Editor
- Added text more clearly describing the validation tests (L105-108, 989-993)
- Added the coefficient of efficiency test to Extended Data Table 1
- Provided more details about how the confidence intervals around the Superposed Epoch Analysis are calculated, and the impact that this has on the results (particularly those shown in Fig. 4)
 - Detailed explanation provided in this rebuttal, as well as addition of extra text to the Methods (L205-209, 730-732)
- Edited the captions of Extended Data Fig. 5 and Extended Data Table 1, to make it clear that these items show skill scores calculated using independent calibration (1951-2000) and validation (1900-1950) intervals
- Added more statistical details to the caption of Extended Data Fig. 3
- On all relevant figures and captions, clearly stated if correlations are significant or not (e.g. Extended Data Figs. 3,6)
- Added more detail to the main text about how we are able to use the Iso2k water isotope proxy network to skillfully reconstruct Δ SLP (L64-71, 79-80)
- Clarified our proposed explanation of the period of low inter-method agreement at the start of the 19th century, with respect to the similar period of low inter-method agreement in Sanchez et al. (2021) (L124-128)
- Changed the abbreviation for 'Pairwise Comparison' from 'PC' to 'PaiCo' throughout

The main text is now ~ 3800 words, and the Methods are now ~4700 words. Please see below for detailed point-by-point responses to Referee comments.

Referees' comments:

Referee #2 (Remarks to the Author):

Dear authors,

I would like to revisit my initial assessment of your manuscript's suitability for publication in Nature. After carefully reviewing your responses to my comments and those of other reviewers and the revisions you have made, I have reevaluated my initial judgment.

Your clarifications and the additional analyses you have incorporated, such as including data from eight additional PMIP3/4 models, strengthen the comparison and provide a better understanding of inter-model variability in the Δ SLP response to volcanic eruptions.

I appreciate your acknowledgement that the underlying physical explanation for the observed shift in PWC frequency is beyond the scope of this paper and that it is now a topic of ongoing research in your group. I understand that the main focus of your work is to provide empirical evidence of PWC changes and use model data to compare with previous work. Your effort to minimise uncertainties in the Δ SLP reconstruction using multiple statistical methods and datasets is commendable, as it strengthens the basis for studying the dynamics of PWC variability and its response to anthropogenic and volcanic forcings.

It is good to know that detailed investigations of the anthropogenic and other external forcings' influences on modelled PWC are currently underway in your group. I believe this will provide valuable insights into the complex interactions driving PWC variability.

Given these improvements and considering the potential impact of your reconstructed Δ SLP time series on future research, I am confident your paper is well-suited for publication in Nature. I hope you understand that my revised opinion is based on the strength of your responses and the revisions made to the manuscript.

I look forward to seeing your work published in Nature and its contribution to our understanding of PWC dynamics.

Referee #3 (Remarks to the Author):

The authors have addressed my comments, I am happy with the revised manuscript and therefore recommend it's publication as it is.

Referee #4 (Remarks to the Author):

Dear authors of the manuscript "Forced changes in the Pacific Walker Circulation over the past millennium", I appreciate your diligent efforts in revising the manuscript and addressing my comments. The manuscript is now significantly clearer than before. I understand that the focus of this manuscript is to explore the Pacific Walker Circulation (PWC) variability using proxy records. The innovative aspect of this manuscript lies in utilizing the same indicator with different types of proxy records to reconstruct this circulation index. Analyzing the attribution of PWC variability trends during 1992-2011 CE from a paleoclimate perspective is an excellent example of "The past is the key to the present". Moreover, the significant application of this PWC reconstruction is to demonstrate the El Niño-like response to volcanic eruptions, which is also a hotspot issue. I recommend this manuscript is accepted for publication after a minor revision.

Minor comments:

1. How should one interpret the good negative correlation between PWC and ENSO during the instrumental period, but a more complicated relationship before the instrumental period, as shown in Figure 6c? Previous studies show the PWC is coupled with ENSO variability (e.g., (Rasmusson and Carpenter, 1982)).

Response:

Unfortunately, we expect that this is at least partly due to decreasing reconstruction skill back through time. For example, ENSO reconstructions derived from a single site are particularly susceptible to non-stationary teleconnections (e.g. Batehup et al., 2015). This is evident from Extended Data Fig. 6d, which demonstrates that the ENSO reconstructions themselves are poorly correlated.

The degree to which the changing correlations between PWC and ENSO is due to a) changing relationship between the two, and b) decreased reconstruction skill is difficult to assess. However, we are currently

attempting to investigate this point via detailed comparisons with the Last Millennium Reanalysis (LMR; Tardif et al., 2019). The LMR is a data-assimilation based ensemble reconstruction, using CCSM4 for the model prior. Our investigation includes analysis of the temporal relationships between the two reconstruction ensembles, and comparison of spectral character, as well as supporting analyses using climate models. This detailed investigation is beyond the scope of this particular paper, but will be the topic of an upcoming publication.

References

Batehup, R., McGregor, S., and Gallant, A. J. E.: The influence of non-stationary teleconnections on palaeoclimate reconstructions of ENSO variance using a pseudoproxy framework, *Clim. Past*, 11, 1733–1749, <https://doi.org/10.5194/cp-11-1733-2015>, 2015.

Tardif, R., G. J. Hakim, W. A. Perkins, K. A. Horlick, M. P. Erb, J. Emile-Geay, D. M. Anderson, E. J. Steig, and D. Noone, 2019: Last Millennium Reanalysis with an expanded proxy database and seasonal proxy modeling. *Clim. Past*, 15, 1251-1273, doi:10.5194/cp-15-1251-2019.

2. Why do these records primarily reflect PWC variability but not other indicators? The authors cited previous studies to explain it, but it is still suggested to provide a more detailed explanation about "PWC-related changes in moisture source and transport length, and a PWC- or ENSO-driven 'amount effect' in tropical regions." In fact, some of these proxy records (e.g., the $\delta^{18}\text{O}$ records of coral and tree-ring) also show a strong correlation with the ENSO index during the instrumental period.

Additionally, I agree that the dominant mode of observed global interannual precipitation may be strongly related to PWC, however, the $\delta^{18}\text{O}$ records from different proxies cannot be equated to precipitation $\delta^{18}\text{O}$. In other words, the climate significance of different oxygen isotope indicators is not uniform and clear (e.g., (Cheng et al., 2019; Shi et al., 2022; Xu et al., 2021)). It is suggested to clarify the climate significance of oxygen isotopes of different proxies.

Response:

The referee may have slightly misunderstood the sentence at Line 64 - Falster et al. (2021) find a correlation between the dominant mode of precipitation $\delta^{18}\text{O}$ and the PWC (not precipitation amount). We have re-written the start of this paragraph to avoid this confusion (L64-71). At the site level, in some cases, the relationship between the PWC and precipitation $\delta^{18}\text{O}$ is a result of the amount effect, but in others it is due to changes in e.g. moisture source and/or transport pathway. But also, as the referee states, proxy $\delta^{18}\text{O}$ does not always directly reflect precipitation $\delta^{18}\text{O}$. A detailed description of the interpretation of different proxies is outside the scope of this manuscript (those details are available in Konecky et al, 2020), but we have added a sentence at Line 79-80 stating that this is an additional complication.

References

Konecky, B. L., et al.: The Iso2k database: a global compilation of paleo- $\delta^{18}\text{O}$ and $\delta^2\text{H}$ records to aid understanding of Common Era climate, *Earth Syst. Sci. Data*, 12, 2261–2288, 2020

3. Lines 116-118, how should this reason be understood? Shouldn't volcanic activity as a strong external forcings improve the consistency of results?

Response:

Sanchez et al. (2021) use data assimilation methods to reconstruct ENSO variability. They find poor inter-method agreement across 1809 to 1830, and attribute this to a 'period of intense volcanism'. They conclude that the poor agreement is largely due to a cluster of proxies located in the South Pacific Convergence Zone driving

SST covariance patterns associated with ENSO to produce unrealistically warm SST in the Niño 3.4 region after volcanic eruptions. This is not generic to all eruptions, but rather due to a combination of a) closely-spaced eruptions, and b) the geographical distribution of proxy data during this interval.

In the case of our reconstruction, the 1800-2000 reconstruction segment includes a coral $\delta^{18}\text{O}$ record in the South Pacific Convergence Zone region, which is strongly anti-correlated with the PWC, and which did not contribute to previous segments (Extended Data Fig. 2). It is possible that the same combination of a) closely-spaced eruptions, and b) the geographical distribution of proxy data had disparate influences on the way in which each of our methods reconstructs ΔSLP (depending on the particular covariance structures). This is one of the reasons we chose a multi-method reconstruction approach - hopefully the 'correct' ΔSLP falls somewhere within the ensemble's range.

We have edited the text to better outline this reasoning (L124-128).

References

Sanchez, S. C., G. J. Hakim, and C. P. Saenger, 2021: Climate Model Teleconnection Patterns Govern the Niño-3.4 Response to Early Nineteenth-Century Volcanism in Coral-Based Data Assimilation Reconstructions. *J. Climate*, 34, 1863–1880

4. Lines 126-128, is there an issue when comparing two power spectra of sequences with different time periods? Different lengths of two sequences will result in different power spectra.

Response:

We accounted for this by comparing industrial-era (1850-2000) spectra with the distribution of spectral power in all possible 150-year periods prior to 1850, still showing the proportion of ensemble members with power in each period. This is described in the Methods (Section 2.6.1). We refer readers to the Methods on Line 139.

6. Since PWC may be related to Interdecadal Pacific Oscillation (IPO) variability, it is suggested to compare it with multiple Pacific Decadal Oscillation (PDO) reconstructions, as the IPO includes the PDO, and the PDO has a better expression from middle-low latitude proxies than just one IPO reconstruction from high-latitude proxies.

Response:

We performed our comparisons with the IPO rather than the PDO for two reasons: 1) it has a more geographically-extensive footprint, and 2) Vance et al. (2022) is the most recently-published publicly-available reconstruction of decadal-scale Pacific variability covering the 1200-2000 interval. Additionally, Henley et al. (2011) determined that given the high correlation between PDO and IPO timeseries, they are essentially indistinguishable on palaeoclimate timescales.

We agree that reconstruction of the IPO from a single high-latitude proxy is sub-optimal (e.g. Batehup et al., 2015). However, Wise (2015) and Henley (2017) demonstrated that published reconstructions of the PDO and IPO have low inter-reconstruction coherence prior to the 20th century (e.g. Henley (2017) Table 2). Therefore, comparing our PWC reconstruction to several PDO/IPO indices is unlikely to provide information that is more climatically meaningful than what is currently shown in Extended Data Fig. 6.

We also agree with the Referee that the relationship between the PWC and low-frequency modes of variability is worth investigating further. However this remains a challenge, given the lack of coherence between existing

PDO/IPO reconstructions. Our PWC reconstruction ensemble will be publicly-available, facilitating comparison with future IPO/PDO reconstructions as they become available.

References

Vance, T.R., Kiem, A.S., Jong, L.M. et al. Pacific decadal variability over the last 2000 years and implications for climatic risk. *Commun Earth Environ* 3, 33 (2022).

Henley, B. J., M. A. Thyer, G. Kuczera, and S. W. Franks (2011), Climate-informed stochastic hydrological modeling: Incorporating decadal-scale variability using paleo data, *Water Resources Research*.

Batehup, R., McGregor, S., and Gallant, A. J. E.: The influence of non-stationary teleconnections on palaeoclimate reconstructions of ENSO variance using a pseudoproxy framework, *Clim. Past*, 11, 1733–1749, <https://doi.org/10.5194/cp-11-1733-2015>, 2015.

Wise, E.K. (2015), Tropical Pacific and Northern Hemisphere influences on the coherence of Pacific Decadal Oscillation reconstructions. *Int. J. Climatol.*, 35: 154-160.

Henley, B. (2017), Pacific decadal climate variability: Indices, patterns and tropical-extratropical interactions. *Global and Planetary Change*, Volume 155, Pages 42-55.

7. Solar activity, as a source of Earth's energy, should also be mentioned in the text.

Response:

We have added a statement about the possible influence of solar variability at Line 134.

Following the editor's requirement, I also assessed the response to referee #1.

Referee #1's primary concern is that the 1992-2011 CE period is not covered in the reconstruction (800-2000 CE). I believe this concern is insignificant, as there is a significant relationship between the reconstructed PWC and the instrumental PWC during the instrumental period. The authors have also included a new figure comparing them, which **sufficiently addresses this concern.**

The second concern from Referee #1 is important. I am unable to understand the statement "changes in variance through time are not strongly affected by the variance correlation." I believe variance scaling should be applied to the instrumental period, not the entire time series, and then any variance changes before the instrumental period can be maintained relative to the variance of the instrumental data. Regardless, **the non-nested version of the reconstruction does not affect the conclusion, which is sufficient to address this concern.**

Response:

During the variance scaling, changes in variance throughout the reconstruction are maintained. That is, if the variance in the instrumental period is changed by a certain factor, then that same factor is applied across the entire time period. Hence, variance in the calibration period of the reconstruction matches variance in the instrumental data over the same period, then any changes back through time are also maintained.

However, as Referee 4 states, our main results are reflected in the non-nested version of the reconstruction, which addresses Referee 1's original concern.

3. Except for the first two main comments, the following minor comments are revised according to the Referee #1's comments. **All responses are acceptable to me.**

References

- Cheng, H., Zhang, H., Zhao, J., Li, H., Ning, Y., Kathayat, G., 2019. Chinese stalagmite paleoclimate researches: A review and perspective. *Sci. China Earth Sci.* 62, 1489-1513.
- Rasmusson, E.M., Carpenter, T.H., 1982. Variations in tropical sea surface temperature and surface wind fields associated with the Southern Oscillation/El Niño. *Mon. Weather Rev.* 110, 354-384.
- Shi, F., Goosse, H., Li, J., Yin, Q., Ljungqvist, F.C., Lian, T., Sun, C., Wang, L., Wu, Z., Li, J., Zhao, S., Xu, C., Liu, W., Liu, T., Nakatsuka, T., Guo, Z., 2022. Interdecadal to multidecadal variability of East Asian Summer Monsoon over the past half millennium. *J. Geophys. Res. [Atmos.]* 127, e2022JD037260.
- Xu, C., Zhao, Q., An, W., Wang, S., Tan, N., Sano, M., Nakatsuka, T., Borhara, K., Guo, Z., 2021. Tree-ring oxygen isotope across monsoon Asia: Common signal and local influence. *Quat. Sci. Rev.* 269, 107156.

Referee #5 (Remarks to the Author):

Summary:

Falster et al. use a network of water isotope records to reconstruct the strength of the Pacific Walker Circulation (PWC) from 1200-2000 CE. The main findings are 1) the recent (1992-2011 CE) PWC strengthening is anomalous but not outside the range of natural variability, 2) strong volcanic eruptions trigger an El Niño-like PWC weakening in subsequent years, and 3) there is not a clear relationship between PWC strength and global mean surface temperature (GMST).

A key strength of this study is how thoroughly the authors address many sources of uncertainty inherent in paleoclimate reconstructions. As mentioned in my first review, this study uses 5 statistical reconstruction techniques, 3 observational products as a training set, and a Monte Carlo-style approach for adding chronological uncertainty in the proxy records. This approach is comprehensive and addresses many of the most common uncertainties in paleoclimate reconstructions. The resulting Δ SLP reconstructions provides important context about the range of PWC variability and will likely support many follow-up studies that seek to better understand the mechanisms driving PWC variability and its response to external forcing. Overall, the analysis is novel and will of broad interest to readers interested in paleoclimate, tropical climate variability and change, and climate dynamics.

Review:

The authors did substantial work to address the comments from the 5 referees and the editor. This included restructuring the manuscript and extended data to improve readability, reducing the amount of material in the supporting information to only 1 table, clarifying the text (particularly in the methods), promoting the GMST analysis to the main text, and adding model data from PMIP3/PMIP4 Last Millennium simulations to Figure 5. The codes for the analyses were made privately available to the reviewers and a public DOI will be available upon publication.

In the revised manuscript and response to the referees, the authors clearly state that the goal of this study is to provide evidence of PWC changes through time as opposed to a holistic mechanistic understanding of why PWC responds to anthropogenic and volcanic forcings. Given the amount of effort that went into quantifying

uncertainties, generating the Δ SLP reconstructions, and characterizing PWC changes through time (and its response to external forcing), I agree with the authors' assessment that the current scope of work is sufficiently comprehensive for publication.

The authors thoroughly addressed my 6 major comments as well as the minor comments/suggestions. They did this by clarifying text in the main text and the methods, adding Extended Data Fig. 3 (observed Δ SLP with the proxy-inferred reconstructions) and Extended Data Fig. 1 (proxy availability through time), and answering my questions about seasonality in the proxy records and uncertainty in the observed Δ SLP products. They also modified the statistical testing in Fig. 2D by performing K-S tests on all the individual ensemble members. They find that the pre- and post-1850 mean Δ SLP is not significantly different in 81% of the 4800 ensemble members. The boxplot of p-values in Fig. 2D is a welcome addition that nicely summarizes these findings. In response to my comments and those of Referee #3, the authors also repeated several analyses using a non-nested version of the Δ SLP reconstruction. I was also pleased to see the addition of the PMIP3/4 simulations in the superposed epoch analysis, and the general consistency of the results in Fig. 5.

In summary, the above additions and methodological changes strengthened the study and further supported the main conclusions. The layout changes, particularly the restructured methods, extended data, and supplementary information, also greatly improved the overall readability of the manuscript. The manuscript is well-written, and the analyses/figures remain high-quality.

Referee #6 (Remarks to the Author):

Original comments from Referee 1 are in black text; *author responses to original Referee 1 comments are in italicised red text*. Reviewer 6 notes on Referee 1 comments and author responses are in blue text, with those in Referee 6's view requiring further rebuttal and revision in purple text (as per the Referee's convention). Author responses to Referee 6 are in green text.

A. Summary of the key results

Falster et al.'s "Forced changes in the Pacific Walker Circulation over the past millennium" describes a highly original reconstruction and assessment of Walker Circulation variability using a network of long proxy records in the Iso2k network.

The chief conclusions of this analysis are: 1) the recent 1992-2011 trends in the Walker Circulation are not unprecedented over the last 800 years, 2) Volcanic eruptions prompt an observable Weaker Walker response, 3). The relationship between global mean surface temperature and Walker Circulation is not as straightforward as many studies have suggested. The two important indices can vary considerably.

This manuscript will be of interest to readers of Nature who are interested in climate dynamics, paleoclimatology, and historical climate change. I recommend this manuscript for publication with minor revisions.

B. Originality and significance

This analysis is very novel and illustrates the utility of a new water isotope dataset (Konecky et al. 2020) to examine a long standing question in the scientific literature. This analysis does an admirable job of examining the uncertainty in their reconstructions. The authors examine the sensitivity of their method to dating errors, calibration dataset, temporal period of reconstruction and reconstruction methods—and clearly illustrate the

sensitivity of their results to these choices. It is difficult to envision how the authors could have done more to test the robustness of their reconstruction to arbitrary methodological choices.

Response:

Thanks - we feel that this is one of the key strengths & novelties of the paper :).

As indicated specifically below, I think there are opportunities for independent validation exercise results to be further described and discussed.

Response:

We have addressed Referee 6's specific concerns as outlined below. Many concerns about the independent validation are addressed by additional explanatory text, as well as clearer figure captions for:

- Extended Data Fig. 5, which shows results for validation tests performed on an independent interval (calibration 1951-2000, validation 1900-1950), for all individual ensemble members
- Extended Data Table 1, which shows results for validation tests performed on an independent interval in the third column

This work would be very relevant to several fields. This work also serves to highlight the utility of water isotope data.

One suggestion for the authors is to amplify the fact that this analysis uses such a novel water isotope dataset. This is **really** exciting (and novel!) and I am afraid that this important nuance might be lost on readers who have never heard of the Iso2k project.

Response:

We have added text throughout the manuscript to emphasise the novelty of the water isotope dataset (and its use in paleoclimate reconstruction). In particular, at Line 71 we now state "The reconstruction leverages the recently created Iso2k database, an innovative global synthesis of water isotope proxy records. Iso2k includes data from diverse archive types, and allows ready integration of water isotopic signals into palaeoclimate reconstructions."

OK.

C. Data and Methodology

The data used and methodology used in this manuscript are both very reasonable and very original. The new Iso2k dataset (Konecky et al. 2020) used heightens the originality and significance of this study.

D. Appropriate use of statistics

I did not attempt to repeat any statistical tests, but all tests and results used in the manuscript appear logical. All figures illustrate an admirable use of error bars and significance testing. Furthermore, the authors take supreme care to test the sensitivity of their results to reconstruction methods. As an example of this, a 1992-2011 was a key analysis period in their study (due to previously published work in the scientific literature),

but this period begins just after Pinatubo. The authors examined the sensitivity of this trend to the volcanic eruption by examining the 20 year trends after all volcanic eruptions and concluding no strong signal.

My largest concern with this study is that the 1992-2011 period is of chief importance to the analysis, but is a period not covered by the authors reconstruction (800-2000AD). The authors repeatedly compare the observed instrumental trend over the 1992-2011 period to the proxy reconstructed trends and draw major conclusions from this comparison. I understand that proxy records grow sparse after the year 2000, but it would be helpful to see just how well the reconstruction method compares with the instrumental observations over the 20th century. (Perhaps add a panel to Figure 3)? The authors state that there is high correlation between the two (Supplementary Table 2, $r \sim 0.8$), but it would be helpful to see the agreement between the reconstruction and all instrumental products over this important period. I think readers will question if the authors are indeed comparing apples to apples with the present discussion.

Response:

We have added the suggested figure (Extended Data Fig. 3), showing correspondence between the reconstruction and observed Δ SLP over the 20th century. This also shows how observed Δ SLP continues past the end of the reconstructions.

OK. But also: define the colors in EDF3B in the caption (suggestion: use them to denote significance given effective degrees of freedom; also in other instances with similar presentation noted in specific comments below); put into labeling or caption the root mean square difference (RMSD) between ERA20C, ICOADS, HadSLP dSLP, and for the RMSD between reconstructed dSLP and 20th century observed dSLP. This will also meet the second largest concern by showing that the calibrated residual amplitude is no bigger than the mean amplitude difference between historical dSLP estimates. The authors could report the median across estimates for the different reconstruction methods in EDF3, and for the 20y trend in dSLP in Fig 3. See below for other instances in which significance tests are missing.

Response:

All correlations presented in Extended Data Fig. 3b are significant ($p < 0.05$) when accounting for serial autocorrelation; we have now stated this in the figure caption. We have also added a colourbar, as well as the mean estimates for the different reconstruction methods.

The RMSE between reconstructed and 20th century observed Δ SLP is shown in Extended Data Figs. 4-5 (individual ensemble members), and Extended Data Table 1 (ensemble median; 0.27). We have now added that information to the Extended Data Fig. 3 caption, along with mean RMSE between Δ SLP calculated from the three gridded products (0.3).

My second largest concern with the methodology used in the study regards the variance adjustment applied to the reconstructions to maintain the variance of the 20th century throughout the 800-2000 reconstruction interval. The motivations for this variance adjustment are highly logical—nearly all proxy reconstructions feature a loss in variance with time (as the available proxy fraction dwindles), but my concern lies in the documented change in ENSO variability over the last millennium and how that may influence the Walker Circulation reconstruction. Both Grothe et al. 2020 and Cobb et al. 2013 demonstrate an increase in Nino 3.4 variability over the last few decades. At the very least, the authors could comment on how the assumed 20th century variance may influence the results (and cite these studies).

Response:

When constructing the reconstruction methodology, we sought to balance true variance changes through time with artificial variance loss back through time due to chronological uncertainty and proxy-specific effects. We performed extensive sensitivity testing on this point. However, changes in variance through time are not strongly affected by the variance correction, as a variance scaling is applied to the entire timeseries. That is, any variance changes through time are maintained. We have edited the Methods text to make this clearer (L620-624).

Although I disagree with the reviewer about the legitimacy of variance adjustment, because it hides uncertainty arising from changes in data availability, I think the author's rebuttal is good. It would be better if it were performed in validation rather than calibration, in other words, comparison of reconstructed with observed dSLP for a period not used for calibrating the reconstruction. Are validation-period variance changes successfully reconstructed? This could be demonstrated by adding validation scores and coefficient of efficiency (CE) to Extended Data Table 1 for the 1951-2000 calibration/1900-1950 validation exercise. For CE, the difference is referencing to the validation rather than the calibration mean; this is relevant for assessing skill at decadal timescale mean changes, in this case.

Response:

Response Extended Data Fig. 5 and Extended Data Table 1 (third column) show skills scores calculated on independent validation intervals. We have clarified that in the relevant captions. Extended Data Table 1 in particular demonstrates that validation-period variance in the reconstruction ensemble median is successfully reconstructed. We have added CE scores to Extended Data Table 1 as suggested.

Additionally, following related concerns from several referees about the temporal nesting methodology and how that might affect variance, we repeated two key analyses, using a non-nested version of the reconstruction, i.e., a reconstruction using the same set of proxies throughout. All conclusions remain robust in this context (see below), and we have added one of these analyses as a new Extended Data Figure (ED Fig. 7, L673-693).

Comparing Fig 2AB to EDF7A and considering EDF7B, and the interpretation that the significant reconstructed spectral power is at interannual frequencies, I agree.

But it seems that the Fig 2 results are sensitive to nesting. This is from comparing Fig 2C (1850-2000, nested) with EDF7D (1850-2000, non-nested). The proportions of ensemble members are roughly half as large as with the nested results, despite the fact that plotting proportions should adjust for different sample sizes, and despite the fact that the first nest is for 1860-2000, and despite the fact that the calibration interval is 1900-2000, or $\sim 2/3$ of the period of analysis. And the relative proportions within the interpreted 2-9yr periodicities are different. This needs some clarification and further analysis of validation results.

Response:

Although the 1) overall proportion of ensemble members with significant periodicities is lower, and 2) the relative proportions are slightly different, the main overall finding from Extended Data Fig. 7c-d (industrial-era shift to lower-frequency variability) is unchanged. Additionally, following the Referee's concern, we checked over the code. In re-inspecting this, we found a small error (using a 99% significance cutoff instead of 95% as per the main text), and have re-made the figure. Extended Data Fig. 7c-d are now more similar to Fig. 2b-c,

which makes more sense given the similarity of Extended Data Fig. 7a to Fig. 2a. Nevertheless, the proportions remain slightly different, and we have explicitly stated this difference in the main text (L140-142).

The other main finding from Extended Data Fig. 7—that there is minimal power at multi-decadal frequencies, and this is unaffected by the nesting approach—also remains robust, despite the relative proportions at each periodicity not being identical to Fig. 2a.

E. Conclusions

The conclusions of this manuscript are well grounded (aside from the concern about the reconstruction parisons with the instrumental record over the 1992-2011 period). The authors describe their conclusions in the context of present literature and their results are easily interpretable.

I did not personally try to replicate any of the analysis, but the authors are very clear about the methodology and data used, thus their results seem reproducible. Furthermore, the authors state that they will provide access to their reconstruction upon publication—and offer to provide code upon request.

Response:

We now provide all code necessary to replicate our findings. Code will be available from the following Zenodo repository: doi.org/10.5281/zenodo.7742761. {REDACTED}.

Good. The url exists and is a downloadable zipfile. I didn't want to compromise my anonymity by accessing it, and actually testing the code/data repository entry goes beyond the scope of the review Request remit I was given.

F. Suggested Improvements

I have major and minor suggestions for improvement and have listed them below. Major Points

The first two points are discussed in the “Appropriate Use of Statistics” section, but I will briefly outline them again (apologies for repetition).

First: I worry about the comparison of the proxy reconstruction to the instrumental archives over the 1992-2011 period. I think that adding a figure to show the consistency of the reconstruction results to the HADSLP, ICOADS, and ERA20C reanalysis products over the 20th century would help abate the concern that it is not entirely appropriate to directly compare these two sources of data.

Response:

We have added this figure as a new Extended Data item (ED Fig. 3).

OK. See notes above, this could be improved, and validation results added and discussed. But changes in means across decades is not where the skill of the reconstruction lies (Fig 2).

Response:

We have more clearly captioned the Extended Data figures reporting validation skill test results (calibration and validation on independent intervals; Extended Data Fig. 5 and Extended Data Table 1 column 3). We have

added the CE test to Extended Data Table 1, and also made the suggested changes to Extended Data Fig. 3. The validation results are discussed in the Methods ('Assessment of reconstruction skill' section), but we have now added to the main text describing the validation test (L105-108, 112-113).

Secondly: Adjusting the variance of the reconstruction to that observed in the 20th century causes some concern. I suggest that the authors comment that they acknowledge that they may be incorporating some bias with this approach.

Response:

We have added details and additional commentary on the variance adjustment to the Methods.

OK.

Finally: The authors include some discussion about the utility of examining Walker Circulation Variability with a few different variables, but set up a contrast between examining the SST component of Walker Circulation change vs the atmospheric/ SLP component (eg. Line 195, but also in a few other places). I think a sentence or two to clarify that discussion would be helpful to readers. Few studies examining the climate response to volcanic eruptions use raw SST, instead they use a "relative SST" and examine the Nino3.4 SST after subtracting the tropical mean (due to the expectation that volcanic aerosols will cause cooling globally and mask the tropical Pacific response, eg. Khodri et al. 2017, Predybaylo et al. 2020, Zuo et al. 2018, many others). This is virtually impossible when using single paleoclimate proxy record that only monitors conditions at a single location.

Response:

We have re-done the SEA on model SST in the Nino 3.4 region, but now using the relative SST as suggested (results shown in Fig. 5). We have added text describing this approach in the Methods (L764- 768). We have also added additional discussion of this point to the main text (L284-285).

OK. But the description of the results at l. 274, "This is evident in both our reconstruction ensemble and climate model simulations, although the significant anomaly lasts ~three years in the reconstruction compared with one year in the model simulations." doesn't seem consistent with what's actually shown in those figures. The abstract doesn't reflect that very well either. Revise.

Response:

We have revised the text to better describe what is shown in Figs. 4 and 5 (L285-289). We also point out to the reader that the negative anomalies at -1 and +3 years (Fig. 4d) are likely due at least in part to the chronological uncertainty incorporated into the reconstructions, which 'smears' the response out in time (L205-207 - more detailed provided below).

Minor Points

Line 22—"the 4800 member ensemble" phrase is just confusing without introducing what the ensemble represents. How about something like, "We use a new paleoproxy-derived 4800 member ensemble"

Response:

We added 'paleoproxy-derived' to the introductory sentence (L22). OK.

Line 40: A reference should consider adding to this discussion is the Seager et al. 2019 paper that illustrates some of this mismatch between observations and models may be entirely due to biases in climate models.

Response:

We have added this reference (L42). OK.

Line 50: What is your specific definition for Walker Circulation? Suggest that the authors say “hydroclimate (rather than SLP directly)” or something like that for clarity.

Response:

We have edited the text for clarity (L51-52).

Revise sentence at l 33 to clearly defined the PWC.

Response:

We avoided including a specific metric of the PWC in the first sentence, as there are multiple ways of quantifying PWC strength (e.g. Kosovelj & Zapoltnik 2023); we used Δ SLP as it provides the most ‘accurate’ representation whilst also allowing the longest possible calibration interval. We state later (L58-59) that we used Δ SLP to quantify the PWC.

Nevertheless, we have added a more specific definition of the PWC at Line 31-33. The two sentences in place of the original first sentence now read: “*The Pacific Walker Circulation (PWC) is the zonal component of atmospheric circulation over the tropical Pacific. The PWC can be characterised by a sea level pressure gradient across the equatorial Pacific, with deep convection over the Indo-Pacific warm pool, subsidence over the equatorial eastern Pacific, upper tropospheric westerlies, and surface easterlies (the Pacific trade winds).*”

References

Kosovelj, K.; Zapoltnik, Ž. Indices of Pacific Walker Circulation Strength. *Atmosphere* 2023, 14, 397.

Line 53: Again, be clear with what this 4800 member ensemble entails

Response:

We added a brief statement introducing the sources of uncertainty that make up the ensemble (L56-58). OK.

Line 54: As the 1992-2011 period has been mentioned, it would help to clarify why the reconstruction does not extend to year 2100

Response:

Assuming that the referee means ‘extend to the year 2011’, we added some clarifying text as to this choice in the Methods (L492-493). In short, it is because many of the proxy records stop around 2000, such that the proxy data availability would have been greatly decreased.

OK. But even better would have been to say how many or what fraction of the records extend to 2011. How many is it? If ~14, this is similar to what was available for 1860-2000, maybe worth the exercise.

Response:

Eight records extend to 2011 (~14%). We could have reconstructed past 2000, by dropping some of the records that only have data to 2000 for the most recent subset. However, we chose to maintain the 2000 cutoff such that there is more consistency across the different subsets. Prior to calculating the reconstructions, we performed

extensive sensitivity testing around this point, i.e., the optimal start and end years to balance having as long a reconstruction as possible while including the largest amount of proxy data.

Line 55: If specifically citing a study, its far less confusing for the reader if you write XX et al. 13 rather than just the superscripted number

Response:

We have changed all instances of this in the text.

OK.

Line 60: All readers would appreciate a little more information about “the dominant mode of global interannual precipitation $\delta 18\text{O}$ carries a strong imprint of delta SLP”. Over what years? What was the proxy distribution? How do the authors define “strong imprint”? Perhaps a sentence or two more would help readers better appreciate the work you have put into this task!

Response:

We have added extra text to provide more information. Specifically, from L64: “The dominant mode of observed global interannual precipitation $\delta 18\text{O}$ over 1982–2015 carries a strong imprint of ΔSLP , even though many individual records are not strongly correlated with ΔSLP . This imprint arises from multiple well-documented processes, including PWC-related changes in moisture source and transport length, and a PWC- or ENSO-driven ‘amount effect’ in tropical regions.”

"strong imprint" remains undefined. Define it in the first revised sentence at l. 64.

Response:

We have edited the paragraph to better define ‘strong imprint’, and provide more information about the theoretical link between water isotopes and the PWC (L64-71).

Line 65: I would include a few more details about the proxy dataset in the main text. Are these records at least annually resolved? What types of proxies compose these records?

Response:

We have added more details about the proxy dataset to the main text, including more information about the Iso2k database (L67-74).

OK. the rebuttal is most reflected in EDFs 1,2; cite them here.

Response:

We have added a citation of Extended Data Figs. 1-2 at Line 76.

Lines 70-90: Absolutely love that you did all of this sensitivity testing—and so clearly! This is one of my favorite parts of the paper. I was curious how you went from 59 proxies to a 4800 member ensemble. Could you provide the reader with this detail? We have 5 reconstruction methods, 3 calibration products, and some sort of age adjustment? Am I missing anything?

Response:

It’s one of our favorite parts too :). That information is provided in the Methods (Section 2.3.1.1) - we have added a pointer at this location to refer the reader to the Methods (L99-100), and edited the text to make this more clear to the reader.

At l. 96, if 'simulateBAM' is based on Comboul et al (2014), this reference should be cited and the specific "banded" added before "age-depth" on l. 96.

Comboul, M., Emile-Geay, J., Evans, M. N., Mirnateghi, N., Cobb, K. M., and Thompson, D. M.: A probabilistic model of chronological errors in layer-counted climate proxies: applications to annually banded coral archives, *Clim. Past*, 10, 825–841, <https://doi.org/10.5194/cp-10-825-2014>, 2014.

Response:

We added this citation at Line 103, and added the word 'banded' as requested (L102).

Line 95: How well do the instrumental products agree over the 1900-2022 interval?

Response:

This is an excellent question, and one that we spent a lot of time looking into at the start of this project. The short answer is 'not as well as one might hope', which is also the case for other gridded observational products (e.g., SST). A full discussion of this question is outside the scope of this paper, but to allow readers to assess this for themselves, we have added Extended Data Fig. 3 showing the 1900-2010 CE period (the limits of ERA20C), with the ΔSLP from a) the reconstruction ensemble, and b) the three instrumental products. We also state this in the text (L102).

OK, but see suggestions previously for adding information to this figure in support of uncertainty analysis.

Response:

We have added the suggested information to Extended Data Fig. 3 as outlined above.

Line 95: If you train the products on the 20th century instrumental record, aren't you forcing them to agree with the instrumental trends over the 20th century? Perhaps there is a way to check this influence

-- I see that you did this in the Supplemental Materials! Great! Clearly referencing that additional analysis here would be helpful!

Response:

We have added this information to the main text (L104-105).

Incomplete; see notes prior and later on expanding discussion of validation results.

Response:

Extended Data Fig. 5 and Extended Data Table 1 (third column) show results of skill tests performed using independent calibration and validation intervals. We have clarified that in the two captions, and also clearly stated this in the main text so the reader is not left with the same concerns (L105-108).

Line 99: "this decrease in skill back through time is a common but rarely emphasized feature of paleoclimate reconstructions" – I'm not sure what you mean here. Could you clarify?

Response:

We have edited the wording for clarity (L108-110) - "this decrease in skill back through time is a challenge for most palaeoclimate reconstructions, due to decreased data coverage and increased chronological uncertainty".

I think it's better to base this on the results presented and simply say "this decrease in skill back through time is due to decreased data coverage and increased chronological uncertainty".

Response:

We have made the suggested change (L116-117).

Line 105: Personally, I think some of this inter-method disagreement is really interesting. Why might these methods have strong disagreement over particular intervals? Changes in traditional covariance patterns between proxies and the Walker Circulation? Sanchez et al. 2021 documented similar disagreement in tropical Pacific reconstructions over the 1800-1850 periods when using coral-based paleo data assimilation and attributed the breakdown to different patterns of covariability during a period highly influenced by volcanic eruptions.

Response:

Agreed! We assume that the periods of lower/higher agreement are due at least in part to changes in the covariances between the proxies and the Walker circulation - that is one of the reasons we used the different reconstruction methods. Many thanks for the reference - we have added this comparison with Sanchez et al's work to the main text (L116-118).

I think the reviewer is confusing two aspects of covariance: covariance in the climate system, e.g. between a forcing and a response; and in the paleoclimatic reconstruction method, e.g. the use of covariance in a regression coefficient estimate. In the data assimilation of Sanchez et al (2021), these two things are represented independently by a climate model and a data model, respectively.

In the authors' work, nonlocal regression statistically conflates climate and data models. The reconstruction methods used in the manuscript all require covariance estimates and the stationarity assumption, so the author's response is insufficient here.

I would revise the rebuttal and text to say that covariation within the climate system may vary with forcing (Sanchez et al 2021), and reconstructed differences may arise from changes in covariance in the climate system or differences between reconstruction algorithms and how they treat bias (e.g. the distinction of opPCA vs FiPCA). A citation for this could be

J Gergis, R Neukom, AJE Gallant, DJ Karoly, Australasian temperature reconstructions spanning the last millennium, *Journal of Climate* 29 (15), 5365-5392. 2016, doi: 10.1175/JCLI-D-13-00781.1

or another reference of their choice that uses a variety of statistical methods to perform a nonlocal statistical paleoclimate reconstruction of a target time series.

It's also unclear from the Methods how the number of retained PCs for PCR was determined without a stopping criterion.

Minor, but helpful: at l. 83 it would be better to use the existing abbreviation for pairwise comparison based paleoclimate reconstruction, PaiCo, instead of PC, which is too similar to PC and PCR.

Response:

We have made the suggested changes, to more clearly differentiate between non-stationarities in covariance structures in the climate system, and differences in the various reconstruction methods (L124-128). Explaining this difference without getting into the details of Sanchez et al. (2021)'s work (and data assimilation) was something that we had struggled with, so we appreciate this suggestion from the Referee!

We added more detail to the section of the Methods that describes how the number of retained PCs was determined (L587-589, Section 2.3.2.3). We have also changed 'PC' to 'PaiCo' throughout the text and figures.

Line 107/Figure 2: How do some of these frequencies have zero power? Are the authors only plotting the periodicities with significant power? If so, be sure to say it! It is not immediately clear to the reader.

Response:

Yes, that is correct - we are only showing periodicities with significant power. This is stated in the figure caption, but we have clarified it in the text.

OK.

Line 114-115: Really happy the authors put in the sentence “ the difference between the two distributions is within the range calculated between any other two intervals of equivalent length...”! I was curious if the PDFs had a more distinct difference if you compared slightly different intervals, eg. 1200-1900 vs 1900-2000?

Response:

Prior to submitting the paper, we thoroughly investigated this exact point, including stepping that 'split point' progressively further toward the present to see the change in the differences between the two distributions (see below). We agree that this is a really interesting analysis, however we do not have space (in text or figures) to cover all the nuance of this particular point, and instead distilled this analysis into just the information relevant to our question of whether there is a detectable anthropogenic influence on the PWC.

Additionally, we re-did the statistical analysis addressing this point such that the results are more intuitive (Fig. 2d, L132-134), and added the option for readers to play around with similar tests using the provided code.

OK, but what could the 1900-2000 calibration period contribute to this problem?

Response:

We are unsure what Referee 6 is asking here but will try to provide some additional clarity on the calibration period in this response. The comparison period shown in the paper is pre- versus post-1850. So yes: most of the data in the post-1850 distribution has higher skill than the data in the pre-1850 distribution. This disparity increases as the 'split point' steps forward from 1850 to 1900. In the original response to Referee 1, we showed a figure with split points every 20 years from 1830 to 1990, given their interest was in whether the difference between the two PDFs is strongly dependent on the two time periods being considered. As that figure showed, there is minimal change in the difference between the two PDFs if the split point is at 1900 versus 1850.

Line 122: Could also cite the Seager et al. 2019 paper (mentioned earlier) here

Response:

We have added this citation (L141). OK.

Line 123: Could potentially discuss tropical Pacific Decadal Variability (eg. Power et al. 2021) here

Response:

We have added some discussion of the Pacific decadal variability in the paragraph above (L124-126), and also added Extended Data Fig. 6b comparing our reconstruction with Vance et al.'s (2022) ice core-based reconstruction of the Interdecadal Pacific Oscillation.

OK. But in EF6b, the significance of the correlations should be indicated and the text revised accordingly; for example if a correlation is not significant it should not be described as a weak correlation; or else its p-value should be estimated.

Response:

Extended Data Fig. 6b only shows significant correlations, but we had failed to include that in the caption - we have now added it, and clarified this in the Methods (L797). We also added a denotation of significance to the correlations shown in Extended Data Fig. 6d.

Line 142-146: This is so cool! I had been worried about the potential influence of Pinatubo on the results given the importance of the 1992-2011 period, but I feel much more comfortable given this analysis. I am still a little worried about the potential apples to oranges comparison as detailed in the Major Points section.

Response:

Comparison of observed or modelled modern/future changes to a paleoclimate reconstruction is reasonably common. In this particular case, we argue that the demonstrated strong agreement between the reconstruction and the observations in the calibration period means that this comparison is valid. To make this more clear, we have added Extended Data Fig. 3 zoomed in on this period of overlap, where the reader can more clearly see the close correspondence between the observations and the reconstruction.

As for EF6B, EF3B ought to be controlled for statistical significance. Here I expect all but a few of these correlations over the calibration period are significant, and might indicate that the differences between the reconstruction techniques, which are all related to one another, are not.

Response:

All correlations reported in Extended Data Fig. 3b are significant ($p < 0.05$), and we have now included this in the figure caption.

Line 149: I again am wondering about the resolution of the proxies considered in this network. Are they all annual or sub annual?

Response:

Yes, they are all sub-annually or annually resolved. This is mentioned in the Methods, and we have now added it to the main text (L68).

OK.

Line 177: “reassigned major eruption”—this is great!

:)

Line 165: how large is this trend?

Response:

At the suggestion of Referee 3, we have added a second axis to the panels of Fig. 4, showing a random sample of the ensemble PWC response to volcanic eruptions. This shows the magnitude of the response.

OK.

Line 172: If specifically referencing a study, please use the name of the study and then superscript it instead of just using the superscripted numbers

Response:

We have changed all instances of this in the text. OK.

Line 190-192: Consider re-writing this sentence for clarity “if all volcanic eruptions with...” I’m not totally sure I follow what you are saying.

Response:

We have re-written this paragraph for better clarity (and also added PMIP3/4 models to this analysis) - L216-220 “When applying the above SEA approach to the CESM1 LME (using the 25 strongest eruptions; Methods), 9 of the 13 CESM1 LME ensemble members produce a significant negative Δ SLP anomaly the year following a volcanic eruption (Fig. 5a), with Δ SLP anomaly magnitudes similar to those occurring during an average El Niño event”.

OK, but see next note.

Response:

Please see the response below.

Line 191: How is significance defined?

Response:

Significance is defined the same way as for the reconstruction. We have added the citation of the method we used to determine significance to make this clear (L218).

Too cryptic a response. Explain in rebuttal what the 'double-bootstrap' method of Rao et al (2019) works and how it's used, because you spend some time explaining later in rebuttal and revised text how it might identify significant warmings prior to volcanic forcing.

Response:

Traditionally, superposed epoch analysis (SEA) has been viewed as a method for comparing changes in the mean of composite timeseries (those composites being a summary of a climatic variable before and after volcanic eruptions). However, this is a slightly reductive view of SEA: more accurately, it is about comparing the condition mean of post-event years with the distribution of all years. The double-bootstrap uncertainty quantification method developed by Rao et al. (2019) accounts for this subtle difference.

Specifically, to define the confidence intervals, we generated 1000 ‘pseudo-composite’ matrices, each constructed using randomly-chosen years of the full reconstruction as the ‘eruption’ year (two fewer than the number of eruptions being considered in the SEA). These were centred in the same manner as the actual composite matrices (by subtracting the pre-event mean). Then, the 2.5/97.5 percentiles were calculated as the significance thresholds needed to be exceeded for the SEA response to be deemed significant. The confidence intervals prior to the eruption are relatively narrow because by design, the mean for those three years is zero. The intervals widen slightly after the eruption as the conditional distributions revert to the unconditional distribution. As now stated in the text (L207-209), ‘significant’ volcanic responses may arise in year -2 relative

to the eruption, due to the chronological uncertainty incorporated in the reconstruction. Given the pre-eruption mean is calculated with all three years, if year -1 has unusually low Δ SLP, this may predispose the preceding year of the SEA to be outside the confidence interval on the opposite side. This can be seen most clearly in Panel D of Extended Data Fig. 4 (thin grey lines).

The ‘double’ component of the ‘double bootstrap’ arises from confidence intervals calculated for the response itself. This is generated simply by resampling (without replacement) the specific set of eruption years used to calculate each composite. This resampling is performed 100 times (or as many times as possible, with unique combinations of eruptions years).

We have added text to the Methods (Section 2.7, L729-732) to provide more details of our application of Rao et al. (2019)’s approach.

Line 195: Consider writing “Fewer LME ensemble members” to help the readers stay on point as you also call your proxy reconstruction an ensemble

Response:

We have made the suggested edit (L223).

OK.

Line 218: Again, this is so cool!

Response:

We have promoted the associated Extended Data figure to the main text to emphasise this point (now Fig. 6).

I think the authors mean Fig 5, as referenced in the text at and near l. 218? OK - except that the significance test is again important here, and needs description in rebuttal and I think in the text.

Response:

I think there may be a little confusion here around the original line numbers, and the line numbers in the revised manuscript. In any case, please see above for a more complete description of the SEA significance test.

Line 225: Why pCO₂ instead of CO₂? I think CO₂ is more intuitive for readers

Response:

We have removed the rho in both instances (L258, L260).

OK. rho? p? I leave this up to the copy editors.

Response:

Line 271 & 273 in the revised manuscript

Line 248: Did the authors remove the tropical mean from this? See major point for more discussion

Response:

We have re-done this analysis, now assessing relative SSTs in all cases (L764-768, L284-285).

Hard to evaluate without seeing a side-by-side comparison of original and revised results. Did the interpretation change? Please explain in rebuttal if and how the interpretation changed as a result.

Response:

Correcting our Niño 3.4 calculations by removing the tropical mean made minimal difference to the results originally presented in Fig. 5. In the original submission, we only showed results of the SEA performed on Niño 3.4 and Δ SLP from the Community Earth System Model Last Millennium Ensemble.

In the revised submission, we added data from 8 PMIP3/4 models. Interestingly, the degree of disparity between the results of the SEA performed using a) raw Niño 3.4 SST and b) Niño 3.4 SST with the tropical mean removed varied strongly by model. In models with a strong difference, there was a strong (and significant) *negative* SST anomaly in the years following volcanic eruptions. This is due to the global cooling induced by volcanic eruptions. Removing the tropical mean (as suggested by Referee 1) allows inspection of the tropical Pacific response independent of the global signal.

In summary: our original interpretations did not change as a result of this correction.

Line 259: This manuscript describes a very original and exciting analysis. However, I don't think that the authors have emphasized just how original this source of data is! Many scientists and working groups are interested in finding uses for water isotope data—and this is a great illustration! If there is space, I would definitely hammer on this point a little more.

Response:

We have added text to remind readers how useful water isotope proxy data is. For example, at line 290 “Finally, our novel use of water isotope proxy data to reconstruct atmospheric variability, including explicit incorporation of uncertainty from the training dataset, reconstruction method, and age-depth models....”.
OK but IMHO unnecessary.

Figure 2B. I would make two boxes to indicate just where your delta SLP index comes from

Response:

We have added the two boxes to Figure 1.

OK. This is valuable because it shows that not a single observation comes from the target areas. Worth noting that this suggests the use of a nonlocal reconstruction approach, and why the Iso2k network is nonetheless expected to produce skillful reconstructions for Δ SLP from these regions. See also note below.

Response:

It is not a new approach to perform statistical reconstructions of a climate mode by using data outside the region used to calculate that mode's index. For example, this is a common approach for reconstructions of tropical Pacific variability (e.g. D'Arrigo et al., 2005; Dätwyler et al., 2019; Vance et al., 2022, tree ring-based ENSO reconstructions).

We expect the water isotope proxy records of the Iso2k network to produce skill Δ SLP reconstructions due to the demonstrated dominant influence of the PWC on global interannual precipitation $\delta^{18}\text{O}$ (Falster et al., 2021). The underlying reason for this is that the PWC directly or indirectly influences atmospheric circulation processes in many parts of the world—more so than any other globally-relevant climate mode. Atmospheric circulation processes in turn affect the isotopic composition of precipitation. These effects vary according to local/region factors. For example, PWC-related changes in precipitation amount in the tropics may manifest in precipitation $\delta^{18}\text{O}$ via the amount effect. The PWC also affects moisture sources or transport pathways in the

extra-tropics, and this too affects precipitation $\delta^{18}\text{O}$ in locations well outside the two boxes used to define the ΔSLP index.

Additionally, our network approach - incorporating many records - buffers against the possibility of changing teleconnections, which helps address the fact that there are no records in the ΔSLP boxes (Batehup et al., 2015).

References

D'Arrigo, R., E. Cook, R. Wilson, R. Allan and M. Mann. 2005. On the variability of ENSO over the past six centuries. *Geophysical Research Letters*, 32(3)

Dätwyler, C, Abram, NJ, Grosjean, M, Wahl, ER, Neukom, R. El Niño–Southern Oscillation variability, teleconnection changes and responses to large volcanic eruptions since AD 1000. *Int J Climatol*. 2019; 39: 2711– 2724.

Vance, T.R., Kiem, A.S., Jong, L.M. et al. Pacific decadal variability over the last 2000 years and implications for climatic risk. *Commun Earth Environ* 3, 33 (2022).

Falster, G., B. Konecky, M. Madhavan, S. Stevenson, and S. Coats, 2021: Imprint of the Pacific Walker Circulation in Global Precipitation $\delta^{18}\text{O}$. *J. Climate*, 34, 8579–8597.

Batehup, R., McGregor, S., and Gallant, A. J. E.: The influence of non-stationary teleconnections on palaeoclimate reconstructions of ENSO variance using a pseudoproxy framework, *Clim. Past*, 11, 1733–1749, <https://doi.org/10.5194/cp-11-1733-2015>, 2015.

Figure 3. These distributions are very gaussian. Is that expected? How well do these indices agree with the reconstruction?

Response:

The distributions include data from all 4800 ensemble members. Given the large number of samples, we expect the distributions to tend to a normal distribution. When examining skill scores, both individual ensemble members (Extended Data Fig. 4a) and the reconstruction ensemble mean (Extended Data Fig. 3b) agree very well with the target indices. With the new Extended Data Fig. 3, readers can now see this for themselves.

I suspect the reason the reconstructions are Gaussian is because (a) the observations are gaussian or have been standardized, and (b) the reconstruction methods, being covariance-based, impose this. For the skill scores (EF4a, EF3b) this suggests the differences between results using reconstruction methods and different target SLP fields are not very important. It also suggests more analysis and testing using independent validation experiments, e.g. 1951-2000 calibration, reporting the 1900-1950 validation scores in EDT1, and other tests, e.g. with independent subsets of proxies (I suspect independent reconstruction targets are not available.)

Response:

This point—differences between results using reconstruction methods and different target SLP fields are not very important—is an important conclusion of the paper. That is, although there may be small differences in results from ensemble members calculated using different targets/methods (or perhaps, as the referee suggests, no difference), our main findings are robust to these differences.

Irregardless, we provide these independent validation tests in Extended Data Fig. 5, as well as the third column of Extended Data Table 1 (calibration 1951-2000, validation 1900-1950). We acknowledge that we did not clearly signpost this information. We have clarified the relevant figure captions, and added a statement in the main text about these independent tests (L105-108)

Figure 4: In many of these plots, it looks as though there is a consistent Stronger Walker state observed two years prior to the eruptions in all periods. This is wild! It is quite large in subsets B, C, and D. It might be worth commenting on this unexpected result.

Response:

We have added text with a proposed explanation for this results (L195-197).

This further illustrates the need for a detailed description of the significance testing approach, and for more reliance on independent validation.

Response:

Please see above for a more detailed description of how we calculated the confidence intervals, and how this method may have caused this unexpected result.

Figure 5A/B/C: How large are these SLP anomalies relative to a traditional El Nino/ La Nina Event?

Response:

They are of very similar magnitude: Δ SLP anomalies during El Niño events in the CESM-LME range from around -0.8 to -4 hPa. We have added that information to the main text (L219-220).

OK, no way for me to evaluate.

Figure 5D/E/F: Have you subtracted the tropical mean from these Niño 3.4 SST estimations? Most of the studies that look at the ENSO response do remove the global tropical mean from SSTs (something very difficult to do with a single proxy record)

Response:

We have re-done this analysis, first subtracting the tropical mean from the Niño 3.4 SST anomalies. We have also added text explaining and discussing this point (L764-768, L284-285). Many thanks for pointing out this error in our original calculations.

OK. Please add the same level of explanation in the text to the Rao et al (2019) based significance testing.

Response:

Please see above for a more detailed description of how we calculated the confidence intervals; this description has been added to the text at L207-209 and L730-732.

Figure 5G/H/I: Love that you included this! So cool!

Line 498: It would be great if you included a table with each proxy record and cite the individual authors with the database. Also, what resolution are these proxies?

Response:

This table is unfortunately too large to be included in the Extended Data, but is available (including authors of each individual dataset) as supplementary information. All records are annually or sub-annually resolved, and we have added this information to the main text (L68).

OK; I don't have the SI.

Line 509: “Model simulations” instead of “Model data”

Response:

We have made the suggested change (L479).

OK.

Line 529—great! I don't think that I got this from the main text. Do you include a table of the skill?

Response:

That table was in the supplementary materials, but has now been promoted to Extended Data Table 1.

If I read the EDT1 caption correctly, this doesn't include 1900-1950 validation scores, but it should: please add, discuss if necessary the presence or absence of evidence for artificial skill.

Response:

Our caption was unclear—the third column of Extended Data Table 1 shows 1900-1950 validation scores. We have edited the caption to make this clear. Comparing values in columns 2 (calibration & validation 1900-2000) and 3 (calibration 1951-2000, validation 1900-1950) demonstrates that the reconstruction remains skillful across all tests when using an independent interval.

Line 535: Do you use a January-December year or a tropical (May-April) year when converting records to annual resolution? It's very important to be clear on what was done.

Response:

We used a calendar year (Jan-Dec) to match the other proxy data. We have added this clarification to the text (L508).

OK: it would have been better to use a tropical year definition, but this may not make much difference. Put the resolution or averaging for the records in question into the appropriate location in the SI.

Response:

If a record is sub-annually resolved, this is stated in the SI table. We have added a statement that the averaging for these records was performed over a calendar year.

Line 645: If referencing the study by name, please include the author's name, not just the superscript number.

Response:

We have changed all instances of this in the text.

OK.

Line 656: Matching the SLP variance to that observed over the twentieth century could be a big problem. I discuss this in the major points section.

Response:

This is a valid concern, and one that we spent a lot of time investigating. The method we used is a commonly-used approach for variance scaling in palaeoclimate reconstructions. Additionally, variance changes through time are preserved, and we have added text to the Methods expanding on this point (L623-624).

OK; previously assessed.

Line 741: the years 1992-2011 are superscripted

Response:

We have fixed this formatting error.

OK.

Line 743: just mention the authors and then superscript if directly mentioning a study.

Response:

We have changed all instances of this in the text.

OK.

Line 826: Suggest adding the of the method here-- eg. PAGES2k GMST (#66)

Response:

We added the in-text reference to PAGES 2k (L778).

OK.

G. References

The authors did a good job of including relevant, recent references. However, there were a few papers that merit mention in this discussion (mentioned previously in this review).

Response:

Many thanks for these excellent suggestions. We are limited by the 50-reference limit, but have added most of the below references, in place of some of the references originally included.

OK.

H. Clarity and Context

There are a few places where the authors could improve the clarity of their manuscript to help the readers understand the novelty of the work and significance of their results. I have generally commented on this in the Suggested Improvements Section.

However, there are a few points that would add a lot of clarity to the manuscript in general :

Firstly, A more in depth description of the proxies that make up the reconstruction. Table S1 cites the Iso2k database, but does not directly reference the individual authors who made many of these proxy datasets. This is a small point, but many paleo people that aren't familiar with the ISO2k database would get much more out of this table if you directly wrote "Bagnato et al. 2005 (14)" instead of #14 and the ISO2k ID number. I'm also a little confused about the "unknown" resolution or seasonality. Does this mean variable resolution?

Response:

We have added more informative citations to the table. For seasonality, 'unknown' refers to records where the authors of the primary dataset did not claim any particular seasonality for their data, but also did not

explicitly state that the record represents an annually integrated signal. We have added clarifying text to the table caption.

OK. I don't have Table S1, cannot comment further.

Secondly, I think that it would help the readers if the authors included their specific definition of Walker Circulation Variability in the main text (presently included in the methods). Suggestion that the authors add two boxes to Figure 1 to highlight the regions where SLP anomalies are considered. It'd really help the clarity if this was directly addressed this early on (I think early on in the manuscript the authors clarify that their index isn't related to hydroclimate or SST, but don't say exactly what it is). I also think that adding a sentence to describe just how these proxies mechanistically relate to Walker Circulation Variability would be extremely helpful for readers unfamiliar with the Falster et al. Journal of Climate paper.

Response:

We have added the index reference boxes to the map in Figure 1. We also added text summarising the main findings of the Falster et al. JCLim paper, in terms of how water isotopes relate to the Walker circulation (L64-67).

This needs revision and more explanation. The reviewer stated it well: "adding a sentence to describe just how these proxies mechanistically relate to Walker Circulation Variability would be extremely helpful for readers unfamiliar with the Falster et al. Journal of Climate paper." As noted previously, this is important, as all the proxies are nonlocal to the target dSLP regions indicated by the black boxes.

Response:

We have added more detailed text outlining the reason we expect globally-distributed water isotope records to contain information about the Pacific Walker circulation (L64-71).

Reviewer Reports on the Second Revision:

Referees' comments:

Referee #6 (Remarks to the Author):

Response:

We have addressed the residual concerns of Referee 4, as well as additional comments from Referee 6, and the Editor's requirements.

In summary, we have:

- Made all formatting changes requested by the Editor
- Added text more clearly describing the validation tests (L105-108, 989-993)
- Added the coefficient of efficiency test to Extended Data Table 1
- Provided more details about how the confidence intervals around the Superposed Epoch Analysis are calculated, and the impact that this has on the results (particularly those shown in Fig. 4)
 - Detailed explanation provided in this rebuttal, as well as addition of extra text to the Methods (L205-209, 730-732)
- Edited the captions of Extended Data Fig. 5 and Extended Data Table 1, to make it clear that these items show skill scores calculated using independent calibration (1951-2000) and validation (1900-1950) intervals
- Added more statistical details to the caption of Extended Data Fig. 3
- On all relevant figures and captions, clearly stated if correlations are significant or not (e.g. Extended Data Figs. 3,6)
- Added more detail to the main text about how we are able to use the Iso2k water isotope proxy network to skillfully reconstruct Δ SLP (L64-71, 79-80)
- Clarified our proposed explanation of the period of low inter-method agreement at the start of the 19th century, with respect to the similar period of low inter-method agreement in Sanchez et al. (2021) (L124-128)
- Changed the abbreviation for 'Pairwise Comparison' from 'PC' to 'PaiCo' throughout

The main text is now ~ 3800 words, and the Methods are now ~4700 words. Please see below for detailed point-by-point responses to Referee comments.

Referees' comments:

Referee #2 (Remarks to the Author):

Dear authors,

I would like to revisit my initial assessment of your manuscript's suitability for publication in Nature. After carefully reviewing your responses to my comments and those of other reviewers and the revisions you have made, I have reevaluated my initial judgment.

Your clarifications and the additional analyses you have incorporated, such as including data from eight additional PMIP3/4 models, strengthen the comparison and provide a better understanding of inter-model variability in the Δ SLP response to volcanic eruptions.

I appreciate your acknowledgement that the underlying physical explanation for the observed shift in PWC frequency is beyond the scope of this paper and that it is now a topic of ongoing research in your group. I understand that the main focus of your work is to provide empirical evidence of PWC changes and use model data to compare with previous work. Your effort to minimise uncertainties in the Δ SLP reconstruction using multiple statistical methods and datasets is commendable, as it strengthens the basis for studying the dynamics of PWC variability and its response to anthropogenic and volcanic forcings.

It is good to know that detailed investigations of the anthropogenic and other external forcings' influences on modelled PWC are currently underway in your group. I believe this will provide valuable insights into the complex interactions driving PWC variability.

Given these improvements and considering the potential impact of your reconstructed Δ SLP time series on future research, I am confident your paper is well-suited for publication in Nature. I hope you understand that my revised opinion is based on the strength of your responses and the revisions made to the manuscript.

I look forward to seeing your work published in Nature and its contribution to our understanding of PWC dynamics.

Referee #3 (Remarks to the Author):

The authors have addressed my comments, I am happy with the revised manuscript and therefore recommend its publication as it is.

Referee #4 (Remarks to the Author):

Dear authors of the manuscript "Forced changes in the Pacific Walker Circulation over the past millennium", I appreciate your diligent efforts in revising the manuscript and addressing my comments. The manuscript is now significantly clearer than before. I understand that the focus of this manuscript is to explore the Pacific Walker Circulation (PWC) variability using proxy records. The innovative aspect of this manuscript lies in utilizing the same indicator with different types of proxy records to reconstruct this circulation index. Analyzing the attribution of PWC variability trends during 1992-2011 CE from a paleoclimate perspective is an excellent example of "The past is the key to the present". Moreover, the significant application of this PWC reconstruction is to demonstrate the El Niño-like response to volcanic eruptions, which is also a hotspot issue. I recommend this manuscript is accepted for publication after a minor revision.

Minor comments:

1. How should one interpret the good negative correlation between PWC and ENSO during the instrumental period, but a more complicated relationship before the instrumental period, as shown in Figure 6c? Previous studies show the PWC is coupled with ENSO variability (e.g., (Rasmusson and Carpenter, 1982)).

Response:

Unfortunately, we expect that this is at least partly due to decreasing reconstruction skill back through time. For example, ENSO reconstructions derived from a single site are particularly susceptible to non-stationary teleconnections (e.g. Batehup et al., 2015). This is evident from Extended Data Fig. 6d, which demonstrates that the ENSO reconstructions themselves are poorly correlated.

The degree to which the changing correlations between PWC and ENSO is due to a) changing relationship between the two, and b) decreased reconstruction skill is difficult to assess. However, we are currently attempting to investigate this point via detailed comparisons with the Last Millennium Reanalysis (LMR; Tardif et al., 2019). The LMR is a data-assimilation based ensemble reconstruction, using CCSM4 for the model prior. Our investigation includes analysis of the temporal relationships between the two reconstruction ensembles, and comparison of spectral character, as well as supporting analyses using climate models. This detailed investigation is beyond the scope of this particular paper, but will be the topic of an upcoming publication.

References

Batehup, R., McGregor, S., and Gallant, A. J. E.: The influence of non-stationary teleconnections on palaeoclimate reconstructions of ENSO variance using a pseudoproxy framework, *Clim. Past*, 11, 1733–1749, <https://doi.org/10.5194/cp-11-1733-2015>, 2015.

Tardif, R., G. J. Hakim, W. A. Perkins, K. A. Horlick, M. P. Erb, J. Emile-Geay, D. M. Anderson, E. J. Steig, and D. Noone, 2019: Last Millennium Reanalysis with an expanded proxy database and seasonal proxy modeling. *Clim. Past*, 15, 1251-1273, doi:10.5194/cp-15-1251-2019.

2. Why do these records primarily reflect PWC variability but not other indicators? The authors cited previous studies to explain it, but it is still suggested to provide a more detailed explanation about "PWC-related changes in moisture source and transport length, and a PWC- or ENSO-driven 'amount effect' in tropical regions." In fact, some of these proxy records (e.g., the $\delta^{18}\text{O}$ records of coral and tree-ring) also show a strong correlation with the ENSO index during the instrumental period.

Additionally, I agree that the dominant mode of observed global interannual precipitation may be strongly related to PWC, however, the $\delta^{18}\text{O}$ records from different proxies cannot be equated to precipitation $\delta^{18}\text{O}$. In other words, the climate significance of different oxygen isotope indicators is not uniform and clear (e.g., (Cheng et al., 2019; Shi et al., 2022; Xu et al., 2021)). It is suggested to clarify the climate significance of oxygen isotopes of different proxies.

Response:

The referee may have slightly misunderstood the sentence at Line 64 - Falster et al. (2021) find a correlation between the dominant mode of precipitation $\delta^{18}\text{O}$ and the PWC (not precipitation amount). We have re-written the start of this paragraph to avoid this confusion (L64-71). At the site level, in some cases, the relationship between the PWC and precipitation $\delta^{18}\text{O}$ is a result of the amount effect, but in others it is due to changes in e.g. moisture source and/or transport pathway. But also, as the referee states, proxy $\delta^{18}\text{O}$ does not always directly reflect precipitation $\delta^{18}\text{O}$. A detailed description of the interpretation of different proxies is outside the scope of

this manuscript (those details are available in Konecky et al, 2020), but we have added a sentence at Line 79-80 stating that this is an additional complication.

References

Konecky, B. L., et al.: The Iso2k database: a global compilation of paleo- $\delta^{18}\text{O}$ and $\delta^2\text{H}$ records to aid understanding of Common Era climate, *Earth Syst. Sci. Data*, 12, 2261–2288, 2020

3. Lines 116-118, how should this reason be understood? Shouldn't volcanic activity as a strong external forcings improve the consistency of results?

Response:

Sanchez et al. (2021) use data assimilation methods to reconstruct ENSO variability. They find poor inter-method agreement across 1809 to 1830, and attribute this to a ‘period of intense volcanism’. They conclude that the poor agreement is largely due to a cluster of proxies located in the South Pacific Convergence Zone driving SST covariance patterns associated with ENSO to produce unrealistically warm SST in the Nino 3.4 region after volcanic eruptions. This is not generic to all eruptions, but rather due to a combination of a) closely-spaced eruptions, and b) the geographical distribution of proxy data during this interval.

In the case of our reconstruction, the 1800-2000 reconstruction segment includes a coral $\delta^{18}\text{O}$ record in the South Pacific Convergence Zone region, which is strongly anti-correlated with the PWC, and which did not contribute to previous segments (Extended Data Fig. 2). It is possible that the same combination of a) closely-spaced eruptions, and b) the geographical distribution of proxy data had disparate influences on the way in which each of our methods reconstructs ΔSLP (depending on the particular covariance structures). This is one of the reasons we chose a multi-method reconstruction approach - hopefully the ‘correct’ ΔSLP falls somewhere within the ensemble’s range.

We have edited the text to better outline this reasoning (L124-128).

References

Sanchez, S. C., G. J. Hakim, and C. P. Saenger, 2021: Climate Model Teleconnection Patterns Govern the Niño-3.4 Response to Early Nineteenth-Century Volcanism in Coral-Based Data Assimilation Reconstructions. *J. Climate*, 34, 1863–1880

4. Lines 126-128, is there an issue when comparing two power spectra of sequences with different time periods? Different lengths of two sequences will result in different power spectra.

Response:

We accounted for this by comparing industrial-era (1850-2000) spectra with the distribution of spectral power in all possible 150-year periods prior to 1850, still showing the proportion of ensemble members with power in each period. This is described in the Methods (Section 2.6.1). We refer readers to the Methods on Line 139.

6. Since PWC may be related to Interdecadal Pacific Oscillation (IPO) variability, it is suggested to compare it with multiple Pacific Decadal Oscillation (PDO) reconstructions, as the IPO includes the PDO, and the PDO has a better expression from middle-low latitude proxies than just one IPO reconstruction from high-latitude proxies.

Response:

We performed our comparisons with the IPO rather than the PDO for two reasons: 1) it has a more geographically-extensive footprint, and 2) Vance et al. (2022) is the most recently-published publicly-available

reconstruction of decadal-scale Pacific variability covering the 1200-2000 interval. Additionally, Henley et al. (2011) determined that given the high correlation between PDO and IPO timeseries, they are essentially indistinguishable on palaeoclimate timescales.

We agree that reconstruction of the IPO from a single high-latitude proxy is sub-optimal (e.g. Batehup et al., 2015). However, Wise (2015) and Henley (2017) demonstrated that published reconstructions of the PDO and IPO have low inter-reconstruction coherence prior to the 20th century (e.g. Henley (2017) Table 2). Therefore, comparing our PWC reconstruction to several PDO/IPO indices is unlikely to provide information that is more climatically meaningful than what is currently shown in Extended Data Fig. 6.

We also agree with the Referee that the relationship between the PWC and low-frequency modes of variability is worth investigating further. However this remains a challenge, given the lack of coherence between existing PDO/IPO reconstructions. Our PWC reconstruction ensemble will be publicly-available, facilitating comparison with future IPO/PDO reconstructions as they become available.

References

Vance, T.R., Kiem, A.S., Jong, L.M. et al. Pacific decadal variability over the last 2000 years and implications for climatic risk. *Commun Earth Environ* 3, 33 (2022).

Henley, B. J., M. A. Thyer, G. Kuczera, and S. W. Franks (2011), Climate-informed stochastic hydrological modeling: Incorporating decadal-scale variability using paleo data, *Water Resources Research*.

Batehup, R., McGregor, S., and Gallant, A. J. E.: The influence of non-stationary teleconnections on palaeoclimate reconstructions of ENSO variance using a pseudoproxy framework, *Clim. Past*, 11, 1733–1749, <https://doi.org/10.5194/cp-11-1733-2015>, 2015.

Wise, E.K. (2015), Tropical Pacific and Northern Hemisphere influences on the coherence of Pacific Decadal Oscillation reconstructions. *Int. J. Climatol.*, 35: 154-160.

Henley, B. (2017), Pacific decadal climate variability: Indices, patterns and tropical-extratropical interactions. *Global and Planetary Change*, Volume 155, Pages 42-55.

7. Solar activity, as a source of Earth's energy, should also be mentioned in the text.

Response:

We have added a statement about the possible influence of solar variability at Line 134.

Following the editor's requirement, I also assessed the response to referee #1.

Referee #1's primary concern is that the 1992-2011 CE period is not covered in the reconstruction (800-2000 CE). I believe this concern is insignificant, as there is a significant relationship between the reconstructed PWC and the instrumental PWC during the instrumental period. The authors have also included a new figure comparing them, which **sufficiently addresses this concern**.

The second concern from Referee #1 is important. I am unable to understand the statement "changes in variance through time are not strongly affected by the variance correlation." I believe variance scaling should be applied to the instrumental period, not the entire time series, and then any variance changes before the instrumental

period can be maintained relative to the variance of the instrumental data. Regardless, **the non-nested version of the reconstruction does not affect the conclusion, which is sufficient to address this concern.**

Response:

During the variance scaling, changes in variance throughout the reconstruction are maintained. That is, if the variance in the instrumental period is changed by a certain factor, then that same factor is applied across the entire time period. Hence, variance in the calibration period of the reconstruction matches variance in the instrumental data over the same period, then any changes back through time are also maintained.

However, as Referee 4 states, our main results are reflected in the non-nested version of the reconstruction, which addresses Referee 1's original concern.

3. Except for the first two main comments, the following minor comments are revised according to the Referee #1's comments. **All responses are acceptable to me.**

References

- Cheng, H., Zhang, H., Zhao, J., Li, H., Ning, Y., Kathayat, G., 2019. Chinese stalagmite paleoclimate researches: A review and perspective. *Sci. China Earth Sci.* 62, 1489-1513.
- Rasmusson, E.M., Carpenter, T.H., 1982. Variations in tropical sea surface temperature and surface wind fields associated with the Southern Oscillation/El Niño. *Mon. Weather Rev.* 110, 354-384.
- Shi, F., Goosse, H., Li, J., Yin, Q., Ljungqvist, F.C., Lian, T., Sun, C., Wang, L., Wu, Z., Li, J., Zhao, S., Xu, C., Liu, W., Liu, T., Nakatsuka, T., Guo, Z., 2022. Interdecadal to multidecadal variability of East Asian Summer Monsoon over the past half millennium. *J. Geophys. Res. [Atmos.]* 127, e2022JD037260.
- Xu, C., Zhao, Q., An, W., Wang, S., Tan, N., Sano, M., Nakatsuka, T., Borhara, K., Guo, Z., 2021. Tree-ring oxygen isotope across monsoon Asia: Common signal and local influence. *Quat. Sci. Rev.* 269, 107156.

Referee #5 (Remarks to the Author):

Summary:

Falster et al. use a network of water isotope records to reconstruct the strength of the Pacific Walker Circulation (PWC) from 1200-2000 CE. The main findings are 1) the recent (1992-2011 CE) PWC strengthening is anomalous but not outside the range of natural variability, 2) strong volcanic eruptions trigger an El Niño-like PWC weakening in subsequent years, and 3) there is not a clear relationship between PWC strength and global mean surface temperature (GMST).

A key strength of this study is how thoroughly the authors address many sources of uncertainty inherent in paleoclimate reconstructions. As mentioned in my first review, this study uses 5 statistical reconstruction techniques, 3 observational products as a training set, and a Monte Carlo-style approach for adding chronological uncertainty in the proxy records. This approach is comprehensive and addresses many of the most common uncertainties in paleoclimate reconstructions. The resulting Δ SLP reconstructions provides important context about the range of PWC variability and will likely support many follow-up studies that seek to better understand the mechanisms driving PWC variability and its response to external forcing. Overall, the analysis is novel and will of broad interest to readers interested in paleoclimate, tropical climate variability and change, and climate dynamics.

Review:

The authors did substantial work to address the comments from the 5 referees and the editor. This included restructuring the manuscript and extended data to improve readability, reducing the amount of material in the supporting information to only 1 table, clarifying the text (particularly in the methods), promoting the GMST analysis to the main text, and adding model data from PMIP3/PMIP4 Last Millennium simulations to Figure 5. The codes for the analyses were made privately available to the reviewers and a public DOI will be available upon publication.

In the revised manuscript and response to the referees, the authors clearly state that the goal of this study is to provide evidence of PWC changes through time as opposed to a holistic mechanistic understanding of why PWC responds to anthropogenic and volcanic forcings. Given the amount of effort that went into quantifying uncertainties, generating the Δ SLP reconstructions, and characterizing PWC changes through time (and its response to external forcing), I agree with the authors' assessment that the current scope of work is sufficiently comprehensive for publication.

The authors thoroughly addressed my 6 major comments as well as the minor comments/suggestions. They did this by clarifying text in the main text and the methods, adding Extended Data Fig. 3 (observed Δ SLP with the proxy-inferred reconstructions) and Extended Data Fig. 1 (proxy availability through time), and answering my questions about seasonality in the proxy records and uncertainty in the observed Δ SLP products. They also modified the statistical testing in Fig. 2D by performing K-S tests on all the individual ensemble members. They find that the pre- and post-1850 mean Δ SLP is not significantly different in 81% of the 4800 ensemble members. The boxplot of p-values in Fig. 2D is a welcome addition that nicely summarizes these findings. In response to my comments and those of Referee #3, the authors also repeated several analyses using a non-nested version of the Δ SLP reconstruction. I was also pleased to see the addition of the PMIP3/4 simulations in the superposed epoch analysis, and the general consistency of the results in Fig. 5.

In summary, the above additions and methodological changes strengthened the study and further supported the main conclusions. The layout changes, particularly the restructured methods, extended data, and supplementary information, also greatly improved the overall readability of the manuscript. The manuscript is well-written, and the analyses/figures remain high-quality.

Referee #6 (Remarks to the Author):

Original comments from Referee 1 are in black text; *author responses to original Referee 1 comments are in italicised red text*. Reviewer 6 notes on Referee 1 comments and author responses are in blue text, with those in Referee 6's view requiring further rebuttal and revision in purple text (as per the Referee's convention). Author responses to Referee 6 are in green text. Reviewer 6 notes on those Author responses are in bold blue text following the green text, with those in Referee 6's view requiring further revision in bold purple text.

A. Summary of the key results

Falster et al.'s "Forced changes in the Pacific Walker Circulation over the past millennium" describes a highly original reconstruction and assessment of Walker Circulation variability using a network of long proxy records in the Iso2k network.

The chief conclusions of this analysis are: 1) the recent 1992-2011 trends in the Walker Circulation are not unprecedented over the last 800 years, 2) Volcanic eruptions prompt an observable Weaker Walker response, 3). The relationship between global mean surface temperature and Walker Circulation is not as straightforward as many studies have suggested. The two important indices can vary considerably.

This manuscript will be of interest to readers of Nature who are interested in climate dynamics, paleoclimatology, and historical climate change. I recommend this manuscript for publication with minor revisions.

B. Originality and significance

This analysis is very novel and illustrates the utility of a new water isotope dataset (Konecky et al. 2020) to examine a long standing question in the scientific literature. This analysis does an admirable job of examining the uncertainty in their reconstructions. The authors examine the sensitivity of their method to dating errors, calibration dataset, temporal period of reconstruction and reconstruction methods—and clearly illustrate the sensitivity of their results to these choices. It is difficult to envision how the authors could have done more to test the robustness of their reconstruction to arbitrary methodological choices.

Response:

Thanks - we feel that this is one of the key strengths & novelties of the paper :).

As indicated specifically below, I think there are opportunities for independent validation exercise results to be further described and discussed.

Response:

We have addressed Referee 6's specific concerns as outlined below. Many concerns about the independent validation are addressed by additional explanatory text, as well as clearer figure captions for:

- Extended Data Fig. 5, which shows results for validation tests performed on an independent interval (calibration 1951-2000, validation 1900-1950), for all individual ensemble members
- Extended Data Table 1, which shows results for validation tests performed on an independent interval in the third column

OK. Well done and there are now only a few instances where I would suggest further revisions (in bold purple).

This work would be very relevant to several fields. This work also serves to highlight the utility of water isotope data.

One suggestion for the authors is to amplify the fact that this analysis uses such a novel water isotope dataset. This is **really** exciting (and novel!) and I am afraid that this important nuance might be lost on readers who have never heard of the Iso2k project.

Response:

We have added text throughout the manuscript to emphasise the novelty of the water isotope dataset (and its use in paleoclimate reconstruction). In particular, at Line 71 we now state “The reconstruction leverages the recently created Iso2k database, an innovative global synthesis of water isotope proxy records. Iso2k includes data from diverse archive types, and allows ready integration of water isotopic signals into palaeoclimate reconstructions.”

OK.

C. Data and Methodology

The data used and methodology used in this manuscript are both very reasonable and very original. The new Iso2k dataset (Konecky et al. 2020) used heightens the originality and significance of this study.

D. Appropriate use of statistics

I did not attempt to repeat any statistical tests, but all tests and results used in the manuscript appear logical. All figures illustrate an admirable use of error bars and significance testing. Furthermore, the authors take supreme care to test the sensitivity of their results to reconstruction methods. As an example of this, a 1992-2011 was a key analysis period in their study (due to previously published work in the scientific literature), but this period begins just after Pinatubo. The authors examined the sensitivity of this trend to the volcanic eruption by examining the 20 year trends after all volcanic eruptions and concluding no strong signal.

My largest concern with this study is that the 1992-2011 period is of chief importance to the analysis, but is a period not covered by the authors reconstruction (800-2000AD). The authors repeatedly compare the observed instrumental trend over the 1992-2011 period to the proxy reconstructed trends and draw major conclusions from this comparison. I understand that proxy records grow sparse after the year 2000, but it would be helpful to see just how well the reconstruction method compares with the instrumental observations over the 20th century. (Perhaps add a panel to Figure 3)? The authors state that there is high correlation between the two (Supplementary Table 2, $r \sim 0.8$), but it would be helpful to see the agreement between the reconstruction and all instrumental products over this important period. I think readers will question if the authors are indeed comparing apples to apples with the present discussion.

Response:

We have added the suggested figure (Extended Data Fig. 3), showing correspondence between the reconstruction and observed ΔSLP over the 20th century. This also shows how observed ΔSLP continues past the end of the reconstructions.

OK. But also: define the colors in EDF3B in the caption (suggestion: use them to denote significance given effective degrees of freedom; also in other instances with similar presentation noted in specific comments below); put into labeling or caption the root mean square difference (RMSD) between ERA20C, ICOADS, HadSLP dSLP, and for the RMSD between reconstructed dSLP and 20th century observed dSLP. This will also meet the second largest concern by showing that the calibrated residual amplitude is no bigger than the mean amplitude difference between historical dSLP estimates. The authors could report the median across estimates for the different reconstruction methods in EDF3, and for the 20y trend in dSLP in Fig 3. See below for other instances in which significance tests are missing.

Response:

All correlations presented in Extended Data Fig. 3b are significant ($p < 0.05$) when accounting for serial autocorrelation; we have now stated this in the figure caption. We have also added a colourbar, as well as the mean estimates for the different reconstruction methods.

OK. Better would have been for the colorbar to reflect significance of the correlation coefficients shown, to add that much more information.

The RMSE between reconstructed and 20th century observed Δ SLP is shown in Extended Data Figs. 4-5 (individual ensemble members), and Extended Data Table 1 (ensemble median; 0.27). We have now added that information to the Extended Data Fig. 3 caption, along with mean RMSE between Δ SLP calculated from the three gridded products (0.3).

OK.

My second largest concern with the methodology used in the study regards the variance adjustment applied to the reconstructions to maintain the variance of the 20th century throughout the 800-2000 reconstruction interval. The motivations for this variance adjustment are highly logical—nearly all proxy reconstructions feature a loss in variance with time (as the available proxy fraction dwindles), but my concern lies in the documented change in ENSO variability over the last millennium and how that may influence the Walker Circulation reconstruction. Both Grothe et al. 2020 and Cobb et al. 2013 demonstrate an increase in Niño 3.4 variability over the last few decades. At the very least, the authors could comment on how the assumed 20th century variance may influence the results (and cite these studies).

Response:

When constructing the reconstruction methodology, we sought to balance true variance changes through time with artificial variance loss back through time due to chronological uncertainty and proxy-specific effects. We performed extensive sensitivity testing on this point. However, changes in variance through time are not strongly affected by the variance correction, as a variance scaling is applied to the entire timeseries. That is, any variance changes through time are maintained. We have edited the Methods text to make this clearer (L620-624).

Although I disagree with the reviewer about the legitimacy of variance adjustment, because it hides uncertainty arising from changes in data availability, I think the author's rebuttal is good. It would be better if it were performed in validation rather than calibration, in other words, comparison of reconstructed with observed dSLP for a period not used for calibrating the reconstruction. Are validation-period variance changes successfully reconstructed? This could be demonstrated by adding validation scores and coefficient of efficiency (CE) to Extended Data Table 1 for the 1951-2000 calibration/1900-1950 validation exercise. For CE, the difference is referencing to the validation rather than the calibration mean; this is relevant for assessing skill at decadal timescale mean changes, in this case.

Response:

Response Extended Data Fig. 5 and Extended Data Table 1 (third column) show skills scores calculated on independent validation intervals. We have clarified that in the relevant captions. Extended Data Table 1 in particular demonstrates that validation-period variance in the reconstruction ensemble median is successfully reconstructed. We have added CE scores to Extended Data Table 1 as suggested.

OK. And the CE results suggest either the decade-to-decade variations are reliably reconstructed, and/or they are small relative to the interannual variations, a point made elsewhere already in text.

Additionally, following related concerns from several referees about the temporal nesting methodology and how that might affect variance, we repeated two key analyses, using a non-nested version of the reconstruction, i.e., a reconstruction using the same set of proxies throughout. All conclusions remain robust in this context (see below), and we have added one of these analyses as a new Extended Data Figure (ED Fig. 7, L673-693).

Comparing Fig 2AB to EDF7A and considering EDF7B, and the interpretation that the significant reconstructed spectral power is at interannual frequencies, I agree.

But it seems that the Fig 2 results are sensitive to nesting. This is from comparing Fig 2C (1850-2000, nested) with EDF7D (1850-2000, non-nested). The proportions of ensemble members are roughly half as large as with the nested results, despite the fact that plotting proportions should adjust for different sample sizes, and despite the fact that the first nest is for 1860-2000, and despite the fact that the calibration interval is 1900-2000, or $\sim 2/3$ of the period of analysis. And the relative proportions within the interpreted 2-9yr periodicities are different. This needs some clarification and further analysis of validation results.

Response:

Although the 1) overall proportion of ensemble members with significant periodicities is lower, and 2) the relative proportions are slightly different, the main overall finding from Extended Data Fig. 7c-d (industrial-era shift to lower-frequency variability) is unchanged. Additionally, following the Referee's concern, we checked over the code. In re-inspecting this, we found a small error (using a 99% significance cutoff instead of 95% as per the main text), and have re-made the figure. Extended Data Fig. 7c-d are now more similar to Fig. 2b-c, which makes more sense given the similarity of Extended Data Fig. 7a to Fig. 2a. Nevertheless, the proportions remain slightly different, and we have explicitly stated this difference in the main text (L140-142).

The other main finding from Extended Data Fig. 7—that there is minimal power at multi-decadal frequencies, and this is unaffected by the nesting approach—also remains robust, despite the relative proportions at each periodicity not being identical to Fig. 2a.

OK. Good work.

E. Conclusions

The conclusions of this manuscript are well grounded (aside from the concern about the reconstruction comparisons with the instrumental record over the 1992-2011 period). The authors describe their conclusions in the context of present literature and their results are easily interpretable.

I did not personally try to replicate any of the analysis, but the authors are very clear about the methodology and data used, thus their results seem reproducible. Furthermore, the authors state that they will provide access to their reconstruction upon publication—and offer to provide code upon request.

Response:

We now provide all code necessary to replicate our findings. Code will be available from the following Zenodo repository: doi.org/10.5281/zenodo.7742761. {REDACTED}.

Good. The url exists and is a downloadable zipfile. I didn't want to compromise my anonymity by accessing it, and actually testing the code/data repository entry goes beyond the scope of the review Request remit I was given.

F. Suggested Improvements

I have major and minor suggestions for improvement and have listed them below. Major Points

The first two points are discussed in the “Appropriate Use of Statistics” section, but I will briefly outline them again (apologies for repetition).

First: I worry about the comparison of the proxy reconstruction to the instrumental archives over the 1992-2011 period. I think that adding a figure to show the consistency of the reconstruction results to the HADSLP, ICOADS, and ERA20C reanalysis products over the 20th century would help abate the concern that it is not entirely appropriate to directly compare these two sources of data.

Response:

We have added this figure as a new Extended Data item (ED Fig. 3).

OK. See notes above, this could be improved, and validation results added and discussed. But changes in means across decades is not where the skill of the reconstruction lies (Fig 2).

Response:

We have more clearly captioned the Extended Data figures reporting validation skill test results (calibration and validation on independent intervals; Extended Data Fig. 5 and Extended Data Table 1 column 3). We have added the CE test to Extended Data Table 1, and also made the suggested changes to Extended Data Fig. 3. The validation results are discussed in the Methods (‘Assessment of reconstruction skill’ section), but we have now added to the main text describing the validation test (L105-108, 112-113).

OK. Good work.

Secondly: Adjusting the variance of the reconstruction to that observed in the 20th century causes some concern. I suggest that the authors comment that they acknowledge that they may be incorporating some bias with this approach.

Response:

We have added details and additional commentary on the variance adjustment to the Methods.

OK.

Finally: The authors include some discussion about the utility of examining Walker Circulation Variability with a few different variables, but set up a contrast between examining the SST component of Walker Circulation change vs the atmospheric/ SLP component (eg. Line 195, but also in a few other places). I think a sentence or two to clarify that discussion would be helpful to readers. Few studies examining the climate response to volcanic eruptions use raw SST, instead they use a “relative SST” and examine the Nino3.4 SST after subtracting the tropical mean (due to the expectation that volcanic aerosols will cause cooling globally and mask the tropical Pacific response, eg. Khodri et al. 2017, Predybaylo et al. 2020, Zuo et al. 2018, many others). This

is virtually impossible when using single paleoclimate proxy record that only monitors conditions at a single location.

Response:

We have re-done the SEA on model SST in the Nino 3.4 region, but now using the relative SST as suggested (results shown in Fig. 5). We have added text describing this approach in the Methods (L764- 768). We have also added additional discussion of this point to the main text (L284-285).

OK. But the description of the results at l. 274, "This is evident in both our reconstruction ensemble and climate model simulations, although the significant anomaly lasts ~three years in the reconstruction compared with one year in the model simulations." doesn't seem consistent with what's actually shown in those figures. The abstract doesn't reflect that very well either. Revise.

Response:

We have revised the text to better describe what is shown in Figs. 4 and 5 (L285-289). We also point out to the reader that the negative anomalies at -1 and +3 years (Fig. 4d) are likely due at least in part to the chronological uncertainty incorporated into the reconstructions, which 'smears' the response out in time (L205-207 - more detailed provided below).

OK. Good work!

Minor Points

Line 22—"the 4800 member ensemble" phrase is just confusing without introducing what the ensemble represents. How about something like, "We use a new paleoproxy-derived 4800 member ensemble"

Response:

We added 'paleoproxy-derived' to the introductory sentence (L22). OK.

Line 40: A reference should consider adding to this discussion is the Seager et al. 2019 paper that illustrates some of this mismatch between observations and models may be entirely due to biases in climate models.

Response:

We have added this reference (L42). OK.

Line 50: What is your specific definition for Walker Circulation? Suggest that the authors say "hydroclimate (rather than SLP directly)" or something like that for clarity.

Response:

We have edited the text for clarity (L51-52).

Revise sentence at l 33 to clearly defined the PWC.

Response:

We avoided including a specific metric of the PWC in the first sentence, as there are multiple ways of quantifying PWC strength (e.g. Kosovelj & Zaplotnik 2023); we used Δ SLP as it provides the most 'accurate' representation whilst also allowing the longest possible calibration interval. We state later (L58-59) that we used Δ SLP to quantify the PWC.

Nevertheless, we have added a more specific definition of the PWC at Line 31-33. The two sentences in place of the original first sentence now read: “*The Pacific Walker Circulation (PWC) is the zonal component of atmospheric circulation over the tropical Pacific. The PWC can be characterised by a sea level pressure gradient across the equatorial Pacific, with deep convection over the Indo-Pacific warm pool, subsidence over the equatorial eastern Pacific, upper tropospheric westerlies, and surface easterlies (the Pacific trade winds).*”

References

Kosovelj, K.; Zaplotnik, Ž. Indices of Pacific Walker Circulation Strength. *Atmosphere* 2023, 14, 397.

OK. “...may be characterised...” may be better grammar. This revision clearly defines terms for the reader of your paper.

Line 53: Again, be clear with what this 4800 member ensemble entails

Response:

We added a brief statement introducing the sources of uncertainty that make up the ensemble (L56-58). OK.

Line 54: As the 1992-2011 period has been mentioned, it would help to clarify why the reconstruction does not extend to year 2100

Response:

Assuming that the referee means ‘extend to the year 2011’, we added some clarifying text as to this choice in the Methods (L492-493). In short, it is because many of the proxy records stop around 2000, such that the proxy data availability would have been greatly decreased.

OK. But even better would have been to say how many or what fraction of the records extend to 2011. How many is it? If ~14, this is similar to what was available for 1860-2000, maybe worth the exercise.

Response:

Eight records extend to 2011 (~14%). We could have reconstructed past 2000, by dropping some of the records that only have data to 2000 for the most recent subset. However, we chose to maintain the 2000 cutoff such that there is more consistency across the different subsets. Prior to calculating the reconstructions, we performed extensive sensitivity testing around this point, i.e., the optimal start and end years to balance having as long a reconstruction as possible while including the largest amount of proxy data.

OK. And there are also the public data and code availability statements.

Line 55: If specifically citing a study, its far less confusing for the reader if you write XX et al. 13 rather than just the superscripted number

Response:

We have changed all instances of this in the text.

OK.

Line 60: All readers would appreciate a little more information about “the dominant mode of global interannual precipitation d18O carries a strong imprint of delta SLP”. Over what years? What was the proxy

distribution? How do the authors define “strong imprint”? Perhaps a sentence or two more would help readers better appreciate the work you have put into this task!

Response:

We have added extra text to provide more information. Specifically, from L64: “The dominant mode of observed global interannual precipitation $\delta^{18}\text{O}$ over 1982–2015 carries a strong imprint of ΔSLP , even though many individual records are not strongly correlated with ΔSLP . This imprint arises from multiple well-documented processes, including PWC-related changes in moisture source and transport length, and a PWC- or ENSO-driven ‘amount effect’ in tropical regions.”

"strong imprint" remains undefined. Define it in the first revised sentence at l. 64.

Response:

We have edited the paragraph to better define ‘strong imprint’, and provide more information about the theoretical link between water isotopes and the PWC (L64-71).

I think “strongly” is too ambiguous. Better at l. 64 would be to state specifics of correlation amplitude and significance from ref. 22 as already cited: “The first mode of observed global interannual precipitation $\delta^{18}\text{O}$ over 1982–2015 is significantly ($p < 0.05$) correlated with and explains 55% of the ΔSLP variance. This is the case even though many *individual* precipitation $\delta^{18}\text{O}$ records are not highly or significantly correlated with ΔSLP .” The latter statement is a consequence of finding large scale patterns in records with random-normal error, and could be deleted to save space if need be.

Line 65: I would include a few more details about the proxy dataset in the main text. Are these records at least annually resolved? What types of proxies compose these records?

Response:

We have added more details about the proxy dataset to the main text, including more information about the Iso2k database (L67-74).

OK. the rebuttal is most reflected in EDFs 1,2; cite them here.

Response:

We have added a citation of Extended Data Figs. 1-2 at Line 76.

OK.

Lines 70-90: Absolutely love that you did all of this sensitivity testing—and so clearly! This is one of my favorite parts of the paper. I was curious how you went from 59 proxies to a 4800 member ensemble. Could you provide the reader with this detail? We have 5 reconstruction methods, 3 calibration products, and some sort of age adjustment? Am I missing anything?

Response:

It’s one of our favorite parts too :). That information is provided in the Methods (Section 2.3.1.1) - we have added a pointer at this location to refer the reader to the Methods (L99-100), and edited the text to make this more clear to the reader.

At l. 96, if 'simulateBAM' is based on Comboul et al (2014), this reference should be cited and the specific "banded" added before "age-depth" on l. 96.

Comboul, M., Emile-Geay, J., Evans, M. N., Mirnateghi, N., Cobb, K. M., and Thompson, D. M.: A probabilistic model of chronological errors in layer-counted climate proxies: applications to annually banded coral archives, *Clim. Past*, 10, 825–841, <https://doi.org/10.5194/cp-10-825-2014>, 2014.

Response:

We added this citation at Line 103, and added the word 'banded' as requested (L102).

OK.

Line 95: How well do the instrumental products agree over the 1900-2022 interval?

Response:

This is an excellent question, and one that we spent a lot of time looking into at the start of this project. The short answer is 'not as well as one might hope', which is also the case for other gridded observational products (e.g., SST). A full discussion of this question is outside the scope of this paper, but to allow readers to assess this for themselves, we have added Extended Data Fig. 3 showing the 1900-2010 CE period (the limits of ERA20C), with the Δ SLP from a) the reconstruction ensemble, and b) the three instrumental products. We also state this in the text (L102).

OK, but see suggestions previously for adding information to this figure in support of uncertainty analysis.

Response:

We have added the suggested information to Extended Data Fig. 3 as outlined above.

OK.

Line 95: If you train the products on the 20th century instrumental record, aren't you forcing them to agree with the instrumental trends over the 20th century? Perhaps there is a way to check this influence

-- I see that you did this in the Supplemental Materials! Great! Clearly referencing that additional analysis here would be helpful!

Response:

We have added this information to the main text (L104-105).

Incomplete; see notes prior and later on expanding discussion of validation results.

Response:

Extended Data Fig. 5 and Extended Data Table 1 (third column) show results of skill tests performed using independent calibration and validation intervals. We have clarified that in the two captions, and also clearly stated this in the main text so the reader is not left with the same concerns (L105-108).

OK.

Line 99: “this decrease in skill back through time is a common but rarely emphasized feature of paleoclimate reconstructions” – I’m not sure what you mean here. Could you clarify?

Response:

We have edited the wording for clarity (L108-110) - “this decrease in skill back through time is a challenge for most palaeoclimate reconstructions, due to decreased data coverage and increased chronological uncertainty”.

I think it's better to base this on the results presented and simply say "this decrease in skill back through time is due to decreased data coverage and increased chronological uncertainty”.

Response:

We have made the suggested change (L116-117).

OK.

Line 105: Personally, I think some of this inter-method disagreement is really interesting. Why might these methods have strong disagreement over particular intervals? Changes in traditional covariance patterns between proxies and the Walker Circulation? Sanchez et al. 2021 documented similar disagreement in tropical Pacific reconstructions over the 1800-1850 periods when using coral-based paleo data assimilation and attributed the breakdown to different patterns of covariability during a period highly influenced by volcanic eruptions.

Response:

Agreed! We assume that the periods of lower/higher agreement are due at least in part to changes in the covariances between the proxies and the Walker circulation - that is one of the reasons we used the different reconstruction methods. Many thanks for the reference - we have added this comparison with Sanchez et al's work to the main text (L116-118).

I think the reviewer is confusing two aspects of covariance: covariance in the climate system, e.g. between a forcing and a response; and in the paleoclimatic reconstruction method, e.g. the use of covariance in a regression coefficient estimate. In the data assimilation of Sanchez et al (2021), these two things are represented independently by a climate model and a data model, respectively.

In the authors' work, nonlocal regression statistically conflates climate and data models. The reconstruction methods used in the manuscript all require covariance estimates and the stationarity assumption, so the author's response is insufficient here.

I would revise the rebuttal and text to say that covariation within the climate system may vary with forcing (Sanchez et al 2021), and reconstructed differences may arise from changes in covariance in the climate system or differences between reconstruction algorithms and how they treat bias (e.g. the distinction of opPCA vs FiPCA). A citation for this could be

J Gergis, R Neukom, AJE Gallant, DJ Karoly, Australasian temperature reconstructions spanning the last millennium, *Journal of Climate* 29 (15), 5365-5392. 2016, doi: 10.1175/JCLI-D-13-00781.1

or another reference of their choice that uses a variety of statistical methods to perform a nonlocal statistical paleoclimate reconstruction of a target time series.

It's also unclear from the Methods how the number of retained PCs for PCR was determined without a stopping criterion.

Minor, but helpful: at l. 83 it would be better to use the existing abbreviation for pairwise comparison based paleoclimate reconstruction, PaiCo, instead of PC, which is too similar to PC and PCR.

Response:

We have made the suggested changes, to more clearly differentiate between non-stationarities in covariance structures in the climate system, and differences in the various reconstruction methods (L124-128). Explaining this difference without getting into the details of Sanchez et al. (2021)'s work (and data assimilation) was something that we had struggled with, so we appreciate this suggestion from the Referee!

We added more detail to the section of the Methods that describes how the number of retained PCs was determined (L587-589, Section 2.3.2.3). We have also changed 'PC' to 'PaiCo' throughout the text and figures.

OK.

Line 107/Figure 2: How do some of these frequencies have zero power? Are the authors only plotting the periodicities with significant power? If so, be sure to say it! It is not immediately clear to the reader.

Response:

Yes, that is correct - we are only showing periodicities with significant power. This is stated in the figure caption, but we have clarified it in the text.

OK.

Line 114-115: Really happy the authors put in the sentence “ the difference between the two distributions is within the range calculated between any other two intervals of equivalent length...”! I was curious if the PDFs had a more distinct difference if you compared slightly different intervals, eg. 1200-1900 vs 1900-2000?

Response:

Prior to submitting the paper, we thoroughly investigated this exact point, including stepping that 'split point' progressively further toward the present to see the change in the differences between the two distributions (see below). We agree that this is a really interesting analysis, however we do not have space (in text or figures) to cover all the nuance of this particular point, and instead distilled this analysis into just the information relevant to our question of whether there is a detectable anthropogenic influence on the PWC.

Additionally, we re-did the statistical analysis addressing this point such that the results are more intuitive (Fig. 2d, L132-134), and added the option for readers to play around with similar tests using the provided code.

OK, but what could the 1900-2000 calibration period contribute to this problem?

Response:

We are unsure what Referee 6 is asking here but will try to provide some additional clarity on the calibration period in this response. The comparison period shown in the paper is pre- versus post-1850. So yes: most of the data in the post-1850 distribution has higher skill than the data in the pre-1850 distribution. This disparity increases as the 'split point' steps forward from 1850 to 1900. In the original response to Referee 1, we showed a figure with split points every 20 years from 1830 to 1990, given their interest was in whether the difference

between the two PDFs is strongly dependent on the two time periods being considered. As that figure showed, there is minimal change in the difference between the two PDFs if the split point is at 1900 versus 1850.

OK. I apologize for not being clearer, but yes, the question was whether the 1900-2000 calibration period forces a difference in skill relative to that for data available prior to 1900. As Reviewer 1 originally asked: “I was curious if the PDFs had a more distinct difference if you compared slightly different intervals, eg. 1200-1900 vs 1900-2000?” I think you’ve addressed this with the rebuttal: “As that figure showed, there is minimal change in the difference between the two PDFs if the split point is at 1900 versus 1850.”

Line 122: Could also cite the Seager et al. 2019 paper (mentioned earlier) here

Response:

We have added this citation (L141). OK.

Line 123: Could potentially discuss tropical Pacific Decadal Variability (eg. Power et al. 2021) here

Response:

We have added some discussion of the Pacific decadal variability in the paragraph above (L124-126), and also added Extended Data Fig. 6b comparing our reconstruction with Vance et al.’s (2022) ice core-based reconstruction of the Interdecadal Pacific Oscillation.

OK. But in EF6b, the significance of the correlations should be indicated and the text revised accordingly; for example if a correlation is not significant it should not be described as a weak correlation; or else its p-value should be estimated.

Response:

Extended Data Fig. 6b only shows significant correlations, but we had failed to include that in the caption - we have now added it, and clarified this in the Methods (L797). We also added a denotation of significance to the correlations shown in Extended Data Fig. 6d.

OK.

Line 142-146: This is so cool! I had been worried about the potential influence of Pinatubo on the results given the importance of the 1992-2011 period, but I feel much more comfortable given this analysis. I am still a little worried about the potential apples to oranges comparison as detailed in the Major Points section.

Response:

Comparison of observed or modelled modern/future changes to a paleoclimate reconstruction is reasonably common. In this particular case, we argue that the demonstrated strong agreement between the reconstruction and the observations in the calibration period means that this comparison is valid. To make this more clear, we have added Extended Data Fig. 3 zoomed in on this period of overlap, where the reader can more clearly see the close correspondence between the observations and the reconstruction.

As for EF6B, EF3B ought to be controlled for statistical significance. Here I expect all but a few of these correlations over the calibration period are significant, and might indicate that the differences between the reconstruction techniques, which are all related to one another, are not.

Response:

All correlations reported in Extended Data Fig. 3b are significant ($p < 0.05$), and we have now included this in the figure caption.

OK. And that the RMSE for reconstruction medians are about the same amplitude as the RMSE amongst gridded instrumental product estimates (prior rebuttal point) further supports your point.

Line 149: I again am wondering about the resolution of the proxies considered in this network. Are they all annual or sub annual?

Response:

Yes, they are all sub-annually or annually resolved. This is mentioned in the Methods, and we have now added it to the main text (L68).

OK.

Line 177: “reassigned major eruption”—this is great!

:)

Line 165: how large is this trend?

Response:

At the suggestion of Referee 3, we have added a second axis to the panels of Fig. 4, showing a random sample of the ensemble PWC response to volcanic eruptions. This shows the magnitude of the response.

OK.

Line 172: If specifically referencing a study, please use the name of the study and then superscript it instead of just using the superscripted numbers

Response:

We have changed all instances of this in the text. OK.

Line 190-192: Consider re-writing this sentence for clarity “if all volcanic eruptions with...” I’m not totally sure I follow what you are saying.

Response:

We have re-written this paragraph for better clarity (and also added PMIP3/4 models to this analysis) - L216-220 “When applying the above SEA approach to the CESM1 LME (using the 25 strongest eruptions; Methods), 9 of the 13 CESM1 LME ensemble members produce a significant negative Δ SLP anomaly the year following a volcanic eruption (Fig. 5a), with Δ SLP anomaly magnitudes similar to those occurring during an average El Niño event”.

OK, but see next note.

Response:

Please see the response below.

Line 191: How is significance defined?

Response:

Significance is defined the same way as for the reconstruction. We have added the citation of the method we used to determine significance to make this clear (L218).

Too cryptic a response. Explain in rebuttal what the 'double-bootstrap' method of Rao et al (2019) works and how it's used, because you spend some time explaining later in rebuttal and revised text how it might identify significant warmings prior to volcanic forcing.

Response:

Traditionally, superposed epoch analysis (SEA) has been viewed as a method for comparing changes in the mean of composite timeseries (those composites being a summary of a climatic variable before and after volcanic eruptions). However, this is a slightly reductive view of SEA: more accurately, it is about comparing the condition mean of post-event years with the distribution of all years. The double-bootstrap uncertainty quantification method developed by Rao et al. (2019) accounts for this subtle difference.

Specifically, to define the confidence intervals, we generated 1000 'pseudo-composite' matrices, each constructed using randomly-chosen years of the full reconstruction as the 'eruption' year (two fewer than the number of eruptions being considered in the SEA). These were centred in the same manner as the actual composite matrices (by subtracting the pre-event mean). Then, the 2.5/97.5 percentiles were calculated as the significance thresholds needed to be exceeded for the SEA response to be deemed significant. The confidence intervals prior to the eruption are relatively narrow because by design, the mean for those three years is zero. The intervals widen slightly after the eruption as the conditional distributions revert to the unconditional distribution. As now stated in the text (L207-209), 'significant' volcanic responses may arise in year -2 relative to the eruption, due to the chronological uncertainty incorporated in the reconstruction. Given the pre-eruption mean is calculated with all three years, if year -1 has unusually low Δ SLP, this may predispose the preceding year of the SEA to be outside the confidence interval on the opposite side. This can be seen most clearly in Panel D of Extended Data Fig. 4 (thin grey lines).

The 'double' component of the 'double bootstrap' arises from confidence intervals calculated for the response itself. This is generated simply by resampling (without replacement) the specific set of eruption years used to calculate each composite. This resampling is performed 100 times (or as many times as possible, with unique combinations of eruptions years).

We have added text to the Methods (Section 2.7, L729-732) to provide more details of our application of Rao et al. (2019)'s approach.

OK. And I appreciate the extended explanation in the rebuttal.

Line 195: Consider writing "Fewer LME ensemble members" to help the readers stay on point as you also call your proxy reconstruction an ensemble

Response:

We have made the suggested edit (L223).

OK.

Line 218: Again, this is so cool!

Response:

We have promoted the associated Extended Data figure to the main text to emphasise this point (now Fig. 6).

I think the authors mean Fig 5, as referenced in the text at and near l. 218? OK - except that the significance test is again important here, and needs description in rebuttal and I think in the text.

Response:

I think there may be a little confusion here around the original line numbers, and the line numbers in the revised manuscript. In any case, please see above for a more complete description of the SEA significance test.

OK.

Line 225: Why pCO₂ instead of CO₂? I think CO₂ is more intuitive for readers

Response:

We have removed the rho in both instances (L258, L260).

OK. rho? p? I leave this up to the copy editors.

Response:

Line 271 & 273 in the revised manuscript

Line 248: Did the authors remove the tropical mean from this? See major point for more discussion

Response:

We have re-done this analysis, now assessing relative SSTs in all cases (L764-768, L284-285).

Hard to evaluate without seeing a side-by-side comparison of original and revised results. Did the interpretation change? Please explain in rebuttal if and how the interpretation changed as a result.

Response:

Correcting our Niño 3.4 calculations by removing the tropical mean made minimal difference to the results originally presented in Fig. 5. In the original submission, we only showed results of the SEA performed on Niño 3.4 and Δ SLP from the Community Earth System Model Last Millennium Ensemble.

In the revised submission, we added data from 8 PMIP3/4 models. Interestingly, the degree of disparity between the results of the SEA performed using a) raw Niño 3.4 SST and b) Niño 3.4 SST with the tropical mean removed varied strongly by model. In models with a strong difference, there was a strong (and significant) *negative* SST anomaly in the years following volcanic eruptions. This is due to the global cooling induced by volcanic eruptions. Removing the tropical mean (as suggested by Referee 1) allows inspection of the tropical Pacific response independent of the global signal.

In summary: our original interpretations did not change as a result of this correction.

OK.

Line 259: This manuscript describes a very original and exciting analysis. However, I don't think that the authors have emphasized just how original this source of data is! Many scientists and working groups are interested in finding uses for water isotope data—and this is a great illustration! If there is space, I would definitely hammer on this point a little more.

Response:

We have added text to remind readers how useful water isotope proxy data is. For example, at line 290 “Finally, our novel use of water isotope proxy data to reconstruct atmospheric variability, including explicit incorporation of uncertainty from the training dataset, reconstruction method, and age-depth models....”.
OK but IMHO unnecessary.

Figure 2B. I would make two boxes to indicate just where your delta SLP index comes from

Response:

We have added the two boxes to Figure 1.

OK. This is valuable because it shows that not a single observation comes from the target areas. Worth noting that this suggests the use of a nonlocal reconstruction approach, and why the Iso2k network is nonetheless expected to produce skillful reconstructions for dSLP from these regions. See also note below.

Response:

It is not a new approach to perform statistical reconstructions of a climate mode by using data outside the region used to calculate that mode’s index. For example, this is a common approach for reconstructions of tropical Pacific variability (e.g. D’Arrigo et al., 2005; Dätwyler et al., 2019; Vance et al., 2022, tree ring-based ENSO reconstructions).

We expect the water isotope proxy records of the Iso2k network to produce skill Δ SLP reconstructions due to the demonstrated dominant influence of the PWC on global interannual precipitation $\delta^{18}\text{O}$ (Falster et al., 2021). The underlying reason for this is that the PWC directly or indirectly influences atmospheric circulation processes in many parts of the world—more so than any other globally-relevant climate mode. Atmospheric circulation processes in turn affect the isotopic composition of precipitation. These effects vary according to local/region factors. For example, PWC-related changes in precipitation amount in the tropics may manifest in precipitation $\delta^{18}\text{O}$ via the amount effect. The PWC also affects moisture sources or transport pathways in the extra-tropics, and this too affects precipitation $\delta^{18}\text{O}$ in locations well outside the two boxes used to define the Δ SLP index.

Additionally, our network approach - incorporating many records - buffers against the possibility of changing teleconnections, which helps address the fact that there are no records in the Δ SLP boxes (Batehup et al., 2015).

References

D’Arrigo, R., E. Cook, R. Wilson, R. Allan and M. Mann. 2005. On the variability of ENSO over the past six centuries. *Geophysical Research Letters*, 32(3)

Dätwyler, C, Abram, NJ, Grosjean, M, Wahl, ER, Neukom, R. El Niño–Southern Oscillation variability, teleconnection changes and responses to large volcanic eruptions since AD 1000. *Int J Climatol*. 2019; 39: 2711– 2724.

Vance, T.R., Kiem, A.S., Jong, L.M. et al. Pacific decadal variability over the last 2000 years and implications for climatic risk. *Commun Earth Environ* 3, 33 (2022).

Falster, G., B. Konecky, M. Madhavan, S. Stevenson, and S. Coats, 2021: Imprint of the Pacific Walker Circulation in Global Precipitation $\delta^{18}\text{O}$. *J. Climate*, 34, 8579–8597.

Batehup, R., McGregor, S., and Gallant, A. J. E.: The influence of non-stationary teleconnections on palaeoclimate reconstructions of ENSO variance using a pseudoproxy framework, *Clim. Past*, 11, 1733–1749, <https://doi.org/10.5194/cp-11-1733-2015>, 2015.

See also, for example, Smerdon et al (2010) and Wang et al (2014), which show that skill outside of directly observed areas is often overestimated, and bias underestimated.

To address this issue, please add a partial sentence stating that the reconstruction of the Δ SLP index is by nonlocal estimators requires a nonlocal reconstruction approach, but is nonetheless expected to produce skillful reconstructions for Δ SLP. For instance, this could be done at l. 65 as: “This is the case even though many *individual* precipitation $\delta^{18}\text{O}$ records are not strongly correlated with Δ SLP, and supports the use of a nonlocal reconstruction approach.”

Figure 3. These distributions are very gaussian. Is that expected? How well do these indices agree with the reconstruction?

Response:

The distributions include data from all 4800 ensemble members. Given the large number of samples, we expect the distributions to tend to a normal distribution. When examining skill scores, both individual ensemble members (Extended Data Fig. 4a) and the reconstruction ensemble mean (Extended Data Fig. 3b) agree very well with the target indices. With the new Extended Data Fig. 3, readers can now see this for themselves.

I suspect the reason the reconstructions are Gaussian is because (a) the observations are gaussian or have been standardized, and (b) the reconstruction methods, being covariance-based, impose this. For the skill scores (EF4a, EF3b) this suggests the differences between results using reconstruction methods and different target SLP fields are not very important. It also suggests more analysis and testing using independent validation experiments, e.g. 1951-2000 calibration, reporting the 1900-1950 validation scores in EDT1, and other tests, e.g. with independent subsets of proxies (I suspect independent reconstruction targets are not available.)

Response:

This point—differences between results using reconstruction methods and different target SLP fields are not very important—is an important conclusion of the paper. That is, although there may be small differences in results from ensemble members calculated using different targets/methods (or perhaps, as the referee suggests, no difference), our main findings are robust to these differences.

Irregardless, we provide these independent validation tests in Extended Data Fig. 5, as well as the third column of Extended Data Table 1 (calibration 1951-2000, validation 1900-1950). We acknowledge that we did not clearly signpost this information. We have clarified the relevant figure captions, and added a statement in the main text about these independent tests (L105-108)

OK.

Figure 4: In many of these plots, it looks as though there is a consistent Stronger Walker state observed two years prior to the eruptions in all periods. This is wild! It is quite large in subsets B, C, and D. It might be worth commenting on this unexpected result.

Response:

We have added text with a proposed explanation for this results (L195-197).

This further illustrates the need for a detailed description of the significance testing approach, and for more reliance on independent validation.

Response:

Please see above for a more detailed description of how we calculated the confidence intervals, and how this method may have caused this unexpected result.

OK.

Figure 5A/B/C: How large are these SLP anomalies relative to a traditional El Niño/ La Niña Event?

Response:

They are of very similar magnitude: Δ SLP anomalies during El Niño events in the CESM-LME range from around -0.8 to -4 hPa. We have added that information to the main text (L219-220).

OK, no way for me to evaluate.

Figure 5D/E/F: Have you subtracted the tropical mean from these Niño 3.4 SST estimations? Most of the studies that look at the ENSO response do remove the global tropical mean from SSTs (something very difficult to do with a single proxy record)

Response:

We have re-done this analysis, first subtracting the tropical mean from the Niño 3.4 SST anomalies. We have also added text explaining and discussing this point (L764-768, L284-285). Many thanks for pointing out this error in our original calculations.

OK. Please add the same level of explanation in the text to the Rao et al (2019) based significance testing.

Response:

Please see above for a more detailed description of how we calculated the confidence intervals; this description has been added to the text at L207-209 and L730-732.

OK.

Figure 5G/H/I: Love that you included this! So cool!

Line 498: It would be great if you included a table with each proxy record and cite the individual authors with the database. Also, what resolution are these proxies?

Response:

This table is unfortunately too large to be included in the Extended Data, but is available (including authors of each individual dataset) as supplementary information. All records are annually or sub-annually resolved, and we have added this information to the main text (L68).

OK; I don't have the SI.

OK.

Line 509: “Model simulations” instead of “Model data”

Response:

We have made the suggested change (L479).

OK.

Line 529—great! I don’t think that I got this from the main text. Do you include a table of the skill?

Response:

That table was in the supplementary materials, but has now been promoted to Extended Data Table 1.

If I read the EDT1 caption correctly, this doesn't include 1900-1950 validation scores, but it should: please add, discuss if necessary the presence or absence of evidence for artificial skill.

Response:

Our caption was unclear—the third column of Extended Data Table 1 shows 1900-1950 validation scores. We have edited the caption to make this clear. Comparing values in columns 2 (calibration & validation 1900-2000) and 3 (calibration 1951-2000, validation 1900-1950) demonstrates that the reconstruction remains skillful across all tests when using an independent interval.

OK, but this is still not clear from first reading of EDT1. It would be easy and clearer to insert a third column and label it “Calibration 1951-2000 CE”, and give the calibration scores. Then the fourth column is the existing third column, with header “Validation 1900-1950 CE”, and all is completely clear to the reader.

Line 535: Do you use a January-December year or a tropical (May-April) year when converting records to annual resolution? It’s very important to be clear on what was done.

Response:

We used a calendar year (Jan-Dec) to match the other proxy data. We have added this clarification to the text (L508).

OK: it would have been better to use a tropical year definition, but this may not make much difference. Put the resolution or averaging for the records in question into the appropriate location in the SI.

Response:

If a record is sub-annually resolved, this is stated in the SI table. We have added a statement that the averaging for these records was performed over a calendar year.

OK.

Line 645: If referencing the study by name, please include the author’s name, not just the superscript number.

Response:

We have changed all instances of this in the text.

OK.

Line 656: Matching the SLP variance to that observed over the twentieth century could be a big problem. I discuss this in the major points section.

Response:

This is a valid concern, and one that we spent a lot of time investigating. The method we used is a commonly-used approach for variance scaling in palaeoclimate reconstructions. Additionally, variance changes through time are preserved, and we have added text to the Methods expanding on this point (L623-624).

OK; previously assessed.

Line 741: the years 1992-2011 are superscripted

Response:

We have fixed this formatting error.

OK.

Line 743: just mention the authors and then superscript if directly mentioning a study.

Response:

We have changed all instances of this in the text.

OK.

Line 826: Suggest adding the of the method here-- eg. PAGES2k GMST (#66)

Response:

We added the in-text reference to PAGES 2k (L778).

OK.

G. References

The authors did a good job of including relevant, recent references. However, there were a few papers that merit mention in this discussion (mentioned previously in this review).

Response:

Many thanks for these excellent suggestions. We are limited by the 50-reference limit, but have added most of the below references, in place of some of the references originally included.

OK.

H. Clarity and Context

There are a few places where the authors could improve the clarity of their manuscript to help the readers understand the novelty of the work and significance of their results. I have generally commented on this in the Suggested Improvements Section.

However, there are a few points that would add a lot of clarity to the manuscript in general :

Firstly, A more in depth description of the proxies that make up the reconstruction. Table S1 cites the Iso2k database, but does not directly reference the individual authors who made many of these proxy datasets. This is a small point, but many paleo people that aren't familiar with the ISO2k database would get much more out of this table if you directly wrote "Bagnato et al. 2005 (14)" instead of #14 and the ISO2k ID number. I'm also a little confused about the "unknown" resolution or seasonality. Does this mean variable resolution?

Response:

We have added more informative citations to the table. For seasonality, 'unknown' refers to records where the authors of the primary dataset did not claim any particular seasonality for their data, but also did not explicitly state that the record represents an annually integrated signal. We have added clarifying text to the table caption.

OK. I don't have Table S1, cannot comment further.

OK

Secondly, I think that it would help the readers if the authors included their specific definition of Walker Circulation Variability in the main text (presently included in the methods). Suggestion that the authors add two boxes to Figure 1 to highlight the regions where SLP anomalies are considered. It'd really help the clarity if this was directly addressed this early on (I think early on in the manuscript the authors clarify that their index isn't related to hydroclimate or SST, but don't say exactly what it is). I also think that adding a sentence to describe just how these proxies mechanistically relate to Walker Circulation Variability would be extremely helpful for readers unfamiliar with the Falster et al. Journal of Climate paper.

Response:

We have added the index reference boxes to the map in Figure 1. We also added text summarising the main findings of the Falster et al. JCLim paper, in terms of how water isotopes relate to the Walker circulation (L64-67).

This needs revision and more explanation. The reviewer stated it well: " adding a sentence to describe just how these proxies mechanistically relate to Walker Circulation Variability would be extremely helpful for readers unfamiliar with the Falster et al. Journal of Climate paper." As noted previously, this is important, as all the proxies are nonlocal to the target dSLP regions indicated by the black boxes.

Response:

We have added more detailed text outlining the reason we expect globally-distributed water isotope records to contain information about the Pacific Walker circulation (L64-71).

OK, good; and see prior request for partial sentence on nonlocal reconstruction to be added at l. 65.

Author Rebuttals to Second Revision:

Response to Referee 6 for the revised manuscript “**Forced changes in the Pacific Walker Circulation over the past millennium**”, submitted to *Nature* by Falster et al.

We have made all additional revisions exactly as suggested by Referee 6.

Specifically, we have:

- Re-coloured Extended Data Figure 3b so the colours now reflect the significance of the correlation coefficients shown. We updated the corresponding figure caption (L1283).
- Changed ‘can’ to ‘may’ (L41).
- Made the suggested changes at L74-77 (exactly as outlined by the Referee). The first two sentences of the paragraph now read “*The first mode of observed global interannual precipitation $\delta^{18}O$ over 1982–2015 is significantly ($p < 0.05$) correlated with and explains 55% of the ΔSLP variance²². This is the case even though many individual precipitation $\delta^{18}O$ records are not highly or significantly correlated with ΔSLP ²², and supports the use of a nonlocal reconstruction approach.*”
- Edited Extended Data Table 1 to include an additional column ‘Calibration 1951-2000’ between columns ‘Calibration 1900-2000’ and ‘Validation 1900-1950’. We also updated the associated caption (L1408-1410).

These edits address all comments written in **bold purple text** in ‘6_reviewer_attachment_1_1686051405.pdf’.